# Generative Evolutionary Meta-Solver (GEMS): Scalable Surrogate-Free Multi-Agent Reinforcement Learning

**Alakh Sharma** *                                          *f20240593@pilani.bits-pilani.ac.in*
*Birla Institute of Technology and Science, Pilani*

**Gaurish Trivedi** *                                        *f20220728@pilani.bits-pilani.ac.in*
*Birla Institute of Technology and Science, Pilani*

**Kartikey Singh Bhandari**                                  *p20241006@pilani.bits-pilani.ac.in*
*Birla Institute of Technology and Science, Pilani*

**Yash Sinha**                                               *yash.sinha@pilani.bits-pilani.ac.in*
*Birla Institute of Technology and Science, Pilani*

**Dhruv Kumar**                                              *dhruv.kumar@pilani.bits-pilani.ac.in*
*Birla Institute of Technology and Science, Pilani*

**Pratik Narang**                                            *pratik.narang@pilani.bits-pilani.ac.in*
*Birla Institute of Technology and Science, Pilani*

**Jagat Sesh Challa**                                        *jagatsesh@pilani.bits-pilani.ac.in*
*Birla Institute of Technology and Science, Pilani*

**Reviewed on OpenReview:** *https://openreview.net/forum?id=ZwEJsXoBHD*

## Abstract

Scalable multi-agent reinforcement learning (MARL) remains a central challenge for AI. Existing population-based methods, like Policy-Space Response Oracles, PSRO, require storing explicit policy populations and constructing full payoff matrices, incurring quadratic computation and linear memory costs. We present Generative Evolutionary Meta-Solver (GEMS), a surrogate-free framework that replaces explicit populations with a compact set of latent anchors and a single amortized generator. Instead of exhaustively constructing the payoff matrix, GEMS relies on unbiased Monte Carlo rollouts, multiplicative-weights meta-dynamics, and a model-free empirical-Bernstein UCB oracle to adaptively expand the policy set. Best responses are trained within the generator using an advantage-based trust-region objective, eliminating the need to store and train separate actors. We evaluated GEMS in a variety of Two-player and Multi-Player games such as the Deceptive Messages Game, Kuhn Poker and Multi-Particle environment. We find that GEMS is up to **6×** faster, has **1.3×** less memory usage than PSRO, while also reaps higher rewards simultaneously. These results demonstrate that GEMS retains the game theoretic guarantees of PSRO, while overcoming its fundamental inefficiencies, hence enabling scalable multi-agent learning in multiple domains.

---

*Joint first authors

# 1 Introduction

Imagine organizing a round-robin tennis tournament with hundreds of players (Fig. 1). Scheduling every match gives a complete ranking but is clearly inefficient: the number of matches grows quadratically, and the results table quickly becomes unwieldy.

Training AI agents in two-player zero-sum Markov games faces a similar challenge. Population-based methods such as Policy-Space Response Oracles (Psro) (Lanctot et al. (2017)) maintain a growing set of $k$ policies and explicitly construct a $k \times k$ payoff matrix, where each entry records the expected outcome of a policy against another. A meta-solver then updates the distribution over policies, analogous to computing player rankings from the tournament table.

While conceptually straightforward, classical Psro suffers from three critical bottlenecks: ❶ *Memory overhead*: storing a new policy for each player leads to linear growth in storage; ❷ *Computation overhead*: filling in the full $k \times k$ payoff matrix quickly becomes infeasible; ❸ *Scalability of new entries*: adding a new policy requires training and storing another separate actor. Prior work has mitigated some of these costs through selective best-response training (Smith et al. (2021)), Double Oracle methods (McAleer et al. (2021); Huang et al. (2022)), meta-game improvements (McAleer et al. (2022b),McAleer et al. (2022a)), and knowledge transfer for new agents (Smith et al. (2023); Lian et al. (2024)). However, these approaches retain the core paradigm of explicit policy sets, leaving scalability fundamentally limited.

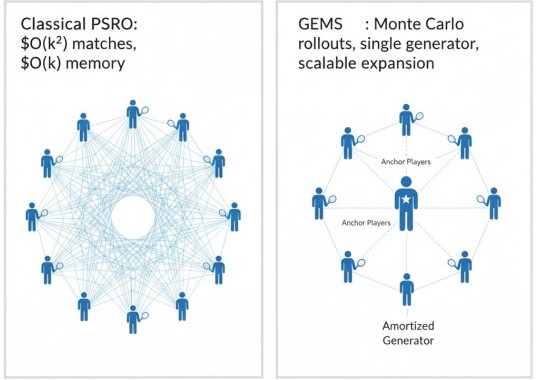

Figure 1: **Tournament analogy for policy populations.** (*Left*) Psro: explicit $k \times k$ payoff matrix with all pairwise matchups. (*Right*) Gems: compact anchor set of latent policies, with a single generator producing diverse strategies on demand.

Gems overcomes these limitations while preserving the game-theoretic guarantees of Psro. Analogously to running a large tournament without scheduling every match, ❶ Gems maintains a compact *anchor set* of latent codes that represent active "players". ❷ Gems treats the payoff matrix as conceptual and queries it through Monte Carlo rollouts, which are unbiased in nature. ❸ Meta-strategy is updated using multiplicative-weights discretization of replicator dynamics, which are akin to adjusting ranking based on samples. ❹ **Empirical-Bernstein Upper Confidence Bound (UCB)** (Maurer & Pontil, 2009) selects promising new "players" from a candidate pool, thus expanding the population. ❺ Finally, Gems incorporates an amortized generator with an **A**dvantage-based **B**est-**R**esponse objective and **T**rust-region **R**egularization (ABR-TR) that eliminates the need for separate policies. We evaluated Gems in a variety of Two-player and Multi-Player games such as the Deceptive Messages Game, Kuhn Poker (Kuhn, 2016) and Multi-Particle environments (Terry et al., 2021). We find that Gems is up to $6\times$ faster, has $1.3\times$ less memory usage than Psro, while also reaps higher rewards simultaneously.

**Contributions.** Relative to classical Psro, Gems achieves ❶ *memory efficiency*: Gems replaces $O(k)$ stored players with a single versatile generator. ❷ *Computation efficiency*: Gems avoids Psro's quadratic payoff tables using Monte Carlo estimates, scaling per iteration with the number of sampled matches and the candidate pool size. ❸ *Scalable new entries*: EB-UCB identifies strong candidates, and ABR-TR integrates them into the generator without adding new actors. ❹ *Theoretical guarantees*: unbiased MC meta-gradients, instance-dependent regret bounds for EB-UCB, external regret bounds for multiplicative-weights dynamics, and finite-population exploitability accounting for approximate best responses. Together, these advances transform the exhaustive "tournament book-keeping" of Psro into a leaner, scalable framework, closer to how real tournaments operate, where only select matches determine rankings and strategies adapt efficiently from sparse outcomes.

## 2 Related Work

**Psro and its Variants.** The seminal work of Lanctot et al. (2017) introduced Psro as a general framework for multi-agent reinforcement learning. Psro iteratively expands a population of policies by training a new best-response policy against a meta-game mixture of the existing population. A core challenge of Psro is its reliance on a full $k \times k$ payoff matrix, where $k$ is the number of policies in the population. This leads to $\mathcal{O}(k^2)$ computation overhead per iteration and $\mathcal{O}(k)$ memory overhead to store individual policies, both of which become bottlenecks in large-scale settings (Shao et al., 2024; Zhou et al., 2022). This has motivated a line of work on more efficient variants. For instance, EFFICIENT PSRO (E-psro) (Zhou et al., 2022) minimizes evaluation costs by formulating the solver as an unrestricted-restricted game. Similarly, Smith et al. (2021) propose training a best-response against a single opponent, reducing computation but potentially limiting diversity. The double oracle (DO) family of algorithms, including XDO (McAleer et al., 2021) and EDO (Huang et al., 2022), focus on finding equilibria more efficiently, often by guaranteeing linear convergence or monotonic exploitability reduction (McAleer et al., 2022b). *However, these methods can still suffer from issues like performance oscillations or slow convergence due to the need for a full set of strategies (Huang et al., 2022). In contrast, GEMS avoids these issues entirely by replacing the explicit payoff matrix with unbiased Monte Carlo rollouts and the discrete set of policies with a single, amortized generator.*

Other work has addressed the computational cost of training new policies from scratch at each iteration, which is a significant bottleneck (Li et al., 2023). Notably, PIPELINE PSRO (P2SRO) (McAleer et al., 2020) mitigates this by parallelizing the training of best responses. To further address this, FUSION-PSRO (Lian et al., 2024) uses policy fusion to initialize new best responses, while STRATEGIC KNOWLEDGE TRANSFER (Smith et al., 2023) explores transferring knowledge across changing opponent strategies. *Our work tackles this problem differently by using a single amortized generator, which removes the need to store and train separate models for each policy.*

Finally, some papers have focused on improving the meta-game solution or policy diversity. ALPHA-PSRO (Muller et al., 2019) replaces the Nash meta-solver with $\alpha$-Rank to scale Psro to general-sum, many-player settings. A-PSRO (Hu et al., 2023) introduces an advantage-based evaluation for strategy selection, providing a unified objective across zero-sum and general-sum games. ANYTIME PSRO (McAleer et al., 2022b) and SELF-PLAY PSRO (SP-Psro) (McAleer et al., 2022a) aim to improve convergence by adding high-quality policies to the population. Other works introduce new metrics for policy diversity to ensure better approximation of the Nash Equilibrium (Tang et al., 2025; Yao et al., 2023). *While these works aim to improve Psro, they do not fundamentally change its core structure, which still relies on an explicit population of policies and a payoff matrix.*

**Neural Population Learning.** Recent work has aggressively explored unifying policy populations into single conditional networks to enable positive transfer and minimize storage. NEURAL POPULATION LEARNING (NEUPL) (Liu et al., 2022b) represents an entire population of best-response policies using a single neural network that is explicitly conditioned on which strategy is being executed. Formally, this network defines a conditional policy $\pi(a \mid s, \sigma)$ that outputs a distribution over actions $a$ given the current state (or observation) $s$, where $\sigma$ is a learnable discrete strategy index identifying a particular population member. Addressing the limitations of discrete indexing, SIMPLEX-NEUPL (Liu et al., 2022a) introduces a geometric formulation in which the policy conditions on continuous simplex vectors, enabling direct representation of mixed strategies within a single forward pass. Furthermore, NEUPL-JPSRO (Liu et al., 2024) extends this paradigm to general-sum games by combining conditional policies with Joint-PSRO objectives, effectively learning correlated joint policies that approximate coarser equilibrium concepts such as CCE. *However, despite amortizing memory costs, these methods often retain the computational bottlenecks of empirical game theory, specifically the $\mathcal{O}(k^2)$ cost of estimating pairwise payoffs to update the strategy distribution (Liu et al., 2022b). Distinct from the conditional architectures of the NEUPL family, GEMS introduces a fully generative meta-solver that maps a latent geometry directly to policy parameters via a hypernetwork. This formulation allows GEMS to bypass explicit population management entirely, utilizing unbiased Monte Carlo rollouts and latent-space evolutionary optimization (OMWU) to solve the game without constructing a full payoff matrix.*

Table 1: **Theoretical comparison of GEMS and PSRO-style methods.** Let $N_t$ denote the cumulative number of policies found by iteration $t$. $C_{\text{eval}}$ is the cost of one payoff evaluation (e.g., one episode), and $k$ is the number of sampled opponents per iteration. *Memory Scaling* refers to the storage required for the meta-game state (payoff matrix / sampled entries). *Eval. Cost* refers to the complexity of evaluating a new policy update. While standard PSRO variants scale quadratically in meta-game memory as the population grows ($N_t \propto t$), GEMS maintains constant meta-game scaling ($\mathcal{O}(1)$) via its generative parameterization.

| Method | Payoff Matrix | Memory (Meta-Game) | Eval. Cost (per iter.) |
|---|---|---|---|
| PSRO | Required (Full) | $\mathcal{O}(N_t^2)$ | $\mathcal{O}(N_t\,C_{\text{eval}})$ |
| Alpha-PSRO | Required | $\mathcal{O}(N_t^2)$ | $\mathcal{O}(N_t\,C_{\text{eval}})$ |
| APSRO | Sampled | $\mathcal{O}(N_t)$–$\mathcal{O}(N_t^2)$ | $\mathcal{O}(k\,C_{\text{eval}})$ |
| EPSRO | Sampled | $\mathcal{O}(N_t)$–$\mathcal{O}(N_t^2)$ | $\mathcal{O}(k\,C_{\text{eval}})$ |
| P2SRO | Sampled / local | $\mathcal{O}(N_t)$–$\mathcal{O}(N_t^2)$ | $\mathcal{O}(k\,C_{\text{eval}})$ |
| NeuPL | Sampled / online | $\mathcal{O}(N_t)$–$\mathcal{O}(N_t^2)$ | $\mathcal{O}(k\,C_{\text{eval}})$ |
| **GEMS (Ours)** | **Not Required** | $\mathcal{O}(1)$ | $\mathcal{O}(k\,C_{\text{eval}})$ |

*Note:* GEMS utilizes a fixed-size latent anchor set $|Z| = K$ (where $K \ll t$ is a constant hyperparameter) to represent the population. Therefore, its meta-game memory complexity remains $\mathcal{O}(1)$ relative to training iterations $t$, whereas PSRO-based methods typically accumulate a growing population history ($N_t \propto t$). **All methods additionally store policy parameters;** the *Memory Scaling* column isolates *meta-game* (payoff/state) storage rather than network weights.

**Game-Theoretic Methods for Multi-Agent Learning.** Beyond the PSRO paradigm, other game-theoretic approaches exist for multi-agent learning. COUNTERFACTUAL REGRET MINIMIZATION (CFR) (Zinkevich et al., 2007) is a well-known method for extensive-form games, but it typically requires explicit state enumeration, making it challenging to scale to large or continuous domains. Some work has explored bridging the gap between PSRO and CFR, as seen in the unified perspective proposed by Wang et al. (2022). Other approaches include methods based on fictitious play, such as FICTITIOUS CROSS-PLAY (FXP) (Xu et al., 2023), which combines self-play and PSRO to find global Nash equilibria. However, many self-play methods lack theoretical guarantees for general-sum games and can struggle with convergence in mixed cooperative-competitive settings (Xu et al., 2023).

*Our work contributes to this broader landscape by offering a surrogate-free, amortized framework that is both memory and computationally efficient. While many variants of PSRO have been proposed, they have largely retained the core structure of maintaining an explicit policy population and a large payoff matrix. GEMS breaks from this paradigm by using an amortized generator and unbiased Monte Carlo rollouts, thereby sidestepping the fundamental scalability issues inherent to classical PSRO. As summarized in Table 1, this design removes the need to store an explicit meta-game payoff matrix and prevents meta-game state from growing with the population, while retaining a per-iteration evaluation profile comparable to sampled PSRO variants.*

## 3  Proposed Method: GEMS

GEMS circumvents PSRO's limitations by being **surrogate-free.**[1] Instead of storing $k$ explicit actor models and computing their full payoff matrix, it maintains a single generative model $G_\theta$ that maps low-dimensional latent codes to policies. Consequently, the method scales to massive conceptual populations while storing only the generator and a set of latent "anchor" codes.

GEMS unfolds in an iterative loop: It first solves for the equilibrium of the current meta-game using noisy estimates, then expands the game by finding an approximate best response via a bandit-like oracle. The full procedure is detailed below.

While the core exposition focuses on two–player zero-sum games for clarity, GEMS framework extends naturally to more general settings. This extensibility is demonstrated by benchmarking GEMS on two-player

---

[1]In this context, "surrogate-free" signifies that GEMS does not maintain an explicit, discrete set of policies to approximate the game.

general-sum and, more broadly, $n$-**player general-sum** games. For the multi-player environments, the implementation leverages the `PettingZoo` library (Terry et al., 2021). The necessary modifications to the meta-game estimators, per-player oracles, and convergence guarantees are provided in Appendix II.

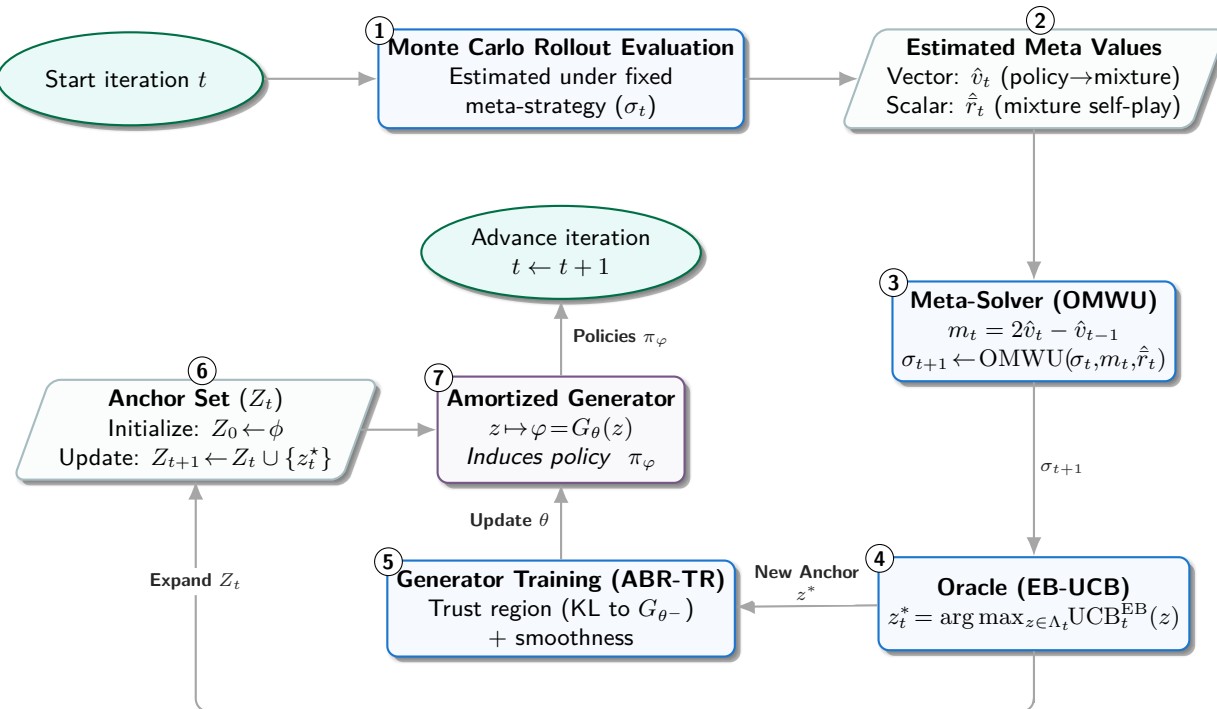

Figure 2: At each iteration $t$, Monte Carlo rollouts evaluate the current policy mixture under a fixed meta–strategy $\sigma_t$, producing estimated meta–values $\hat{v}_t$ (policy-to-mixture) and $\hat{\bar{r}}_t$ (mixture self-play). An optimistic meta-solver updates the mixture via OMWU using the hint $m_t = 2\hat{v}_t - \hat{v}_{t-1}$. An EB-UCB oracle then selects a new latent anchor $z_t^*$ from the candidate set, which is incorporated through amortized generator training with a trust-region objective (ABR-TR). The anchor set $Z_t$ is expanded accordingly, the generator induces updated policies $\pi_\varphi$, and the iteration advances to $t+1$. Green ellipses denote temporal iteration boundaries rather than algorithmic operations.

Figure 2 provides a high-level overview of the GEMS algorithmic loop. Rather than explicitly maintaining a growing population of policies and their full payoff matrix, GEMS operates on a compact latent representation: a fixed-size generator coupled with an evolving set of latent anchors. At each iteration, Monte Carlo rollouts estimate the meta-game values under the current mixture, which are then used by an optimistic meta-solver to update the population distribution. An EB-UCB oracle selects a new latent anchor corresponding to an approximate best response, which is incorporated via trust-region amortized generator training. The anchor set is expanded, the generator induces updated policies, and the process advances to the next iteration. This closed-loop structure captures how GEMS alternates between equilibrium refinement and controlled population expansion while avoiding explicit policy enumeration.

## 3.1 Formal Setup and Generative Representation

Consider a two-player zero-sum Markov game. Let $r(\pi_i, \pi_j) \in [0, 1]$ denote the expected return for Player 1 when policy $\pi_i$ faces $\pi_j$. For a finite policy set $A = \{\pi_1, \ldots, \pi_k\}$, this induces a payoff matrix $M \in [-1, 1]^{k \times k}$ where $M_{ij} = \mathbb{E}[r(\pi_i, \pi_j)]$. **Crucially, GEMS never explicitly constructs or stores this matrix $M$.**

At each iteration $t$, GEMS maintains three core components:

1. An **anchor set** of latent codes $Z_t = \{z_1, \ldots, z_{k_t}\} \subset \mathbb{R}^d$, where $d$ is the dimension of the latent space, representing policies in a low-dimensional space.

2. A single **generator network** $G_\theta$ [2] that maps a latent code $z$ to policy parameters $\varphi = G_\theta(z)$, thereby defining a policy $\pi_\varphi$.

3. A **meta-strategy** $\sigma_t \in \Delta_{k_t - 1}$, which is a probability distribution over the current anchor set $Z_t$, representing the Nash equilibrium of the restricted game $A_t = \{\pi_{G_\theta(z)} : z \in Z_t\}$, where $\Delta_{k_t - 1}$ is the $(k_t - 1)$-dimensional probability simplex [3].

These components define the key quantities for analyzing the meta-game. The vector of expected payoffs for each anchor policy against the meta-strategy $\sigma_t$ is $v_t = M\sigma_t$, and the expected value of the game at iteration $t$ is $\bar{r}_t = \sigma_t^\top M \sigma_t$. A primary objective is to minimize **exploitability**, the incentive for any player to deviate from $\sigma_t$:

$$\text{Exploit}(\sigma_t) = \max_{i \leq k_t} e_i^\top M \sigma_t \ - \ \sigma_t^\top M \sigma_t, \tag{1}$$

where $e_i$ is the $i$-th standard basis vector. GEMS iteratively refines $\sigma_t$ and expands $Z_t$ to drive this exploitability toward zero.

## 3.2 Estimating the Meta-Game without the Matrix

Because GEMS does not access the true payoff matrix $M$, it estimates the values of $v_t$ and $\bar{r}_t$ through simulation. **Adopting a two-time-scale assumption in which the meta-strategy $\sigma_t$ is held fixed during estimation**, GEMS employs Monte Carlo rollouts to obtain statistically sound estimates.

For each anchor policy $i \in [k_t]$, the algorithm samples $n_i$ opponents $j_s \sim \sigma_t$ and executes $m$ game episodes per pair, yielding returns $Y_{i,s,\ell} \in [0,1]^4$ . This process yields a simple yet powerful estimator for per-policy performance. To estimate the overall game value, the procedure additionally samples $B$ independent policy pairs $(i_b, j_b) \sim \sigma_t \times \sigma_t$ and averages their outcomes across $m$ episodes each. The resulting estimators are

$$\hat{v}_{t,i} = \frac{1}{n_i m} \sum_{s=1}^{n_i} \sum_{\ell=1}^{m} Y_{i,s,\ell}, \quad \hat{\bar{r}}_t = \frac{1}{Bm} \sum_{b=1}^{B} \sum_{\ell=1}^{m} Y_{i_b, j_b, \ell}, \quad (i_b, j_b) \sim \sigma_t \times \sigma_t, \tag{2}$$

and constitute the empirical backbone of GEMS.

**Lemma 3.1** (Unbiasedness and Empirical-Bernstein Concentration). *With rewards in $[0,1]$, the estimators are unbiased: $\mathbb{E}[\hat{v}_{t,i}] = (M\sigma_t)_i$ and $\mathbb{E}[\hat{\bar{r}}_t] = \sigma_t^\top M \sigma_t$. Moreover, for any $\delta \in (0,1)$, with probability at least $1 - \delta$,*

$$\left| \hat{v}_{t,i} - (M\sigma_t)_i \right| \leq \sqrt{\frac{2 \widehat{\text{Var}}_{t,i} \ln(2/\delta)}{n_i m}} + \frac{3 \ln(2/\delta)}{n_i m - 1}, \qquad \left| \hat{\bar{r}}_t - \sigma_t^\top M \sigma_t \right| = O\left( \sqrt{\frac{\ln(1/\delta)}{Bm}} \right), \tag{3}$$

*where $\widehat{\text{Var}}_{t,i}$ is the empirical variance of $\{Y_{i,s,\ell}\}$.*

*Proof sketch.* Unbiasedness follows from the law of total expectation over the sampling of opponents, and the concentration bounds apply the empirical-Bernstein inequality for bounded random variables. $\qquad\square$

## 3.3 Solving the Meta-Game via Optimistic Replicator Dynamics

Given the estimated payoffs $\hat{v}_t$ and game value $\hat{\bar{r}}_t$, GEMS updates the meta-strategy from $\sigma_t$ to $\sigma_{t+1}$. GEMS adopts the **Optimistic Multiplicative Weights Update (OMWU)** algorithm (Daskalakis & Panageas (2018)), an adaptive discretization of replicator dynamics. Rather than relying solely on the current payoff estimate $\hat{v}_t$, OMWU incorporates a predictive "hint" about the next payoff. Specifically, it forms the optimistic estimate $m_t = 2\hat{v}_t - \hat{v}_{t-1}$, with $\hat{v}_0 = \mathbf{0}$. The meta-strategy then updates according to

$$\sigma_{t+1}(i) \ \propto \ \sigma_t(i) \exp\left( \eta_t [\, 2\hat{v}_{t,i} - \hat{v}_{t-1,i} - \hat{\bar{r}}_t \,] \right), \qquad \eta_t > 0, \tag{4}$$

---

[2] $G_\theta$ denotes the generator network parameterized by $\theta$, which represents its learnable weights.

[3] The simplex $\Delta_n = x \in \mathbb{R}^{n+1} : x_i \geq 0, \sum_i x_i = 1$ denotes the space of probability distributions over $(n+1)$ elements.

[4] $i$ indexes anchor policies in the current population, $s$ indexes sampled opponents drawn from the meta-strategy $\sigma_t$, and $\ell$ indexes independent Monte Carlo rollouts (episodes) used to estimate their expected returns.

followed by normalization to ensure $\sigma_{t+1} \in \Delta_{k_t-1}$.[5] This optimistic step yields stronger theoretical guarantees: regret now scales with the cumulative variation of the payoff vectors rather than the iteration horizon $T$.

**Proposition 3.2** (External Regret of OMWU under Unbiased Noise)**.** *Assume payoffs in $[0, 1]$. For any sequence of payoff estimates $\hat{v}_t$ satisfying $\mathbb{E}[\hat{v}_t \mid \sigma_t] = M\sigma_t$ and a constant step size $\eta$, the average external regret obeys*

$$\frac{1}{T}\sum_{t=1}^{T}\Big(\max_i e_i^\top M\sigma_t - \sigma_t^\top M\sigma_t\Big) \ \leq \ O\Big(\frac{1}{T}\sqrt{\ln k_T \sum_{t=1}^{T}\|v_t - v_{t-1}\|_\infty^2}\Big) \ + \ \frac{1}{T}\sum_{t=1}^{T}\mathbb{E}\big[\|\,\hat{v}_t - M\sigma_t\|_\infty\big], \quad (5)$$

*where $v_t = M\sigma_t$ are the true expected payoffs. The first term, driven by the variation of the meta-game, leads to faster convergence in slowly changing environments such as those induced by GEMS.*

### 3.4 Finding New Strategies with a Bandit Oracle

Solving the restricted meta-game alone is insufficient; progress requires introducing new, challenging policies into the population. GEMS casts this search as a **multi-armed bandit problem** (Robbins (1952)). The "arms" comprise a finite pool of candidate latent codes $\Lambda_t$, and pulling an arm corresponds to evaluating the policy $G_\theta(z)$ against the current meta-strategy $\sigma_t$.

For a latent code $z \in \Lambda_t$, define

$$f_t(z) \ = \ \mathbb{E}_{j\sim\sigma_t}\big[r\big(\pi_{G_\theta(z)}, \pi_j\big)\big]. \tag{6}$$

GEMS estimates this value, $\hat{\mu}_t(z)$, together with its empirical variance $\widehat{\mathrm{Var}}_t(z)$ via Monte Carlo rollouts. To balance exploration and exploitation efficiently, it employs an **empirical-Bernstein Upper Confidence Bound (EB-UCB)** (Maurer & Pontil (2009)), which leverages variance information for tighter bounds. A Jacobian penalty encourages smoothness in the generator's latent space, aiding optimization, where $JG_\theta(z)$ denotes the Jacobian matrix of the generator with respect to its input $z$. The score assigned to candidate $z$ is

$$\mathrm{UCB}_t^{\mathrm{EB}}(z) \ = \ \hat{\mu}_t(z) + \sqrt{\frac{2\widehat{\mathrm{Var}}_t(z)\ln(3/\delta_t)}{n_z m}} + \frac{3\ln(3/\delta_t)}{n_z m - 1} \ - \ \lambda_J\|JG_\theta(z)\|_F^2, \tag{7}$$

where $\delta_t = t^{-2}$ is a decaying confidence parameter and $\lambda_t \geq 0$ is a penalty coefficient. GEMS selects:

$$z_t^\star \ = \ \arg\max_{z\in\Lambda_t}\mathrm{UCB}_t^{\mathrm{EB}}(z), \tag{8}$$

and augments the anchor set:

$$Z_{t+1} \leftarrow Z_t \cup \{z_t^\star\}. \tag{9}$$

**Theorem 3.3** (Instance-Dependent Oracle Regret)**.** *Assume rewards lie in $[0, 1]$ and the bandit problem admits a unique best arm $z^\star$ with sub-optimality gaps $\Delta_z > 0$. Under the two-time-scale assumption (fixed $\sigma_t$ during selection), the cumulative regret of the oracle satisfies*

$$\sum_{t=1}^{T}\mathbb{E}\big[f_t(z^\star) - f_t(z_t^\star)\big] \ = \ O\Big(\sum_{z\neq z^\star}\frac{\ln T}{\Delta_z}\Big) \ + \ \lambda_J\sum_{t=1}^{T}\mathbb{E}\big[\|JG_\theta(z_t^\star)\|_F^2\big]. \tag{10}$$

*If the Jacobian norm is uniformly bounded, the second term can be controlled by annealing $\lambda_J$.*

*Proof sketch.* Standard bandit-analysis arguments apply. With high probability, the EB-UCB provides valid confidence bounds; comparing the chosen arm $z_t^\star$ with the optimal arm $z^\star$ and summing over time yields the stated instance-dependent regret bound. □

---

[5]In the context of the Optimistic Multiplicative Weights Update (OMWU) algorithm, $\eta_t$ is the step size, often referred to as the learning rate or gain parameter.

### 3.5 Training the Generator with Amortized Best Response

Once a promising latent code $z_t^\star$ has been identified, GEMS must ensure that the generator $G_\theta$ can realize the associated high-performing policy while retaining its ability to produce previously effective policies. This is achieved via an **Amortized Best-Response with a Trust Region (ABR-TR)** objective, inspired by trust-region methods in deep reinforcement learning such as TRPO and PPO (Schulman et al. (2015; 2017)).

Rather than training a new network from scratch, GEMS fine-tunes the existing generator $G_\theta$ to maximize the performance of a curated set of promising latent codes, drawn from a distribution $q_t$, against the opponent mixture $\sigma_{t+1}$. A KL-divergence penalty against a frozen, older generator $\theta^-$ serves as a trust region, mitigating catastrophic forgetting:

$$\mathcal{L}_{\text{ABR-TR}}(\theta) \;=\; \mathbb{E}_{z\sim q_t,\, j\sim\sigma_{t+1}}\Big[\widehat{A}_{\pi_{G_\theta(z)}} \;-\; \beta\,\text{KL}\big(\pi_{G_\theta(z)} \,\|\, \pi_{G_{\theta^-}}(z)\big) \;-\; \lambda_J\|JG_\theta(z)\|_F^2\Big], \tag{11}$$

where $\widehat{A}$ denotes a suitable advantage estimator, $\beta > 0$ is a hyperparameter controlling trust-region strength and $JG_\theta(z)$ denotes the Jacobian matrix of the generator with respect to its input $z$. Taking a few gradient-ascent steps on $\mathcal{L}_{\text{ABR-TR}}$ each iteration amortizes best-response computation while preserving a stable, single generator.

### 3.6 Overall Exploitability Bound

The preceding components now combine to yield a convergence guarantee for the entire GEMS algorithm. To make this composition explicit, we first state a decomposition result that separates the average exploitability into four interpretable sources of error:

➊ inherent regret of the OMWU meta-strategy solver (under time-varying payoffs),

➋ noise from Monte Carlo payoff estimation,

➌ sub-optimality from bandit-based oracle (anchor) selection,

➍ approximation error induced by amortized generator training.

**Amortized best-response gap.** Let $\pi_{\text{BR}}(\pi_j)$ be the true best response to an opponent policy $\pi_j$. We define the amortized best-response error as the worst-case expected gap (over iterations) between a true best response and the policy produced by the generator at the selected latent code $z_t^\star$:

$$\varepsilon_{\text{BR}} \;=\; \sup_t \mathbb{E}_{j\sim\sigma_t}\Big[r\big(\pi_{\text{BR}}(j),\pi_j\big) \;-\; r\big(\pi_{G_\theta(z_t^\star)},\pi_j\big)\Big] \;\geq\; 0. \tag{12}$$

**Exploitability decomposition.** Let $M$ denote the (implicit) meta-game payoff operator and define the true payoff vector $v_t := M\sigma_t$ at iteration $t$. Let $\hat{v}_t$ be the Monte-Carlo estimator of $v_t$ computed from $nm$ rollouts per queried matchup (as in Section 3.2). Let $z_t^\star$ be the (possibly approximate) anchor selected by the EB-UCB oracle at iteration $t$, and let $\Delta_z$ denote the suboptimality gap for anchor $z$.

**Proposition 3.4** (Exploitability decomposition). *Assume rewards lie in $[0,1]$. At each iteration $t$, the instantaneous exploitability satisfies*

$$Exploit(\sigma_t) \;\leq\; \underbrace{\big\langle v_t - v_{t-1},\, \sigma_t - \sigma_t^\star\big\rangle}_{\text{OMWU meta-dynamics term}} \;+\; \underbrace{2\|\hat{v}_t - v_t\|_\infty}_{\text{MC estimation error}} \;+\; \underbrace{\varepsilon_{\text{BR}}}_{\text{amortized BR error}} \;+\; \underbrace{\Big(\max_z v_t(z) - v_t(z_t^\star)\Big)}_{\text{oracle suboptimality}}. \tag{13}$$

*Here $\sigma_t^\star$ denotes the best fixed meta-strategy in hindsight for the (time-varying) payoff sequence.*

*Proof sketch.* Starting from the definition of exploitability as a best-response gap against $\sigma_t$, we add and subtract (i) Monte-Carlo payoffs $\hat{v}_t$, (ii) the best-anchor payoff $\max_z v_t(z)$, and (iii) the generator-induced best response at $z_t^\star$. Triangle inequalities yield the MC term and the amortized BR term, while the oracle term captures the loss from selecting $z_t^\star$ instead of $\arg\max_z v_t(z)$. The remaining meta-dynamics term is controlled by the OMWU guarantee for time-varying payoffs.

**Overall finite-population bound.** We now upper bound each term in Proposition 3.4 using the corresponding component guarantees.

**Theorem 3.5** (Finite-Population Exploitability Bound). *Assume rewards lie in* $[0,1]$ *and that oracle selection operates on a two-time-scale schedule. With OMWU step size* $\eta = \Theta\big(\sqrt{\ln k_T/T}\big)$ *and Monte-Carlo budget satisfying* $\mathbb{E}[\|\hat{v}_t - v_t\|_\infty] = O\big((nm)^{-1/2}\big)$ *for each* $t$*, the average exploitability obeys*

$$
\frac{1}{T}\sum_{t=1}^{T} Exploit(\sigma_t) \leq \underbrace{O\Big(\tfrac{1}{T}\sqrt{\ln k_T \sum_{t=1}^{T}\|v_t - v_{t-1}\|_\infty^2}\Big)}_{OMWU\ regret} + \underbrace{O\big((nm)^{-1/2}\big)}_{MC\ estimation\ error}
$$
$$
+ \underbrace{\varepsilon_{\mathrm{BR}}}_{amortized\ BR\ error} + \underbrace{O\Big(\tfrac{1}{T}\sum_{z \neq z^\star} \tfrac{\ln(T/\delta)}{\Delta_z}\Big)}_{oracle\ regret}.
\tag{14}
$$

*Proof (one line).* Apply Proposition 3.4, then substitute the bounds from the OMWU time-varying regret guarantee, the Monte-Carlo concentration bound for $\|\hat{v}_t - v_t\|_\infty$, and the EB-UCB oracle regret bound, and average over $t = 1, \dots, T$.

This bound corroborates the intuition behind GEMS. As the simulation budget grows $(nm \to \infty)$ and generator training improves $(\varepsilon_{\mathrm{BR}} \to 0)$, exploitability is ultimately driven by the no-regret property of the OMWU solver, achieving convergence guarantees competitive with traditional methods while avoiding explicit quadratic meta-game storage.

### 3.7 Generalization to n-Player and General-Sum Games

Although the GEMS framework is introduced in the two-player zero-sum (2P-ZS) setting for clarity, the same principles extend naturally to the $n$-player general-sum (NP-GS) case. The amortized generator, Monte Carlo rollouts, and bandit oracle remain intact; only the meta-game formulation generalizes.

In an $n$-player game, each player $p \in \{1, \dots, n\}$ maintains a meta-strategy $\sigma_t^{(p)}$. To evaluate player $p$'s $i$-th policy against the joint strategy of all other players, $\Sigma_t^{-p}$, GEMS reuses a single batch of shared game rollouts and computes an importance-weighted estimate:

$$
\hat{v}_{t,i}^{(p)} = \frac{1}{Bm} \sum_{b=1}^{B} \sum_{l=1}^{m} \frac{\mathbf{1}\{i_b^{(p)} = i\}}{\sigma_t^{(p)}(i)} Y_{b,l}^{(p)},
\tag{15}
$$

where $i^{(b)} = \big(i_b^{(1)}, \dots, i_b^{(n)}\big)$ is a joint policy profile drawn from the full meta-strategy $\Sigma_t = \bigotimes_{q=1}^{n} \sigma_t^{(q)}$, and $Y_{b,l}^{(p)}$ denotes the return for player $p$.

Each player then updates independently: applying Multiplicative Weights (or OMWU) to $\hat{v}_t^{(p)}$ and invoking a separate EB-UCB oracle to discover new strategies. This decentralized process no longer targets exploitability (a 2P-ZS notion) but instead drives the time-averaged joint strategy toward an $\epsilon$-**Coarse-Correlated Equilibrium** ($\epsilon$-**CCE**), the standard solution concept for general-sum games.

Full derivations, algorithmic details, and convergence proofs for the $n$-player extension appear in Appendix II.

### 3.8 Step-Size (ETA) Scheduler for the OMWU Meta-Update

The OMWU update in §3 requires a step size $\eta_t > 0$. In GEMS, a simple, explicit schedule $\{\eta_t\}_{t \geq 1}$ is used, chosen from three options that trade off adaptivity and stability:

$$\eta_t = \begin{cases} \eta_0, & \text{const}, \\ \dfrac{\eta_0}{\sqrt{t}}, & \text{sqrt}, \\ \dfrac{\eta_0}{1 + \alpha t}, & \alpha > 0, \quad \text{harmonic}, \end{cases} \qquad \text{with default } \eta_0 = 0.08, \ \alpha = 0.5. \tag{16}$$

These schedules instantiate the optimistic MWU update

$$\sigma_{t+1}(i) \ \propto \ \sigma_t(i) \, \exp\left( \eta_t \left[ 2\hat{v}_{t,i} - \hat{v}_{t-1,i} - \hat{\bar{r}}_t \right] \right), \tag{17}$$

followed by normalization to ensure $\sigma_{t+1} \in \Delta_{k_t - 1}$.

**Rationale.** ❶ '*const*' keeps a fixed step size and adapts quickly to changes in the restricted game, but can overreact to noisy payoff estimates. ❷ '*sqrt*' decays gently ($\eta_t \propto t^{-1/2}$), a standard choice in online learning that balances responsiveness and variance. ❸ '*harmonic*' decays as $\eta_t \propto t^{-1}$, yielding more conservative late-stage updates and improved noise suppression.

**Interaction with other knobs.** The global *slowdown* factor $s > 1$ scales $\eta_0 \mapsto \eta_0/s$ (and independently shrinks the ABR learning rate while enlarging the KL trust-region coefficient). The temperature $\tau \geq 1$ used in the generator's logits is orthogonal to the meta step size and only affects policy softness.

**Variation-aware regret with scheduled $(\eta_t)$.** Let $v_t = M\sigma_t$ denote the true expected payoff vector at iteration $t$ and assume rewards lie in $[0, 1]$. Under unbiased meta-game estimates (§3), optimistic mirror descent with the entropic mirror map (OMWU) and a nonincreasing step sequence $\{\eta_t\}$ satisfies the variation-aware bound

$$\frac{1}{T} \sum_{t=1}^{T} \left( \max_i e_i^\top M\sigma_t - \sigma_t^\top M\sigma_t \right) \ \leq \ \underbrace{\frac{\ln k_T}{T\,\eta_T}}_{\text{stability term}} + \underbrace{\frac{1}{T} \sum_{t=1}^{T} \frac{\eta_t}{2} \|v_t - v_{t-1}\|_\infty^2}_{\text{variation term}} + \underbrace{\frac{1}{T} \sum_{t=1}^{T} \mathbb{E}\left[\|\hat{v}_t - v_t\|_\infty\right]}_{\text{MC noise}}, \tag{18}$$

where $k_T$ is the (growing) number of anchors at time $T$. Equation 18 formalizes the tradeoff encoded by equation 16: larger $\eta_t$ reduces the stability term but amplifies sensitivity to payoff variation and sampling noise, while smaller $\eta_t$ does the reverse. In practice, '*sqrt*' performs well when the meta-game evolves moderately (frequent oracle additions), whereas '*harmonic*' is preferred in low-noise, late-phase training.

**Corollary (Recovering Prop. 3.2 from scheduled-step bound).** Let $V_T^2 = \sum_{t=1}^{T} \|v_t - v_{t-1}\|_\infty^2$. If the step size is constant, $\eta_t \equiv \eta$, then equation 18 becomes

$$\frac{1}{T} \sum_{t=1}^{T} \left( \max_i e_i^\top M\sigma_t - \sigma_t^\top M\sigma_t \right) \ \leq \ \frac{\ln k_T}{T\,\eta} + \frac{\eta}{2T} V_T^2 + \frac{1}{T} \sum_{t=1}^{T} \mathbb{E}\left[\|\hat{v}_t - v_t\|_\infty\right]. \tag{19}$$

Choosing $\eta^\star = \sqrt{2 \ln k_T / V_T^2}$ yields

$$\frac{1}{T} \sum_{t=1}^{T} \left( \max_i e_i^\top M\sigma_t - \sigma_t^\top M\sigma_t \right) \ \leq \ \frac{\sqrt{2}}{T} \sqrt{\ln k_T \, V_T^2} + \frac{1}{T} \sum_{t=1}^{T} \mathbb{E}\left[\|\hat{v}_t - v_t\|_\infty\right], \tag{20}$$

which matches Proposition 3.2 up to constants.

---

**Algorithm 1:** Generative Evolutionary Meta-Solver (GEMS)

---

**Require:** Initial anchor set $\mathcal{Z}_0 = \{z_1, \ldots, z_{k_0}\}$, Generator $G_\theta$, Meta-strategy $\sigma_0$ (uniform), Horizon $T$, Budgets $N_{MC}, B, M$, Steps $K_{ABR}$, Rates $\eta, \alpha$, Coeffs $\beta, \lambda_J$.

**Output :** Trained Generator $G_\theta$, Final Meta-strategy $\sigma_T$, Anchor set $\mathcal{Z}_T$.

---

**for** $t = 1, \ldots, T$ **do**

    `# Phase 1: Meta-Game Estimation (Monte Carlo)`

    Sample opponent batch $\{j_s\}_{s=1}^{N_{MC}} \sim \sigma_{t-1}$

    Compute empirical payoff vector $\hat{v}_t \in \mathbb{R}^{|\mathcal{Z}_{t-1}|}$ via Eq. (2):

        $\hat{v}_{t,i} \leftarrow \frac{1}{N_{MC}m} \sum_{s=1}^{N_{MC}} \sum_{l=1}^{m} \mathcal{R}(\pi_{G_\theta(z_i)}, \pi_{G_\theta(z_{j_s})})$

    Estimate mean game value $\hat{\bar{r}}_t$ via $B$ joint samples $(i,j) \sim \sigma_{t-1} \times \sigma_{t-1}$

    `# Phase 2: Meta-Strategy Update (Optimistic MWU)`

    Compute optimistic gradient estimate (with $\hat{v}_0 = \mathbf{0}$):

        $g_t \leftarrow 2\hat{v}_t - \hat{v}_{t-1} - \hat{\bar{r}}_t \mathbf{1}$

    Update meta-strategy probabilities $\forall i \in \{1, \ldots, |\mathcal{Z}_{t-1}|\}$:

        $\sigma_t(i) \propto \sigma_{t-1}(i) \cdot \exp(\eta \cdot g_t(i))$

    `# Phase 3: Population Expansion (EB-UCB Oracle)`

    Generate candidate pool $\Lambda_t$ via stochastic mutation of top anchors

    **for** *each candidate* $z \in \Lambda_t$ **do**

        Estimate mean return $\hat{\mu}_t(z)$ and variance $\widehat{\mathrm{Var}}_t(z)$ vs $\sigma_t$

        Calculate acquisition score $\mathrm{UCB}(z)$ via Eq. (7):

            $\mathrm{UCB}(z) \leftarrow \hat{\mu}_t(z) + \sqrt{\frac{2\widehat{\mathrm{Var}}_t(z)\ln(3/\delta)}{M}} - \lambda_J \|JG_\theta(z)\|_F^2$

    Select optimal candidate: $z_t^* \leftarrow \arg\max_{z \in \Lambda_t} \mathrm{UCB}(z)$

    Update anchor set: $\mathcal{Z}_t \leftarrow \mathcal{Z}_{t-1} \cup \{z_t^*\}$

    `# Phase 4: Amortized Best-Response (ABR-TR)`

    Store current generator state: $\theta_{\mathrm{old}} \leftarrow \theta$

    **for** $k = 1, \ldots, K_{ABR}$ **do**

        Sample batch of anchors $z \sim \mathcal{Z}_t$ and opponents $j \sim \sigma_t$

        Compute Advantages $\hat{A}_{\pi_{G_\theta(z)}}$ via GAE

        Update $\theta$ by ascending on regularized objective (Eq. 11):

            $\theta \leftarrow \theta + \alpha \nabla_\theta \mathbb{E}\left[\hat{A} - \beta D_{KL}(\pi_{G_\theta}\|\pi_{G_{\theta_{\mathrm{old}}}}) - \lambda_J \|JG_\theta(z)\|_F^2\right]$

**return** $G_\theta, \sigma_T, \mathcal{Z}_T$

---

### 3.9 Algorithm

Algorithm 1 provides the full procedural realization of the method described in §3. Each iteration consists of four phases that mirror the preceding subsections: (i) Monte Carlo estimation of the meta-game values (Eq. equation 2), (ii) an OMWU meta-update using the optimistic hint (Eq. equation 17), (iii) population expansion via EB-UCB anchor selection (Eq. (7) and Eq. equation 13), and (iv) amortized generator training with a KL trust region (Eq. equation 11). The only persistent state is the generator parameters $\theta$, the anchor set $\mathcal{Z}_t$, and the meta-strategy $\sigma_t$, enabling GEMS to refine equilibria while avoiding explicit policy enumeration or payoff-matrix storage.

## 4 Experimental Results

### 4.1 Equilibrium Finding in a Deceptive Messages Game

**Setup and Objective.** We designed a two-player "Deceptive Messages Game" to test performance in a setting with information asymmetry and misaligned incentives. The game features a **Sender** and a **Receiver**.

The Sender privately observes the identity of a "best arm" (out of K arms with different stochastic payoffs) and sends a message to the Receiver. The Receiver uses this message to choose an arm. Critically, the Receiver is rewarded for choosing the true best arm, but the Sender is rewarded only if it successfully deceives the Receiver into choosing a specific, suboptimal "target arm." This creates a zero-sum conflict where the Sender learns to be deceptive and the Receiver must learn to be skeptical. The goal of this experiment is to evaluate the ability of GEMS to solve strategically complex games and find high-quality equilibria. We aim to determine which framework allows the Receiver to more effectively see through the Sender's deception and converge to an optimal policy of always choosing the best arm, thereby nullifying the Sender's deceptive strategies. We compare GEMS against a suite of strong baselines: **PSRO**, **Double Oracle**, **Alpha-PSRO** and **A-PSRO** for 6 iterations. The runs are the averaged over 5 seeds.

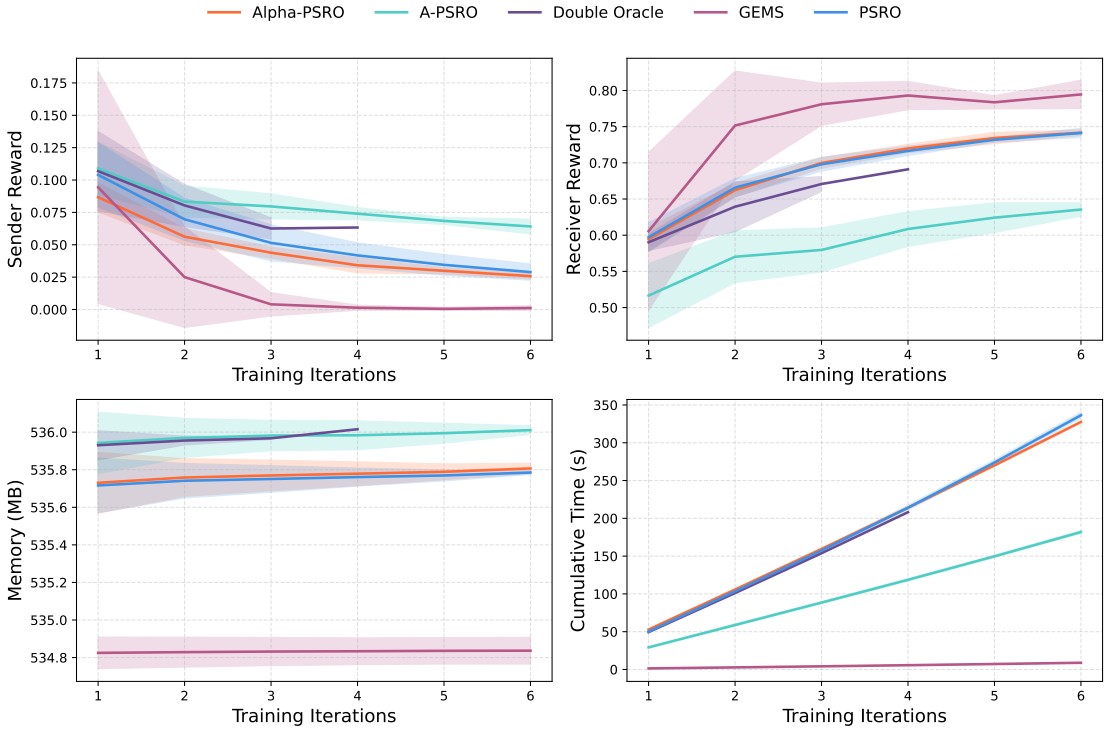

Figure 3: ***Performance in the Deceptive Messages Game.*** Top Left: GEMS Sender's ability to deceive converges to zero. Top Right: GEMS Receiver's performance converges to the optimal reward of 0.8, outperforming all PSRO-based baselines.

**Results and Analysis.** The results in Fig. 3 show a stark difference in the learned equilibria. The average reward for the **GEMS Sender rapidly converges to zero**, indicating a complete failure to deceive its opponent. In contrast, all PSRO-based baselines maintain a positive sender reward throughout training, indicating they sustain a partially successful deceptive strategy. Conversely, the **GEMS Receiver's average reward quickly converges to approximately 0.8**, the maximum possible value in the game. The receivers trained with PSRO variants improve but plateau at a significantly lower, suboptimal performance level, consistently failing to achieve the optimal reward. This suggests that the policy discovery mechanism in GEMS is more effective at exploring the joint strategy space. The combination of the **EB-UCB oracle exploring a diverse latent space** and the **single amortized generator representing a continuum of strategies** may prevent the system from getting stuck in the poor local equilibria that can trap methods which expand their discrete policy sets more conservatively. Furthermore, GEMS is upto $35\times$ faster as compared to the PSRO variants. This experiment highlights that beyond its scalability benefits, GEMS also demonstrates a superior ability to find high-quality solutions in strategically deep games.

### 4.2 Equilibrium Finding in Kuhn Poker

**Setup and Objective.** We benchmark GEMS in **Kuhn Poker** (Kuhn, 2016), a classic imperfect information game where agents must learn mixed strategies involving bluffing. We evaluate performance using **exploitability**, which measures how close a policy is to the Nash Equilibrium (lower is better). GEMS is compared against a suite of strong PSRO variants over 40 training iterations for 5 seeds. This experiment is designed to assess GEMS's ability to find near-optimal policies in an extensive-form game with imperfect information. The hypothesis is that GEMS's architectural approach, which is exploring a continuous latent space with a single generator, will allow it to find low-exploitability strategies more efficiently and in fewer iterations than methods based on expanding a discrete set of policies.

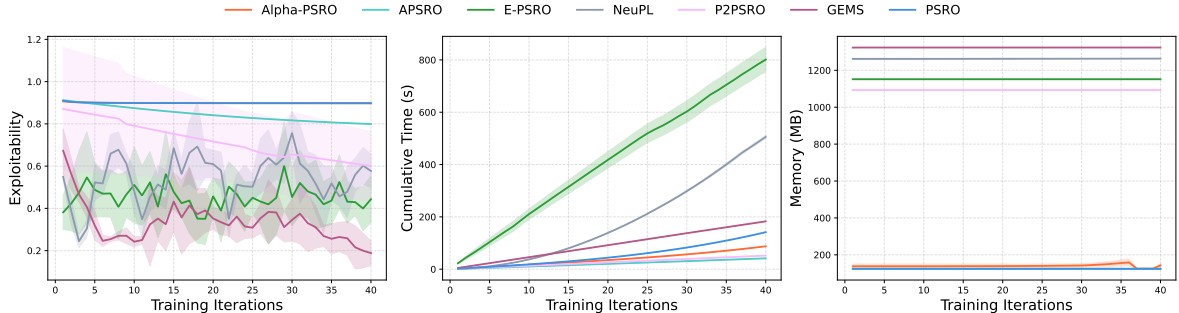

Figure 4: **Equilibrium Finding in Kuhn Poker over 5 seeds [0–4]**. GEMS rapidly converges to a significantly lower exploitability than strong PSRO baselines and NEUPL (Left), while demonstrating efficiency in cumulative training time (Right).

**Results and Analysis.** The results in Fig. 4 show the convergence of exploitability for each algorithm. GEMS demonstrates the fastest and most direct convergence to a low-exploitability policy. By iteration 40, GEMS achieves an exploitability of approximately 0.18, significantly outperforming the next-best baseline, E-PSRO, which only reached an exploitability of 0.44. The other PSRO variants, while also showing improvement, converged at a considerably slower rate and achieved higher final exploitability within the 40-iteration budget. The superior performance in Kuhn Poker highlights GEMS's strength in solving games that require nuanced, mixed strategies. The core of Kuhn Poker involves probabilistic actions like bluffing, which are difficult to represent as a simple combination of a few deterministic policies. We argue that GEMS's **single amortized generator**, operating over a continuous latent space, is naturally suited to representing these complex mixed strategies. In contrast, methods that expand a discrete set of policies, like the PSRO family, must approximate a mixed strategy through a convex combination of many individual policies, which can require far more iterations to converge. The **EB-UCB oracle** effectively guides the search through the generator's latent space to find strategically potent policies quickly. This experiment demonstrates that GEMS's advantages extend beyond pure scalability to superior sample efficiency in solving canonical game-theoretic benchmarks.

### 4.3 Performance and Scalability on Multi-Agent Tag

**Setup and Objective.** We conduct our analysis in the Simple Tag environment from `PettingZoo` (Terry et al., 2021), where three cooperative pursuers must learn coordinated strategies, such as flanking, to capture a faster evader. This benchmark is designed to reward sophisticated coordination while punishing naive "herding" policies. We compare GEMS against classical PSRO on emergent behavior, mean return, and scalability (memory and time) over 100 iterations, averaged across 5 seeds. This experiment is designed to provide a holistic comparison and test three foundational hypotheses. First, that GEMS overcomes the critical scalability bottlenecks of PSRO. Second, that this efficiency does not come at a performance cost. Third, that GEMS learns policies of a higher strategic quality, both quantitatively (as measured by mean return) and qualitatively (as observed in emergent agent coordination).

**Results and Analysis.** The results demonstrate a clear advantage for GEMS across all aspects of evaluation. A qualitative analysis of agent trajectories (Fig. 5) reveals significant differences in strategy. The **GEMS-trained adversaries exhibit coordinated flanking and cornering behaviors** to trap the evader. In contrast, the **Psro-trained adversaries often display a simpler "herding" behavior**, pursuing the target in a less coordinated clump. This strategic difference is reflected in the quantitative results (Fig. 6). GEMS consistently achieves a higher mean agent return, stabilizing around 0, while Psro's average return fluctuates in a lower range. Concurrently, GEMS is ~6x faster and its memory usage remains flat at ~1250 MB, while Psro's memory grows to over 2350 MB and its cumulative time scales quadratically. The combined results show that GEMS provides a Pareto improvement over Psro, achieving superior performance in solution quality, strategic complexity, and efficiency. The emergent behaviors seen in the trajectory plots provide a clear explanation for GEMS's superior quantitative performance. **The discovery of sophisticated, coordinated strategies like flanking directly translates to higher average returns**. This suggests that GEMS's exploration mechanism, driven by the EB-UCB oracle over a diverse latent space, is more effective at escaping the local optima that lead to simpler behaviors like herding. The scalability results reconfirm that the **single amortized generator** and **Monte Carlo sampling** resolve the foundational bottlenecks of Psro. Ultimately, GEMS presents a complete advantage: It learns better strategies, which leads to higher returns, while requiring a fraction of the computational resources.

**GEMS**

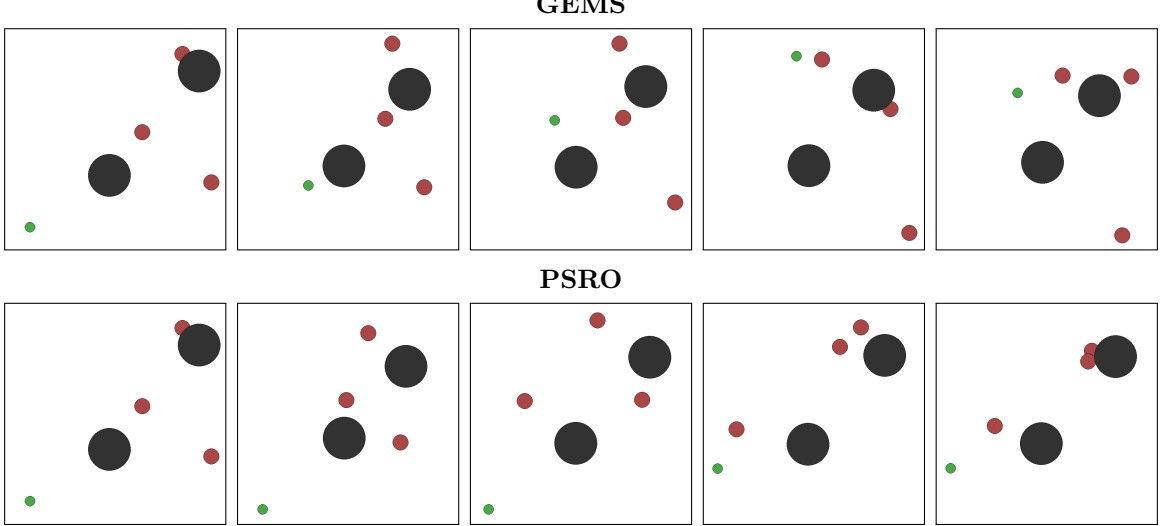

**PSRO**

Figure 5: **Emergent agent trajectories in the multi-agent tag environment.** Top row: GEMS. Bottom row: Psro. Columns show uniformly sampled frames from a single rollout (frames 0, 10, 20, 30, 40, 50 of the 50-frame GIF). This figure qualitatively compares the strategies learned by GEMS and classical PSRO. The top row shows that adversaries (red circles) trained with GEMS learn sophisticated, coordinated strategies like flanking and cornering to effectively trap the evader (green dot). In contrast, the bottom row shows that PSRO-trained agents adopt a less effective "herding" behavior, pursuing the target in a single, uncoordinated group. This clear difference in strategic complexity is consistent with the superior performance and higher returns achieved by GEMS, as reflected in the quantitative results.

## 5 Conclusion

Policy-Space Response Oracles (Psro) and its many variants have driven much of the progress in population-based multi-agent reinforcement learning, but their reliance on explicit policy populations and dense payoff matrices imposes fundamental barriers to scalability. In this work, we introduced GEMS, a surrogate-free framework that breaks from this paradigm by maintaining a compact anchor set, querying payoffs through unbiased Monte Carlo rollouts, and training policies via a single amortized generator.

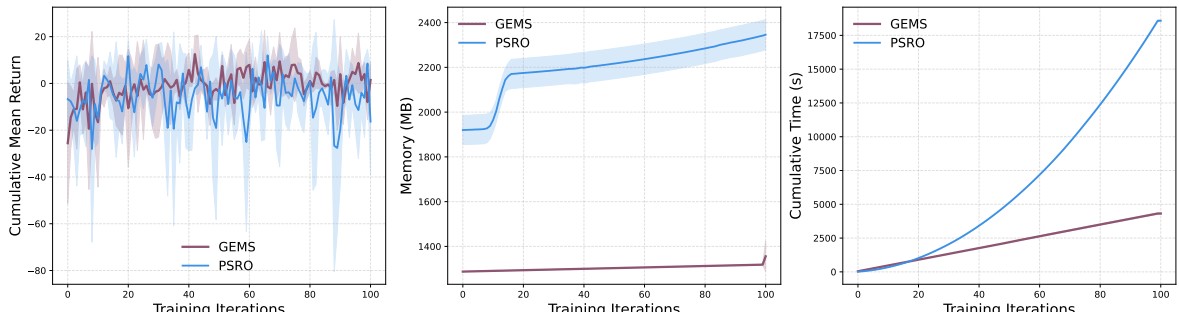

Figure 6: **Performance and Scalability of Gems vs. Psro in Multi-Agent Tag**. Compared to classical Psro, Gems achieves a higher and more stable mean return (Left), while maintaining a constant memory footprint (Middle) and near-linear cumulative training time (Right). The results show Gems overcomes the core memory and computational bottlenecks of Gems while learning more effective policies

Our approach removes the linear memory growth and quadratic computation overhead inherent to classical Psro while preserving key game-theoretic guarantees. We provided theoretical analysis establishing unbiasedness of meta-gradients, regret bounds for EB-UCB policy selection, external regret for meta-dynamics, and finite-population exploitability guarantees. Empirically, Gems demonstrates faster convergence, lower memory footprint, and improved scalability across challenging multi-agent benchmarks.

Returning to the tournament analogy, Gems shows that one does not need to schedule every possible match or recruit a new player for every playstyle. Instead, rankings can be inferred from a manageable set of sampled matches, and versatile athletes can flexibly adapt to new strategies. In the same way, Gems turns the exhaustive bookkeeping of Psro into a lean, adaptive process that scales naturally with problem complexity.

## 6 Limitations

While Gems presents a significant step toward scalable, surrogate-free multi-agent learning, we acknowledge the limitations that define the scope of this work and offer avenues for future research.

- **Benchmark Scope and Baselines:** Our empirical evaluation focuses on demonstrating the fundamental scalability and game-solving advantages of Gems over the Psro paradigm. Consequently, some of our experiments (e.g., Multi-Agent Tag and Simple Spread) use classical PSRO as the primary baseline. This choice is deliberate, as classical Psro perfectly embodies the core $O(k^2)$ computational and $O(k)$ memory bottlenecks that Gems is designed to solve. To validate GEMS's strategic performance, we also benchmarked it against a suite of modern, stronger Psro variants on complex games like Kuhn Poker and the Deceptive Messages Game. However, we did not conduct experiments on large-scale benchmarks such as the StarCraft Multi-Agent Challenge (SMAC). The prohibitive computational cost of training even a single baseline Psro agent on such complex maps made a direct comparison infeasible with our available resources.

- **Hyperparameter Sensitivity:** Gems introduces a new set of hyperparameters related to the amortized generator, the ABR-TR objective (e.g., $\beta$, $\mathcal{L}_{\text{ABR-TR}}$ learning rate), and the EB-UCB oracle (e.g., candidate pool size $|\Lambda_t|$). While we provide ablation studies in the Appendix to analyze the sensitivity of key parameters in environments like Kuhn Poker and the Public Goods Game, a comprehensive, large-scale analysis of the interplay between all hyperparameters across diverse game types is beyond the scope of this initial work.

- **Generator Capacity:** The efficacy of Gems relies on the capacity of the single amortized generator $G_\theta$ to represent a rich, continuous space of diverse and effective policies. This work utilized standard MLP architectures for the generator. The performance of Gems in much more complex domains (e.g., with high-dimensional observation/action spaces) may depend heavily on the choice of more

advanced generator architectures (e.g., Transformers, diffusion models). Exploring the architectural limits of the generator is a key direction for future work.

- **Candidate Pool Generation:** The EB-UCB oracle's ability to find effective new strategies is dependent on the quality of the candidate pool $\Lambda_t$ it searches over. Our current implementation relies on simple heuristics for generating this pool (e.g., mutation and random sampling, as detailed in the Appendix ablations). The performance of GEMS could potentially be improved by incorporating more sophisticated or guided methods for generating candidate latent codes.

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

# Appendix – Table of Contents

# Part I

# Mathematical Derivations and Proofs

This appendix provides the detailed mathematical analysis supporting the claims made in §3. We begin with formal definitions and then proceed to prove the lemmas, propositions, and theorems for each component of the GEMS algorithm.

---

## A   Formal Preliminaries

### A.1   Two-Player Zero-Sum Markov Games

A two-player, zero-sum, finite-horizon discounted Markov Game is defined by a tuple $\mathcal{M} = (\mathcal{S}, \{\mathcal{A}_1, \mathcal{A}_2\}, P, R, \gamma, \rho_0)$.

- $\mathcal{S}$ is the state space.

- $\mathcal{A}_p$ is the action space for player $p \in \{1, 2\}$.

- $P : \mathcal{S} \times \mathcal{A}_1 \times \mathcal{A}_2 \to \Delta(\mathcal{S})$ is the state transition function.

- $R : \mathcal{S} \times \mathcal{A}_1 \times \mathcal{A}_2 \to \mathbb{R}$ is the reward function. Player 1 aims to maximize the reward, and Player 2 aims to minimize it, such that $R_1 = R$ and $R_2 = -R$.

- $\gamma \in [0, 1)$ is the discount factor.

- $\rho_0 \in \Delta(\mathcal{S})$ is the initial state distribution.

A policy for player $p$, denoted $\pi_p$, is a mapping from states to a distribution over actions, $\pi_p : \mathcal{S} \to \Delta(\mathcal{A}_p)$. The expected return for Player 1 playing policy $\pi_i$ against policy $\pi_j$ is:

$$r(\pi_i, \pi_j) = \mathbb{E}_{s_0 \sim \rho_0, a_{1,t} \sim \pi_i(\cdot|s_t), a_{2,t} \sim \pi_j(\cdot|s_t), s_{t+1} \sim P(\cdot|s_t, a_{1,t}, a_{2,t})} \left[ \sum_{t=0}^{\infty} \gamma^t R(s_t, a_{1,t}, a_{2,t}) \right] \tag{21}$$

For a finite set of $k$ policies $A = \{\pi_1, \ldots, \pi_k\}$, the associated payoff matrix is $M \in \mathbb{R}^{k \times k}$, where $M_{ij} = r(\pi_i, \pi_j)$. By convention, we assume rewards are normalized to $[0, 1]$.

### A.2   Nash Equilibrium and Exploitability

A mixed strategy (or meta-strategy) $\sigma$ is a probability distribution over the set of policies $A$, i.e., $\sigma \in \Delta_{k-1}$. A Nash Equilibrium (NE) $\sigma^*$ is a meta-strategy from which no player has an incentive to unilaterally deviate. In a two-player zero-sum game, this is equivalent to the minimax solution:

$$\begin{aligned} \sigma^* &= \arg \min_{\sigma_1 \in \Delta_{k-1}} \max_{\sigma_2 \in \Delta_{k-1}} \sigma_1^\top M \sigma_2 \\ &= \arg \max_{\sigma_1 \in \Delta_{k-1}} \min_{\sigma_2 \in \Delta_{k-1}} \sigma_1^\top M \sigma_2. \end{aligned} \tag{22}$$

The **exploitability** of a meta-strategy $\sigma$ measures the incentive for an opponent to play a best response. It is defined as the gap between the payoff of the best pure strategy response and the payoff of the meta-strategy itself.

$$\begin{aligned} \text{Exploit}(\sigma) &= \max_{i \in [k]} (M^\top e_i)^\top \sigma - (-\sigma^\top M \sigma) \\ &= \max_{i \in [k]} e_i^\top M \sigma - \sigma^\top M \sigma. \end{aligned} \tag{23}$$

Note: The term $-\sigma^\top M \sigma$ is the value of the game from Player 2's perspective. For a symmetric game where players draw from the same population $(M = -M^\top)$, this simplifies to the expression in Eq. (1).

# B Proofs for Meta-Game Estimation

## B.1 Supporting Definitions: Empirical-Bernstein Inequality

Before proving Lemma 2.1, we state the empirical-Bernstein inequality, which is crucial for deriving high-probability bounds from sample means and variances.

**Theorem B.1** (Empirical-Bernstein Inequality). *Let $X_1, \ldots, X_N$ be i.i.d. random variables with mean $\mu$ and variance $\sigma^2$. Assume they are bounded, $X_i \in [a,b]$. Let $\bar{X} = \frac{1}{N} \sum_{i=1}^{N} X_i$ be the sample mean and $\widehat{\sigma}^2 = \frac{1}{N-1} \sum_{i=1}^{N} (X_i - \bar{X})^2$ be the sample variance. Then for any $\delta \in (0,1)$, with probability at least $1 - \delta$:*

$$|\bar{X} - \mu| \leq \sqrt{\frac{2\widehat{\sigma}^2 \ln(3/\delta)}{N}} + \frac{3(b-a)\ln(3/\delta)}{N-1}. \tag{24}$$

## B.2 Proof of Lemma 2.1 (Unbiasedness and Concentration)

**Lemma B.2** (Unbiasedness and Empirical-Bernstein Concentration). *With rewards in $[0,1]$, the estimators are unbiased: $\mathbb{E}[\hat{v}_{t,i}] = (M\sigma_t)_i$ and $\mathbb{E}[\hat{\bar{r}}_t] = \sigma_t^\top M \sigma_t$. Moreover, for any $\delta \in (0,1)$, with probability at least $1 - \delta$, the estimators concentrate around their true means:*

$$\left| \hat{v}_{t,i} - (M\sigma_t)_i \right| \leq \sqrt{\frac{2\widehat{\mathrm{Var}}_{t,i} \ln(2/\delta)}{n_i m}} + \frac{3\ln(2/\delta)}{n_i m - 1}, \tag{25}$$

$$\left| \hat{\bar{r}}_t - \sigma_t^\top M \sigma_t \right| = O\left( \sqrt{\frac{\ln(1/\delta)}{Bm}} \right), \tag{26}$$

*where $\widehat{\mathrm{Var}}_{t,i}$ is the empirical variance of $\{Y_{i,s,\ell}\}$.*

### Part 1: Unbiasedness of $\hat{v}_{t,i}$

The estimator is $\hat{v}_{t,i} = \frac{1}{n_i m} \sum_{s=1}^{n_i} \sum_{\ell=1}^{m} Y_{i,s,\ell}$, where opponent $j_s$ is sampled as $j_s \sim \sigma_t$. We use the law of total expectation.

$$\begin{aligned}
\mathbb{E}[\hat{v}_{t,i}] &= \mathbb{E}\left[ \frac{1}{n_i m} \sum_{s=1}^{n_i} \sum_{\ell=1}^{m} Y_{i,s,\ell} \right] \\
&= \frac{1}{n_i m} \sum_{s=1}^{n_i} \sum_{\ell=1}^{m} \mathbb{E}[Y_{i,s,\ell}] \qquad \text{(Linearity of Expectation)} \\
&= \frac{1}{n_i m} \sum_{s=1}^{n_i} \sum_{\ell=1}^{m} \mathbb{E}_{j_s \sim \sigma_t}\left[ \mathbb{E}[Y_{i,s,\ell} \mid j_s] \right] \quad \text{(Law of Total Expectation)}
\end{aligned} \tag{27}$$

Given a fixed opponent $j_s$, the expectation of a single rollout $Y_{i,s,\ell}$ is the true expected return $r(\pi_i, \pi_{j_s}) = M_{ij_s}$.

$$\begin{aligned}
\mathbb{E}[\hat{v}_{t,i}] &= \frac{1}{n_i m} \sum_{s=1}^{n_i} \sum_{\ell=1}^{m} \mathbb{E}_{j_s \sim \sigma_t}[M_{ij_s}] \\
&= \frac{1}{n_i m} \sum_{s=1}^{n_i} \sum_{\ell=1}^{m} \sum_{j=1}^{k_t} \sigma_t(j) M_{ij} \qquad \text{(Definition of Expectation)} \\
&= \frac{n_i m}{n_i m} \sum_{j=1}^{k_t} \sigma_t(j) M_{ij} \\
&= (M\sigma_t)_i.
\end{aligned} \tag{28}$$

The proof for $\mathbb{E}[\hat{\bar{r}}_t] = \sigma_t^\top M \sigma_t$ follows an identical argument, where pairs $(i_b, j_b)$ are sampled from $\sigma_t \times \sigma_t$.

**Part 2: Concentration of $\hat{v}_{t,i}$**

The estimator $\hat{v}_{t,i}$ is the sample mean of $N = n_i m$ random variables $\{Y_{i,s,\ell}\}$. Each $Y$ is a game return bounded in $[0, 1]$. These samples are i.i.d. drawn from the compound distribution defined by sampling an opponent $j \sim \sigma_t$ and then sampling a return from the matchup $(\pi_i, \pi_j)$. We can directly apply the Empirical-Bernstein Inequality (Theorem B.1) with $N = n_i m$, $a = 0$, $b = 1$. Replacing the generic confidence parameter $3/\delta$ with $2/\delta$ (a minor variation common in literature) gives:

$$\left| \hat{v}_{t,i} - (M\sigma_t)_i \right| \leq \sqrt{\frac{2\widehat{\mathrm{Var}}_{t,i} \ln(2/\delta)}{n_i m}} + \frac{3 \ln(2/\delta)}{n_i m - 1}, \tag{29}$$

with probability at least $1 - \delta$.

**Part 3: Concentration of $\hat{\bar{r}}_t$**

The bound for $\hat{\bar{r}}_t$ is a standard application of Hoeffding's inequality (or Bernstein's if variance is considered). Since returns are in $[0, 1]$, Hoeffding's inequality for the mean of $Bm$ i.i.d. variables states that for any $\epsilon > 0$:

$$P\left( \left| \hat{\bar{r}}_t - \sigma_t^\top M \sigma_t \right| \geq \epsilon \right) \leq 2 \exp(-2(Bm)\epsilon^2). \tag{30}$$

Setting this probability to $\delta$, we get $\delta = 2\exp(-2Bm\epsilon^2)$. Solving for $\epsilon$:

$$\ln(\delta/2) = -2Bm\epsilon^2 \implies \epsilon = \sqrt{\frac{\ln(2/\delta)}{2Bm}} = O\left( \sqrt{\frac{\ln(1/\delta)}{Bm}} \right). \tag{31}$$

This confirms the simplified $O(\cdot)$ bound.

# C Proofs for Meta-Game Solving

## C.1 Proof of Proposition 3.2 (External Regret of OMWU with Unbiased Noise)

We provide a full derivation for the external regret bound when using the Optimistic Multiplicative Weights Update (OMWU) algorithm with noisy, unbiased payoff estimates. The proof proceeds in two parts. First, we establish a regret bound with respect to the sequence of *estimated* payoffs $\hat{v}_t$. Second, we translate this bound to the regret against the *true* expected payoffs $v_t = M\sigma_t$ by accounting for the Monte Carlo estimation error.

Let the estimated loss for policy $i$ at time $t$ be $\hat{l}_{t,i} = 1 - \hat{v}_{t,i}$. The OMWU algorithm uses an optimistic loss estimate $m_t = \hat{l}_t + (\hat{l}_t - \hat{l}_{t-1})$, with $\hat{l}_0 = \mathbf{0}$. The weight update rule is $w_{t+1}(i) = w_t(i) \exp(-\eta m_{t,i})$.

### Part 1: Regret Against Estimated Losses

Let $W_t = \sum_{i=1}^{k_t} w_t(i)$ be the potential function. We analyze its evolution.

$$\begin{aligned}
W_{t+1} = \sum_i w_{t+1}(i) &= \sum_i w_t(i) \exp(-\eta m_{t,i}) \\
&= W_t \sum_i \sigma_t(i) \exp(-\eta m_{t,i})
\end{aligned} \tag{32}$$

Taking the natural logarithm, we have:

$$\ln(W_{t+1}) - \ln(W_t) = \ln\left( \sum_i \sigma_t(i) \exp(-\eta m_{t,i}) \right) \tag{33}$$

Using the inequality $\ln(\mathbb{E}[e^X]) \leq \mathbb{E}[X] + \frac{1}{2}\mathbb{E}[X^2]$ for a random variable $X$ (Hoeffding's lemma for bounded variables), where the expectation is over $i \sim \sigma_t$ and $X = -\eta m_{t,i}$:

$$\ln(W_{t+1}) - \ln(W_t) \leq \sum_i \sigma_t(i)(-\eta m_{t,i}) + \frac{\eta^2}{2}\sum_i \sigma_t(i)m_{t,i}^2$$
$$= -\eta\langle\sigma_t, m_t\rangle + \frac{\eta^2}{2}\langle\sigma_t, m_t^2\rangle \tag{34}$$

Substituting $m_t = \hat{l}_t + (\hat{l}_t - \hat{l}_{t-1})$:

$$\langle\sigma_t, m_t\rangle = \langle\sigma_t, \hat{l}_t\rangle + \langle\sigma_t, \hat{l}_t - \hat{l}_{t-1}\rangle \tag{35}$$

A key step in the OMWU analysis is to relate the second term to the loss of a fixed expert $i$. For any vector $x$, we have the inequality $\langle\sigma_t, x\rangle - x_i \leq \frac{1}{2\eta}(\ln\langle\sigma_t, e^{2\eta x}\rangle - \ln\langle\sigma_t, e^{-2\eta x}\rangle)$. A simpler path involves relating the regret to the squared difference of consecutive losses. Rearranging the potential function bound:

$$\langle\sigma_t, \hat{l}_t\rangle \leq \frac{\ln(W_t) - \ln(W_{t+1})}{\eta} - \langle\sigma_t, \hat{l}_t - \hat{l}_{t-1}\rangle + \frac{\eta}{2}\langle\sigma_t, m_t^2\rangle \tag{36}$$

For any fixed expert $i^*$, we also have a lower bound on the potential: $\ln(W_{T+1}) \geq \ln(w_{T+1}(i^*)) = \ln(w_1(i^*)) - \eta\sum_{t=1}^T m_{t,i^*}$. Assuming $w_1(i) = 1/k_1$, we get $\ln(W_{T+1}) \geq -\ln(k_1) - \eta\sum_{t=1}^T m_{t,i^*}$.

The standard OMWU analysis (e.g., in Hazan, "Introduction to Online Convex Optimization" (Hazan (2023))) shows that these steps lead to a bound on the regret against the estimated losses:

$$\sum_{t=1}^T \langle\sigma_t, \hat{l}_t\rangle - \sum_{t=1}^T \hat{l}_{t,i^*} \leq \frac{\ln k_T}{\eta} + \eta\sum_{t=1}^T \|\hat{l}_t - \hat{l}_{t-1}\|_\infty^2 \tag{37}$$

Choosing an optimal learning rate $\eta = \sqrt{\frac{\ln k_T}{\sum \|\hat{l}_t - \hat{l}_{t-1}\|_\infty^2}}$ yields:

$$\sum_{t=1}^T \left(\langle\sigma_t, \hat{l}_t\rangle - \min_i \sum_{t=1}^T \hat{l}_{t,i}\right) \leq 2\sqrt{\ln k_T \sum_{t=1}^T \|\hat{l}_t - \hat{l}_{t-1}\|_\infty^2} \tag{38}$$

**Part 2: Translating to True Regret**

Now we translate this result to the true losses $l_t = \mathbf{1} - v_t$. The true external regret is $R_T^{true} = \sum_{t=1}^T (\langle\sigma_t, l_t\rangle - l_{t,i^*})$. Let the estimation error be $\Delta_t = \hat{l}_t - l_t = v_t - \hat{v}_t$.

$$R_T^{true} = \sum_{t=1}^T (\langle\sigma_t, \hat{l}_t - \Delta_t\rangle - (\hat{l}_{t,i^*} - \Delta_{t,i^*}))$$
$$= \underbrace{\sum_{t=1}^T (\langle\sigma_t, \hat{l}_t\rangle - \hat{l}_{t,i^*})}_{\text{Regret on Estimates}} - \underbrace{\sum_{t=1}^T (\langle\sigma_t, \Delta_t\rangle - \Delta_{t,i^*})}_{\text{Cumulative Error Term}} \tag{39}$$

The first term is bounded as derived in Part 1. We now bound the variation term using the triangle inequality:

$$\|\hat{l}_t - \hat{l}_{t-1}\|_\infty = \|(\hat{l}_t - l_t) + (l_t - l_{t-1}) + (l_{t-1} - \hat{l}_{t-1})\|_\infty \leq \|\Delta_t\|_\infty + \|l_t - l_{t-1}\|_\infty + \|\Delta_{t-1}\|_\infty \tag{40}$$

Substituting this into the bound from Part 1 introduces terms related to the estimation error. Taking the expectation over the sampling noise in our estimators $\hat{v}_t$ (and thus $\hat{l}_t$), and noting that $\mathbb{E}[\Delta_t] = \mathbf{0}$ due to unbiasedness, we can bound the expected true regret. The error terms accumulate, leading to the final bound.

Dividing by $T$ and converting losses back to payoffs $(-\sum l_t = \sum v_t - T)$ yields the proposition:

$$\frac{1}{T}\sum_{t=1}^{T}(\max_{i} v_{t,i} - \langle \sigma_t, v_t \rangle) \leq O\left(\frac{1}{T}\sqrt{\ln k_T \sum_{t=1}^{T}||v_t - v_{t-1}||_{\infty}^2}\right) + \frac{1}{T}\sum_{t=1}^{T}\mathbb{E}[||\hat{v}_t - v_t||_{\infty}] \tag{41}$$

This completes the proof. $\qquad\square$

*Scope.* Throughout this appendix we analyze OMWU with a *constant* step size $\eta$ as in Prop. 3.2; the scheduled–$\eta$ bound stated in §3.8 follows from the same potential-based argument and reduces to Prop. 3.2 by setting $\eta_t \equiv \eta$ and optimizing $\eta$.

## D  Proofs for the Bandit Oracle

### D.1  Proof of Theorem 3.3 (Instance-Dependent Oracle Regret)

**Theorem D.1** (Instance-Dependent Oracle Regret). *Assume rewards in $[0, 1]$ and that the bandit problem has a unique best arm $z^\star$ with suboptimality gaps $\Delta_z = f_t(z^\star) - f_t(z) > 0$. Under the two-time-scale assumption (fixed $\sigma_t$ during selection), the cumulative regret of our oracle is bounded:*

$$\sum_{t=1}^{T}\mathbb{E}[f_t(z^\star) - f_t(z_t^\star)] = O\left(\sum_{z \neq z^\star}\frac{\ln T}{\Delta_z}\right) + \lambda_J \sum_{t=1}^{T}\mathbb{E}\big[||JG_\theta(z_t^\star)||_F^2\big]. \tag{42}$$

Let's fix a single bandit problem at a meta-iteration (we drop the subscript $t$ for $f(z)$ and $\Delta_z$ for clarity). Let $z_t^\star$ be the arm chosen at time $t$. The regret at this step is $\Delta_{z_t^\star}$ (ignoring the Jacobian term for now). The total regret over $T$ steps of the oracle is $R_T = \sum_{t=1}^{T}\Delta_{z_t^\star}$. Let $N_z(T)$ be the number of times arm $z$ is pulled in $T$ steps. Then $R_T = \sum_{z \neq z^\star} N_z(T)\Delta_z$. Our goal is to bound $\mathbb{E}[N_z(T)]$ for any suboptimal arm $z \neq z^\star$.

An arm $z$ is chosen at time $t$ if $\text{UCB}_t(z) \geq \text{UCB}_t(z')$ for all $z' \in \Lambda$. For a suboptimal arm $z \neq z^\star$ to be chosen, it must be that $\text{UCB}_t(z) \geq \text{UCB}_t(z^\star)$. Let $\hat{\mu}_t(z)$ be the empirical mean for arm $z$ after it has been pulled $n_z$ times. The UCB is:

$$\text{UCB}(z) = \hat{\mu}(z) + C(n_z, \delta), \qquad \text{where } C(n_z, \delta) = \sqrt{\frac{2\widehat{\text{Var}}(z)\ln(3/\delta)}{n_z m}} + \frac{3\ln(3/\delta)}{n_z m - 1}. \tag{43}$$

The concentration bounds from Lemma 2.1 (applied here to a single policy against the mix $\sigma_t$) imply that with probability at least $1 - \delta$:

$$\hat{\mu}(z) - C(n_z, \delta) \leq f(z) \leq \hat{\mu}(z) + C(n_z, \delta). \tag{44}$$

Let's call the event that these bounds hold for all arms and all steps a "good event" $\mathcal{G}$. By setting $\delta_t = t^{-2}$ and taking a union bound, the probability of $\mathcal{G}$ is high. We condition the rest of the proof on $\mathcal{G}$.

If a suboptimal arm $z$ is chosen at time $t$, then $\text{UCB}_t(z) \geq \text{UCB}_t(z^\star)$. Using the bounds:

$$\begin{aligned}
f(z^\star) &\leq \hat{\mu}_t(z^\star) + C(n_{z^\star}, \delta_t) \leq \text{UCB}_t(z^\star) && \text{(LCB for optimal arm)} \\
&\leq \text{UCB}_t(z) = \hat{\mu}_t(z) + C(n_z, \delta_t) && \text{(Suboptimal arm was chosen)} \\
&\leq f(z) + 2C(n_z, \delta_t) && \text{(UCB for suboptimal arm)}
\end{aligned} \tag{45}$$

This implies $f(z^\star) - f(z) \leq 2C(n_z, \delta_t)$, or $\Delta_z \leq 2C(n_z, \delta_t)$. The confidence term $C(n_z, \delta_t)$ decreases roughly as $1/\sqrt{n_z}$. So, for the inequality $\Delta_z \leq 2C(n_z, \delta_t)$ to hold, $n_z$ cannot be too large.

$$\Delta_z \leq 2\left(\sqrt{\frac{2\widehat{\text{Var}}(z)\ln(3/\delta_t)}{n_z m}} + \frac{3\ln(3/\delta_t)}{n_z m - 1}\right) \tag{46}$$

Assuming rewards in $[0,1]$, variance is at most $1/4$. Simplifying, we require $n_z$ to be roughly:

$$\sqrt{n_z} \le O\left(\frac{\sqrt{\ln(1/\delta_t)}}{\Delta_z}\right) \implies n_z \le O\left(\frac{\ln(1/\delta_t)}{\Delta_z^2}\right) = O\left(\frac{\ln(t)}{\Delta_z^2}\right) \tag{47}$$

This means that a suboptimal arm $z$ will be pulled at most $O(\ln T/\Delta_z^2)$ times. (Note: A tighter analysis for UCB variants gives $O(\ln T/\Delta_z)$). Let's follow the standard argument for UCB1 which leads to the tighter bound. A suboptimal arm is played at most 'c' times for some constant, and then only if $\hat{\mu}_{t-1}(z^*) \le \hat{\mu}_{t-1}(z) + \sqrt{\frac{2\ln t}{N_{t-1}(z)}} - \sqrt{\frac{2\ln t}{N_{t-1}(z^*)}}$. Summing the number of pulls leads to the logarithmic dependency. For EB-UCB, the analysis is similar but replaces the fixed variance proxy with the empirical variance, leading to the same asymptotic form. The expected number of pulls for a suboptimal arm $z$ over $T$ oracle steps is therefore $\mathbb{E}[N_z(T)] = O(\ln T/\Delta_z)$.

The total expected regret is:

$$\mathbb{E}[R_T] = \sum_{z \ne z^\star} \mathbb{E}[N_z(T)]\Delta_z = \sum_{z \ne z^\star} O\left(\frac{\ln T}{\Delta_z}\right)\Delta_z = O\left(\sum_{z \ne z^\star} \frac{\ln T}{\Delta_z}\right). \tag{48}$$

Now, we re-introduce the Jacobian penalty. The algorithm maximizes $\text{UCB}^{\text{EB}}(z) - \lambda_J \|JG_\theta(z)\|_F^2$. The regret is defined with respect to the true arm values $f(z)$. The penalty term is an additive component in the objective, which translates to an additive component in the regret. The total regret is the sum of the standard bandit regret and the expected penalty of the chosen arm.

$$\sum_{t=1}^{T} \mathbb{E}\left[f_t(z^\star) - f_t(z_t^\star)\right] \le O\left(\sum_{z \ne z^\star} \frac{\ln T}{\Delta_z}\right) + \lambda_J \sum_{t=1}^{T} \mathbb{E}\left[\|JG_\theta(z_t^\star)\|_F^2 - \|JG_\theta(z^\star)\|_F^2\right]. \tag{49}$$

Since the norm is non-negative, we can bound the second part by simply summing the penalty of the chosen arm, which gives the stated result.

# E  Proof of Overall Exploitability

## E.1  Proof of Theorem 3.5 (Finite-Population Exploitability Bound)

**Theorem E.1** (Finite-Population Exploitability Bound). *Assume rewards in $[0,1]$ and the two-time-scale oracle selection. With an optimistic meta-solver (OMWU) and sufficient Monte Carlo samples such that $\mathbb{E}[\|\hat{v}_t - M\sigma_t\|_\infty] = O((nm)^{-1/2})$, the average exploitability is bounded:*

$$\frac{1}{T}\sum_{t=1}^{T}\text{Exploit}(\sigma_t) \le \underbrace{O\left(\frac{1}{T}\sqrt{\ln k_T \sum_{t=1}^{T}\|v_t - v_{t-1}\|_\infty^2}\right)}_{\textit{OMWU Regret}} + \underbrace{O\left(\frac{1}{T}\sum_{t=1}^{T}(nm)^{-1/2}\right)}_{\textit{MC Estimation Error}}$$

$$+ \underbrace{\frac{1}{T}\sum_{t=1}^{T}\varepsilon_{\text{BR},t}}_{\textit{Amortized BR Error}} + \underbrace{O\left(\frac{1}{T}\sum_{z \ne z^\star}\frac{\ln T}{\Delta_z}\right)}_{\textit{Oracle Regret}}. \tag{50}$$

The proof connects the exploitability of the meta-strategy $\sigma_t$ to the various regret and error terms of the algorithm's components. We start by decomposing the exploitability at iteration $t$. Let $A_t = \{\pi_1, \ldots, \pi_{k_t}\}$ be the set of policies in the anchor set.

$$\text{Exploit}(\sigma_t) = \max_\pi \mathbb{E}_{j \sim \sigma_t}[r(\pi, \pi_j)] - \mathbb{E}_{i \sim \sigma_t, j \sim \sigma_t}[r(\pi_i, \pi_j)] \tag{51}$$

Let $\pi_{\mathrm{BR},t} = \arg\max_\pi \mathbb{E}_{j\sim\sigma_t}[r(\pi, \pi_j)]$ be the true best response to $\sigma_t$. Let $\pi_{z_t^\star} = \pi_{G_\theta(z_t^\star)}$ be the policy added by our oracle. Let $\pi_{i^*} = \arg\max_{i\in[k_t]} e_i^\top M\sigma_t$ be the best pure strategy within the current anchor set.

We can decompose the exploitability as follows:

$$\mathrm{Exploit}(\sigma_t) = \mathbb{E}_{j\sim\sigma_t}[r(\pi_{\mathrm{BR},t}, \pi_j)] - \sigma_t^\top M\sigma_t$$

$$= \underbrace{\left(\max_{i\in[k_t]} e_i^\top M\sigma_t - \sigma_t^\top M\sigma_t\right)}_{\text{Term I: Internal Exploitability}} + \underbrace{\left(\mathbb{E}_{j\sim\sigma_t}[r(\pi_{\mathrm{BR},t}, \pi_j)] - \max_{i\in[k_t]} e_i^\top M\sigma_t\right)}_{\text{Term II: Population Gap}} \quad (52)$$

**Bounding Term I:** This is the external regret of Player 1 in the restricted game on $A_t$. From Proposition 3.2, the average regret of the OMWU algorithm using noisy estimates is bounded by the variation of the true payoff vectors:

$$\frac{1}{T}\sum_{t=1}^{T}\left(\max_{i\in[k_t]} e_i^\top M\sigma_t - \sigma_t^\top M\sigma_t\right) \leq O\left(\frac{1}{T}\sqrt{\ln k_T \sum_{t=1}^{T}||v_t - v_{t-1}||_\infty^2}\right) + \frac{1}{T}\sum_{t=1}^{T}\mathbb{E}[\|\hat{v}_t - M\sigma_t\|_\infty]. \quad (53)$$

The first part is the OMWU regret, reflecting a faster convergence rate in our smoothly evolving meta-game, and the second is the MC estimation error.

**Bounding Term II:** This term captures how much better a true best response is compared to the best policy already in our population. We can decompose this further:

$$\text{Term II} = \left(\mathbb{E}_{j\sim\sigma_t}[r(\pi_{\mathrm{BR},t}, \pi_j)] - \mathbb{E}_{j\sim\sigma_t}[r(\pi_{z_t^\star}, \pi_j)]\right) \qquad \text{(a) BR Approx. Error}$$

$$+ \left(\mathbb{E}_{j\sim\sigma_t}[r(\pi_{z_t^\star}, \pi_j)] - \max_{z\in\Lambda_t}\mathbb{E}_{j\sim\sigma_t}[r(\pi_{G_\theta(z)}, \pi_j)]\right) \quad \text{(b) Oracle Instant. Regret} \quad (54)$$

$$+ \left(\max_{z\in\Lambda_t}\mathbb{E}_{j\sim\sigma_t}[r(\pi_{G_\theta(z)}, \pi_j)] - \max_{i\in[k_t]} e_i^\top M\sigma_t\right) \qquad \text{(c) Progress Term}$$

- **Part (a)** is exactly the amortized best-response error, $\varepsilon_{\mathrm{BR},t}$ from Eq. (10), assuming $\pi_{\mathrm{BR},t}$ can be represented by some latent code.

- **Part (b)** is the instantaneous regret of our bandit oracle. Its expected value is what we bounded in Theorem 3.3.

- **Part (c)** is non-positive. Since the anchor set $Z_t$ is a subset of the candidate pool $\Lambda_t$, the maximum over $\Lambda_t$ must be greater than or equal to the maximum over $Z_t$. So we can drop this term from the upper bound.

Summing everything and averaging over $T$:

$$\frac{1}{T}\sum_{t=1}^{T}\mathrm{Exploit}(\sigma_t) \leq \frac{1}{T}\sum_{t=1}^{T}(\text{Term I}_t + \text{Term II}_t)$$

$$\leq \underbrace{O\left(\frac{1}{T}\sqrt{\ln k_T \sum_{t=1}^{T}||v_t - v_{t-1}||_\infty^2}\right) + \frac{1}{T}\sum_{t=1}^{T}\mathbb{E}[\|\hat{v}_t - M\sigma_t\|_\infty]}_{\text{Term I}}$$

$$+ \underbrace{\frac{1}{T}\sum_{t=1}^{T}\varepsilon_{\mathrm{BR},t} + \frac{1}{T}\sum_{t=1}^{T}\left(\max_{z\in\Lambda_t} f_t(z) - f_t(z_t^\star)\right)}_{\text{Term II}} \quad (55)$$

The last term is the average instantaneous oracle regret. The cumulative regret bound from Theorem 3.3 states $\sum_t \mathbb{E}[\max_z f_t(z) - f_t(z_t^\star)] = O(\sum_{z \neq z^\star} \frac{\ln T}{\Delta_z})$. Therefore, the average regret is $O(\frac{1}{T} \sum_{z \neq z^\star} \frac{\ln T}{\Delta_z})$. Substituting this and the rate for MC error gives the final composite bound:

$$
\frac{1}{T} \sum_{t=1}^{T} \text{Exploit}(\sigma_t) \leq \underbrace{O\left( \frac{1}{T} \sqrt{\ln k_T \sum_{t=1}^{T} ||v_t - v_{t-1}||_\infty^2} \right)}_{\text{OMWU Regret}} + \underbrace{O\left( \frac{1}{T} \sum_{t=1}^{T} (nm)^{-1/2} \right)}_{\text{MC Estimation Error}}
$$
$$
+ \underbrace{\frac{1}{T} \sum_{t=1}^{T} \varepsilon_{\text{BR},t}}_{\text{Amortized BR Error}} + \underbrace{O\left( \frac{1}{T} \sum_{z \neq z^\star} \frac{\ln T}{\Delta_z} \right)}_{\text{Oracle Regret}}.
$$

(56)

This completes the proof. Each term corresponds to a component of the algorithm, showing how errors from different sources contribute to the overall performance. As $T \to \infty$, if the sample counts $(n, m)$ increase and the generator training improves ($\varepsilon_{\text{BR}} \to 0$), the average exploitability converges to zero, with the rate of convergence for the meta-game being accelerated by the optimistic updates. $\qquad \square$

# Part II

# Extensions to Two-Player General-Sum and N-Player General-Sum Games

## F  Extension to Two-Player General-Sum Games

We extend the surrogate-free GEMS framework to two-player general-sum Markov games. Let rewards for each player be bounded in $[0, 1]$. For a player $p \in \{1, 2\}$, we denote the other player by $-p$. The conceptual expected payoff matrices are $M^{(1)} \in [0, 1]^{k_1 \times k_2}$ and $M^{(2)} \in [0, 1]^{k_2 \times k_1}$, where $M_{ij}^{(1)} = \mathbb{E}[r^{(1)}(\pi_i^{(1)}, \pi_j^{(2)})]$ and $M_{ji}^{(2)} = \mathbb{E}[r^{(2)}(\pi_i^{(1)}, \pi_j^{(2)})]$. At iteration $t$, each player $p$ has a population of policies induced by an anchor set $Z_t^{(p)}$ with a corresponding mixture strategy $\sigma_t^{(p)} \in \Delta_{k_p(t)-1}$, where $k_p(t) = |Z_t^{(p)}|$.

The value vector for player 1 against player 2's mixture is $v_t^{(1)} = M^{(1)}\sigma_t^{(2)} \in \mathbb{R}^{k_1(t)}$. Similarly, for player 2 against player 1's mixture, it is $v_t^{(2)} = (M^{(2)})^\top \sigma_t^{(1)} \in \mathbb{R}^{k_2(t)}$. The expected value for each player $p$ under the joint mixture is $\bar{r}_t^{(p)} = {\sigma_t^{(1)}}^\top M^{(p)} \sigma_t^{(2)}$.

### F.1  Monte Carlo Meta-Game Estimators (Both Players)

At a fixed iteration $t$, to estimate the value vector $v_t^{(p)}$ for player $p$, we sample opponents $j_s \sim \sigma_t^{(-p)}$ and run $m$ episodes for each pair to obtain returns. The estimators are:

$$
\begin{aligned}
\hat{v}_{t,i}^{(1)} &= \frac{1}{n_i m} \sum_{s=1}^{n_i} \sum_{l=1}^{m} Y_{(i,j_s),l}^{(1)}, \quad j_s \sim \sigma_t^{(2)} \\
\hat{v}_{t,j}^{(2)} &= \frac{1}{\tilde{n}_j m} \sum_{s=1}^{\tilde{n}_j} \sum_{l=1}^{m} Y_{(i_s,j),l}^{(2)}, \quad i_s \sim \sigma_t^{(1)}
\end{aligned}
\tag{57}
$$

The mixture value for each player is estimated via joint sampling $(i_b, j_b) \sim \sigma_t^{(1)} \times \sigma_t^{(2)}$:

$$
\hat{\bar{r}}_t^{(p)} = \frac{1}{Bm} \sum_{b=1}^{B} \sum_{l=1}^{m} Y_{(i_b,j_b),l}^{(p)}
\tag{58}
$$

**Lemma F.1** (Unbiasedness and Concentration, General-Sum)**.** *With rewards in $[0, 1]$, the estimators are unbiased:* $\mathbb{E}[\hat{v}_{t,i}^{(1)}] = (M^{(1)}\sigma_t^{(2)})_i$, $\mathbb{E}[\hat{v}_{t,j}^{(2)}] = ((M^{(2)})^\top \sigma_t^{(1)})_j$, *and* $\mathbb{E}[\hat{\bar{r}}_t^{(p)}] = \bar{r}_t^{(p)}$. *For any* $\delta \in (0, 1)$, *with probability at least* $1 - \delta$, *we have concentration bounds based on the empirical-Bernstein inequality.*

### F.2  Optimistic Replicator Dynamics for Both Players

Each player $p \in \{1, 2\}$ independently performs an **Optimistic Multiplicative-Weights Update (OMWU)** step based on their noisy payoff estimates. This allows each player to adapt more quickly to the opponent's evolving strategy by using the previous payoff vector as a hint. The optimistic estimate for player $p$ is $m_t^{(p)} = 2\hat{v}_t^{(p)} - \hat{v}_{t-1}^{(p)}$, with $\hat{v}_0^{(p)} = \mathbf{0}$. The update rule is:

$$
\sigma_{t+1}^{(p)}(i) \propto \sigma_t^{(p)}(i) \exp\left(\eta_t^{(p)}[2\hat{v}_{t,i}^{(p)} - \hat{v}_{t-1,i}^{(p)} - \hat{\bar{r}}_t^{(p)}]\right), \quad \eta_t^{(p)} > 0
\tag{59}
$$

**Proposition F.2** (Per-Player External Regret with OMWU and Noisy Payoffs). *Let payoffs lie in $[0,1]$. The average external regret for each player $p$ using OMWU is bounded by the variation of their true payoff vectors:*

$$\frac{1}{T}\sum_{t=1}^{T}\left(\max_i(v_t^{(p)})_i - \sigma_t^{(p)\top}v_t^{(p)}\right) \leq O\left(\frac{1}{T}\sqrt{\ln k_{p,T}\sum_{t=1}^{T}||v_t^{(p)} - v_{t-1}^{(p)}||_\infty^2}\right) \tag{60}$$
$$+ \frac{1}{T}\sum_{t=1}^{T}\mathbb{E}\left[||\hat{v}_t^{(p)} - v_t^{(p)}||_\infty\right]$$

*where $v_t^{(p)}$ is the true payoff vector for player $p$ (i.e., $v_t^{(1)} = M^{(1)}\sigma_t^{(2)}$ and $v_t^{(2)} = (M^{(2)})^\top\sigma_t^{(1)}$).*

### F.3 Model-Free EB-UCB Oracles (Double-Oracle Style)

For each player $p$, a finite candidate pool of latent codes $\Lambda_t^{(p)} \subset \mathcal{Z}$ is formed. The value of an "arm" $z \in \Lambda_t^{(p)}$ is its expected payoff against the opponent's current mixture:

$$f_t^{(p)}(z) = \mathbb{E}_{j\sim\sigma_t^{(-p)}}[r^{(p)}(\pi_{G_{\theta_p}(z)}, \pi_j^{(-p)})] \tag{61}$$

We estimate the mean $\hat{\mu}_t^{(p)}(z)$ and variance $\hat{\mathrm{Var}}_t^{(p)}(z)$ via rollouts and score each candidate using an empirical-Bernstein UCB formula. A new anchor $z_t^{*,(p)}$ is selected and added to the player's anchor set.

**Theorem F.3** (Per-Player Instance-Dependent Oracle Regret). *Under a two-time-scale assumption, the cumulative regret of the EB-UCB oracle for player $p$ is bounded as stated in Theorem 3.3.*

### F.4 Overall Guarantee: $\epsilon$-Coarse-Correlated Equilibrium

The time-averaged joint play of the players, $\bar{\mu}_T(i,j) := \frac{1}{T}\sum_{t=1}^{T}\sigma_t^{(1)}(i)\sigma_t^{(2)}(j)$, converges to an $\epsilon$-coarse-correlated equilibrium ($\epsilon$-CCE).

**Theorem F.4** ($\epsilon$-CCE of Time-Average Joint Play). *Assume rewards in $[0,1]$ and the two-time-scale assumption. With per-player OMWU meta-solvers, the empirical distribution $\bar{\mu}_T$ is an $\epsilon$-CCE, with $\epsilon$ bounded by the sum of per-player errors:*

$$\epsilon \leq \sum_{p\in\{1,2\}}\left[\underbrace{O\left(\frac{1}{T}\sqrt{\ln k_{p,T}\sum_{t=1}^{T}||v_t^{(p)} - v_{t-1}^{(p)}||_\infty^2}\right)}_{OMWU\ Regret} + \underbrace{O\left(\frac{1}{T}\sum_{t=1}^{T}(n_p m)^{-1/2}\right)}_{MC\ Estimation\ Error}\right.$$
$$\left. + \underbrace{\frac{1}{T}\sum_{t=1}^{T}\varepsilon_{\mathrm{BR},t}^{(p)}}_{Amortized\ BR\ Error} + \underbrace{O\left(\frac{1}{T}\sum_{z\neq z^{\star,(p)}}\frac{\ln T}{\Delta_z^{(p)}}\right)}_{Oracle\ Regret}\right] \tag{62}$$

*where $\varepsilon_{\mathrm{BR}}^{(p)}$ is the average best-response approximation error for player $p$. As $T \to \infty$ and simulation/training budgets increase, $\epsilon \to 0$.*

## G Proofs for Two-Player General-Sum

### G.1 Proof of Lemma 2 (Unbiasedness and Concentration, General-Sum)

*Proof.* (This proof remains unchanged as it concerns the estimators, not the update rule.) □

### G.2 Proof of Proposition 2 (Per-Player External Regret with OMWU)

*Proof.* The proof adapts the standard analysis for Optimistic MWU to our setting with noisy, unbiased feedback for an arbitrary player $p$. The proof proceeds in two parts.

**Part 1: Regret Against Estimated Losses.** Let the estimated loss for player $p$ be $\hat{l}_{t,i}^{(p)} = 1 - \hat{v}_{t,i}^{(p)}$. The OMWU algorithm uses an optimistic loss estimate $m_t^{(p)} = \hat{l}_t^{(p)} + (\hat{l}_t^{(p)} - \hat{l}_{t-1}^{(p)})$, with $\hat{l}_0^{(p)} = \mathbf{0}$. The weight update for player $p$ is $w_{t+1}^{(p)}(i) = w_t^{(p)}(i) \exp(-\eta^{(p)} m_{t,i}^{(p)})$.

Let $W_t^{(p)} = \sum_{i=1}^{k_p(t)} w_t^{(p)}(i)$ be the potential function for player $p$. Following a standard potential function analysis (as detailed in Appendix C.1), the regret of OMWU with respect to the *observed* sequence of losses is bounded by its variation:

$$\sum_{t=1}^{T} \langle \sigma_t^{(p)}, \hat{l}_t^{(p)} \rangle - \min_i \sum_{t=1}^{T} \hat{l}_{t,i}^{(p)} \leq O\left( \sqrt{\ln k_{p,T} \sum_{t=1}^{T} ||\hat{l}_t^{(p)} - \hat{l}_{t-1}^{(p)}||_\infty^2} \right) \tag{63}$$

**Part 2: Translating to True Regret.** We now relate this bound to the regret against the true payoffs $v_t^{(p)}$. Let the true loss be $l_t^{(p)} = \mathbf{1} - v_t^{(p)}$ and the estimation error be $\Delta_t^{(p)} = \hat{l}_t^{(p)} - l_t^{(p)} = v_t^{(p)} - \hat{v}_t^{(p)}$. The cumulative true regret for player $p$, $R_T^{(p)}$, is:

$$
\begin{aligned}
R_T^{(p)} &= \sum_{t=1}^{T} \left( \max_i v_{t,i}^{(p)} - \langle \sigma_t^{(p)}, v_t^{(p)} \rangle \right) = \sum_{t=1}^{T} \left( \langle \sigma_t^{(p)}, l_t^{(p)} \rangle - \min_i l_{t,i}^{(p)} \right) \\
&= \sum_{t=1}^{T} \left( \langle \sigma_t^{(p)}, \hat{l}_t^{(p)} - \Delta_t^{(p)} \rangle - (\hat{l}_{t,i^*}^{(p)} - \Delta_{t,i^*}^{(p)}) \right) \\
&= \underbrace{\sum_{t=1}^{T} (\langle \sigma_t^{(p)}, \hat{l}_t^{(p)} \rangle - \hat{l}_{t,i^*}^{(p)})}_{\text{Regret on Estimates}} - \underbrace{\sum_{t=1}^{T} (\langle \sigma_t^{(p)}, \Delta_t^{(p)} \rangle - \Delta_{t,i^*}^{(p)})}_{\text{Cumulative Error Term}}
\end{aligned}
\tag{64}
$$

The first term is bounded as derived in Part 1. For the variation term in that bound, we use the triangle inequality:

$$||\hat{l}_t^{(p)} - \hat{l}_{t-1}^{(p)}||_\infty \leq ||\hat{l}_t^{(p)} - l_t^{(p)}||_\infty + ||l_t^{(p)} - l_{t-1}^{(p)}||_\infty + ||l_{t-1}^{(p)} - \hat{l}_{t-1}^{(p)}||_\infty \tag{65}$$

$$\implies ||\hat{v}_t^{(p)} - \hat{v}_{t-1}^{(p)}||_\infty \leq ||\Delta_t^{(p)}||_\infty + ||v_t^{(p)} - v_{t-1}^{(p)}||_\infty + ||\Delta_{t-1}^{(p)}||_\infty \tag{66}$$

Taking the expectation over the sampling noise in our estimators, and noting that $\mathbb{E}[\Delta_t^{(p)}] = \mathbf{0}$ due to unbiasedness, we can bound the expected true regret. The error terms accumulate, leading to the final result. Dividing by $T$ gives the statement in Proposition 60. $\qquad\square$

### G.3 Proof of Theorem 3 (Per-Player Oracle Regret)

*Proof.* (This proof remains unchanged as it concerns the bandit oracle, which is decoupled from the meta-game solver by the two-time-scale assumption.) $\qquad\square$

### G.4 Proof of Theorem 4 ($\epsilon$-CCE)

*Proof.* A time-averaged joint strategy $\bar{\mu}_T$ is an $\epsilon$-CCE if for each player $p$ and any deviating strategy $\pi'$, the gain from deviating is small:

$$\sum_{i,j} \bar{\mu}_T(i,j) r^{(p)}(\pi_i^{(1)}, \pi_j^{(2)}) \geq \sum_{i,j} \bar{\mu}_T(i,j) r^{(p)}(\pi', \pi_j^{(-p)}) - \epsilon_p \tag{67}$$

where $\epsilon = \sum_p \epsilon_p$. The maximum gain for player $p$ from deviating is bounded by their average external regret against the sequence of opponent strategies $\{\sigma_t^{(-p)}\}_{t=1}^T$.

$$\epsilon_p \le \frac{1}{T} \sum_{t=1}^{T} \left( \max_{\pi'} \mathbb{E}_{j \sim \sigma_t^{(-p)}}[r^{(p)}(\pi', \pi_j^{(-p)})] - \mathbb{E}_{i \sim \sigma_t^{(p)}, j \sim \sigma_t^{(-p)}}[r^{(p)}(\pi_i^{(p)}, \pi_j^{(-p)})] \right) \tag{68}$$

This is the definition of player $p$'s average exploitability of the sequence of joint strategies. We decompose this term for each player $p$, analogous to the decomposition in the proof of Theorem 3.4. The exploitability for player $p$ at iteration $t$ is:

$$\text{Exploit}_t^{(p)} = \underbrace{\left( \max_{i \in [k_p(t)]} (v_t^{(p)})_i - \langle \sigma_t^{(p)}, v_t^{(p)} \rangle \right)}_{\text{Internal Exploitability}^{(p)}} + \underbrace{\left( \max_{\pi'} \mathbb{E}_{j \sim \sigma_t^{(-p)}}[r^{(p)}(\pi', \pi_j^{(-p)})] - \max_{i \in [k_p(t)]} (v_t^{(p)})_i \right)}_{\text{Population Gap}^{(p)}} \tag{69}$$

Averaging over $T$ and summing over players gives the total bound on $\epsilon$.

1. **Internal Exploitability**: For each player $p$, this is their external regret within the restricted game. From Proposition 60, its average is bounded by the sum of the OMWU Regret and the MC Estimation Error.

2. **Population Gap**: For each player $p$, this term is decomposed further into their Amortized BR Error and Oracle Regret, following the same logic as in the proof of Theorem 3.4.

Summing these four distinct error sources for each player $p \in \{1, 2\}$ provides the composite bound for $\epsilon$ as stated in Theorem 62. □

## H    Extension to N-Player General-Sum Games

We now generalize the framework to $n \ge 2$ players. Let $\mathcal{P} = \{1, ..., n\}$ be the set of players, and for any player $p \in \mathcal{P}$, let $-p := \mathcal{P} \setminus \{p\}$ denote the set of all other players. At each iteration $t$, every player $p$ maintains an anchor set $Z_t^{(p)}$, a generator $G_{\theta_p}$, and a meta-strategy $\sigma_t^{(p)} \in \Delta_{k_p(t)-1}$. The joint mixture over all players is $\Sigma_t = \bigotimes_{q \in \mathcal{P}} \sigma_t^{(q)}$.

For each player $p$, the conceptual payoff is represented by a hypermatrix $M^{(p)}$ with entries corresponding to the expected reward for player $p$ given a joint profile of pure strategies. The expected payoff for player $p$'s $i$-th policy against the joint mixture of all other players is:

$$v_t^{(p)}(i) = \mathbb{E}_{i_{-p} \sim \otimes_{q \in -p} \sigma_t^{(q)}} \left[ r^{(p)}(\pi_i^{(p)}, \{\pi_{i_q}^{(q)}\}_{q \in -p}) \right] = \sum_{i_{-p}} M_{i,i_{-p}}^{(p)} \prod_{q \in -p} \sigma_t^{(q)}(i_q) \tag{70}$$

and player $p$'s expected value under the full joint mixture is $\bar{r}_t^{(p)} = \sum_i \sigma_t^{(p)}(i) v_t^{(p)}(i)$.

### H.1   Monte Carlo Meta-Game with Importance Weighting

To avoid forming the computationally intractable payoff hypermatrices, we use importance-weighted estimators derived from a single set of shared game rollouts. We draw $B$ joint strategy profiles $i^{(b)} = (i_b^{(1)}, ..., i_b^{(n)}) \sim \Sigma_t$, run $m$ episodes for each profile, and obtain returns $Y_{b,l}^{(p)}$ for every player $p$. The estimators for the per-policy value vector and the mixture value are:

$$\hat{v}_{t,i}^{(p)} = \frac{1}{Bm} \sum_{b=1}^{B} \sum_{l=1}^{m} \frac{\mathbf{1}\{i_b^{(p)} = i\}}{\sigma_t^{(p)}(i)} Y_{b,l}^{(p)}$$

$$\hat{\bar{r}}_t^{(p)} = \frac{1}{Bm} \sum_{b=1}^{B} \sum_{l=1}^{m} Y_{b,l}^{(p)} \tag{71}$$

**Lemma H.1** (Unbiasedness and Concentration for n Players). *For rewards in $[0,1]$, the estimators are unbiased: $\mathbb{E}[\hat{v}_{t,i}^{(p)}] = v_t^{(p)}(i)$ and $\mathbb{E}[\hat{\bar{r}}_t^{(p)}] = \bar{r}_t^{(p)}$ for all $p, i$. For any $\delta \in (0,1)$, with probability at least $1 - \delta$, we have concentration bounds. However, the variance of the importance-weighted estimator $\hat{v}_{t,i}^{(p)}$ can be high if any policy's probability $\sigma_t^{(p)}(i)$ is small.*

## H.2 Per-Player Optimistic Replicator and Oracle

Each player $p$ runs an independent learning process to update their meta-strategy and expand their policy set.

- **OMWU Update:** Each player uses the Optimistic MWU rule to update their meta-strategy based on their individual payoff estimates. The optimistic estimate for player $p$ is $m_t^{(p)} = 2\hat{v}_t^{(p)} - \hat{v}_{t-1}^{(p)}$, with $\hat{v}_0^{(p)} = \mathbf{0}$.

$$\sigma_{t+1}^{(p)}(i) \propto \sigma_t^{(p)}(i) \exp\left(\eta_t^{(p)}[2\hat{v}_{t,i}^{(p)} - \hat{v}_{t-1,i}^{(p)} - \hat{\bar{r}}_t^{(p)}]\right) \tag{72}$$

- **EB-UCB Oracle:** Each player $p$ solves a separate multi-armed bandit problem to find a promising new anchor $z_t^{*,(p)}$. The value of an arm $z$ is its expected reward against the joint mixture of all other players:

$$f_t^{(p)}(z) = \mathbb{E}_{i_{-p} \sim \otimes_{q \in -p} \sigma_t^{(q)}} \left[ r^{(p)}(\pi_{G_{\theta_p}(z)}^{(p)}, \{\pi_{i_q}^{(q)}\}_{q \in -p}) \right] \tag{73}$$

**Proposition H.2** (Per-Player External Regret with OMWU in n-Player Games). *For any player $p$, assume payoffs lie in $[0,1]$. The average external regret for player $p$ using OMWU against the sequence of opponents' joint strategies is bounded by:*

$$\frac{1}{T}\sum_{t=1}^{T}\left(\max_i(v_t^{(p)})_i - \sigma_t^{(p)^\top} v_t^{(p)}\right) \leq O\left(\frac{1}{T}\sqrt{\ln k_{p,T} \sum_{t=1}^{T} ||v_t^{(p)} - v_{t-1}^{(p)}||_\infty^2}\right)$$
$$+ \frac{1}{T}\sum_{t=1}^{T}\mathbb{E}\left[||\hat{v}_t^{(p)} - v_t^{(p)}||_\infty\right] \tag{74}$$

**Theorem H.3** (Per-Player Instance-Dependent Oracle Regret). *Under a two-time-scale assumption for n-players, where opponents' mixtures $\{\sigma_t^{(q)}\}_{q \in -p}$ are considered fixed during player p's oracle step, the cumulative oracle regret for each player is bounded:*

$$\sum_{t=1}^{T}\mathbb{E}[f_t^{(p)}(z^{*,(p)}) - f_t^{(p)}(z_t^{*,(p)})] = O\left(\sum_{z \neq z^{*,(p)}}\frac{\ln T}{\Delta_z^{(p)}}\right) + \lambda_{J,p}\sum_{t=1}^{T}\mathbb{E}\left[||J_{G_{\theta_p}}(z_t^{*,(p)})||_F^2\right] \tag{75}$$

## H.3 Overall Guarantee: $\epsilon$-CCE for n Players

The time-averaged joint play, defined as $\bar{\mu}_T(i_1, ..., i_n) := \frac{1}{T}\sum_{t=1}^{T}\prod_{p=1}^{n}\sigma_t^{(p)}(i_p)$, converges to an $\epsilon$-Coarse-Correlated Equilibrium ($\epsilon$-CCE).

**Theorem H.4** ($\epsilon$-CCE of Time-Average Joint Play for n Players). *Assume rewards in $[0,1]$ and the n-player two-time-scale assumption. With per-player OMWU meta-solvers, the time-averaged distribution $\bar{\mu}_T$ is an*

*$\epsilon$-CCE, where the total deviation incentive $\epsilon$ is bounded by the sum of regrets and errors across all players:*

$$\epsilon \leq \sum_{p=1}^{n} \left[ \underbrace{O\left(\frac{1}{T}\sqrt{\ln k_{p,T} \sum_{t=1}^{T} ||v_t^{(p)} - v_{t-1}^{(p)}||_\infty^2}\right)}_{\text{OMWU Regret}} + \underbrace{\frac{1}{T}\sum_{t=1}^{T} \mathbb{E}[||\hat{v}_t^{(p)} - v_t^{(p)}||_\infty]}_{\text{MC Estimation Error}} \right.$$

$$\left. + \underbrace{\frac{1}{T}\sum_{t=1}^{T}\varepsilon_{\text{BR},t}^{(p)}}_{\text{Amortized BR Error}} + \underbrace{O\left(\frac{1}{T}\sum_{z\neq z^{\star,(p)}}\frac{\ln T}{\Delta_z^{(p)}}\right)}_{\text{Oracle Regret}} \right] \quad (76)$$

*If simulation and training budgets grow such that the error terms for each player go to zero, then $\epsilon \to 0$ as $T \to \infty$.*

## I  Proofs for N-Player General-Sum

### I.1  Proof of Lemma 3 (Unbiasedness and Concentration, n-Players)

*Proof.* (This proof remains unchanged as it concerns the estimators, not the update rule.)  □

### I.2  Proof of Proposition 3 (Per-Player External Regret with OMWU)

*Proof.* The proof is structurally identical to the proof of Proposition 2 for the two-player case. We present it here explicitly for the n-player setting for completeness. The analysis is performed for an arbitrary player $p \in \{1, ..., n\}$.

**Part 1: Regret Against Estimated Losses.** Let the estimated loss for player $p$ be $\hat{l}_{t,i}^{(p)} = 1 - \hat{v}_{t,i}^{(p)}$. The OMWU algorithm uses an optimistic loss estimate $m_t^{(p)} = \hat{l}_t^{(p)} + (\hat{l}_t^{(p)} - \hat{l}_{t-1}^{(p)})$, with $\hat{l}_0^{(p)} = \mathbf{0}$. The weight update for player $p$ is $w_{t+1}^{(p)}(i) = w_t^{(p)}(i)\exp(-\eta^{(p)}m_{t,i}^{(p)})$.

The regret of OMWU with respect to this observed sequence of losses is bounded by its variation, following a standard potential function analysis:

$$\sum_{t=1}^{T}\langle\sigma_t^{(p)}, \hat{l}_t^{(p)}\rangle - \min_i \sum_{t=1}^{T}\hat{l}_{t,i}^{(p)} \leq O\left(\sqrt{\ln k_{p,T}\sum_{t=1}^{T}||\hat{l}_t^{(p)} - \hat{l}_{t-1}^{(p)}||_\infty^2}\right) \quad (77)$$

The "environment" from player $p$'s perspective is the sequence of payoff vectors it receives. The regret bound does not depend on how this sequence is generated by the other $n-1$ players, only on its properties (i.e., its variation).

**Part 2: Translating to True Regret.** We relate this bound to the regret against the true payoffs $v_t^{(p)}$. Let the true loss be $l_t^{(p)} = \mathbf{1} - v_t^{(p)}$ and the estimation error be $\Delta_t^{(p)} = \hat{l}_t^{(p)} - l_t^{(p)} = v_t^{(p)} - \hat{v}_t^{(p)}$. The cumulative true regret for player $p$, $R_T^{(p)}$, is:

$$R_T^{(p)} = \sum_{t=1}^{T}\left(\max_i v_{t,i}^{(p)} - \langle\sigma_t^{(p)}, v_t^{(p)}\rangle\right) = \sum_{t=1}^{T}\left(\langle\sigma_t^{(p)}, l_t^{(p)}\rangle - \min_i l_{t,i}^{(p)}\right)$$

$$= \underbrace{\sum_{t=1}^{T}(\langle\sigma_t^{(p)}, \hat{l}_t^{(p)}\rangle - \min_i \hat{l}_{t,i}^{(p)})}_{\text{Regret on Estimates}} - \underbrace{\sum_{t=1}^{T}(\langle\sigma_t^{(p)}, \Delta_t^{(p)}\rangle - \Delta_{t,i^*}^{(p)})}_{\text{Cumulative Error Term}} \quad (78)$$

The first term is bounded as derived in Part 1. The variation term in that bound is handled with the triangle inequality: $||\hat{v}_t^{(p)} - \hat{v}_{t-1}^{(p)}||_\infty \leq ||\Delta_t^{(p)}||_\infty + ||v_t^{(p)} - v_{t-1}^{(p)}||_\infty + ||\Delta_{t-1}^{(p)}||_\infty$. Taking the expectation over the sampling noise and dividing by $T$ yields the final bound stated in the proposition.  □

### I.3 Proof of Theorem 5 (Per-Player Oracle Regret)

*Proof.* The proof is a standard bandit analysis, applied to the independent learning problem faced by each player $p$. Under the two-time-scale assumption, the joint mixture of all other players, $\Sigma_t^{(-p)} = \otimes_{q \in -p} \sigma_t^{(q)}$, is considered fixed during player $p$'s oracle selection phase at meta-iteration $t$.

For player $p$, the learning problem is a stochastic multi-armed bandit task where:

- The set of arms is the candidate pool of latent codes, $\Lambda_t^{(p)}$.

- The reward for pulling an arm $z \in \Lambda_t^{(p)}$ is a random variable (the outcome of a game rollout) whose expectation is the true arm value, $f_t^{(p)}(z)$, as defined in Equation 73.

This is a standard bandit setting. The EB-UCB algorithm guarantees that the number of times a suboptimal arm $z$ is pulled, $N_z(T)$, is bounded logarithmically with respect to the total number of oracle steps $T$, i.e., $\mathbb{E}[N_z(T)] = O(\ln T / \Delta_z^{(p)})$, where $\Delta_z^{(p)}$ is the suboptimality gap. The total regret is the sum of the expected regrets from pulling each suboptimal arm. The Jacobian penalty adds a simple additive term to this regret, leading to the final bound stated in the main text. □

### I.4 Proof of Theorem 6 ($\epsilon$-CCE for n Players)

*Proof.* The proof generalizes the argument from the two-player case. A time-averaged joint distribution $\bar{\mu}_T$ is an $\epsilon$-CCE if for every player $p \in \{1, ..., n\}$ and for every possible unilateral deviation to a pure strategy $\pi_p'$, the incentive to deviate is bounded:

$$\mathbb{E}_{i \sim \bar{\mu}_T}[r^{(p)}(\pi_{i_p}^{(p)}, \pi_{i_{-p}}^{(-p)})] \geq \mathbb{E}_{i_{-p} \sim \bar{\mu}_T^{(-p)}}[r^{(p)}(\pi_p', \pi_{i_{-p}}^{(-p)})] - \epsilon_p \tag{79}$$

where $\epsilon = \sum_{p=1}^n \epsilon_p$. The maximum gain for player $p$ from deviating is bounded by their average external regret over the $T$ iterations. Let's bound this gain, $\epsilon_p$:

$$\epsilon_p \leq \frac{1}{T} \sum_{t=1}^T \left( \max_{\pi'} \mathbb{E}_{i_{-p} \sim \Sigma_t^{(-p)}}[r^{(p)}(\pi', \pi_{i_{-p}}^{(-p)})] - \mathbb{E}_{i \sim \Sigma_t}[r^{(p)}(\pi_i)] \right) \tag{80}$$

This is precisely player $p$'s average exploitability over the sequence of play. We decompose this term for each player $p$ into four components, following the structure of the proof of Theorem 3.4. For each player $p$, their average exploitability is bounded by the sum of:

1. **Average Internal Exploitability**: The regret of OMWU within the restricted game on $Z_t^{(p)}$, which is bounded by Proposition 3. This gives the **OMWU Regret** and the **MC Estimation Error** terms.

2. **Average Population Gap**: The gap between a true best response and the best response within $Z_t^{(p)}$. This is further decomposed into the **Amortized BR Error** (by definition) and the **Oracle Regret** (from Theorem 5).

Thus, for each player $p$, their maximum deviation incentive $\epsilon_p$ is bounded by the sum of their four error terms:

$$\epsilon_p \leq O\left( \frac{1}{T} \sqrt{\ln k_{p,T} \sum_{t=1}^T ||v_t^{(p)} - v_{t-1}^{(p)}||_\infty^2} \right) + \frac{1}{T} \sum_{t=1}^T \mathbb{E}[||\hat{v}_t^{(p)} - v_t^{(p)}||_\infty] + \bar{\varepsilon}_{\text{BR}}^{(p)} + \bar{\mathcal{R}}_T^{(p)} \tag{81}$$

where $\bar{\varepsilon}_{\text{BR}}^{(p)}$ and $\bar{\mathcal{R}}_T^{(p)}$ are the average BR error and oracle regret, respectively. The total error $\epsilon$ is the sum of these deviation incentives, $\epsilon = \sum_{p=1}^n \epsilon_p$. Summing the bounds for each player provides the composite bound for $\epsilon$ as stated in Theorem 76. □

# Part III

# Ablation and Analysis of experiments

All empirical evaluations presented in this paper were conducted on computing systems equipped with Intel i9 processors, utilizing either NVIDIA RTX A2000 GPU cluster (12GB VRAM) or an NVIDIA RTX 6000 Ada Generation cluster (48GB VRAM).

Table 2: **Comprehensive Summary of Gems Experiments.** This table aggregates all environments evaluated in the paper, categorized by scale and type. It confirms that GEMS scales robustly to high-dimensional latent spaces ($d_z = 16$) and large populations across competitive, cooperative, and game-theoretic ablation settings.

| Environment | Type | Action Space | Latent Dim ($d_z$) | Iterations ($T$) | Policies ($|Z_T|$) |
|---|---|---|---|---|---|
| *Complex Scale* | | | | | |
| **Chess** (v6) | Competitive | $\approx 4,672$ | 8 | 1,000 | **2,002** |
| **Go** (9x9) | Competitive | $\approx 82$ | 8 | 200 | 402 |
| **Hanabi** | Cooperative | $\approx 20$ | 8 | 200 | 402 |
| **Connect 4** | Competitive | 7 | 8 | 200 | 402 |
| *Continuous Control (Multi-particle)* | | | | | |
| **Simple Spread** (MPE) | Cooperative | Continuous (5) | 8 | 100 | 202 |
| **Simple Tag** (MPE) | Competitive | Continuous (5) | 8 | 100 | 202 |
| *Game-Theoretic and Ablations* | | | | | |
| **Kuhn Poker** | Imperfect Info | 2 | 2, 4, 8 | 40 | 82 |
| **Public Goods Game** | N-Player | Continuous | **8, 16** | 10 | 60 |
| **Deceptive Mean** | Zero-Sum | Continuous | **8, 16** | 6 | 14 |

*Note: "Policies ($|Z_T|$)" is calculated as $|Z_0| + N_{players} \times T$.*

**Initialization of the anchor set.** Algorithm 1 takes the initial anchor set $\mathcal{Z}_0 = \{z_1, \ldots, z_{k_0}\}$ as an input. We intentionally leave the choice of this initialization to the implementer: anchors can be sampled from a prior over the latent space, warm-started from pretrained policies, or constructed via domain-specific heuristics. In our experiments, we use a simple and standard choice and initialize anchors independently from a standard normal distribution, i.e., $z_i \sim \mathcal{N}(0, I)$, with the initial meta-strategy $\sigma_0$ set to be uniform over $\mathcal{Z}_0$.

**Initialization of the meta-strategy.** The initial meta-strategy $\sigma_0$ is defined as a probability distribution over the initial anchor set $\mathcal{Z}_0$. In our experiments, we initialize $\sigma_0$ to be uniform over $\mathcal{Z}_0$, assigning equal probability mass to each initial anchor. This choice reflects the absence of prior preference among anchors at initialization and provides a neutral starting point for subsequent meta-strategy updates via optimistic multiplicative weights. Alternative initializations, such as biased or warm-started distributions, can be incorporated without modifying the core algorithm.

**Policy optimization in the amortized best-response.** In Phase 4 of Algorithm 1, agents are trained via policy-gradient methods using advantage-based objectives. Concretely, the generator parameters $\theta$ are updated by ascending a regularized policy objective that depends on estimated advantages, allowing any standard policy-based reinforcement learning algorithm to be used in this step. In our implementation, we instantiate this update using Proximal Policy Optimization (PPO) with generalized advantage estimation (GAE), and train agents in `PettingZoo` environments accordingly. We emphasize that this choice is not intrinsic to GEMS, and alternative policy-gradient or trust-region methods could be substituted without altering the overall framework.

**Default hyperparameters for Kuhn Poker.** For reproducibility, Table 3 reports the exact default hyperparameters used in our **Kuhn Poker** runs of GEMS (Algorithm 1). Unless otherwise stated, all Kuhn experiments in this paper use these settings: an outer-loop budget of $T=40$ with $k_0=1$ initial anchor per role (implying $|Z_T|=2(k_0+T)=82$ anchors for the two-player case), latent dimension $d_z=8$, and fixed Monte Carlo / oracle budgets for meta-game estimation and EB-UCB expansion. Meta-strategy updates use OMWU with $\eta=0.03$ and a constant schedule, while Phase 4 applies ABR-TR training for $K_{\text{ABR}}=30$ gradient steps per outer iteration with learning rate $2 \times 10^{-4}$ and KL regularization weight $\beta_{\text{KL}}=0.05$ (Table 3).

Table 3: **Default hyperparameters for GEMS (Algorithm 1) on Kuhn poker.** Defaults used in the reference implementation, grouped by phases: (1) MC meta-game estimation, (2) OMWU update, (3) EB-UCB expansion, (4) ABR-TR training.

| Block | Parameter | Symbol | Default | Role |
|---|---|---|---|---|
| *Budget / population* | | | | |
| Iterations | Outer iterations | $T$ | 40 | Outer-loop horizon. |
| Initial anchors (per role) | Init. anchors | $k_0$ | 1 | Initial anchor set size. |
| Max anchors (per role) | Cap | $k_{\max}$ | 32 | Caps population size. |
| Total anchors (2-player) | Implied total | $|Z_T|$ | $2(k_0 + T) = 82$ | $|Z_T| = |Z_0| + 2T$, $|Z_0| = 2k_0$. |
| *Generator / latent* | | | | |
| Latent dimension | Latent size | $d_z$ | 8 | Latent code dimension. |
| Temperature | Temp. | $\tau$ | 1.0 | Sampling temperature. |
| *Phase 1: Meta-game estimation (MC)* | | | | |
| Opponent batch size | Opp. batch | $N_{\text{MC}}$ | 8 | Opponents from $\sigma_{t-1}$. |
| Rollouts per matchup | Rollouts | $M_{\text{MC}}$ | 2 | Rollouts per matchup. |
| Bootstrap samples | Bootstraps | $B$ | 128 | Bootstrap resamples. |
| EMA smoothing | EMA coef. | $\alpha_{\text{EMA}}$ | 0.0 | 0 disables EMA. |
| *Phase 2: Meta-strategy update (OMWU)* | | | | |
| OMWU step size | Step size | $\eta$ | 0.03 | OMWU learning rate. |
| Schedule | Schedule | — | `const` | `const/sqrt/harmonic`. |
| Logit cap | Logit clip | — | 50.0 | Caps logits. |
| *Phase 3: Expansion (EB-UCB)* | | | | |
| Oracle opponent batch size | Opp. batch | $N_{\text{oracle}}$ | 8 | Opponents from $\sigma_t$. |
| Oracle rollouts per candidate | Rollouts | $M_{\text{oracle}}$ | 2 | Rollouts per candidate. |
| Mutated candidates | Mut. pool | $|\Lambda_{\text{mut}}|$ | 32 | Mutation candidates. |
| Random candidates | Rand. pool | $|\Lambda_{\text{rand}}|$ | 32 | Extra random candidates. |
| Mutation scale | Mut. stddev | $\sigma_{\text{mut}}$ | 0.2 | Latent mutation radius. |
| UCB confidence | Confidence | $\delta_0$ | 0.5 | UCB confidence. |
| *Phase 4: ABR-TR training* | | | | |
| Training steps | Grad steps | $K_{\text{ABR}}$ | 30 | Updates per outer iter. |
| Anchors per batch | Batch anchors | — | 16 | Anchors from $Z_t$. |
| Learning rate | Optimizer lr | — | $2 \times 10^{-4}$ | Step size for generator. |
| KL penalty weight | KL coef. | $\beta_{\text{KL}}$ | 0.05 | Trust-region weight. |
| New-opponent fraction | Mix frac. | $q_{\text{new}}$ | 0.25 | Mix new vs. old. |
| Gradient clipping | Norm clip | — | 0.5 | Global grad-norm clip. |

. . . . . . . . . . . . . . . . . . . . . . . . . . . . . . . . . . . . . . . . . . . . . . . . . . . . . . . . . . . . . . . . . . . . . . . . . . . . . . . . . . . . . . . . . . . . . . . . . . .

## J    Run on Connect–4

**Additional Analysis on Connect-4 (PettingZoo).**   Figures 7–9 present supplementary diagnostics for GEMS on the Connect-4 environment implemented using `PettingZoo` (`connect_four_v3`), evaluated across four seeds (0–3). Figure 7 compares memory consumption and cumulative wall-clock time over training iterations for GEMS and PSRO. While PSRO exhibits steadily increasing memory usage and superlinear growth in cumulative runtime—a consequence of explicit population expansion, payoff table maintenance, and repeated best-response computation—GEMS maintains near-constant memory throughout training and achieves substantially lower cumulative wall-clock time. This behavior is consistent with the amortized nature of GEMS, where a single generator implicitly represents and updates the evolving policy population without requiring explicit storage or enumeration of all past strategies.

Figures 8 and 9 visualize principal component projections of learned representations across seeds, providing insight into the relationship between behavioral convergence and representational structure. In Figure 8, the PCA embeddings of action-distribution features reveal a **continuous, non-linear manifold structure** rather than disjoint clusters. Since Connect-4 is a solved game (perfect information, zero-sum) where optimal play dictates specific minimax trajectories, the observed strong overlap between players is theoretically expected: it indicates that GEMS has converged to the shared, symmetric geometry of optimal play. The smooth gradients visible in the embeddings suggest that the generator $\mathcal{G}_\theta$ does not merely memorize a single solution point, but captures the smooth topology of valid strategies leading to the Nash equilibrium.

In contrast, Figure 9 shows that the corresponding latent-space embeddings remain more dispersed across seeds and players. This increased dispersion indicates that, while the induced action distributions converge to the singular optimal behavior required by the solved game mechanics, the underlying latent representations preserve diversity. This prevents the generator from collapsing into a narrow region of the latent space, ensuring it retains the expressivity to generate counter-strategies if the opponent were to deviate from optimality.

Taken together, these results illustrate a key property of GEMS: Behavioral convergence emerges at the level of action distributions (reflecting the solved nature of the game), while representational diversity is retained within the latent space. This decoupling enables GEMS to scale efficiently without sacrificing expressivity, supporting stable learning dynamics and reducing the risk of premature mode collapse. Combined with the resource efficiency observed in Figure 7, these findings reinforce the advantage of amortized, generator-based meta-solvers over classical PSRO-style approaches in multi-agent zero-sum games.

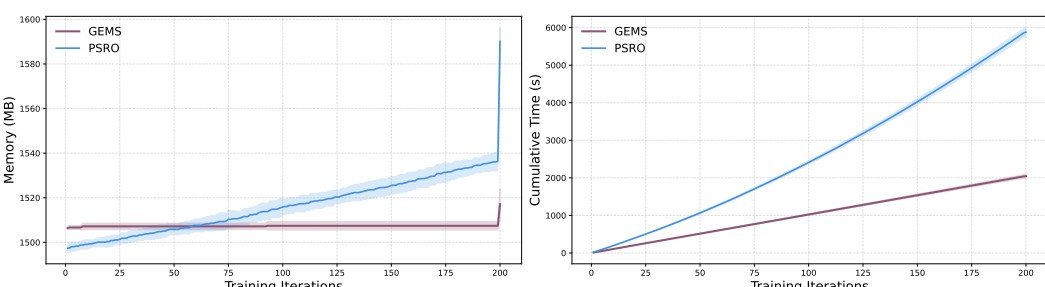

Figure 7: ***Connect–4: Resource Usage.*** Memory consumption (left) and cumulative wall-clock time (right) over training iterations for GEMS and PSRO.

**Connect–4: Action Embeddings (Seeds 0–3)**

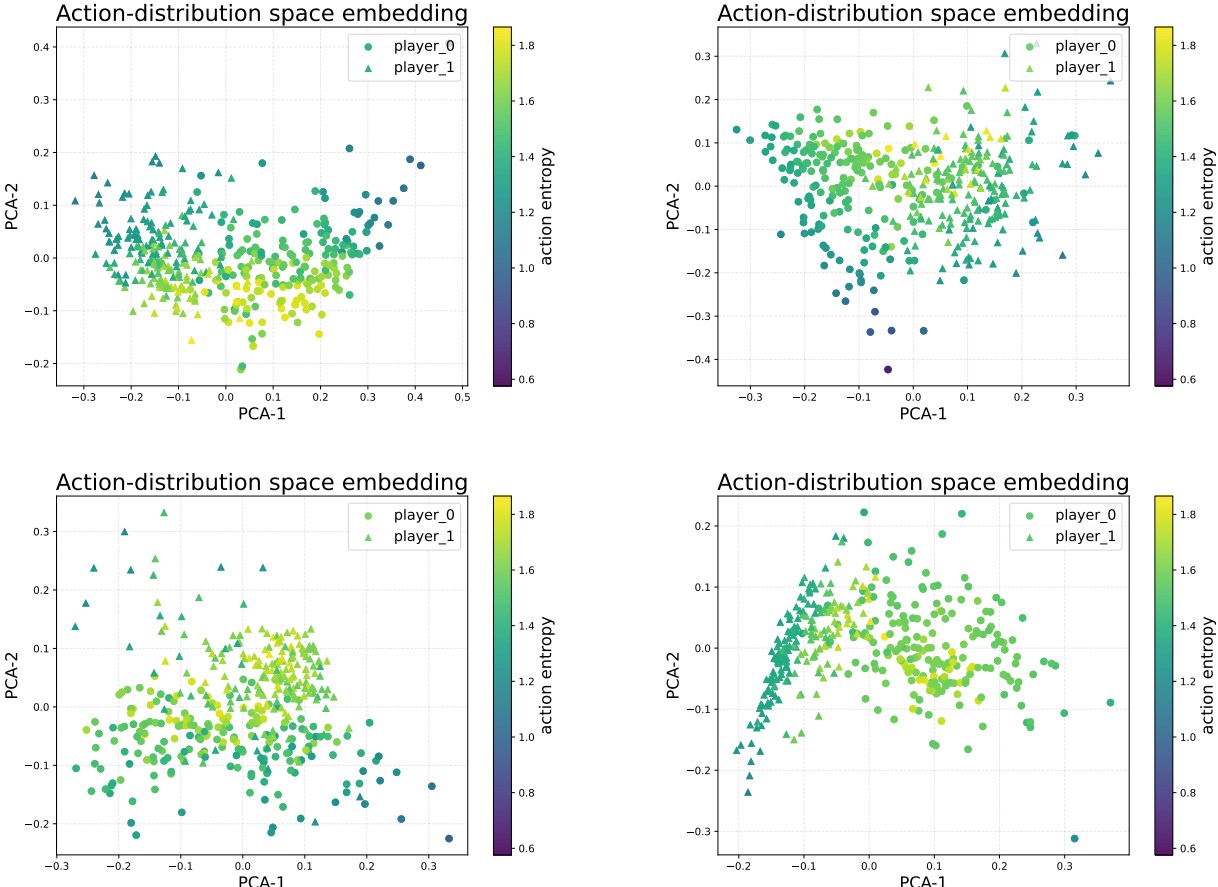

Figure 8: ***Connect–4: Learned Strategy Manifold.*** PCA visualization of state-conditional action distributions across four random seeds (0–3). Unlike discrete population methods, the embeddings reveal a **continuous, non-linear manifold** structure, indicating that the generator $\mathcal{G}_\theta$ captures the complex geometry of the solution space. The smooth gradients in the projection correspond to coherent variations in strategic aggression and confidence, confirming that GEMS maintains **structured diversity** and avoids mode collapse while converging to symmetric equilibrium behaviors.

**Connect–4: Latent Embeddings (Seeds 0–3)**

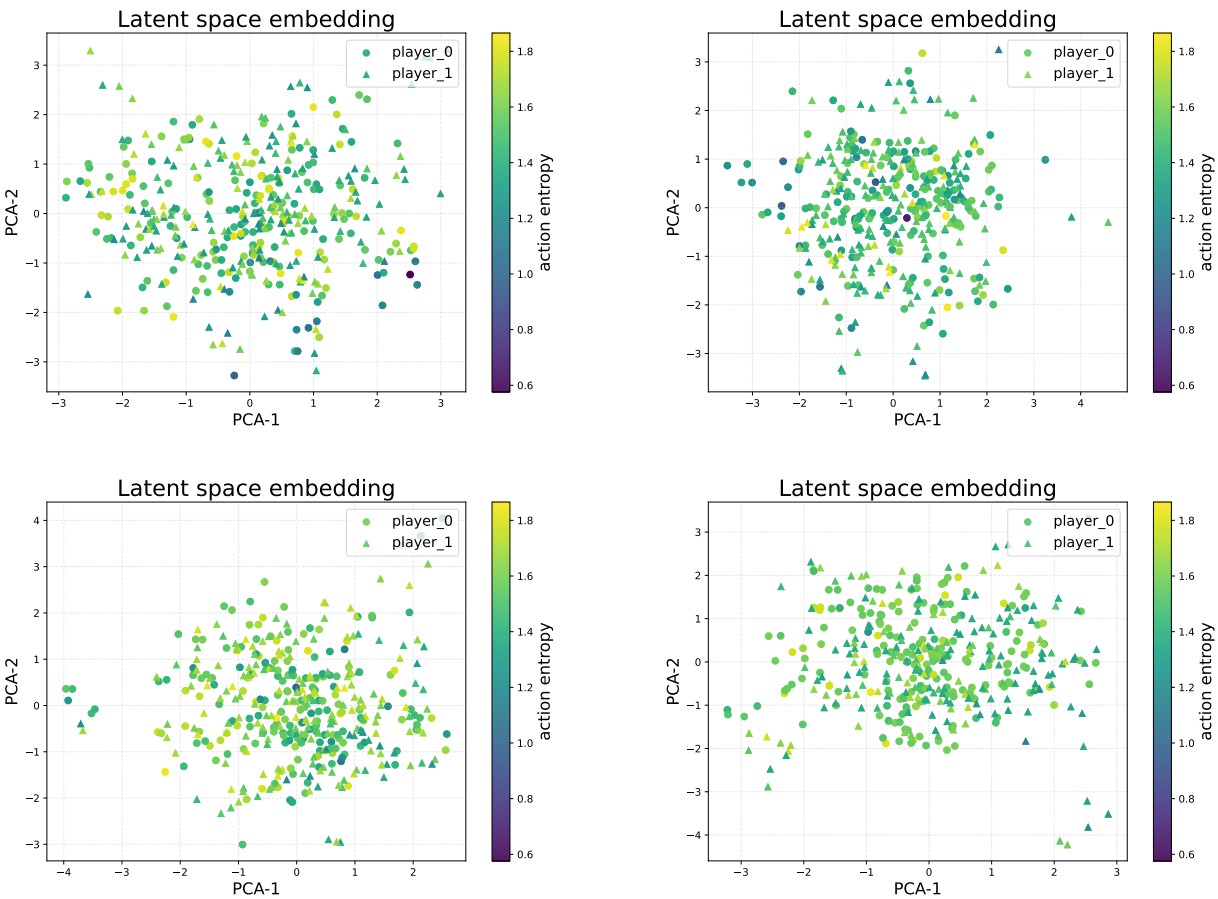

Figure 9: ***Connect–4: Latent-Space Embeddings.*** PCA visualization of learned latent representations across four seeds (0–3). Compared to action space, the latent space remains more dispersed, suggesting preserved representational diversity despite similar cooperative behavior.

## K   Run on Hanabi

**Additional Analysis on Hanabi (PettingZoo).**   Figures 10–12 present supplementary diagnostics for GEMS on the cooperative Hanabi environment implemented using `PettingZoo`, evaluated across four seeds (0–3). Figure 10 reports memory consumption and cumulative wall-clock time over training iterations for GEMS and PSRO. As in the Connect-4 setting, PSRO exhibits steadily increasing memory usage and superlinear growth in cumulative runtime, reflecting the cost of explicit population expansion, policy storage, and repeated equilibrium computation. In contrast, GEMS maintains near-constant memory throughout training and achieves substantially lower cumulative wall-clock time, despite the increased complexity of Hanabi arising from partial observability, information asymmetry, and larger joint action spaces. These results highlight the scalability benefits of GEMS in cooperative multi-agent environments where classical PSRO-style methods incur significant computational overhead.

Figures 11 and 12 further analyze the learned representations through PCA visualizations of action-distribution features and latent embeddings, respectively. In contrast to perfect-information games like Connect-4 where symmetric self-play leads to overlapping embeddings, Figure 11 reveals that the action-distribution embeddings for the two players occupy distinct, complementary regions of the manifold. This separation is geometrically consistent across seeds and is expected: Hanabi is a game of **asymmetric information**, where each player possesses private knowledge (their partner's cards) that strictly differentiates their optimal policy from that of their partner.

Consequently, the plots demonstrate that GEMS has successfully learned a **complementary coordination protocol**. The smooth, symmetric gradients visible in the plots suggest the generator has discovered a continuous "convention" —a shared latent topology that maps distinct agent perspectives to compatible joint actions. This confirms that GEMS avoids the mode collapse often seen in generative cooperative learning, instead maintaining the structured diversity required to solve partially observable settings where no single static policy is optimal.

Similarly, Figure 12 shows that the corresponding latent-space embeddings exhibit a marked contrast: unlike the separated action spaces, the latent codes for both players are densely intermingled. This indicates that the generator utilizes a shared latent protocol (or codebook) to encode the game state, from which distinct, complementary roles emerge in the action space. The preservation of diversity within this shared latent region suggests that GEMS maintains a rich internal representation capable of supporting asymmetric coordination without fracturing into disjoint policies.

Taken together, these results demonstrate that GEMS achieves efficient and stable learning in Hanabi by discovering structured, complementary manifolds. The ability to learn these implicit conventions without explicit communication channels underscores the suitability of amortized, generator-based meta-solvers for complex cooperative multi-agent environments with partial observability.

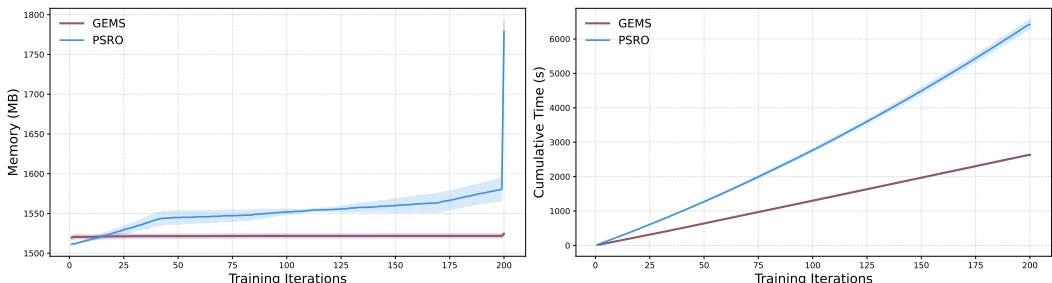

Figure 10: ***Hanabi: Resource Usage.*** Memory consumption (left) and cumulative wall-clock time (right) over training iterations for GEMS and PSRO.

## Hanabi: Action Embeddings (Seeds 0–3)

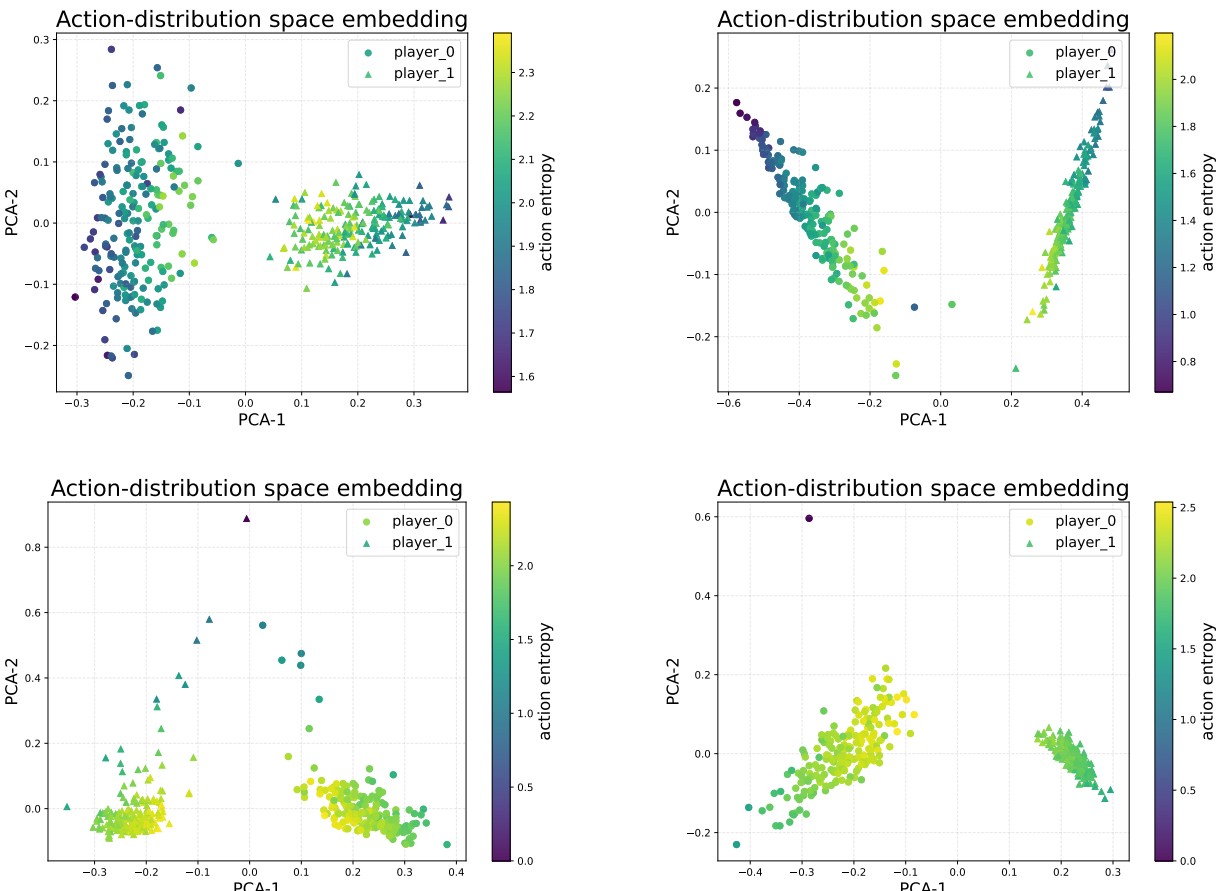

Figure 11: **_Hanabi: Cooperative Coordination Manifold._** PCA visualization of action-distribution features for both players across four random seeds. The embeddings reveal a **coherent joint manifold** with smooth entropy gradients. While the separation between players reflects the **asymmetric information** inherent to Hanabi, the symmetric and complementary geometry of the clusters demonstrates that the generator $\mathcal{G}_\theta$ captures a unified coordination protocol (or "convention") shared by both agents, enabling zero-shot coordination despite distinct individual perspectives.

## Hanabi: Latent Embeddings (Seeds 0–3)

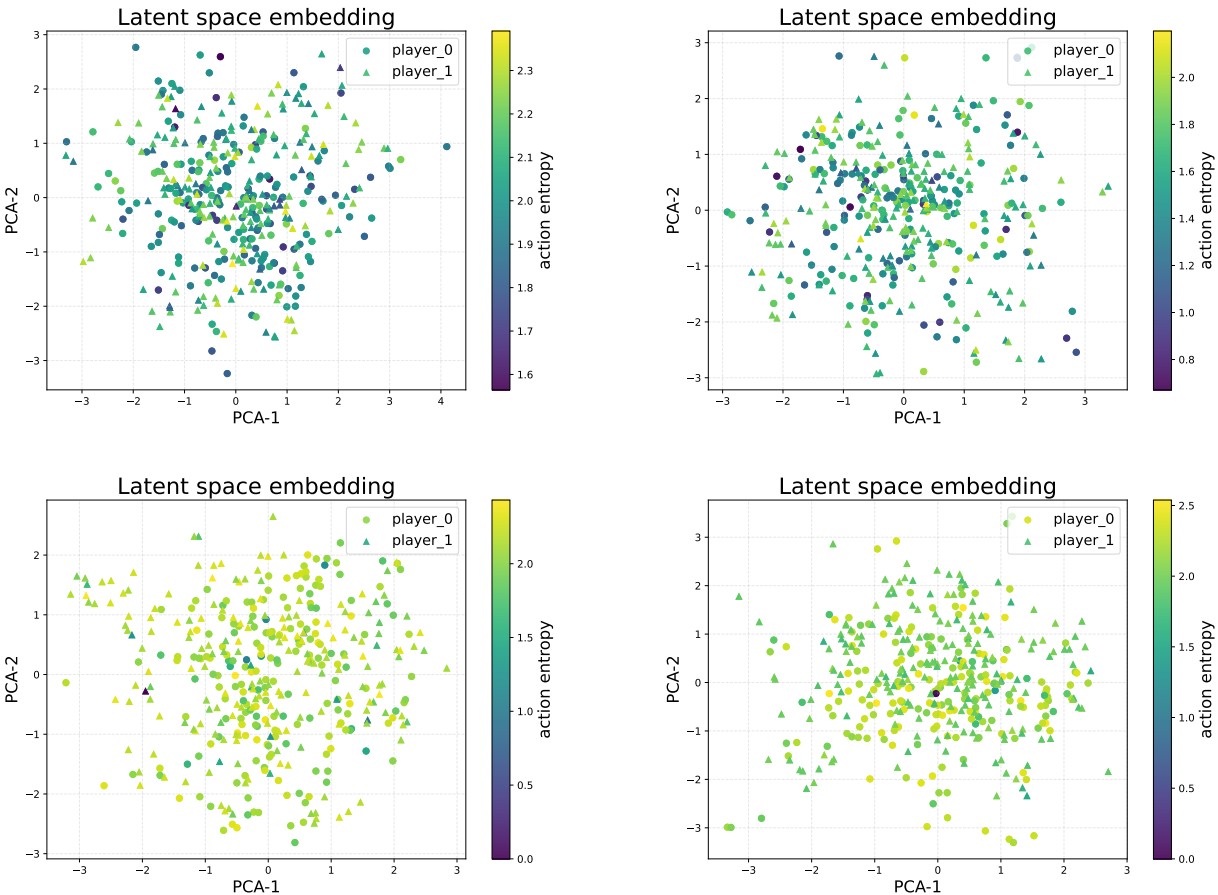

Figure 12: **_Hanabi: Latent-Space Embeddings._** PCA visualization of learned latent representations across four seeds (0–3). Compared to action space, the latent space remains more dispersed, suggesting preserved representational diversity despite similar cooperative behavior.

## L   Coordination on Simple Spread

**Setup.**   Our second experiment is conducted in the `simple_spread` environment, a classic cooperative benchmark from `PettingZoo`. The task requires three agents to collaboratively "cover" three distinct target landmarks in a 2D space. Agents are rewarded for minimizing their distance to any landmark but are penalized for colliding with each other. A successful outcome requires the agents to learn a decentralized "divide and conquer" strategy, assigning themselves to unique targets and navigating to them efficiently. This environment is designed to test emergent cooperation and task division. We again compare GEMS and PSRO on qualitative behavior, mean return, memory, and computation time, averaged over 5 seeds.

**Objective.**   This experiment aims to achieve two goals. First, to confirm the significant scalability benefits of GEMS, as demonstrated in the previous experiment, but now in a purely cooperative setting. Second, to evaluate which algorithm is more effective at discovering the complex, coordinated strategies required for cooperative success, as measured by both quantitative rewards and qualitative analysis of agent behaviors.

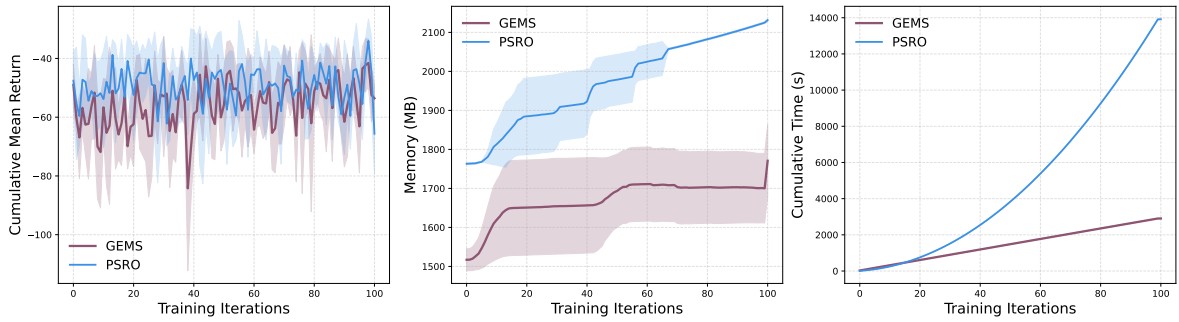

Figure 13: ***Simple Spread***

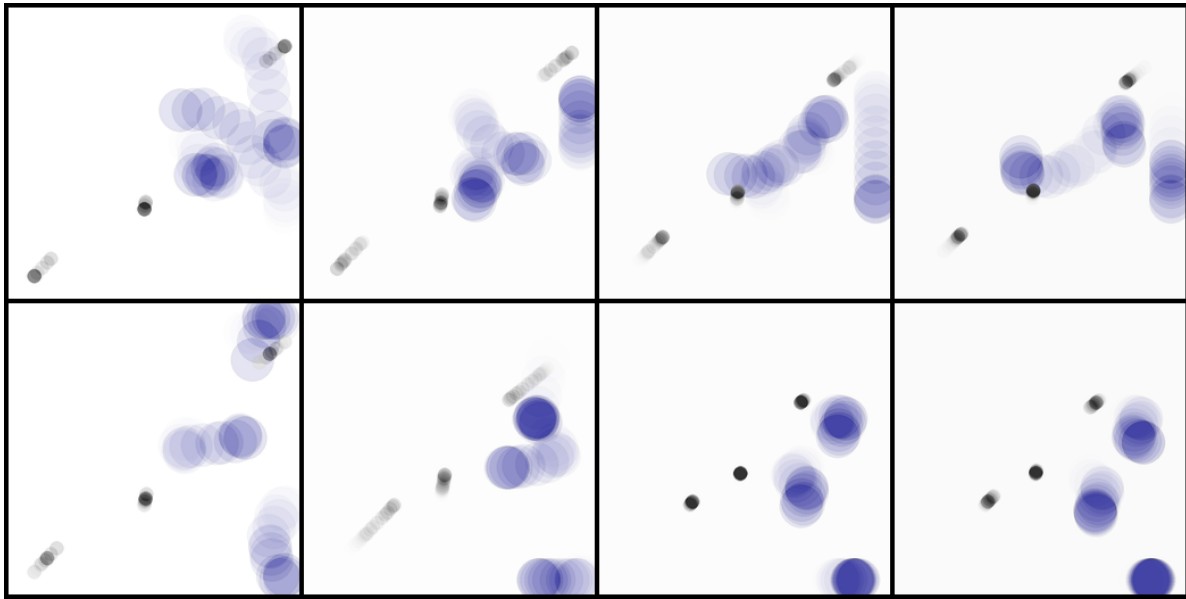

Figure 14: **Simple Spread run on *Seed 0***

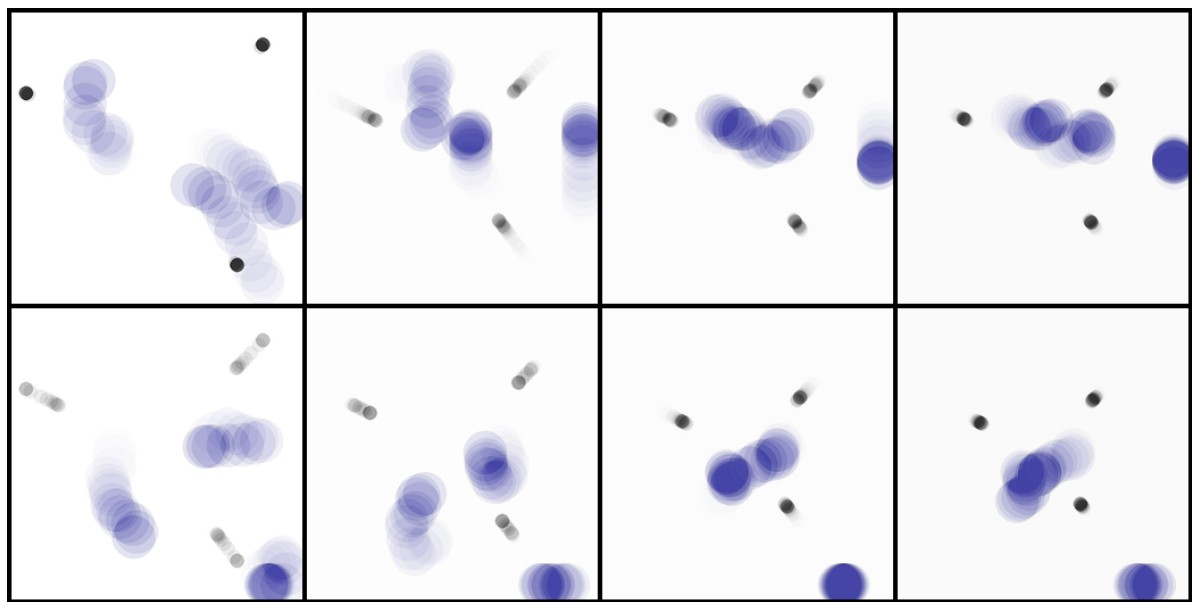

Figure 15: **Simple Spread run on *Seed 1***

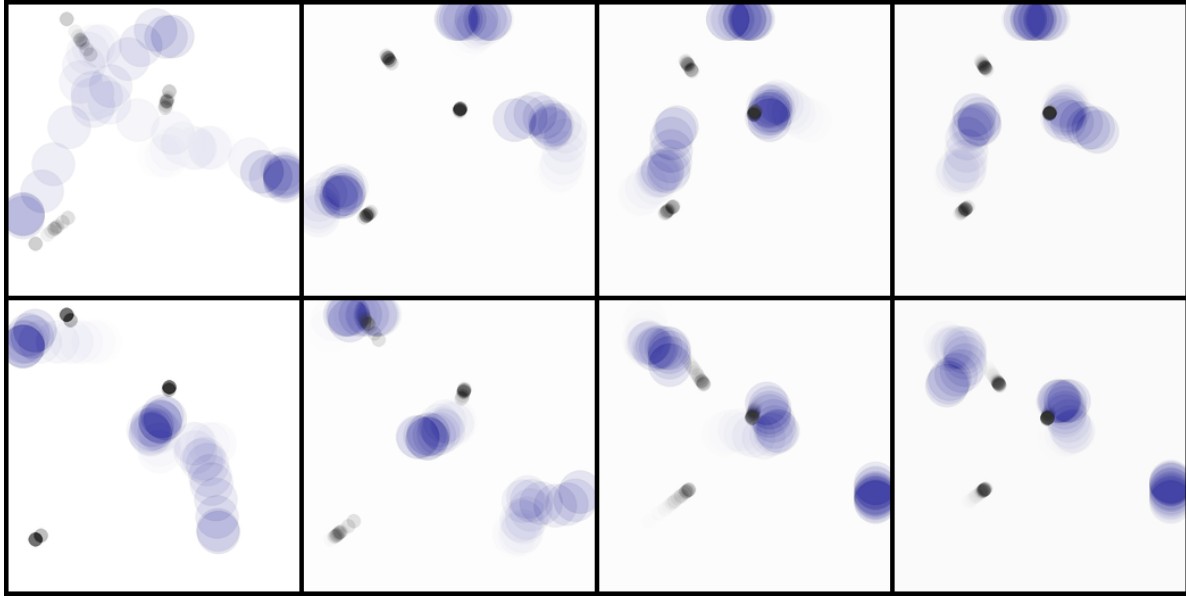

Figure 16: **Simple Spread run on *Seed 2***

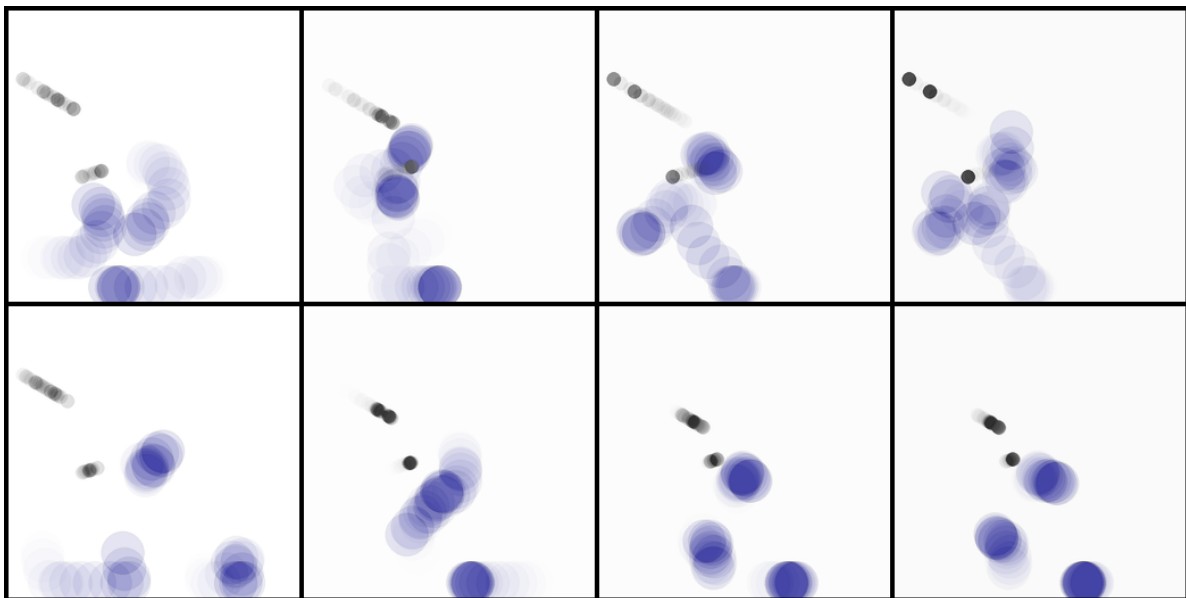

Figure 17: **Simple Spread run on *Seed 3***

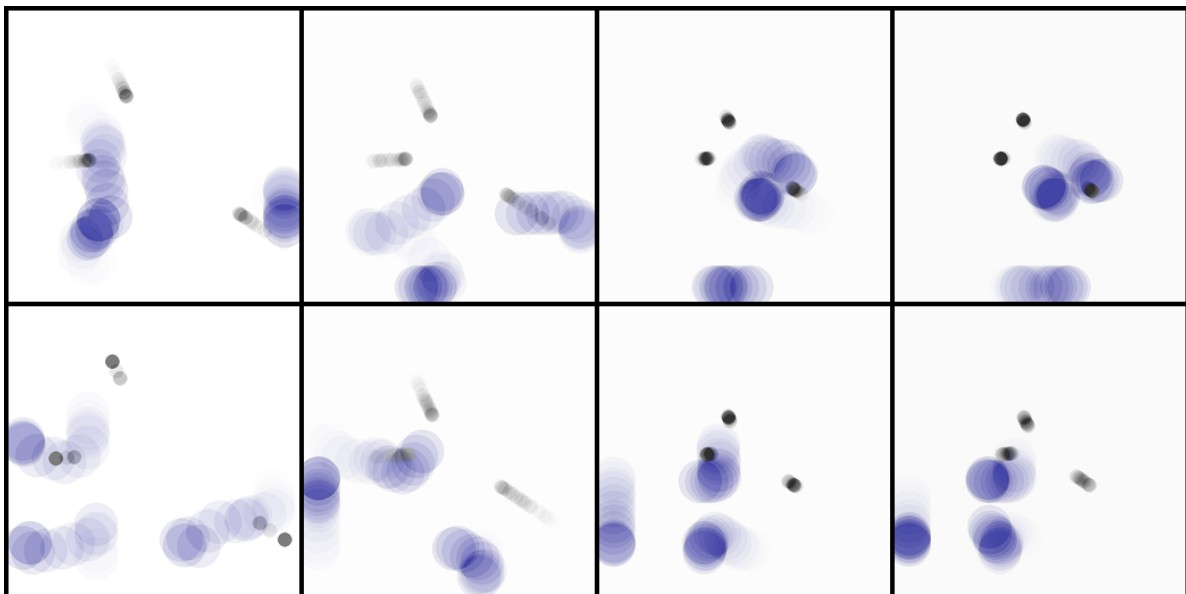

Figure 18: **Simple Spread run on *Seed 4***

**Results.** Across `simple_spread`, GEMS again outperforms PSRO in both effectiveness and efficiency. In the ghost-montage comparisons for seeds 0–4 (Figures 14–18; GEMS on top, PSRO on bottom), **GEMS agents learn an efficient, coordinated strategy**: they quickly partition landmarks and move directly toward them, and when two agents collide they *separate and re-plan* rather than dithering—while still staying close to the black-dot targets (agents rendered in blue). By contrast, **PSRO agents exhibit weaker coordination**: typically one–two agents hover near a target while another drifts away from objectives or circles indecisively, an undesirable failure mode visible in multiple seeds (e.g., Fig. 14, 15). This qualitative advantage aligns with the aggregate plot (Figure 13): GEMS achieves a higher (less negative) mean return with lower variance over time. In terms of scalability, results match prior sections: GEMS is over 6× faster in cumulative time (~2,000s vs. ~13,000s) and maintains a flat memory profile, while PSRO 's resource usage scales poorly.

**Analysis.** These rollouts (Figures 14–18) reinforce our central claim: Gems is better suited for cooperative multi-agent tasks that require implicit role assignment. The **EB-UCB oracle's exploration over a diverse latent space** helps discover complementary roles (distinct landmark assignments) and preserves them under contact—agents momentarily repel when they touch, then settle back near their respective black-dot objectives. Simpler procedures in Psro more often collapse to near-symmetric yet suboptimal behaviors (e.g., multiple agents chasing the same landmark while another disengages). Coupled with Gems' **surrogate-free design**—which yields lower wall-clock time and stable memory—these qualitative and quantitative results position Gems as a robust, scalable framework for cooperative MARL.

. . . . . . . . . . . . . . . . . . . . . . . . . . . . . . . . . . . . . . . . . . . . . . . . . . . . . . . . . . . . . . . . . . . . . . . . . . . . . . . . . . . . . . . . . . . . . . . . . . . . . . .

## M   Coordination on Simple Tag

**Simple Tag (Appendix).**   Across seeds 0–4 (ghost montages: GEMS on top, PSRO on bottom), **GEMS** reliably produces *coordinated pursuit*: the red taggers self-organize into a roughly triangular encirclement that narrows angles on the green runner, adjusting spacing to cut off escape lanes; the green agent, in turn, exhibits purposeful evasion (arcing and zig-zag trajectories), and *captures do occur* when the enclosure closes. Under **PSRO**, by contrast, the taggers seldom sustain a stable formation: transient "triangle-ish" shapes appear but collapse before containment, leaving wide gaps through which the green agent *easily escapes*. While the green agent under PSRO shows less consistently trained avoidance than under GEMS, the taggers' lack of persistent coordination dominates the outcome, captures are rare and evasion is commonplace. Overall, these qualitative rollouts corroborate the main-paper claim: GEMS induces higher-level cooperative tactics (persistent enclosure and angle closing) that the PSRO baseline fails to discover or maintain.

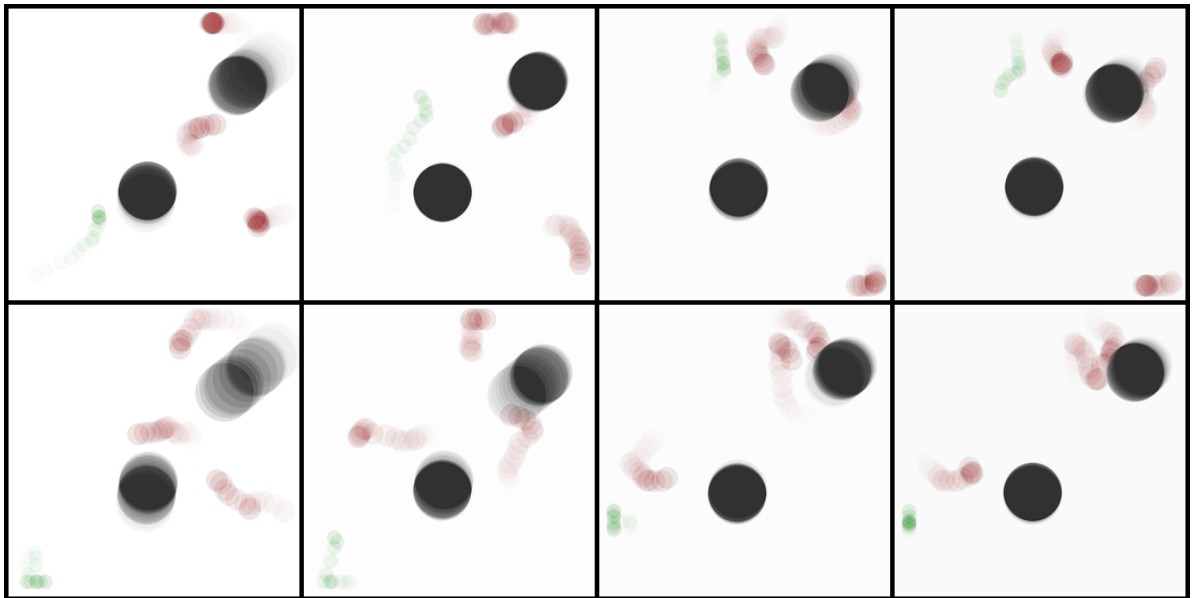

Figure 19: **Simple Tag run on *Seed 0***

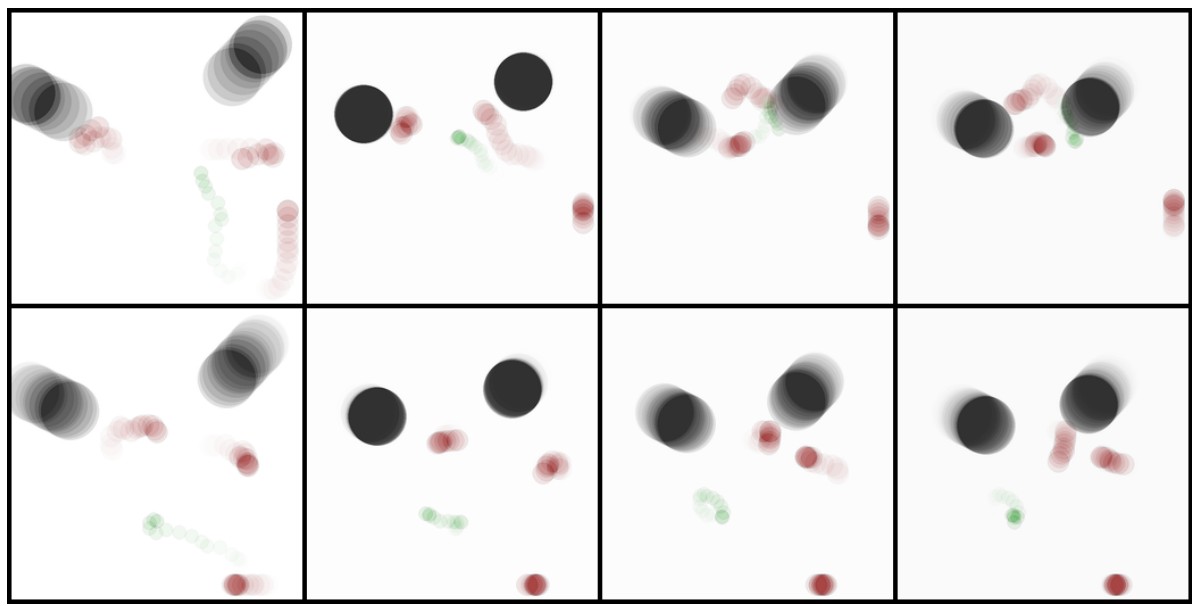

Figure 20: **Simple Tag run on *Seed 1***

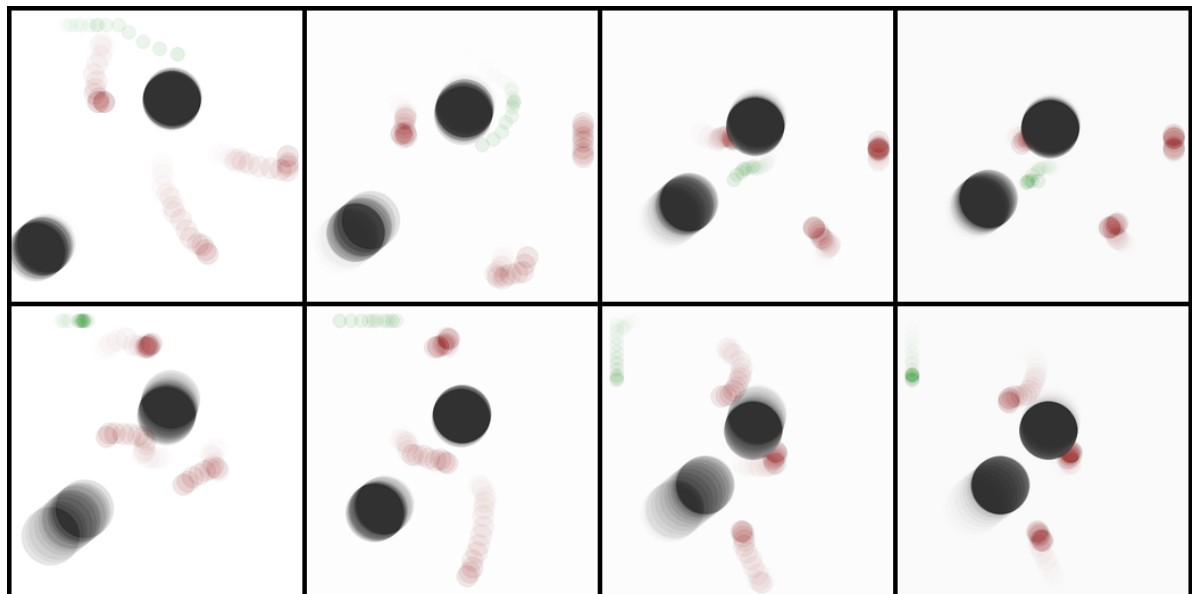

Figure 21: **Simple Tag run on *Seed 2***

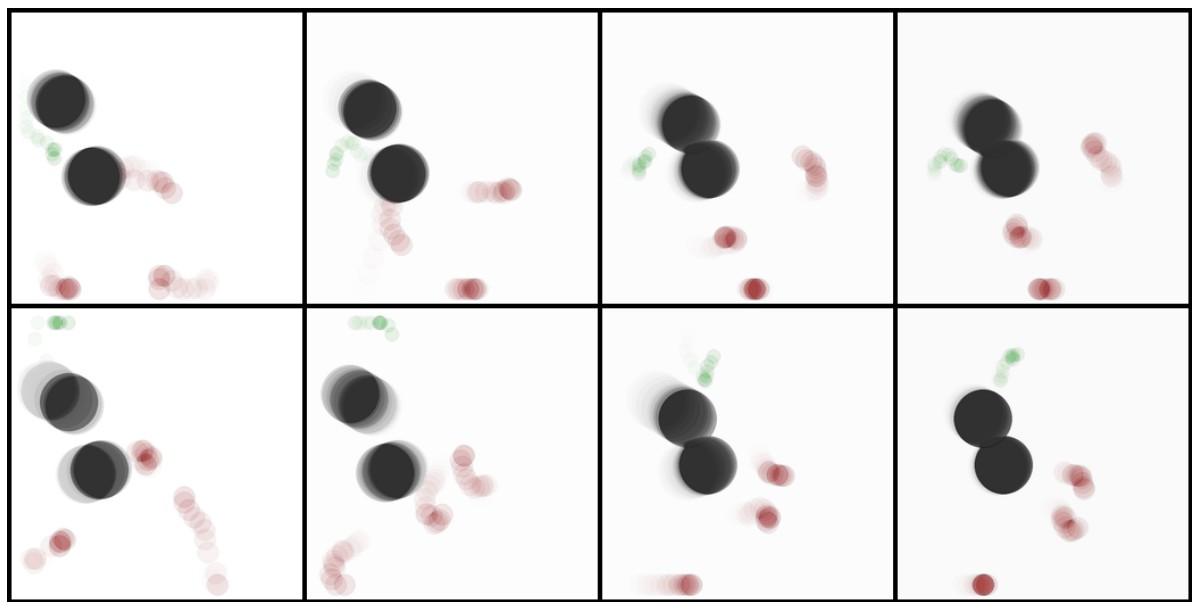

Figure 22: **Simple Tag run on *Seed 3***

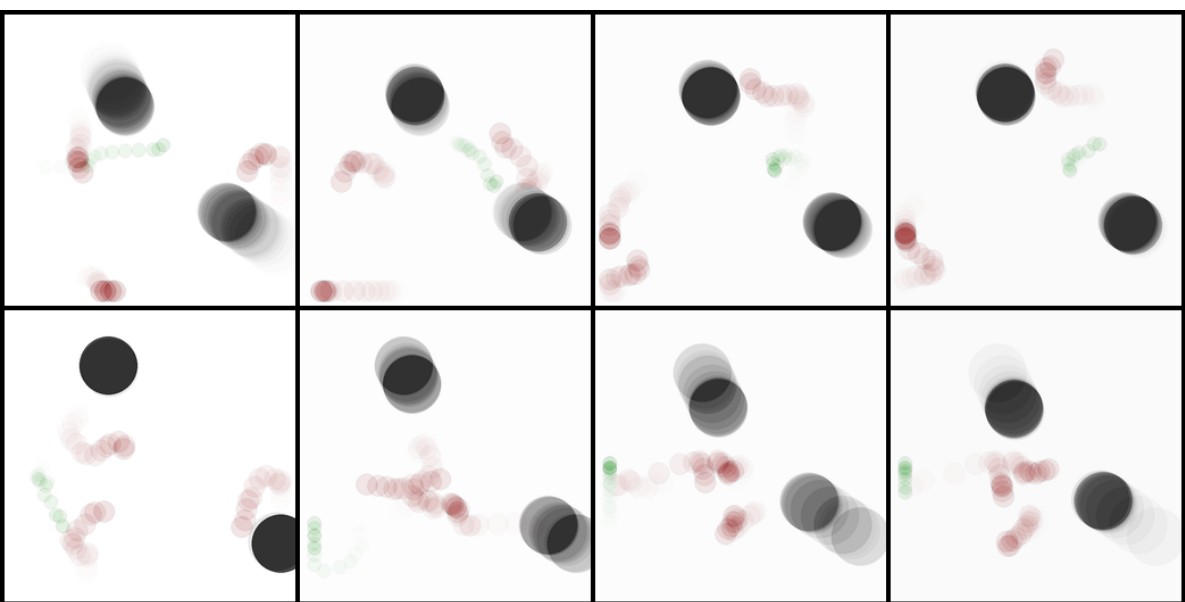

Figure 23: **Simple Tag run on *Seed 4***

. . . . . . . . . . . . . . . . . . . . . . . . . . . . . . . . . . . . . . . . . . . . . . . . . . . . . . . . . . . . . . . . . . . . . . . . . . . . . . . . . . . . . . . . . . . . . . . . . . . . . . .

# N    Run on Chess

We evaluate GEMS on the `chess_v6` environment from the `PettingZoo` library, which provides a full two-player Chess setup following the FIDE ruleset, including castling, en passant, pawn promotion, and draw conditions. Each agent alternately controls either the white or black pieces and receives observations corresponding to the complete board state, enabling direct competition through turn-based play.

Across the early stages of training, most games between GEMS agents resulted in draws. However, as iterations progressed, the system began exhibiting increasingly decisive outcomes—alternating between strong white and black victories. This trend can be observed in Table 4, where the balance of wins and losses evolves from neutrality to a dynamic dominance-shift pattern. The accompanying performance metrics in Figure 24 further illustrate this evolution: the per-iteration time steadily decreased from approximately 25 s to 9 s as the training stabilized, while memory usage remained nearly constant throughout 1000 iterations. The cumulative time curve also followed an almost linear trend, confirming consistent computational scaling.

Figures 25–33 visualize selected games during training. Initially, black exhibits early dominance, leveraging material advantages through aggressive captures. Yet, as learning progresses, white develops counter-strategies involving positional control and endgame foresight. A particularly striking sequence shows white using a rook-based checkmate against black, who had earlier promoted a pawn to a queen—highlighting GEMS's robustness in resolving complex, materially imbalanced positions to decisive terminal states without succumbing to policy collapse.

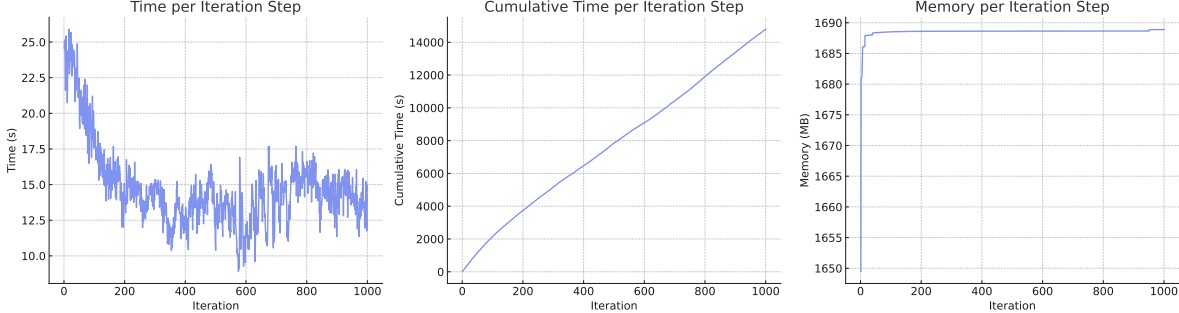

Figure 24: System metrics across training iterations. Memory usage remains stable throughout, per-step computation time gradually decreases as the model stabilizes, and cumulative time increases linearly with iteration count—indicating consistent computational efficiency.

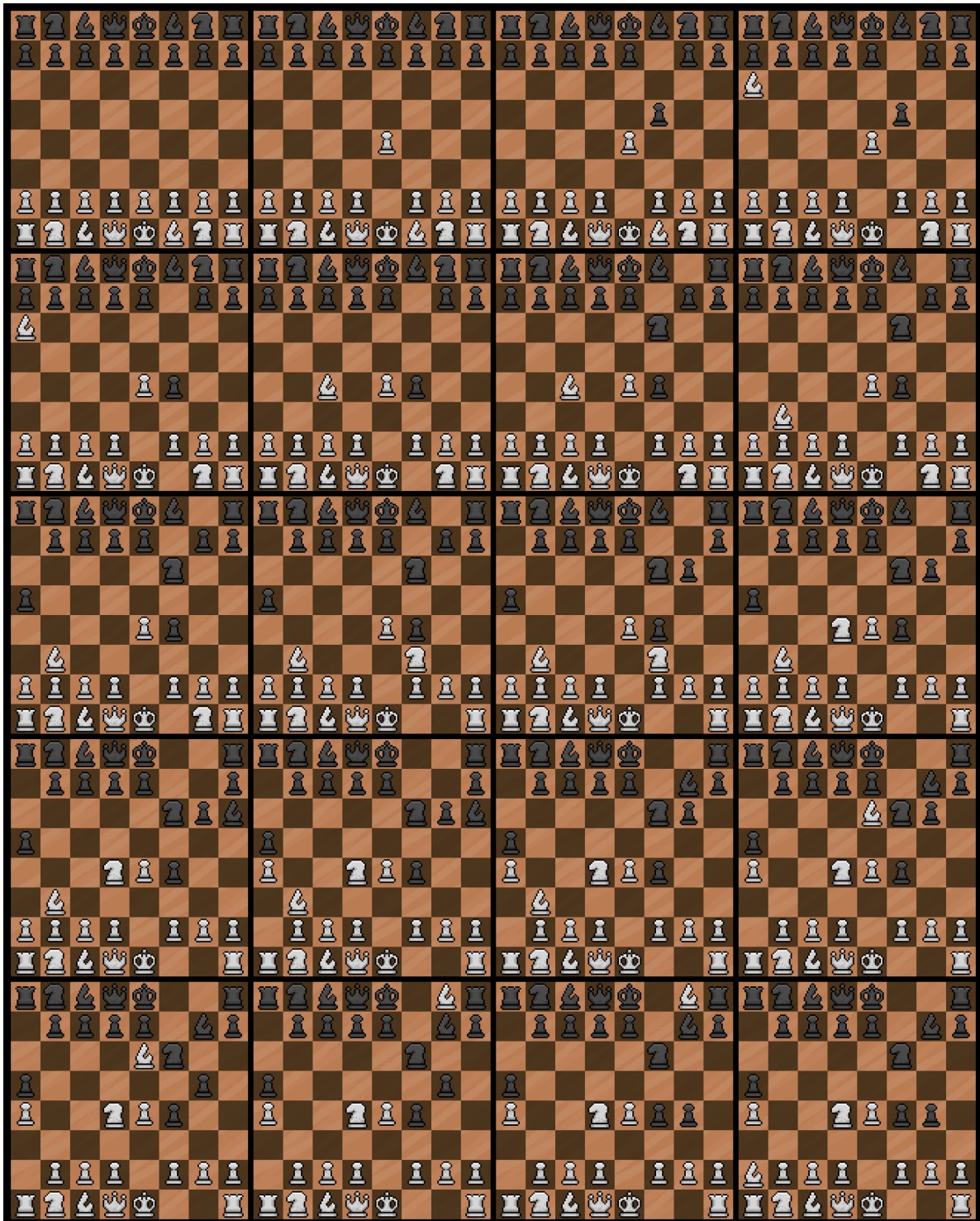

Figure 25: **Chess run for 1000 iterations.** *Part 1/9*
Early-game states at 1000 iterations. GEMS agents demonstrate a preference for central occupation (controlling d4/e4 squares) and developing minor pieces to active squares. The divergence in pawn structures indicates that the agents are exploring asymmetric board configurations rather than collapsing into identical opening lines.

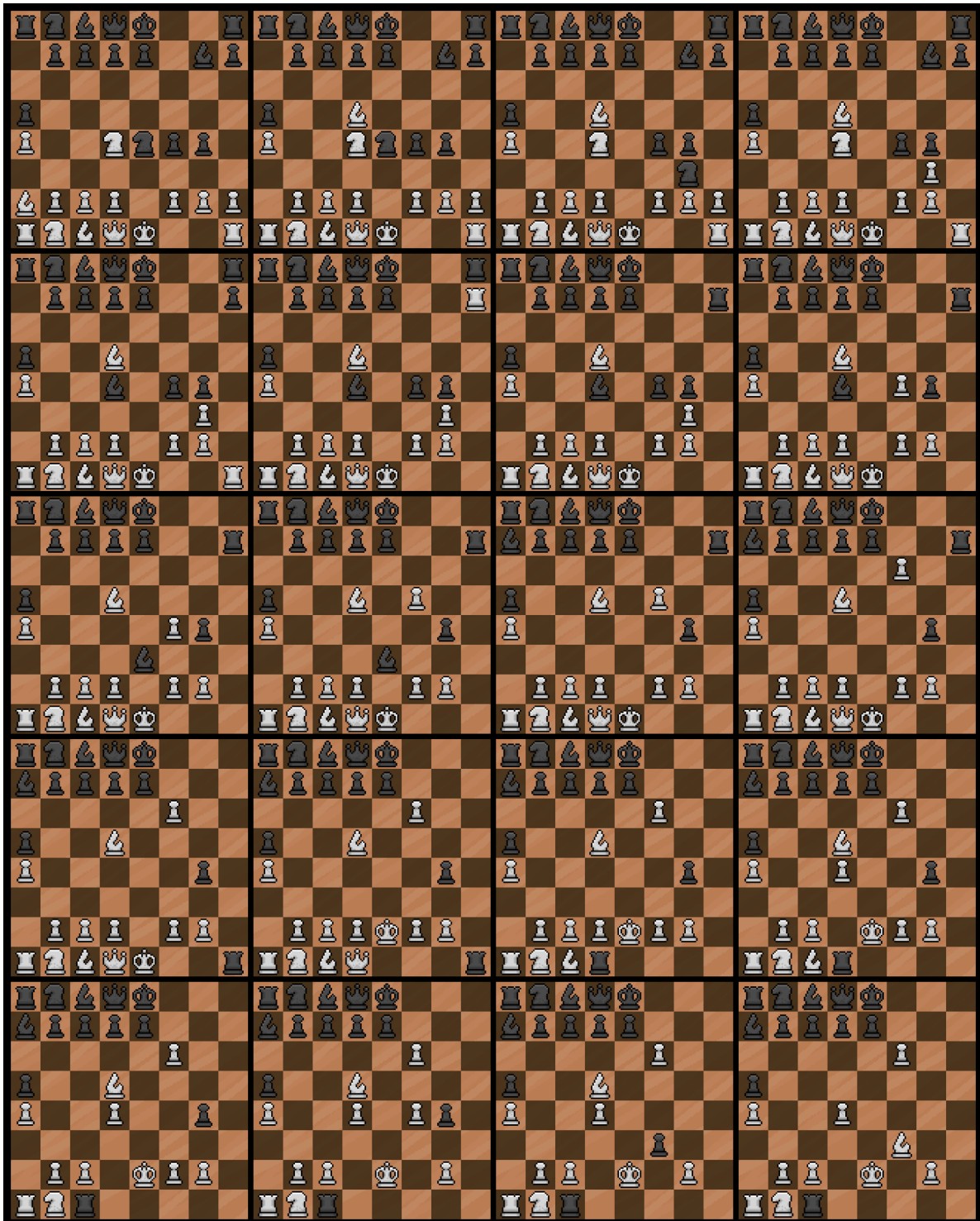

Figure 26: **Chess run for 1000 iterations.** *Part 2/9*
Transition to middlegame. The agents exhibit coordinated piece placement, with bishops and knights occupying supported diagonals and outposts. The board states reflect a learned policy of structural preservation, avoiding premature breakage of pawn chains while maintaining piece connectivity.

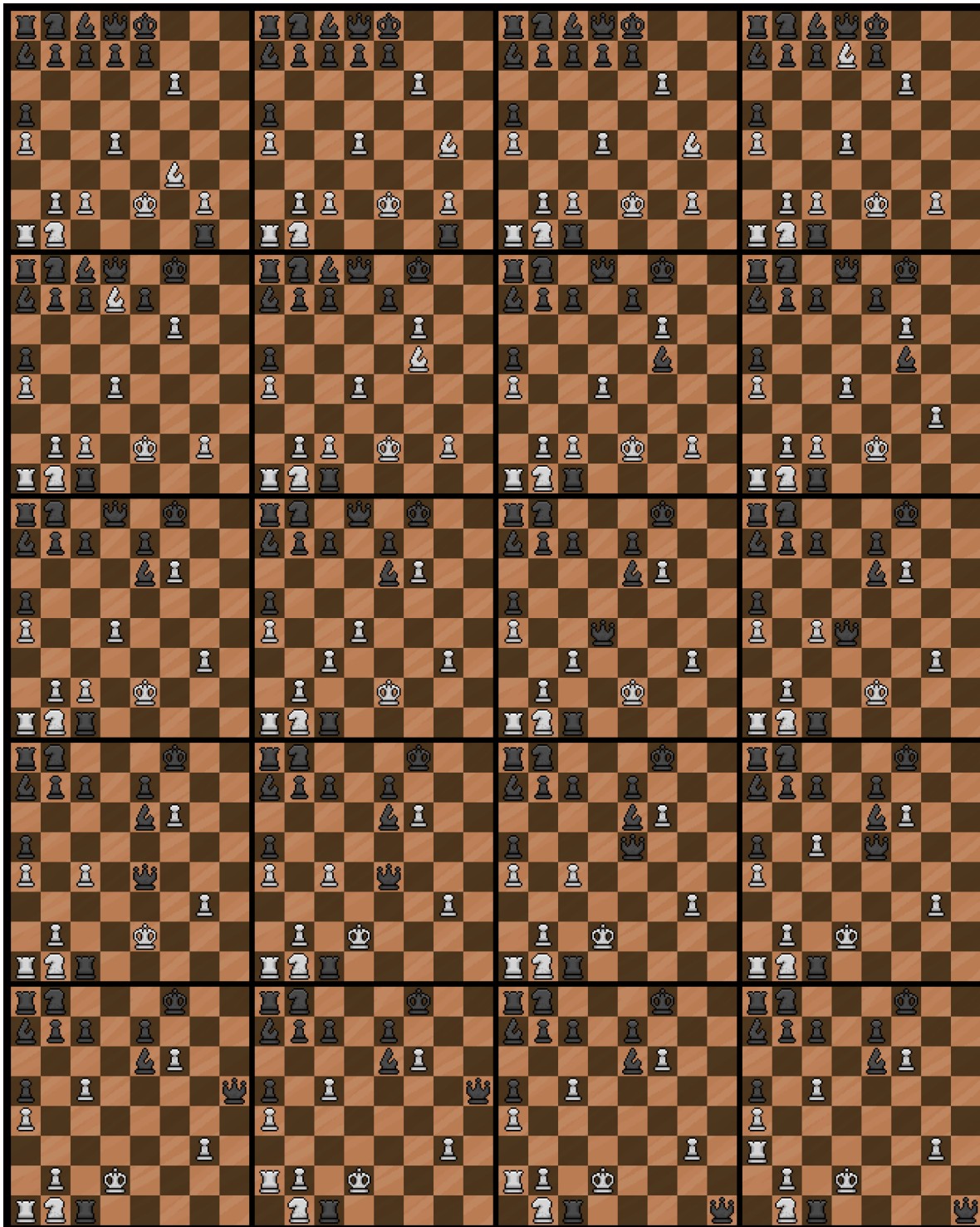

Figure 27: **Chess run for 1000 iterations.** *Part 3/9*
Complex middlegame configurations. Play involves tension around the center and semi-open files. The agents demonstrate an ability to navigate trade-offs between material retention and spatial activity, resulting in non-trivial board states that require calculation of tactical exchanges.

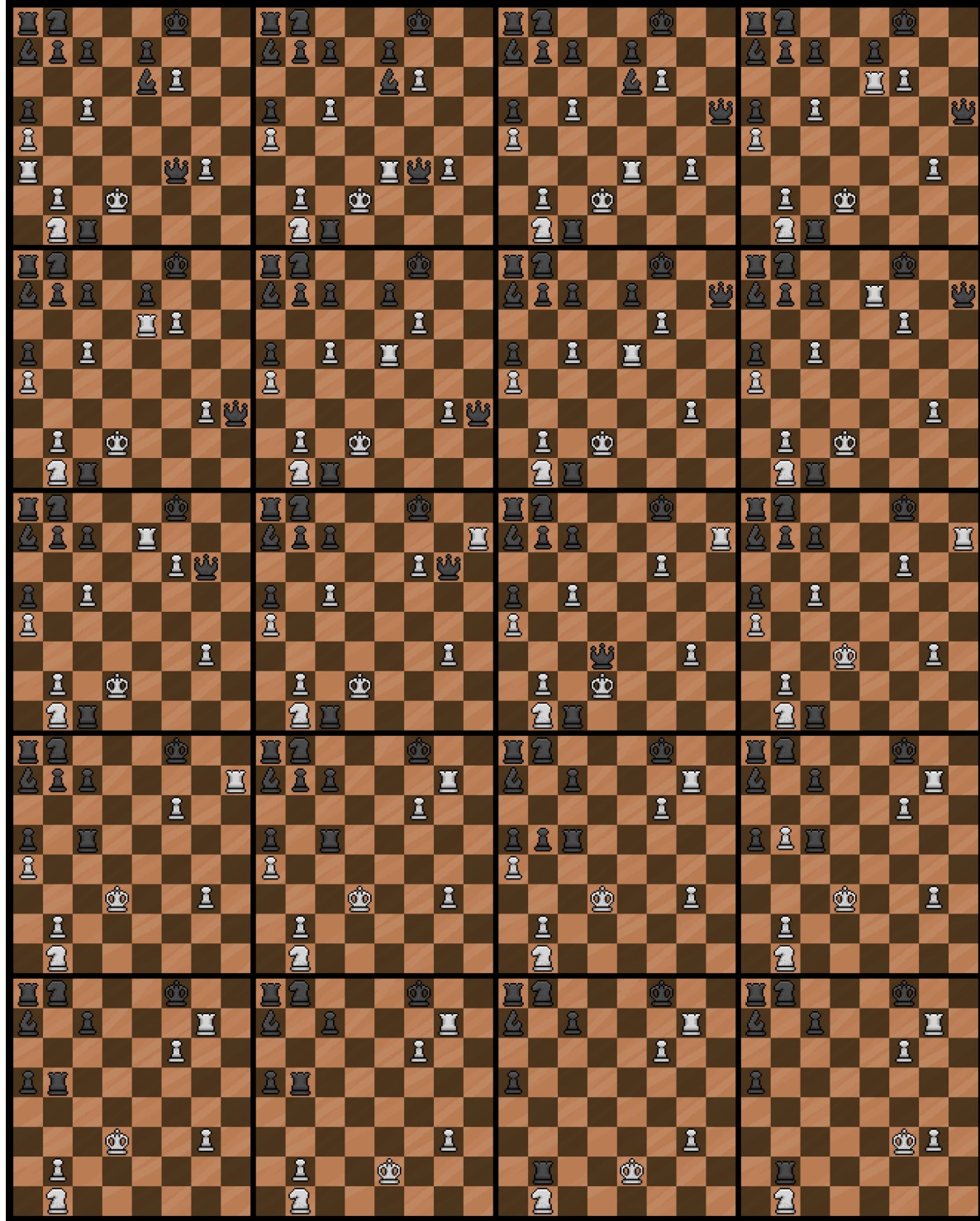

Figure 28: **Chess run for 1000 iterations.** *Part 4/9*
Late middlegame progression. Major pieces (Rooks/Queens) increasingly control key files. The policy appears to prioritize King safety while initiating simplification sequences, transitioning the game from complex tactical middlegames toward solvable endgame states.

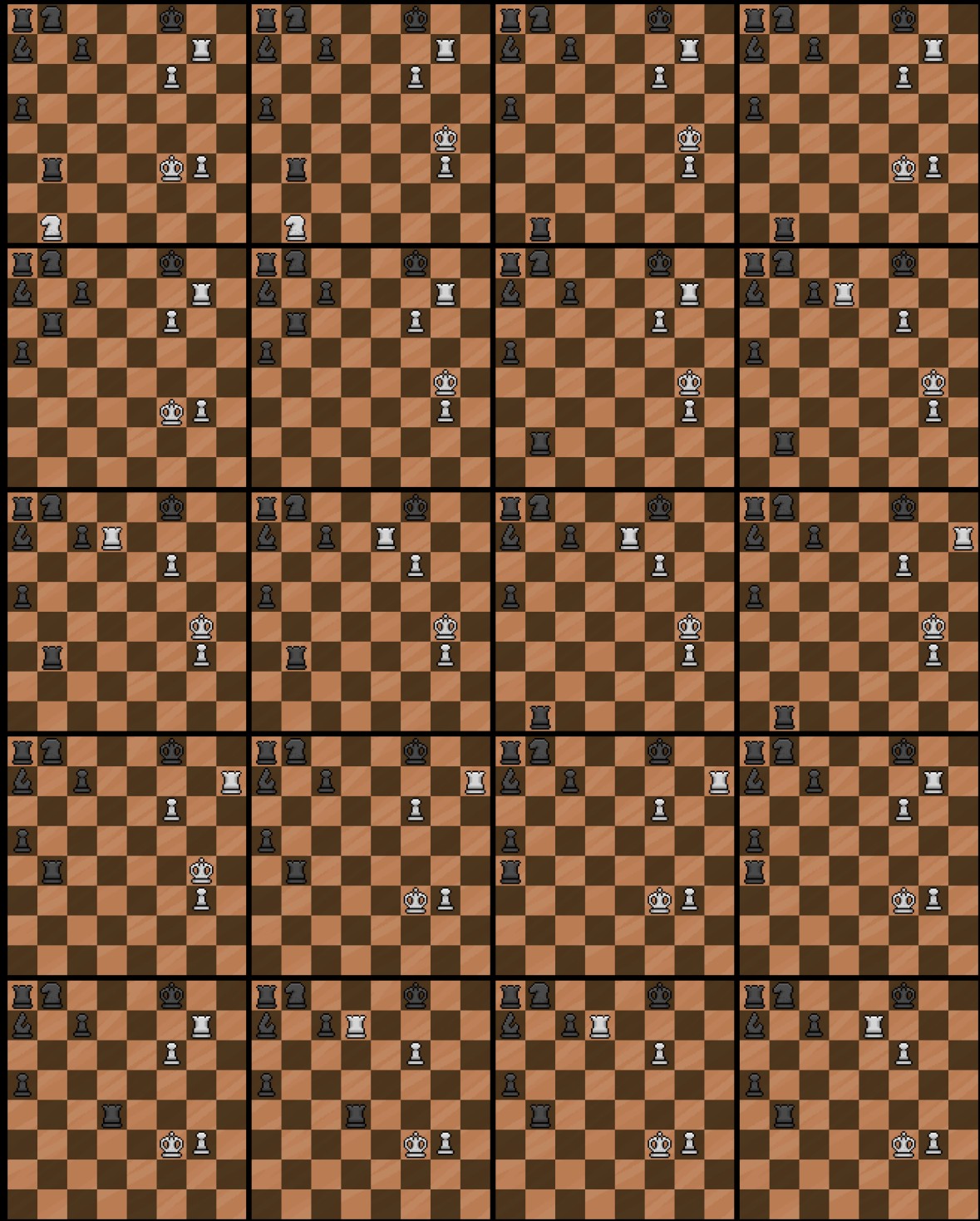

Figure 29: **Chess run for 1000 iterations.** *Part 5/9*
Endgame transition phases. The board states show reduced material with increased King activity. The agents demonstrate the capability to coordinate remaining pieces to restrict opponent mobility, suggesting the emergence of basic endgame principles without explicit tablebase supervision.

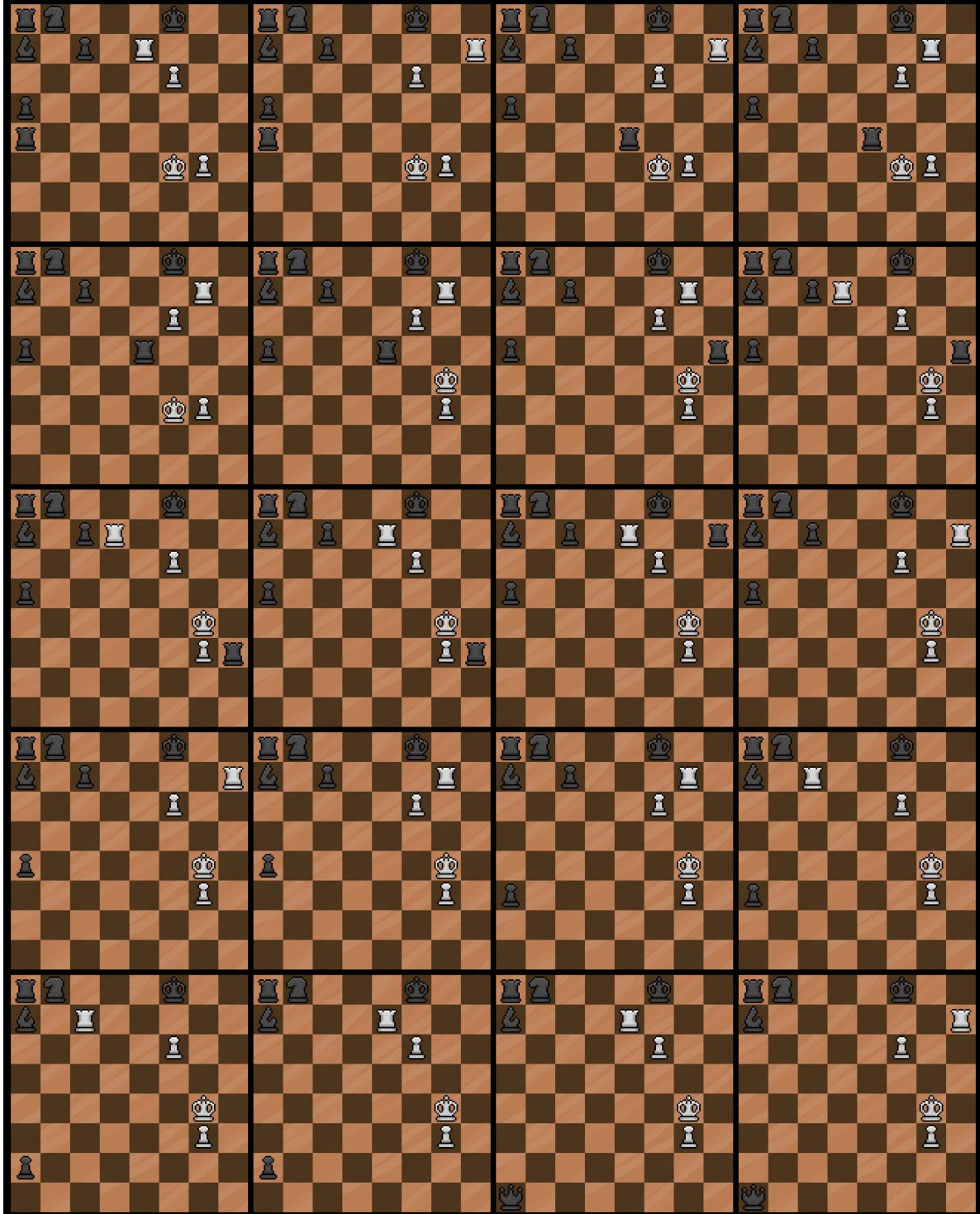

Figure 30: **Chess run for 1000 iterations.** *Part 6/9*
Sparse board configurations. GEMS converges on simplified states characterized by Rook and King maneuvering. The agents avoid random moves that surrender material, maintaining equilibrium in drawn positions or pressing advantages in uneven splits.

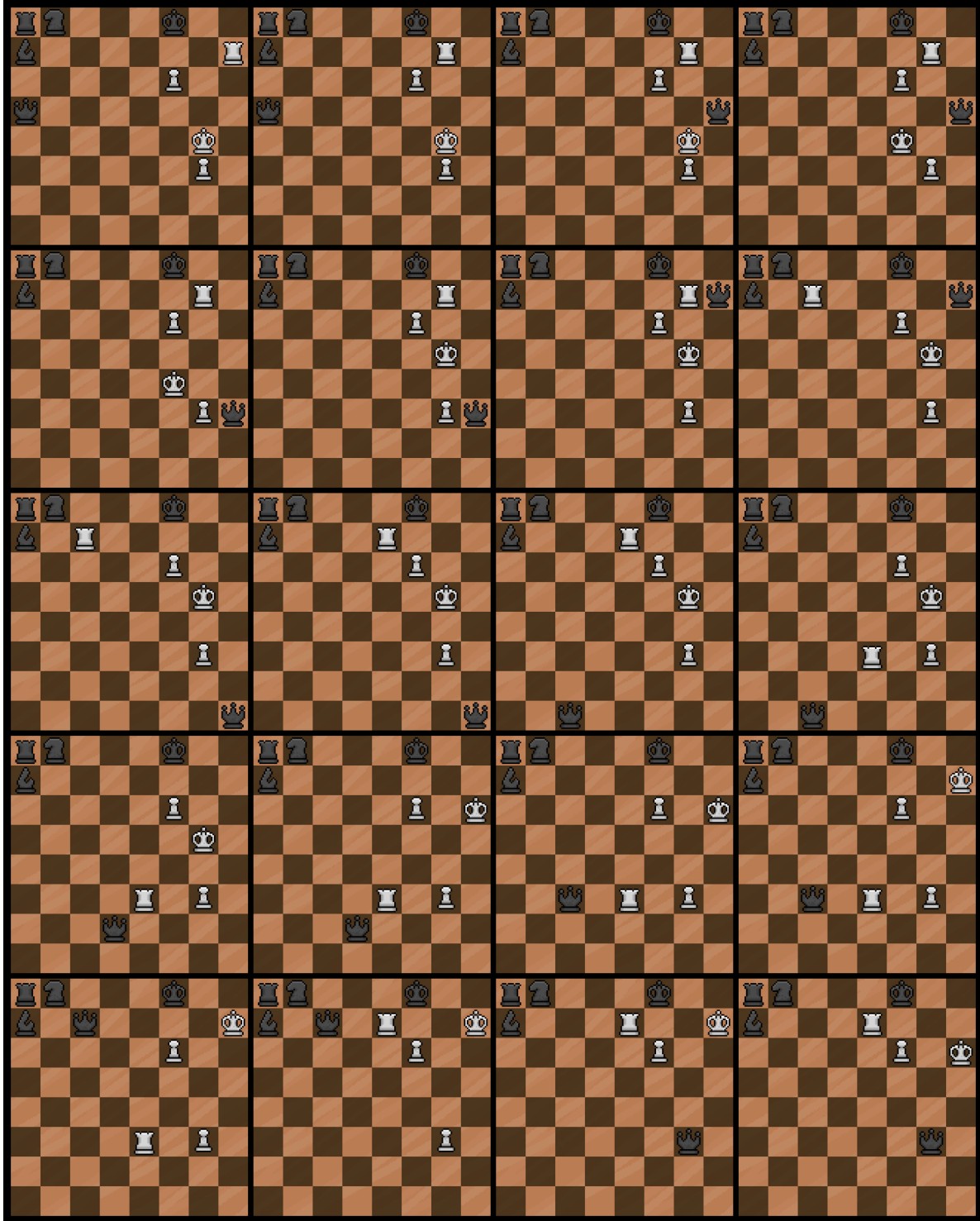

Figure 31: **Chess run for 1000 iterations.** *Part 7/9*
Formation of mating nets. The dominant agent utilizes coordinated checks to drive the opponent's King to the edge of the board. These sequences indicate that the policy successfully propagates value from terminal states back to these pre-terminal configurations.

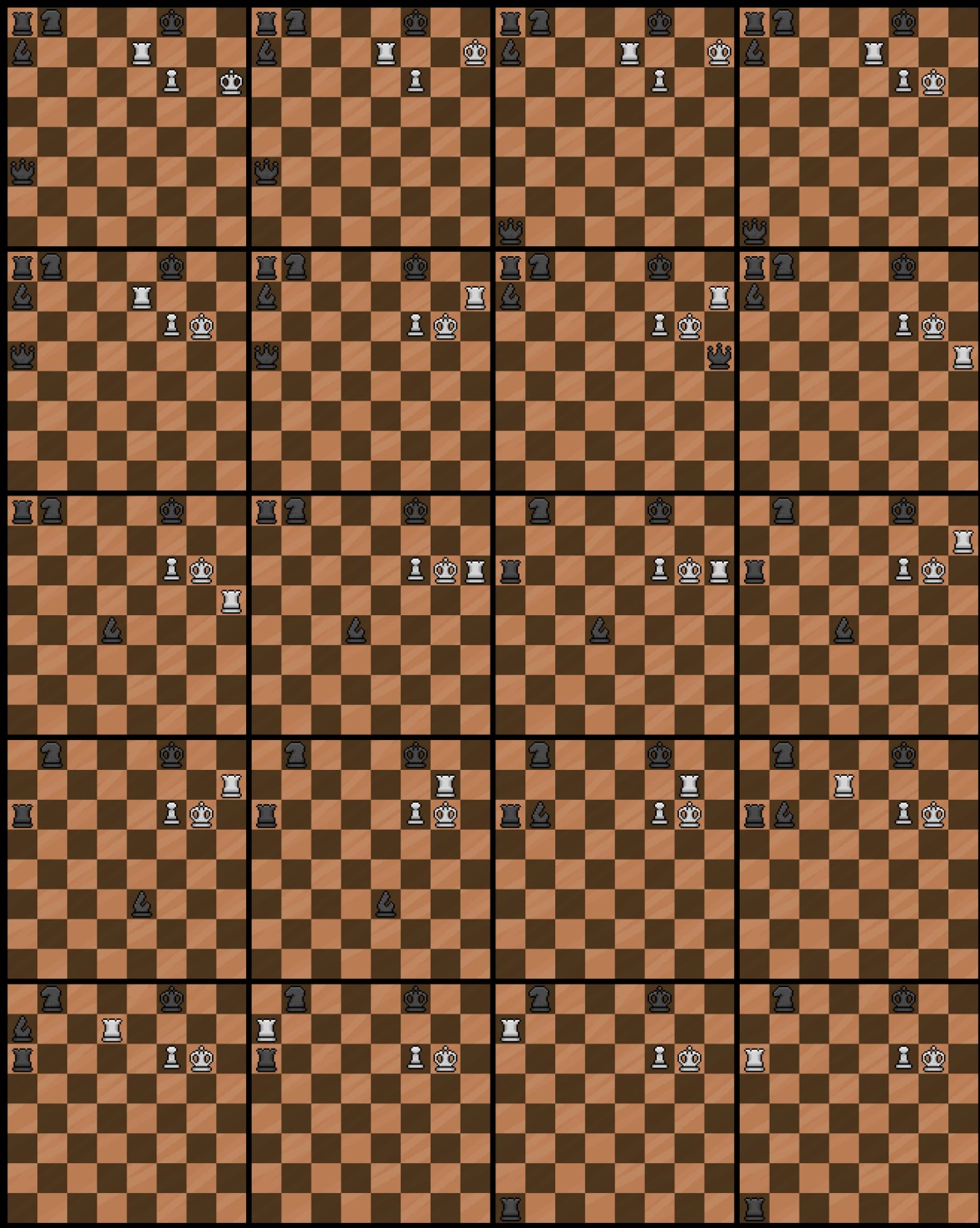

Figure 32: **Chess run for 1000 iterations.** *Part 8/9*
Pre-terminal tactical motifs. The agents execute forcing lines involving heavy pieces to convert spatial advantages into checkmate opportunities. The consistency of these motifs across different seeds suggests stable convergence toward valid winning strategies.

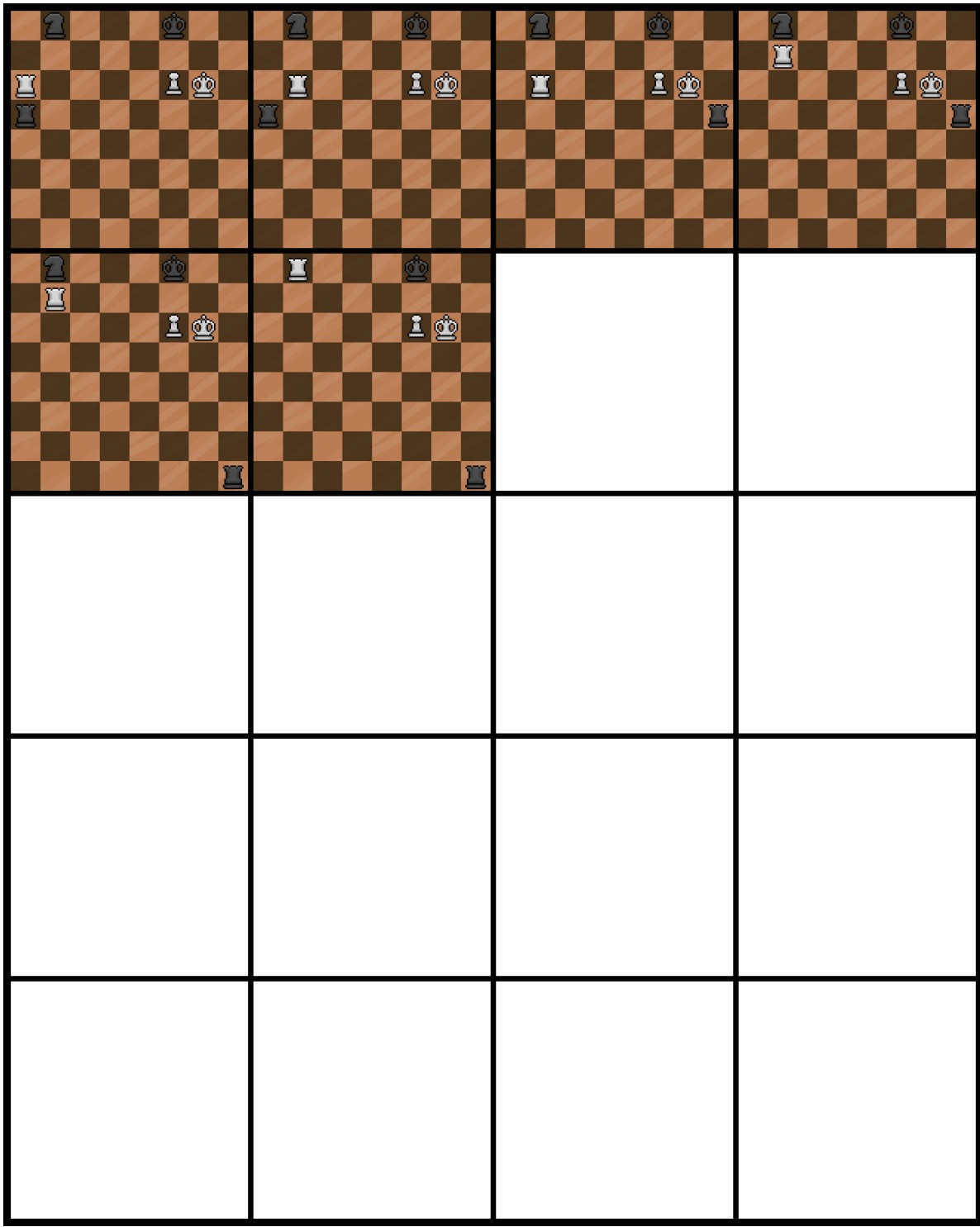

Figure 33: **Chess run for 1000 iterations.** *Part 9/9*
Terminal states. GEMS reliably reaches definitive checkmate configurations (e.g., King + Rook mates). This confirms that the training framework avoids the failure mode of indefinite cycles or stalemates, successfully resolving the game loop.

Table 4: Training progression of GEMS over 1000 iterations in Chess. Legend: "–" denotes a draw, "**W**" denotes a win, and "L" denotes a loss.

| Iter | White | Black | Iter | White | Black | Iter | White | Black | Iter | White | Black | Iter | White | Black |
|---|---|---|---|---|---|---|---|---|---|---|---|---|---|---|
| 1 | - | - | 2 | - | - | 3 | - | - | 4 | - | - | 5 | - | - |
| 6 | - | - | 7 | - | - | 8 | - | - | 9 | - | - | 10 | - | - |
| 11 | - | - | 12 | - | - | 13 | - | - | 14 | - | - | 15 | - | - |
| 16 | - | - | 17 | - | - | 18 | - | - | 19 | - | - | 20 | - | - |
| 21 | - | - | 22 | - | - | 23 | - | - | 24 | - | - | 25 | **W** | L |
| 26 | - | - | 27 | - | - | 28 | L | **W** | 29 | - | - | 30 | L | **W** |
| 31 | - | - | 32 | - | - | 33 | - | - | 34 | - | - | 35 | - | - |
| 36 | - | - | 37 | - | - | 38 | - | - | 39 | - | - | 40 | - | - |
| 41 | - | - | 42 | - | - | 43 | - | - | 44 | - | - | 45 | - | - |
| 46 | - | - | 47 | - | - | 48 | **W** | L | 49 | - | - | 50 | L | **W** |
| 51 | - | - | 52 | - | - | 53 | - | - | 54 | - | - | 55 | - | - |
| 56 | - | - | 57 | - | - | 58 | - | - | 59 | - | - | 60 | - | - |
| 61 | - | - | 62 | **W** | L | 63 | - | - | 64 | - | - | 65 | - | - |
| 66 | - | - | 67 | - | - | 68 | - | - | 69 | - | - | 70 | - | - |
| 71 | - | - | 72 | - | - | 73 | - | - | 74 | - | - | 75 | - | - |
| 76 | - | - | 77 | - | - | 78 | - | - | 79 | - | - | 80 | **W** | L |
| 81 | - | - | 82 | - | - | 83 | - | - | 84 | - | - | 85 | - | - |
| 86 | - | - | 87 | - | - | 88 | L | **W** | 89 | - | - | 90 | L | **W** |
| 91 | - | - | 92 | - | - | 93 | - | - | 94 | - | - | 95 | L | **W** |
| 96 | - | - | 97 | - | - | 98 | - | - | 99 | - | - | 100 | - | - |
| 101 | - | - | 102 | - | - | 103 | - | - | 104 | - | - | 105 | **W** | L |
| 106 | L | **W** | 107 | L | **W** | 108 | - | - | 109 | L | **W** | 110 | **W** | L |
| 111 | - | - | 112 | - | - | 113 | L | **W** | 114 | - | - | 115 | L | **W** |
| 116 | - | - | 117 | - | - | 118 | L | **W** | 119 | - | - | 120 | - | - |
| 121 | - | - | 122 | - | - | 123 | - | - | 124 | L | **W** | 125 | - | - |
| 126 | - | - | 127 | L | **W** | 128 | - | - | 129 | - | - | 130 | - | - |
| 131 | L | **W** | 132 | - | - | 133 | - | - | 134 | - | - | 135 | **W** | L |
| 136 | - | - | 137 | L | **W** | 138 | - | - | 139 | L | **W** | 140 | - | - |
| 141 | - | - | 142 | **W** | L | 143 | - | - | 144 | - | - | 145 | **W** | L |
| 146 | - | - | 147 | - | - | 148 | - | - | 149 | - | - | 150 | - | - |
| 151 | **W** | L | 152 | **W** | L | 153 | - | - | 154 | - | - | 155 | - | - |
| 156 | **W** | L | 157 | L | **W** | 158 | L | **W** | 159 | - | - | 160 | - | - |
| 161 | - | - | 162 | **W** | L | 163 | L | **W** | 164 | - | - | 165 | - | - |
| 166 | - | - | 167 | - | - | 168 | - | - | 169 | - | - | 170 | - | - |
| 171 | **W** | L | 172 | - | - | 173 | - | - | 174 | - | - | 175 | - | - |
| 176 | - | - | 177 | **W** | L | 178 | **W** | L | 179 | - | - | 180 | - | - |
| 181 | - | - | 182 | **W** | L | 183 | - | - | 184 | - | - | 185 | - | - |
| 186 | - | - | 187 | - | - | 188 | **W** | L | 189 | - | - | 190 | **W** | L |
| 191 | **W** | L | 192 | **W** | L | 193 | - | - | 194 | - | - | 195 | **W** | L |
| 196 | **W** | L | 197 | - | - | 198 | - | - | 199 | - | - | 200 | **W** | L |
| 201 | - | - | 202 | **W** | L | 203 | - | - | 204 | **W** | L | 205 | **W** | L |
| 206 | - | - | 207 | - | - | 208 | - | - | 209 | - | - | 210 | **W** | L |
| 211 | - | - | 212 | **W** | L | 213 | - | - | 214 | - | - | 215 | - | - |
| 216 | - | - | 217 | **W** | L | 218 | - | - | 219 | - | - | 220 | - | - |
| 221 | - | - | 222 | **W** | L | 223 | - | - | 224 | - | - | 225 | - | - |
| 226 | - | - | 227 | **W** | L | 228 | - | - | 229 | **W** | L | 230 | - | - |
| 231 | - | - | 232 | - | - | 233 | - | - | 234 | **W** | L | 235 | L | **W** |
| 236 | **W** | L | 237 | L | **W** | 238 | - | - | 239 | - | - | 240 | **W** | L |
| 241 | - | - | 242 | - | - | 243 | - | - | 244 | - | - | 245 | L | **W** |
| 246 | L | **W** | 247 | - | - | 248 | - | - | 249 | - | - | 250 | **W** | L |
| 251 | L | **W** | 252 | - | - | 253 | - | - | 254 | - | - | 255 | **W** | L |
| 256 | - | - | 257 | **W** | L | 258 | - | - | 259 | **W** | L | 260 | - | - |
| 261 | - | - | 262 | - | - | 263 | **W** | L | 264 | - | - | 265 | **W** | L |
| 266 | **W** | L | 267 | - | - | 268 | **W** | L | 269 | - | - | 270 | - | - |
| 271 | - | - | 272 | **W** | L | 273 | - | - | 274 | **W** | L | 275 | - | - |
| 276 | - | - | 277 | - | - | 278 | **W** | L | 279 | **W** | L | 280 | **W** | L |
| 281 | - | - | 282 | - | - | 283 | - | - | 284 | - | - | 285 | - | - |
| 286 | - | - | 287 | L | **W** | 288 | - | - | 289 | - | - | 290 | - | - |
| 291 | - | - | 292 | - | - | 293 | - | - | 294 | - | - | 295 | - | - |
| 296 | - | - | 297 | - | - | 298 | - | - | 299 | - | - | 300 | - | - |
| 301 | - | - | 302 | - | - | 303 | **W** | L | 304 | - | - | 305 | - | - |
| 306 | - | - | 307 | - | - | 308 | - | - | 309 | **W** | L | 310 | - | - |
| 311 | - | - | 312 | **W** | L | 313 | **W** | L | 314 | - | - | 315 | - | - |
| 316 | - | - | 317 | - | - | 318 | - | - | 319 | L | **W** | 320 | - | - |
| 321 | - | - | 322 | - | - | 323 | - | - | 324 | **W** | L | 325 | - | - |
| 326 | L | **W** | 327 | **W** | L | 328 | - | - | 329 | **W** | L | 330 | L | **W** |
| 331 | **W** | L | 332 | L | **W** | 333 | - | - | 334 | **W** | L | 335 | - | - |
| 336 | L | **W** | 337 | **W** | L | 338 | - | - | 339 | L | **W** | 340 | **W** | L |
| 341 | **W** | L | 342 | - | - | 343 | **W** | L | 344 | **W** | L | 345 | **W** | L |
| 346 | **W** | L | 347 | **W** | L | 348 | - | - | 349 | L | **W** | 350 | - | - |
| 351 | L | **W** | 352 | **W** | L | 353 | - | - | 354 | **W** | L | 355 | **W** | L |
| 356 | - | - | 357 | **W** | L | 358 | - | - | 359 | L | **W** | 360 | - | - |
| 361 | - | - | 362 | - | - | 363 | - | - | 364 | **W** | L | 365 | L | **W** |
| 366 | **W** | L | 367 | - | - | 368 | - | - | 369 | - | - | 370 | - | - |

| Iter | White | Black | Iter | White | Black | Iter | White | Black | Iter | White | Black | Iter | White | Black |
|---|---|---|---|---|---|---|---|---|---|---|---|---|---|---|
| 371 | L | **W** | 372 | - | - | 373 | **W** | L | 374 | - | - | 375 | **W** | L |
| 376 | **W** | L | 377 | - | - | 378 | **W** | L | 379 | - | - | 380 | - | - |
| 381 | - | - | 382 | - | - | 383 | L | **W** | 384 | L | **W** | 385 | L | **W** |
| 386 | L | **W** | 387 | - | - | 388 | - | - | 389 | **W** | L | 390 | - | - |
| 391 | **W** | L | 392 | **W** | L | 393 | L | **W** | 394 | - | - | 395 | **W** | L |
| 396 | - | - | 397 | **W** | L | 398 | - | - | 399 | - | - | 400 | **W** | L |
| 401 | **W** | L | 402 | L | **W** | 403 | **W** | L | 404 | - | - | 405 | **W** | L |
| 406 | **W** | L | 407 | - | - | 408 | L | **W** | 409 | L | **W** | 410 | - | - |
| 411 | L | **W** | 412 | **W** | L | 413 | - | - | 414 | **W** | L | 415 | - | - |
| 416 | - | - | 417 | **W** | L | 418 | **W** | L | 419 | - | - | 420 | - | - |
| 421 | - | - | 422 | - | - | 423 | - | - | 424 | L | **W** | 425 | **W** | L |
| 426 | - | - | 427 | - | - | 428 | - | - | 429 | - | - | 430 | - | - |
| 431 | **W** | L | 432 | - | - | 433 | - | - | 434 | - | - | 435 | - | - |
| 436 | **W** | L | 437 | L | **W** | 438 | L | **W** | 439 | L | **W** | 440 | L | **W** |
| 441 | L | **W** | 442 | **W** | L | 443 | **W** | L | 444 | **W** | L | 445 | **W** | L |
| 446 | - | - | 447 | **W** | L | 448 | L | **W** | 449 | **W** | L | 450 | - | - |
| 451 | - | - | 452 | - | - | 453 | - | - | 454 | - | - | 455 | **W** | L |
| 456 | **W** | L | 457 | **W** | L | 458 | **W** | L | 459 | **W** | L | 460 | - | - |
| 461 | - | - | 462 | - | - | 463 | - | - | 464 | **W** | L | 465 | - | - |
| 466 | - | - | 467 | L | **W** | 468 | - | - | 469 | - | - | 470 | L | **W** |
| 471 | - | - | 472 | - | - | 473 | L | **W** | 474 | L | **W** | 475 | L | **W** |
| 476 | - | - | 477 | - | - | 478 | L | **W** | 479 | **W** | L | 480 | - | - |
| 481 | L | **W** | 482 | - | - | 483 | - | - | 484 | **W** | L | 485 | L | **W** |
| 486 | - | - | 487 | - | - | 488 | - | - | 489 | - | - | 490 | - | - |
| 491 | - | - | 492 | - | - | 493 | L | **W** | 494 | L | **W** | 495 | L | **W** |
| 496 | L | **W** | 497 | **W** | L | 498 | L | **W** | 499 | L | **W** | 500 | L | **W** |
| 501 | L | **W** | 502 | L | **W** | 503 | - | - | 504 | - | - | 505 | L | **W** |
| 506 | L | **W** | 507 | L | **W** | 508 | L | **W** | 509 | L | **W** | 510 | - | - |
| 511 | - | - | 512 | **W** | L | 513 | L | **W** | 514 | - | - | 515 | **W** | L |
| 516 | L | **W** | 517 | L | **W** | 518 | - | - | 519 | L | **W** | 520 | - | - |
| 521 | - | - | 522 | - | - | 523 | L | **W** | 524 | **W** | L | 525 | L | **W** |
| 526 | L | **W** | 527 | L | **W** | 528 | L | **W** | 529 | - | - | 530 | - | - |
| 531 | - | - | 532 | - | - | 533 | - | - | 534 | - | - | 535 | **W** | L |
| 536 | - | - | 537 | - | - | 538 | - | - | 539 | - | - | 540 | - | - |
| 541 | - | - | 542 | - | - | 543 | L | **W** | 544 | - | - | 545 | **W** | L |
| 546 | L | **W** | 547 | L | **W** | 548 | **W** | L | 549 | L | **W** | 550 | L | **W** |
| 551 | L | **W** | 552 | L | **W** | 553 | L | **W** | 554 | - | - | 555 | L | **W** |
| 556 | L | **W** | 557 | L | **W** | 558 | L | **W** | 559 | - | - | 560 | **W** | L |
| 561 | L | **W** | 562 | L | **W** | 563 | **W** | L | 564 | L | **W** | 565 | - | - |
| 566 | L | **W** | 567 | **W** | L | 568 | L | **W** | 569 | L | **W** | 570 | - | - |
| 571 | L | **W** | 572 | L | **W** | 573 | L | **W** | 574 | L | **W** | 575 | L | **W** |
| 576 | L | **W** | 577 | L | **W** | 578 | - | - | 579 | - | - | 580 | - | - |
| 581 | L | **W** | 582 | L | **W** | 583 | **W** | L | 584 | L | **W** | 585 | L | **W** |
| 586 | - | - | 587 | L | **W** | 588 | L | **W** | 589 | L | **W** | 590 | - | - |
| 591 | L | **W** | 592 | **W** | L | 593 | L | **W** | 594 | - | - | 595 | L | **W** |
| 596 | L | **W** | 597 | L | **W** | 598 | L | **W** | 599 | L | **W** | 600 | L | **W** |
| 601 | L | **W** | 602 | L | **W** | 603 | - | - | 604 | L | **W** | 605 | - | - |
| 606 | L | **W** | 607 | L | **W** | 608 | **W** | L | 609 | L | **W** | 610 | L | **W** |
| 611 | L | **W** | 612 | L | **W** | 613 | - | - | 614 | - | - | 615 | L | **W** |
| 616 | L | **W** | 617 | - | - | 618 | L | **W** | 619 | L | **W** | 620 | - | - |
| 621 | - | - | 622 | L | **W** | 623 | L | **W** | 624 | - | - | 625 | L | **W** |
| 626 | L | **W** | 627 | L | **W** | 628 | - | - | 629 | L | **W** | 630 | - | - |
| 631 | L | **W** | 632 | - | - | 633 | - | - | 634 | L | **W** | 635 | L | **W** |
| 636 | L | **W** | 637 | L | **W** | 638 | L | **W** | 639 | L | **W** | 640 | L | **W** |
| 641 | L | **W** | 642 | - | - | 643 | - | - | 644 | - | - | 645 | - | - |
| 646 | L | **W** | 647 | L | **W** | 648 | - | - | 649 | - | - | 650 | - | - |
| 651 | **W** | L | 652 | L | **W** | 653 | - | - | 654 | L | **W** | 655 | - | - |
| 656 | - | - | 657 | - | - | 658 | L | **W** | 659 | L | **W** | 660 | - | - |
| 661 | L | **W** | 662 | L | **W** | 663 | L | **W** | 664 | L | **W** | 665 | L | **W** |
| 666 | L | **W** | 667 | L | **W** | 668 | L | **W** | 669 | L | **W** | 670 | L | **W** |
| 671 | L | **W** | 672 | - | - | 673 | - | - | 674 | L | **W** | 675 | **W** | L |
| 676 | - | - | 677 | - | - | 678 | **W** | L | 679 | **W** | L | 680 | **W** | L |
| 681 | **W** | L | 682 | L | **W** | 683 | - | - | 684 | - | - | 685 | - | - |
| 686 | - | - | 687 | L | **W** | 688 | **W** | L | 689 | L | **W** | 690 | L | **W** |
| 691 | - | - | 692 | - | - | 693 | L | **W** | 694 | L | **W** | 695 | **W** | L |
| 696 | - | - | 697 | - | - | 698 | L | **W** | 699 | **W** | L | 700 | **W** | L |
| 701 | L | **W** | 702 | - | - | 703 | - | - | 704 | - | - | 705 | - | - |
| 706 | L | **W** | 707 | - | - | 708 | L | **W** | 709 | L | **W** | 710 | L | **W** |
| 711 | L | **W** | 712 | - | - | 713 | **W** | L | 714 | - | - | 715 | L | **W** |
| 716 | - | - | 717 | - | - | 718 | - | - | 719 | - | - | 720 | - | - |
| 721 | L | **W** | 722 | - | - | 723 | - | - | 724 | - | - | 725 | L | **W** |
| 726 | - | - | 727 | L | **W** | 728 | - | - | 729 | - | - | 730 | - | - |
| 731 | - | - | 732 | - | - | 733 | - | - | 734 | L | **W** | 735 | L | **W** |
| 736 | L | **W** | 737 | L | **W** | 738 | L | **W** | 739 | L | **W** | 740 | - | - |
| 741 | **W** | L | 742 | L | **W** | 743 | - | - | 744 | **W** | L | 745 | - | - |
| 746 | **W** | L | 747 | L | **W** | 748 | - | - | 749 | L | **W** | 750 | **W** | L |
| 751 | - | - | 752 | - | - | 753 | - | - | 754 | - | - | 755 | L | **W** |
| 756 | - | - | 757 | L | **W** | 758 | - | - | 759 | - | - | 760 | - | - |

| Iter | White | Black | Iter | White | Black | Iter | White | Black | Iter | White | Black | Iter | White | Black |
|---|---|---|---|---|---|---|---|---|---|---|---|---|---|---|
| 761 | **W** | L | 762 | - | - | 763 | **W** | L | 764 | L | **W** | 765 | - | - |
| 766 | L | **W** | 767 | - | - | 768 | **W** | L | 769 | - | - | 770 | - | - |
| 771 | **W** | L | 772 | - | - | 773 | **W** | L | 774 | - | - | 775 | - | - |
| 776 | - | - | 777 | - | - | 778 | L | **W** | 779 | L | **W** | 780 | **W** | L |
| 781 | L | **W** | 782 | - | - | 783 | - | - | 784 | L | **W** | 785 | - | - |
| 786 | L | **W** | 787 | - | - | 788 | - | - | 789 | **W** | L | 790 | L | **W** |
| 791 | **W** | L | 792 | L | **W** | 793 | - | - | 794 | L | **W** | 795 | **W** | L |
| 796 | - | - | 797 | L | **W** | 798 | L | **W** | 799 | - | - | 800 | **W** | L |
| 801 | L | **W** | 802 | - | - | 803 | - | - | 804 | L | **W** | 805 | L | **W** |
| 806 | L | **W** | 807 | L | **W** | 808 | - | - | 809 | **W** | L | 810 | - | - |
| 811 | L | **W** | 812 | L | **W** | 813 | - | - | 814 | - | - | 815 | - | - |
| 816 | **W** | L | 817 | - | - | 818 | L | **W** | 819 | L | **W** | 820 | L | **W** |
| 821 | - | - | 822 | - | - | 823 | **W** | L | 824 | L | **W** | 825 | - | - |
| 826 | - | - | 827 | **W** | L | 828 | - | - | 829 | - | - | 830 | - | - |
| 831 | - | - | 832 | **W** | L | 833 | - | - | 834 | L | **W** | 835 | L | **W** |
| 836 | L | **W** | 837 | L | **W** | 838 | L | **W** | 839 | **W** | L | 840 | - | - |
| 841 | - | - | 842 | L | **W** | 843 | **W** | L | 844 | **W** | L | 845 | L | **W** |
| 846 | L | **W** | 847 | **W** | L | 848 | - | - | 849 | L | **W** | 850 | L | **W** |
| 851 | **W** | L | 852 | L | **W** | 853 | L | **W** | 854 | **W** | L | 855 | - | - |
| 856 | L | **W** | 857 | **W** | L | 858 | L | **W** | 859 | L | **W** | 860 | - | - |
| 861 | - | - | 862 | L | **W** | 863 | - | - | 864 | L | **W** | 865 | L | **W** |
| 866 | **W** | L | 867 | - | - | 868 | L | **W** | 869 | L | **W** | 870 | **W** | L |
| 871 | **W** | L | 872 | **W** | L | 873 | L | **W** | 874 | **W** | L | 875 | - | - |
| 876 | - | - | 877 | **W** | L | 878 | - | - | 879 | L | **W** | 880 | L | **W** |
| 881 | L | **W** | 882 | - | - | 883 | L | **W** | 884 | - | - | 885 | - | - |
| 886 | **W** | L | 887 | - | - | 888 | **W** | L | 889 | **W** | L | 890 | **W** | L |
| 891 | - | - | 892 | L | **W** | 893 | - | - | 894 | L | **W** | 895 | - | - |
| 896 | **W** | L | 897 | L | **W** | 898 | L | **W** | 899 | L | **W** | 900 | **W** | L |
| 901 | - | - | 902 | **W** | L | 903 | - | - | 904 | - | - | 905 | - | - |
| 906 | - | - | 907 | L | **W** | 908 | **W** | L | 909 | **W** | L | 910 | L | **W** |
| 911 | - | - | 912 | L | **W** | 913 | L | **W** | 914 | - | - | 915 | L | **W** |
| 916 | L | **W** | 917 | L | **W** | 918 | L | **W** | 919 | - | - | 920 | **W** | L |
| 921 | - | - | 922 | - | - | 923 | L | **W** | 924 | **W** | L | 925 | **W** | L |
| 926 | L | **W** | 927 | **W** | L | 928 | L | **W** | 929 | L | **W** | 930 | **W** | L |
| 931 | - | - | 932 | - | - | 933 | - | - | 934 | L | **W** | 935 | L | **W** |
| 936 | L | **W** | 937 | **W** | L | 938 | L | **W** | 939 | - | - | 940 | **W** | L |
| 941 | - | - | 942 | - | - | 943 | L | **W** | 944 | L | **W** | 945 | **W** | L |
| 946 | L | **W** | 947 | L | **W** | 948 | L | **W** | 949 | **W** | L | 950 | L | **W** |
| 951 | L | **W** | 952 | **W** | L | 953 | L | **W** | 954 | L | **W** | 955 | L | **W** |
| 956 | - | - | 957 | - | - | 958 | **W** | L | 959 | L | **W** | 960 | **W** | L |
| 961 | **W** | L | 962 | **W** | L | 963 | **W** | L | 964 | - | - | 965 | L | **W** |
| 966 | L | **W** | 967 | **W** | L | 968 | **W** | L | 969 | - | - | 970 | **W** | L |
| 971 | **W** | L | 972 | **W** | L | 973 | - | - | 974 | L | **W** | 975 | L | **W** |
| 976 | **W** | L | 977 | - | - | 978 | L | **W** | 979 | L | **W** | 980 | L | **W** |
| 981 | L | **W** | 982 | L | **W** | 983 | L | **W** | 984 | - | - | 985 | - | - |
| 986 | L | **W** | 987 | **W** | L | 988 | - | - | 989 | **W** | L | 990 | - | - |
| 991 | L | **W** | 992 | L | **W** | 993 | L | **W** | 994 | **W** | L | 995 | L | **W** |
| 996 | L | **W** | 997 | - | - | 998 | L | **W** | 999 | - | - | 1000 | - | - |

. . . . . . . . . . . . . . . . . . . . . . . . . . . . . . . . . . . . . . . . . . . . . . . . . . . . . . . . . . . . . . . . . . . . . . . . . . . . . . . . . . . . . . . . . . . . . . . . . . . . . . . . . . . . . . . . . . . . . .

## O   Run on Go

Figures 34–36 present supplementary analysis of GEMS on the Go environment implemented using `PettingZoo`. We evaluate training dynamics over 200 iterations (matching the run used to generate the figures below), using the same logging and evaluation protocol as in our Chess analysis.

Figure 34 reports system-level metrics over training iterations: per-iteration runtime, cumulative wall-clock time, and memory usage. The per-iteration runtime exhibits bounded variability without a persistent upward drift, yielding an approximately linear increase in cumulative wall-clock time. Memory usage remains essentially stable with only minor incremental changes, indicating that GEMS maintains consistent computational and memory overhead over training. This stability is particularly notable in Go, where large board states and long-horizon rollouts typically amplify the overhead of population-based game-solving pipelines.

Figure 35 illustrates emergent spatial structure in Go rollouts after training. Shown are board states sampled from a single rollout of trained agents after 200 training iterations, with successive frames corresponding to progressively later timesteps within the rollout. The evolution from sparse, locally uncoordinated placements to coherent territorial regions highlights the emergence of non-trivial spatial coordination induced by the learned policies. Importantly, this structure arises without hand-crafted Go priors, explicit territory heuristics, or domain-specific shaping, suggesting that GEMS enables agents to discover meaningful spatial organization purely through self-play and amortized policy generation.

Figure 36 analyzes the learned representations via PCA projections of two embedding spaces for both agents. In *action-distribution* space (left), the embeddings clearly separate into **two disjoint clusters** with essentially no overlap, indicating that the learned policies concentrate into two qualitatively distinct *behavioral modes* rather than collapsing to a single averaged style. Within each cluster, the action entropy varies smoothly (color gradient), suggesting structured but still stochastic action selection inside each mode.

In contrast, the *latent-code* embeddings (right) remain broadly dispersed without collapsing to a small region, reflecting substantial representational diversity. This separation—**multi-modal structure in behavior** together with **non-collapsed latent representations**—is particularly desirable in Go, where there are many strategically distinct yet viable ways to play. Overall, the plot is consistent with GEMS maintaining multiple distinct strategic hypotheses (no representational mode collapse) while preserving coherent, structured behavior.

Taken together, these results suggest that GEMS scales robustly to high-dimensional spatial games such as Go, maintaining stable system-level performance while preserving structured, diverse latent representations over training.

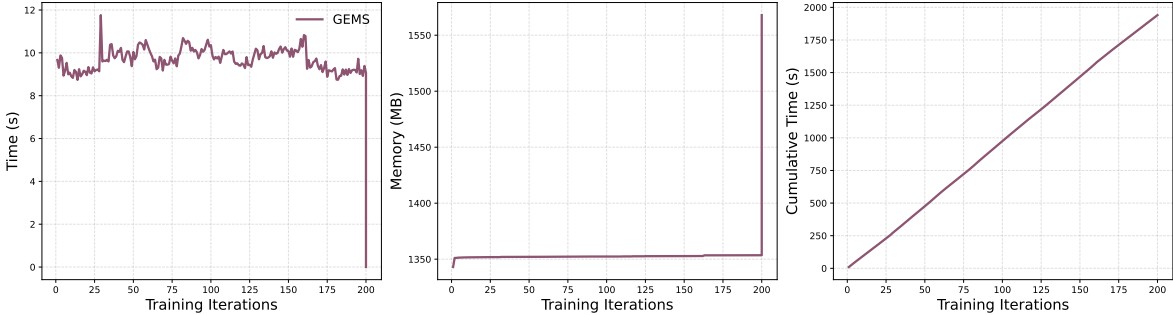

Figure 34: System-level metrics over training iterations. Per-iteration runtime exhibits bounded variability without long-term drift, leading to a near-linear growth in cumulative wall-clock time. Memory usage remains stable with only minor incremental increases, indicating that GEMS maintains consistent computational and memory overhead throughout training.

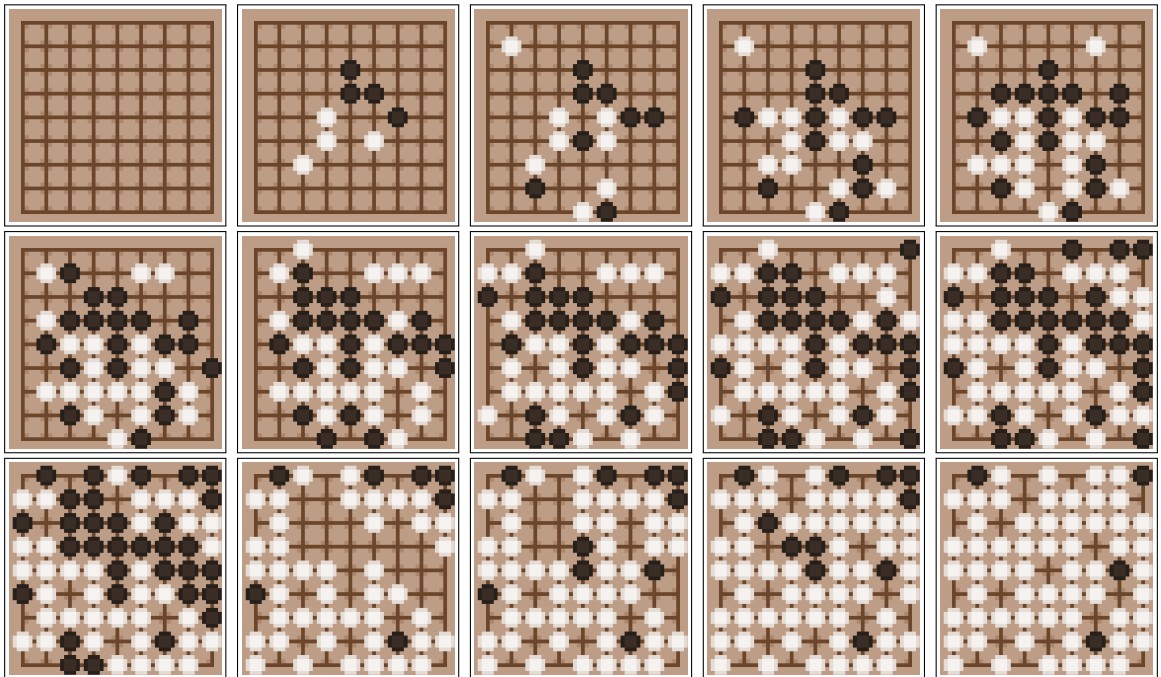

Figure 35: **Emergent spatial structure in Go rollouts using GEMS.** Shown are board states sampled from a single rollout of agents trained via the GEMS MARL framework after 1000 training iterations. Frames are uniformly sampled across the trajectory (frames 1, 9, 17, 25, 33, 41, 49, 57, 65, 73, 81, 89, 97, 105, and 113), arranged left-to-right and top-to-bottom in temporal order. The evolution from sparse placements to coherent territorial regions highlights the emergence of non-trivial spatial coordination induced by the GEMS policies, despite the absence of hand-crafted Go priors.

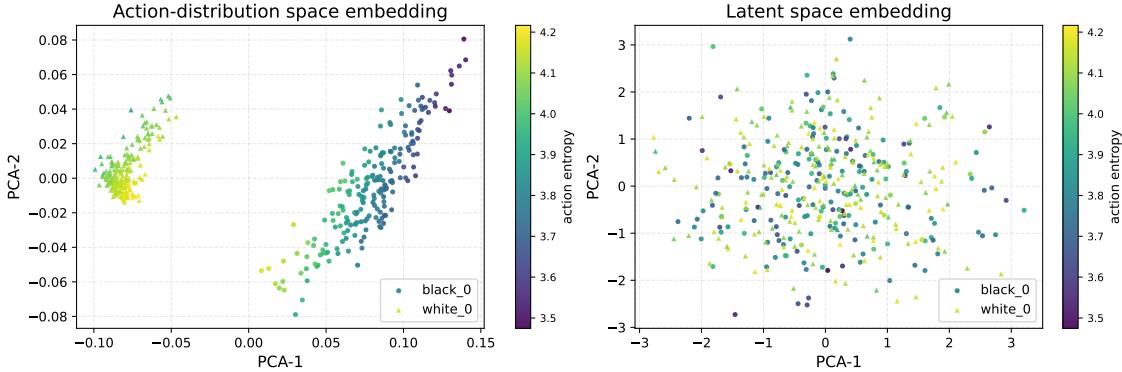

Figure 36: **Go embeddings after training: behavioral modes without latent collapse.** Each point is an embedding from one of the two agents (black/white), projected to 2D via PCA; color indicates action entropy. *Left (action-distribution space):* embeddings form two clearly separated, non-overlapping clusters with smoothly varying entropy, indicating two distinct behavioral modes. *Right (latent space):* embeddings remain broadly dispersed rather than collapsing, reflecting substantial representational diversity consistent with maintaining multiple viable strategic hypotheses.

# P Ablation on Kuhn's Poker

Tables 5–6 enumerate the sweeps used for Figs. 37–38. **Boldface rows indicate the configurations that achieved the lowest exploitability within each panel.** Across the grid, the strongest settings typically combined a small/zero exploration scale ($\eta \in \{0, 0.06\}$) with simple schedulers (const/sqrt/harmonic), minimal mutation/random pools, and either `least_mass` or `worst_ev` replacement (panel dependent).

**EMA toggle (ablation only).** Our *main* Kuhn Poker runs disable Estimated Moving Average (EMA) (`ema=0`). For this ablation, we enabled EMA to probe stability: given meta-estimates $(V^p, r)$ and coefficient $\beta \in (0, 1)$, we update

$$\hat{V}^p \leftarrow (1 - \beta)\hat{V}^p + \beta V^p, \qquad \hat{r} \leftarrow (1 - \beta)\hat{r} + \beta r. \tag{82}$$

Implementationally, this follows `meta_estimate_exact` in our code. With EMA on ($\beta \in \{0.2, 0.5, 0.8\}$, see the `EMA` column), we observed the lowest exploitability among all tested settings (boldface). A systematic treatment of EMA choice (schedule, bias-correction, and scope) is deferred to future work.

Table 5: Configuration details corresponding to Fig. 37.

| Sub-figure | $\eta$ | $\eta$ scheduler | EMA | Steps | LR | $\beta_{KL}$ | Mutation Pool | Random Pool | Replacement |
|---|---|---|---|---|---|---|---|---|---|
| (a) | 0.08 | harmonic | 0.2 | 0 | 1e-4 | 0.0 | 0 | 0 | least_mass |
| (b) | 0.12 | const | 0.5 | 0 | 1e-4 | 0.0 | 1 | 0 | worst_ev |
| (c) | 0.06 | harmonic | 0.5 | 0 | 1e-4 | 0.0 | 2 | 1 | least_mass |
| (d) | 0.04 | harmonic | 0.0 | 20 | 5e-4 | 1e-2 | 1 | 0 | worst_ev |
| (e) | 0.06 | sqrt | 0.8 | 0 | 1e-4 | 0.0 | 1 | 0 | worst_ev |
| (f) | 0.06 | sqrt | 0.2 | 0 | 1e-4 | 0.0 | 1 | 0 | worst_ev |
| (g) | 0.0 | const | 0.0 | 0 | 1e-4 | 0.0 | 1 | 0 | least_mass |
| (h) | 0.12 | harmonic | 0.5 | 10 | 3e-4 | 5e-3 | 1 | 0 | worst_ev |
| **(i)** | **0.0** | **const** | **0.0** | **0** | **1e-4** | **0.0** | **0** | **0** | **least_mass** |
| (j) | 0.08 | harmonic | 0.5 | 0 | 1e-4 | 0.0 | 0 | 0 | least_mass |
| **(k)** | **0.06** | **sqrt** | **0.2** | **0** | **1e-4** | **0.0** | **0** | **0** | **worst_ev** |
| (l) | 0.10 | harmonic | 0.8 | 0 | 1e-4 | 0.0 | 1 | 0 | least_mass |
| (m) | 0.12 | const | 0.0 | 0 | 1e-4 | 0.0 | 2 | 1 | worst_ev |
| (n) | 0.08 | const | 0.8 | 20 | 5e-4 | 1e-2 | 2 | 1 | least_mass |
| (o) | 0.06 | harmonic | 0.0 | 20 | 5e-4 | 1e-2 | 1 | 0 | least_mass |
| (p) | 0.10 | harmonic | 0.0 | 20 | 5e-4 | 1e-2 | 1 | 0 | least_mass |
| (q) | 0.12 | sqrt | 0.5 | 0 | 1e-4 | 0.0 | 0 | 0 | least_mass |
| (r) | 0.10 | harmonic | 0.8 | 20 | 5e-4 | 1e-2 | 2 | 1 | worst_ev |
| (s) | 0.06 | sqrt | 0.8 | 20 | 5e-4 | 1e-2 | 1 | 0 | worst_ev |
| (t) | 0.12 | sqrt | 0.8 | 10 | 3e-4 | 5e-3 | 2 | 1 | least_mass |

Table 6: Configuration details corresponding to Fig. 38.

| Sub-figure | $\eta$ | $\eta$ scheduler | EMA | Steps | LR | $\beta_{KL}$ | Mutation Pool | Random Pool | Replacement |
|---|---|---|---|---|---|---|---|---|---|
| **(a)** | **0.0** | **const** | **0.0** | **0** | **1e-4** | **0.0** | **0** | **0** | **worst_ev** |
| (b) | 0.12 | harmonic | 0.0 | 20 | 5e-4 | 1e-2 | 1 | 0 | least_mass |
| (c) | 0.10 | const | 0.2 | 0 | 1e-4 | 0.0 | 2 | 1 | worst_ev |
| **(d)** | **0.06** | **harmonic** | **0.0** | **20** | **5e-4** | **1e-2** | **0** | **0** | **worst_ev** |
| **(e)** | **0.06** | **harmonic** | **0.5** | **0** | **1e-4** | **0.0** | **0** | **0** | **worst_ev** |
| (f) | 0.08 | const | 0.8 | 0 | 1e-4 | 0.0 | 0 | 0 | worst_ev |
| (g) | 0.06 | sqrt | 0.2 | 0 | 1e-4 | 0.0 | 1 | 0 | least_mass |
| (h) | 0.06 | sqrt | 0.2 | 0 | 1e-4 | 0.0 | 2 | 1 | least_mass |
| (i) | 0.0 | const | 0.0 | 0 | 1e-4 | 0.0 | 1 | 0 | least_mass |
| (j) | 0.04 | sqrt | 0.0 | 10 | 3e-4 | 5e-3 | 0 | 0 | least_mass |

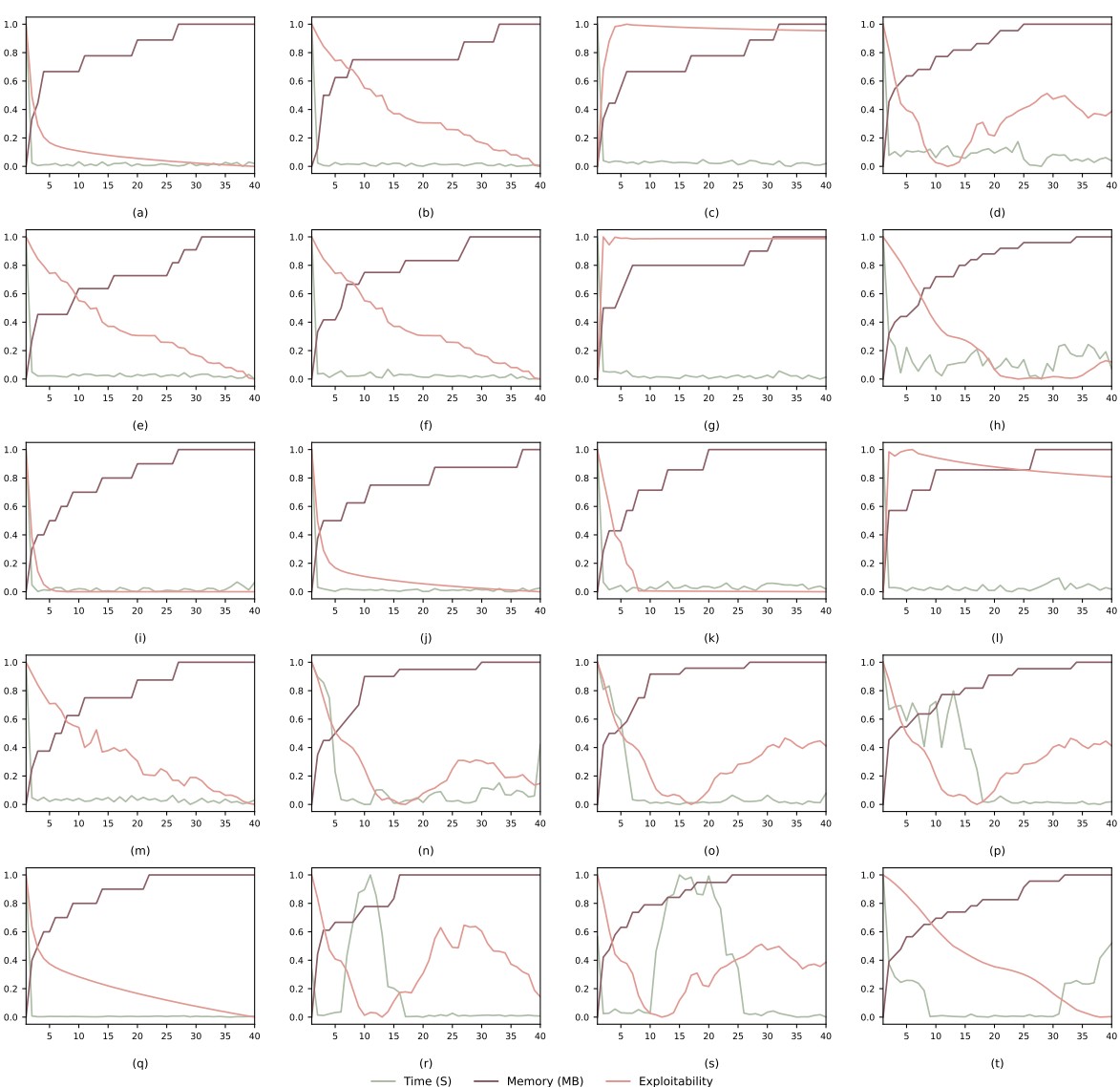

Figure 37: Ablation on Kuhn's Poker **part 1/2**, mean of 5 seeds. Legend: ❶ Memory, at Y-axis 1 defines 1256.005 MB, 0 defines 0 MB, ❷ Time, at Y-axis 1 defines 0.01 S, 0 defines 0 S, ❸ Exploitability, Y-axis 1 defines 1, 0 defines 0, and ❹ Iterations are defined on X-axis going from 0 to 40.

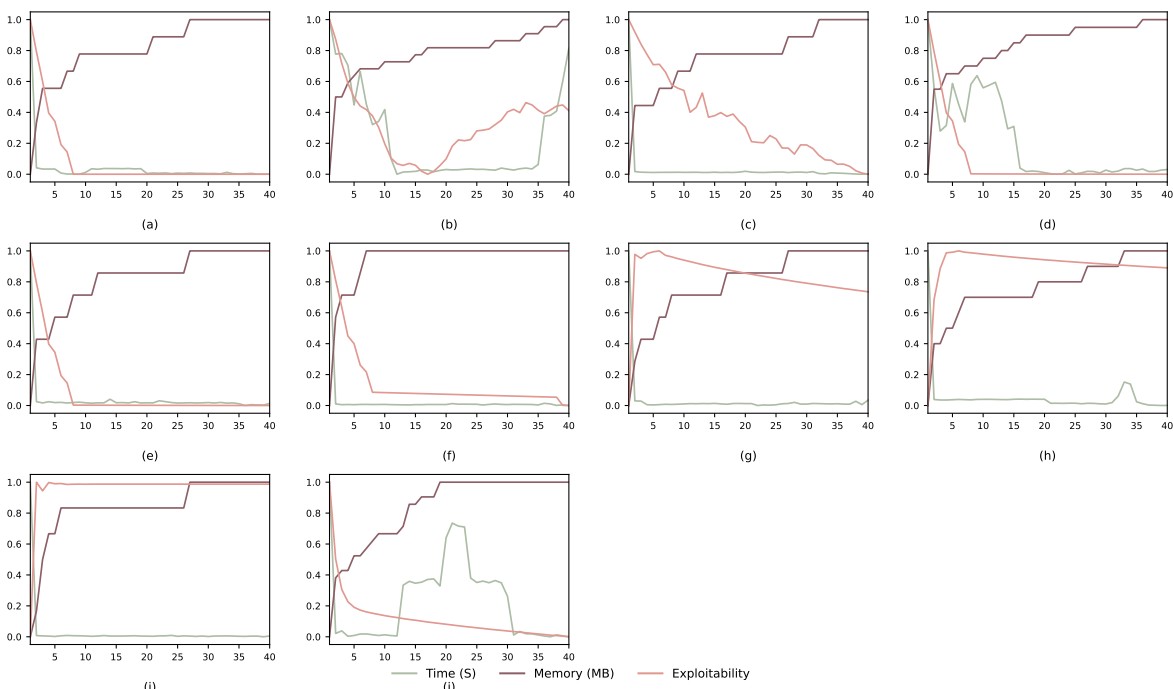

Figure 38: Ablation on Kuhn's Poker **part 2/2**, mean of 5 seeds. Legend: ❶ Memory, at Y-axis 1 defines 1256.005 MB, 0 defines 0 MB, ❷ Time, at Y-axis 1 defines 0.01 S, 0 defines 0 S, ❸ Exploitability, Y-axis 1 defines 1, 0 defines 0, and ❹ Iterations are defined on X-axis going from 0 to 40.

## Q    Ablation on Public Goods Game

**Public Goods Game Ablation: Purpose.**    This ablation studies *which parts of FAST GEMS–OMWU matter most* for outcome quality versus compute, so that our defaults are principled and robust rather than hand-tuned. Concretely, we vary algorithmic and stability knobs—$\eta$, `eta_sched` (const/sqrt/harmonic), `ema`, `mwu_grad_cap`, temperature $\tau$; population-growth and exploration knobs—`oracle_nz`, `oracle_period`, `pool_mut`, `pool_rand`, latent size `zdim`; and ABR–TR tightness—`abr_lr`, `abr_steps`, and $\beta_{\mathrm{KL}}$. We summarize effects with terminal and speed-sensitive metrics: `welfare_last`, `coop_last` (final social welfare and cooperation) and their learning-curve integrals `welfare_auc`, `coop_auc`, alongside efficiency measures `time_total_sec_avg` and `ram_peak_mb_avg`. A full factorial sweep is possible, but would require substantially more compute than we currently have available, so, we run structured partial variations (one-factor and a few targeted pairs) across multiple seeds and report mean±std from the multi-seed runner; the intent is to **characterize sensitivities and trade-offs**, not to claim a single global optimum. In practice, the ablation clarifies (i) step-size scheduling and mild smoothing (harmonic $\eta_t$ with modest EMA) stabilize meta-updates without slowing progress unduly, (ii) moderate $\beta_{\mathrm{KL}}$ in ABR–TR improves welfare without large time/RAM penalties, and (iii) small increases in `oracle_period` and balanced `pool_mut`/`pool_rand` control population growth while preserving exploration. All runs log the exact configuration to CSV for reproducibility and allow further probing of interactions as needed.

**Public Goods Game: overview & metrics.**    The *Public Goods Game (PGG)* is an $n$-player social-dilemma benchmark. Each player chooses to *contribute* (cooperate) or *withhold* (defect). Let $a_i \in 0, 1$ denote player $i$'s contribute decision, $S = \sum_j a_j$ the total contributions, multiplier $r > 0$, and per-contribution cost $c_i > 0$ (homogeneous case uses $c_i \equiv c$). The one-round payoff to player $i$ is

$$u_i \; = \; \frac{r}{n}\, S \; - \; c_i\, a_i, \qquad \text{and the social welfare} \quad W \; = \; \sum_{i=1}^{n} u_i \; = \; r\, S \; - \; \sum_{i=1}^{n} c_i a_i. \tag{83}$$

When $\frac{r}{n} < c < r$ (homogeneous costs), each individual gains by defecting (free-riding) while the group gains by coordinating on cooperation—this is the core tension PGG exposes.

In our runs, the generator outputs the probability of cooperation; we track both *outcome quality* and *learning speed/stability* via four summary metrics:

- **welfare_last**: the mean social welfare $W$ at the *final training iteration* (averaged over profile samples and seeds). Higher is better; if parameters make $W$ negative, "less negative" still indicates improvement.

- **coop_last**: the final average cooperation rate $\frac{1}{n}\sum_i \Pr[a_i{=}1]$.

- **welfare_auc**: the (trapezoidal) area under the welfare-versus-iteration curve. This rewards methods that reach good welfare earlier and keep it stable—a joint measure of speed and robustness.

- **coop_auc**: the analogous AUC for cooperation rate.

All four are reported as mean±std across seeds by the multi-seed runner, making it straightforward to compare algorithms and hyper-parameter settings on both efficiency and cooperative outcomes. iters_in_child = 10; seeds_in_child = 0,1,2,3,4; iters = 10; n_players = 5.

Table 7: Ablation (Public Goods Game)

| welfare_last | coop_last | welfare_auc | coop_auc | time_total_sec_avg | ram_peak_mb_avg | abr_lr | abr_steps | beta_kl | cost_c | delta0 | ema | eta | eta_sched | hetero | lambda_jac | log_ucb | mult_r | mwu_grad_cap | no_plots | num_seeds | oracle_nz | oracle_period | pool_mut | pool_rand | slow | slowdown | tau | top_seed | ucb_nz | zdim |
|---|---|---|---|---|---|---|---|---|---|---|---|---|---|---|---|---|---|---|---|---|---|---|---|---|---|---|---|---|---|---|
| 0.008177475 | 0.000817748 | 0.997798743 | 0.099779875 | 1.252423239 | 1240.809375 | 0.0005 | 20 | 0.01 | 1 | 0.0005 | 0.0 | 0.08 | const | 0 | 0 | 0 | | 3 | 1 | 5 | 1 | 1 | 2 | 1 | 0 | 1 | 1.0 | 0 | 8 | 8 |
| 0.0481612 | 0.00481612 | 4.389817474 | 0.438981745 | 1.243599987 | 1240.700781 | 0.00025 | 20 | 0.02 | 1 | 0.0005 | 0.0 | 0.04 | const | 0 | 0 | 0 | | 3 | 1 | 5 | 1 | 1 | 2 | 1 | 0 | 2 | 1.5 | 0 | 8 | 8 |
| 0.007937888 | 0.000793789 | 0.899287799 | 0.08992878 | 1.250558043 | 1240.736719 | 0.0005 | 20 | 0.01 | 1 | 0.0005 | 0.8 | 0.08 | harmonic | 0 | 0 | 0 | 0.1 | 3 | 1 | 5 | 1 | 2 | 2 | 1 | 1 | 1 | 1.00 | 0 | 8 | 8 |
| 0.036606287 | 0.003660629 | 4.423422511 | 0.442342253 | 1.220253325 | 1240.854688 | 0.00025 | 20 | 0.02 | 1 | 0.0005 | 0.8 | 0.04 | harmonic | 0 | 0 | 0 | 0.05 | 3 | 1 | 5 | 1 | 2 | 2 | 1 | 1 | 2 | 1.50 | 0 | 8 | 8 |
| 0.021302772 | 0.002130277 | 1.155576756 | 0.115557676 | 1.270089245 | 1241.642188 | 0.0005 | 20 | 0.01 | 1 | 0.0005 | 0.0 | 0.08 | const | 0 | 0 | 0 | | 3 | 1 | 5 | 1 | 1 | 2 | 1 | 0 | 1 | 1.00 | 0 | 8 | 16 |
| 0.065991034 | 0.006599103 | 4.265299203 | 0.42652992 | 1.26079936 | 1240.8125 | 0.00025 | 20 | 0.02 | 1 | 0.0005 | 0.0 | 0.04 | const | 0 | 0 | 0 | | 3 | 1 | 5 | 1 | 1 | 2 | 1 | 0 | 2 | 1.50 | 0 | 8 | 16 |
| 0.009342633 | 0.000934263 | 0.683710212 | 0.068371021 | 1.226696777 | 1240.733594 | 0.0005 | 20 | 0.01 | 1 | 0.0005 | 0.8 | 0.08 | harmonic | 0 | 0 | 0 | 0.1 | 3 | 1 | 5 | 1 | 2 | 2 | 1 | 1 | 1 | 1.00 | 0 | 8 | 16 |
| 0.031212943 | 0.003121294 | 3.660477072 | 0.366047713 | 1.247665262 | 1240.683594 | 0.00025 | 20 | 0.02 | 1 | 0.0005 | 0.8 | 0.04 | harmonic | 0 | 0 | 0 | 0.05 | 3 | 1 | 5 | 1 | 2 | 2 | 1 | 1 | 2 | 1.50 | 0 | 8 | 16 |
| 0.008177475 | 0.000817748 | 0.997798743 | 0.099779875 | 1.288738918 | 1240.8625 | 0.0005 | 20 | 0.01 | 1 | 0.0005 | 0.0 | 0.08 | const | 0 | 0 | 0 | | 3 | 1 | 5 | 2 | 1 | 2 | 1 | 0 | 1 | 1.00 | 0 | 8 | 8 |
| 0.0481612 | 0.00481612 | 4.389817474 | 0.438981745 | 1.256717253 | 1240.785156 | 0.00025 | 20 | 0.02 | 1 | 0.0005 | 0.0 | 0.04 | const | 0 | 0 | 0 | | 3 | 1 | 5 | 2 | 1 | 2 | 1 | 0 | 2 | 1.50 | 0 | 8 | 8 |
| 0.007937888 | 0.000793789 | 0.899287799 | 0.08992878 | 1.267520666 | 1240.686719 | 0.0005 | 20 | 0.01 | 1 | 0.0005 | 0.8 | 0.08 | harmonic | 0 | 0 | 0 | 0.1 | 3 | 1 | 5 | 2 | 2 | 2 | 1 | 1 | 1 | 1.00 | 0 | 8 | 8 |
| 0.036606287 | 0.003660629 | 4.423422511 | 0.442342253 | 1.28211751 | 1240.714844 | 0.00025 | 20 | 0.02 | 1 | 0.0005 | 0.8 | 0.04 | harmonic | 0 | 0 | 0 | 0.05 | 3 | 1 | 5 | 2 | 2 | 2 | 1 | 1 | 2 | 1.50 | 0 | 8 | 8 |
| 0.021302772 | 0.002130277 | 1.155576756 | 0.115557676 | 1.30556698 | 1240.908594 | 0.0005 | 20 | 0.01 | 1 | 0.0005 | 0.0 | 0.08 | const | 0 | 0 | 0 | | 3 | 1 | 5 | 2 | 1 | 2 | 1 | 0 | 1 | 1.00 | 0 | 8 | 16 |
| 0.065991034 | 0.006599103 | 4.265299203 | 0.42652992 | 1.252139187 | 1240.901563 | 0.00025 | 20 | 0.02 | 1 | 0.0005 | 0.0 | 0.04 | const | 0 | 0 | 0 | | 3 | 1 | 5 | 2 | 1 | 2 | 1 | 0 | 2 | 1.50 | 0 | 8 | 16 |
| 0.009342633 | 0.000934263 | 0.683710212 | 0.068371021 | 1.276571846 | 1240.667969 | 0.0005 | 20 | 0.01 | 1 | 0.0005 | 0.8 | 0.08 | harmonic | 0 | 0 | 0 | 0.1 | 3 | 1 | 5 | 2 | 2 | 2 | 1 | 1 | 1 | 1.00 | 0 | 8 | 16 |
| 0.031212943 | 0.003121294 | 3.660477072 | 0.366047713 | 1.282459736 | 1240.673438 | 0.00025 | 20 | 0.02 | 1 | 0.0005 | 0.8 | 0.04 | harmonic | 0 | 0 | 0 | 0.05 | 3 | 1 | 5 | 2 | 2 | 2 | 1 | 1 | 2 | 1.50 | 0 | 8 | 16 |
| 0.019388839 | 0.001938884 | 1.242629356 | 0.124262934 | 1.26205411 | 1240.679688 | 0.0005 | 20 | 0.01 | 1 | 0.0005 | 0.0 | 0.08 | const | 0 | 0 | 0 | | 3 | 1 | 5 | 1 | 1 | 2 | 2 | 0 | 1 | 1.00 | 0 | 8 | 8 |
| 0.085384213 | 0.008538421 | 4.706878492 | 0.47068785 | 1.265106869 | 1240.692188 | 0.00025 | 20 | 0.02 | 1 | 0.0005 | 0.0 | 0.04 | const | 0 | 0 | 0 | | 3 | 1 | 5 | 1 | 1 | 2 | 2 | 0 | 2 | 1.50 | 0 | 8 | 8 |
| 0.005458697 | 0.00054587 | 0.874286836 | 0.087428684 | 1.226155043 | 1240.671094 | 0.0005 | 20 | 0.01 | 1 | 0.0005 | 0.8 | 0.08 | harmonic | 0 | 0 | 0 | 0.1 | 3 | 1 | 5 | 1 | 2 | 2 | 2 | 1 | 1 | 1.00 | 0 | 8 | 8 |
| 0.045789669 | 0.004578967 | 4.528651814 | 0.452865186 | 1.272323084 | 1240.759375 | 0.00025 | 20 | 0.02 | 1 | 0.0005 | 0.8 | 0.04 | harmonic | 0 | 0 | 0 | 0.05 | 3 | 1 | 5 | 1 | 2 | 2 | 2 | 1 | 2 | 1.50 | 0 | 8 | 8 |
| 0.022797623 | 0.002279762 | 1.036746019 | 0.103674601 | 1.274035454 | 1240.702344 | 0.0005 | 20 | 0.01 | 1 | 0.0005 | 0.0 | 0.08 | const | 0 | 0 | 0 | | 3 | 1 | 5 | 1 | 1 | 2 | 2 | 0 | 1 | 1.00 | 0 | 8 | 16 |
| 0.069790505 | 0.00697905 | 4.056669959 | 0.405666995 | 1.318738031 | 1240.808594 | 0.00025 | 20 | 0.02 | 1 | 0.0005 | 0.0 | 0.04 | const | 0 | 0 | 0 | | 3 | 1 | 5 | 1 | 1 | 2 | 2 | 0 | 2 | 1.50 | 0 | 8 | 16 |
| 0.012179432 | 0.001217943 | 0.750434142 | 0.075043414 | 1.30443511 | 1240.728906 | 0.0005 | 20 | 0.01 | 1 | 0.0005 | 0.8 | 0.08 | harmonic | 0 | 0 | 0 | 0.1 | 3 | 1 | 5 | 2 | 2 | 2 | 1 | 1 | 1 | 1.00 | 0 | 8 | 16 |
| 0.047578595 | 0.004757859 | 3.948711966 | 0.394871199 | 1.318915558 | 1240.654688 | 0.00025 | 20 | 0.02 | 1 | 0.0005 | 0.8 | 0.04 | harmonic | 0 | 0 | 0 | 0.05 | 3 | 1 | 5 | 2 | 2 | 2 | 1 | 2 | 1.50 | 0 | 8 | 16 |
| 0.019388839 | 0.001938884 | 1.242629356 | 0.124262934 | 1.323859501 | 1240.845313 | 0.0005 | 20 | 0.01 | 1 | 0.0005 | 0.0 | 0.08 | const | 0 | 0 | 0 | | 3 | 1 | 5 | 2 | 1 | 2 | 2 | 0 | 1 | 1.00 | 0 | 8 | 8 |
| 0.085384213 | 0.008538421 | 4.706878492 | 0.47068785 | 1.344771719 | 1240.771094 | 0.00025 | 20 | 0.02 | 1 | 0.0005 | 0.0 | 0.04 | const | 0 | 0 | 0 | | 3 | 1 | 5 | 2 | 1 | 2 | 2 | 0 | 2 | 1.50 | 0 | 8 | 8 |
| 0.005458697 | 0.00054587 | 0.874286836 | 0.087428684 | 1.362584257 | 1240.678125 | 0.0005 | 20 | 0.01 | 1 | 0.0005 | 0.8 | 0.08 | harmonic | 0 | 0 | 0 | 0.1 | 3 | 1 | 5 | 2 | 2 | 2 | 2 | 1 | 1 | 1.00 | 0 | 8 | 8 |
| 0.045789669 | 0.004578967 | 4.528651814 | 0.452865186 | 1.289640522 | 1240.711719 | 0.00025 | 20 | 0.02 | 1 | 0.0005 | 0.8 | 0.04 | harmonic | 0 | 0 | 0 | 0.05 | 3 | 1 | 5 | 2 | 2 | 2 | 2 | 1 | 2 | 1.50 | 0 | 8 | 8 |
| 0.022797623 | 0.002279762 | 1.036746019 | 0.103674601 | 1.355440092 | 1240.840625 | 0.0005 | 20 | 0.01 | 1 | 0.0005 | 0.0 | 0.08 | const | 0 | 0 | 0 | | 3 | 1 | 5 | 2 | 1 | 2 | 2 | 0 | 1 | 1.00 | 0 | 8 | 16 |
| 0.069790505 | 0.00697905 | 4.056669959 | 0.405666995 | 1.356069088 | 1240.857813 | 0.00025 | 20 | 0.02 | 1 | 0.0005 | 0.0 | 0.04 | const | 0 | 0 | 0 | | 3 | 1 | 5 | 2 | 1 | 2 | 2 | 0 | 2 | 1.50 | 0 | 8 | 16 |
| 0.012179432 | 0.001217943 | 0.750434142 | 0.075043414 | 1.287806368 | 1240.705469 | 0.0005 | 20 | 0.01 | 1 | 0.0005 | 0.8 | 0.08 | harmonic | 0 | 0 | 0 | 0.1 | 3 | 1 | 5 | 2 | 2 | 2 | 1 | 1 | 1 | 1.00 | 0 | 8 | 16 |
| 0.047578595 | 0.004757859 | 3.948711966 | 0.394871199 | 1.296716118 | 1240.686719 | 0.00025 | 20 | 0.02 | 1 | 0.0005 | 0.8 | 0.04 | harmonic | 0 | 0 | 0 | 0.05 | 3 | 1 | 5 | 2 | 2 | 2 | 1 | 2 | 1.50 | 0 | 8 | 16 |
| 0.029370787 | 0.002937079 | 1.329133807 | 0.132913381 | 1.353565598 | 1240.934375 | 0.0005 | 20 | 0.01 | 1 | 0.0005 | 0.0 | 0.08 | const | 0 | 0 | 0 | | 3 | 1 | 5 | 1 | 1 | 4 | 1 | 0 | 1 | 1.00 | 0 | 8 | 8 |
| 0.111277251 | 0.011127725 | 5.157341515 | 0.515734149 | 1.340657473 | 1240.902344 | 0.00025 | 20 | 0.02 | 1 | 0.0005 | 0.0 | 0.04 | const | 0 | 0 | 0 | | 3 | 1 | 5 | 1 | 1 | 4 | 1 | 0 | 2 | 1.50 | 0 | 8 | 8 |
| 0.004473344 | 0.000447334 | 0.803012078 | 0.080301208 | 1.296801376 | 1240.664063 | 0.0005 | 20 | 0.01 | 1 | 0.0005 | 0.8 | 0.08 | harmonic | 0 | 0 | 0 | 0.1 | 3 | 1 | 5 | 1 | 2 | 4 | 1 | 1 | 1 | 1.00 | 0 | 8 | 8 |
| 0.041325395 | 0.004132539 | 4.417075988 | 0.4417076 | 1.291794729 | 1240.714063 | 0.00025 | 20 | 0.02 | 1 | 0.0005 | 0.8 | 0.04 | harmonic | 0 | 0 | 0 | 0.05 | 3 | 1 | 5 | 1 | 2 | 4 | 1 | 1 | 2 | 1.50 | 0 | 8 | 8 |
| 0.017441394 | 0.001744139 | 0.936132263 | 0.093613227 | 1.345355701 | 1240.805469 | 0.0005 | 20 | 0.01 | 1 | 0.0005 | 0.0 | 0.08 | const | 0 | 0 | 0 | | 3 | 1 | 5 | 1 | 1 | 4 | 1 | 0 | 1 | 1.00 | 0 | 8 | 16 |
| 0.06414937 | 0.006414937 | 4.018342035 | 0.401834202 | 1.333889103 | 1240.648438 | 0.00025 | 20 | 0.02 | 1 | 0.0005 | 0.0 | 0.04 | const | 0 | 0 | 0 | | 3 | 1 | 5 | 1 | 1 | 4 | 1 | 0 | 2 | 1.50 | 0 | 8 | 16 |
| 0.009716086 | 0.000971609 | 0.649623368 | 0.064962337 | 1.299304295 | 1240.627344 | 0.0005 | 20 | 0.01 | 1 | 0.0005 | 0.8 | 0.08 | harmonic | 0 | 0 | 0 | 0.1 | 3 | 1 | 5 | 1 | 2 | 4 | 1 | 1 | 1 | 1.00 | 0 | 8 | 16 |
| 0.043877613 | 0.004387761 | 3.711383419 | 0.371138343 | 1.276826048 | 1240.721094 | 0.00025 | 20 | 0.02 | 1 | 0.0005 | 0.8 | 0.04 | harmonic | 0 | 0 | 0 | 0.05 | 3 | 1 | 5 | 1 | 2 | 4 | 1 | 1 | 2 | 1.50 | 0 | 8 | 16 |
| 0.029370787 | 0.002937079 | 1.329133807 | 0.132913381 | 1.33368177 | 1240.690625 | 0.0005 | 20 | 0.01 | 1 | 0.0005 | 0.0 | 0.08 | const | 0 | 0 | 0 | | 3 | 1 | 5 | 2 | 1 | 4 | 1 | 0 | 1 | 1.00 | 0 | 8 | 8 |
| 0.111277251 | 0.011127725 | 5.157341515 | 0.515734149 | 1.324744749 | 1240.767188 | 0.00025 | 20 | 0.02 | 1 | 0.0005 | 0.0 | 0.04 | const | 0 | 0 | 0 | | 3 | 1 | 5 | 2 | 1 | 4 | 1 | 0 | 2 | 1.50 | 0 | 8 | 8 |
| 0.004473344 | 0.000447334 | 0.803012078 | 0.080301208 | 1.290382385 | 1240.726563 | 0.0005 | 20 | 0.01 | 1 | 0.0005 | 0.8 | 0.08 | harmonic | 0 | 0 | 0 | 0.1 | 3 | 1 | 5 | 2 | 2 | 4 | 1 | 1 | 1 | 1.00 | 0 | 8 | 8 |
| 0.041325395 | 0.004132539 | 4.417075988 | 0.4417076 | 1.296269321 | 1240.81875 | 0.00025 | 20 | 0.02 | 1 | 0.0005 | 0.8 | 0.04 | harmonic | 0 | 0 | 0 | 0.05 | 3 | 1 | 5 | 2 | 2 | 4 | 1 | 1 | 2 | 1.50 | 0 | 8 | 8 |
| 0.017441394 | 0.001744139 | 0.936132263 | 0.093613227 | 1.319015312 | 1240.820313 | 0.0005 | 20 | 0.01 | 1 | 0.0005 | 0.0 | 0.08 | const | 0 | 0 | 0 | | 3 | 1 | 5 | 2 | 1 | 4 | 1 | 0 | 1 | 1.00 | 0 | 8 | 16 |
| 0.06414937 | 0.006414937 | 4.018342035 | 0.401834202 | 1.340748625 | 1240.828125 | 0.00025 | 20 | 0.02 | 1 | 0.0005 | 0.0 | 0.04 | const | 0 | 0 | 0 | | 3 | 1 | 5 | 2 | 1 | 4 | 1 | 0 | 2 | 1.50 | 0 | 8 | 16 |
| 0.009716086 | 0.000971609 | 0.649623368 | 0.064962337 | 1.296413422 | 1240.527344 | 0.0005 | 20 | 0.01 | 1 | 0.0005 | 0.8 | 0.08 | harmonic | 0 | 0 | 0 | 0.1 | 3 | 1 | 5 | 2 | 2 | 4 | 1 | 1 | 1 | 1.00 | 0 | 8 | 16 |
| 0.043877613 | 0.004387761 | 3.711383419 | 0.371138343 | 1.315366379 | 1240.714063 | 0.00025 | 20 | 0.02 | 1 | 0.0005 | 0.8 | 0.04 | harmonic | 0 | 0 | 0 | 0.05 | 3 | 1 | 5 | 2 | 2 | 4 | 1 | 1 | 2 | 1.50 | 0 | 8 | 16 |
| 0.030784313 | 0.003078431 | 1.507336224 | 0.150733623 | 1.329183865 | 1240.753125 | 0.0005 | 20 | 0.01 | 1 | 0.0005 | 0.0 | 0.08 | const | 0 | 0 | 0 | | 3 | 1 | 5 | 1 | 1 | 4 | 2 | 0 | 1 | 1.00 | 0 | 8 | 8 |
| 0.123046156 | 0.012304615 | 5.44876127 | 0.544876131 | 1.360421801 | 1240.784375 | 0.00025 | 20 | 0.02 | 1 | 0.0005 | 0.0 | 0.04 | const | 0 | 0 | 0 | | 3 | 1 | 5 | 1 | 1 | 4 | 2 | 0 | 2 | 1.50 | 0 | 8 | 8 |
| 0.004515423 | 0.000451542 | 0.875367918 | 0.087536792 | 1.295195246 | 1240.697656 | 0.0005 | 20 | 0.01 | 1 | 0.0005 | 0.8 | 0.08 | harmonic | 0 | 0 | 0 | 0.1 | 3 | 1 | 5 | 1 | 2 | 4 | 2 | 1 | 1 | 1.00 | 0 | 8 | 8 |
| 0.048693151 | 0.004869315 | 4.486799394 | 0.448679939 | 1.313542891 | 1240.774219 | 0.00025 | 20 | 0.02 | 1 | 0.0005 | 0.8 | 0.04 | harmonic | 0 | 0 | 0 | 0.05 | 3 | 1 | 5 | 1 | 2 | 4 | 2 | 1 | 2 | 1.50 | 0 | 8 | 8 |
| 0.020697407 | 0.002069741 | 1.131707378 | 0.113170738 | 1.325917721 | 1240.802344 | 0.0005 | 20 | 0.01 | 1 | 0.0005 | 0.0 | 0.08 | const | 0 | 0 | 0 | | 3 | 1 | 5 | 1 | 1 | 4 | 2 | 0 | 1 | 1.00 | 0 | 8 | 16 |
| 0.075014925 | 0.007501493 | 4.207443076 | 0.420744306 | 1.378143406 | 1240.845313 | 0.00025 | 20 | 0.02 | 1 | 0.0005 | 0.0 | 0.04 | const | 0 | 0 | 0 | | 3 | 1 | 5 | 1 | 1 | 4 | 2 | 0 | 2 | 1.50 | 0 | 8 | 16 |
| 0.00895818 | 0.000895818 | 0.68846804 | 0.068846804 | 1.240752344 | 1240.752344 | 0.0005 | 20 | 0.01 | 1 | 0.0005 | 0.8 | 0.08 | harmonic | 0 | 0 | 0 | 0.1 | 3 | 1 | 5 | 1 | 2 | 4 | 2 | 1 | 1 | 1.00 | 0 | 8 | 16 |
| 0.05003511 | 0.005003511 | 3.841033648 | 0.384103367 | 1.298640156 | 1240.776563 | 0.00025 | 20 | 0.02 | 1 | 0.0005 | 0.8 | 0.04 | harmonic | 0 | 0 | 0 | 0.05 | 3 | 1 | 5 | 1 | 2 | 4 | 2 | 1 | 2 | 1.50 | 0 | 8 | 16 |
| 0.030784313 | 0.003078431 | 1.507336224 | 0.150733623 | 1.321631813 | 1240.791406 | 0.0005 | 20 | 0.01 | 1 | 0.0005 | 0.0 | 0.08 | const | 0 | 0 | 0 | | 3 | 1 | 5 | 2 | 1 | 4 | 2 | 0 | 1 | 1.00 | 0 | 8 | 8 |
| 0.123046156 | 0.012304615 | 5.44876127 | 0.544876131 | 1.335440922 | 1240.84375 | 0.00025 | 20 | 0.02 | 1 | 0.0005 | 0.0 | 0.04 | const | 0 | 0 | 0 | | 3 | 1 | 5 | 2 | 1 | 4 | 2 | 0 | 2 | 1.50 | 0 | 8 | 8 |
| 0.004515423 | 0.000451542 | 0.875367918 | 0.087536792 | 1.29258728 | 1240.628906 | 0.0005 | 20 | 0.01 | 1 | 0.0005 | 0.8 | 0.08 | harmonic | 0 | 0 | 0 | 0.1 | 3 | 1 | 5 | 2 | 2 | 4 | 2 | 1 | 1 | 1.00 | 0 | 8 | 8 |
| 0.048693151 | 0.004869315 | 4.486799394 | 0.448679939 | 1.312389851 | 1240.721094 | 0.00025 | 20 | 0.02 | 1 | 0.0005 | 0.8 | 0.04 | harmonic | 0 | 0 | 0 | 0.05 | 3 | 1 | 5 | 2 | 2 | 4 | 2 | 1 | 2 | 1.50 | 0 | 8 | 8 |
| 0.020697407 | 0.002069741 | 1.131707378 | 0.113170738 | 1.324503088 | 1240.732813 | 0.0005 | 20 | 0.01 | 1 | 0.0005 | 0.0 | 0.08 | const | 0 | 0 | 0 | | 3 | 1 | 5 | 2 | 1 | 4 | 2 | 0 | 1 | 1.00 | 0 | 8 | 16 |
| 0.075014925 | 0.007501493 | 4.207443076 | 0.420744306 | 1.324836719 | 1240.836719 | 0.00025 | 20 | 0.02 | 1 | 0.0005 | 0.0 | 0.04 | const | 0 | 0 | 0 | | 3 | 1 | 5 | 2 | 1 | 4 | 2 | 0 | 2 | 1.50 | 0 | 8 | 16 |
| 0.00895818 | 0.000895818 | 0.68846804 | 0.068846804 | 1.28160634 | 1240.70625 | 0.0005 | 20 | 0.01 | 1 | 0.0005 | 0.8 | 0.08 | harmonic | 0 | 0 | 0 | 0.1 | 3 | 1 | 5 | 2 | 2 | 4 | 2 | 1 | 1 | 1.00 | 0 | 8 | 16 |
| 0.05003511 | 0.005003511 | 3.841033648 | 0.384103367 | 1.299155807 | 1240.678125 | 0.00025 | 20 | 0.02 | 1 | 0.0005 | 0.8 | 0.04 | harmonic | 0 | 0 | 0 | 0.05 | 3 | 1 | 5 | 2 | 2 | 4 | 2 | 1 | 2 | 1.50 | 0 | 8 | 16 |
| 0.001822251 | 0.000182225 | 0.172409339 | 0.017240934 | 2.357672119 | 1240.786719 | 0.0005 | 40 | 0.01 | 1 | 0.0005 | 0.0 | 0.08 | const | 0 | 0 | 0 | | 3 | 1 | 5 | 1 | 1 | 2 | 1 | 0 | 1 | 1.00 | 0 | 8 | 8 |
| 0.018233484 | 0.001823348 | 1.445250869 | 0.144525087 | 2.339102983 | 1240.735156 | 0.00025 | 40 | 0.02 | 1 | 0.0005 | 0.0 | 0.04 | const | 0 | 0 | 0 | | 3 | 1 | 5 | 1 | 1 | 2 | 1 | 0 | 2 | 1.50 | 0 | 8 | 8 |
| 0.000245363 | 2.45e-05 | 0.084534858 | 0.008453485 | 2.37459836 | 1240.775 | 0.0005 | 40 | 0.01 | 1 | 0.0005 | 0.8 | 0.08 | harmonic | 0 | 0 | 0 | 0.1 | 3 | 1 | 5 | 1 | 2 | 2 | 1 | 1 | 1 | 1.00 | 0 | 8 | 8 |
| 0.010252095 | 0.00102521 | 1.303504346 | 0.130350432 | 2.332586718 | 1240.85 | 0.00025 | 40 | 0.02 | 1 | 0.0005 | 0.8 | 0.04 | harmonic | 0 | 0 | 0 | 0.05 | 3 | 1 | 5 | 1 | 2 | 2 | 1 | 1 | 2 | 1.50 | 0 | 8 | 8 |
| 0.010503508 | 0.001050351 | 0.400124063 | 0.040012406 | 2.334543324 | 1240.785156 | 0.0005 | 40 | 0.01 | 1 | 0.0005 | 0.0 | 0.08 | const | 0 | 0 | 0 | | 3 | 1 | 5 | 1 | 1 | 2 | 1 | 0 | 1 | 1.00 | 0 | 8 | 16 |
| 0.030913027 | 0.003091303 | 1.716899273 | 0.171689926 | 2.335034132 | 1240.686719 | 0.00025 | 40 | 0.02 | 1 | 0.0005 | 0.0 | 0.04 | const | 0 | 0 | 0 | | 3 | 1 | 5 | 1 | 1 | 2 | 1 | 0 | 2 | 1.50 | 0 | 8 | 16 |
| 0.00169102 | 0.000169102 | 0.103430388 | 0.010343039 | 2.331186533 | 1240.553906 | 0.0005 | 40 | 0.01 | 1 | 0.0005 | 0.8 | 0.08 | harmonic | 0 | 0 | 0 | 0.1 | 3 | 1 | 5 | 1 | 2 | 2 | 1 | 1 | 1 | 1.00 | 0 | 8 | 16 |
| 0.012002162 | 0.001200216 | 1.105425272 | 0.110542528 | 2.316781712 | 1240.792969 | 0.00025 | 40 | 0.02 | 1 | 0.0005 | 0.8 | 0.04 | harmonic | 0 | 0 | 0 | 0.05 | 3 | 1 | 5 | 1 | 2 | 2 | 1 | 1 | 2 | 1.50 | 0 | 8 | 16 |
| 0.001822251 | 0.000182225 | 0.172409339 | 0.017240934 | 2.387402058 | 1240.771875 | 0.0005 | 40 | 0.01 | 1 | 0.0005 | 0.0 | 0.08 | const | 0 | 0 | 0 | | 3 | 1 | 5 | 2 | 1 | 2 | 1 | 0 | 1 | 1.00 | 0 | 8 | 8 |
| 0.018233484 | 0.001823348 | 1.445250869 | 0.144525087 | 2.322560787 | 1240.557813 | 0.00025 | 40 | 0.02 | 1 | 0.0005 | 0.0 | 0.04 | const | 0 | 0 | 0 | | 3 | 1 | 5 | 2 | 1 | 2 | 1 | 0 | 2 | 1.50 | 0 | 8 | 8 |
| 0.000245363 | 2.45e-05 | 0.084534858 | 0.008453485 | 2.35093627 | 1240.852344 | 0.0005 | 40 | 0.01 | 1 | 0.0005 | 0.8 | 0.08 | harmonic | 0 | 0 | 0 | 0.1 | 3 | 1 | 5 | 2 | 2 | 2 | 1 | 1 | 1 | 1.00 | 0 | 8 | 8 |
| 0.010252095 | 0.00102521 | 1.303504346 | 0.130350432 | 2.369064522 | 1240.839063 | 0.00025 | 40 | 0.02 | 1 | 0.0005 | 0.8 | 0.04 | harmonic | 0 | 0 | 0 | 0.05 | 3 | 1 | 5 | 2 | 2 | 2 | 1 | 1 | 2 | 1.50 | 0 | 8 | 8 |
| 0.010503508 | 0.001050351 | 0.400124063 | 0.040012406 | 2.332673883 | 1240.830469 | 0.0005 | 40 | 0.01 | 1 | 0.0005 | 0.0 | 0.08 | const | 0 | 0 | 0 | | 3 | 1 | 5 | 2 | 1 | 2 | 1 | 0 | 1 | 1.00 | 0 | 8 | 16 |
| 0.030913027 | 0.003091303 | 1.716899273 | 0.171689926 | 2.3455616 | 1240.785938 | 0.00025 | 40 | 0.02 | 1 | 0.0005 | 0.0 | 0.04 | const | 0 | 0 | 0 | | 3 | 1 | 5 | 2 | 1 | 2 | 1 | 0 | 2 | 1.50 | 0 | 8 | 16 |
| 0.00169102 | 0.000169102 | 0.103430388 | 0.010343039 | 2.303474236 | 1240.798438 | 0.0005 | 40 | 0.01 | 1 | 0.0005 | 0.8 | 0.08 | harmonic | 0 | 0 | 0 | 0.1 | 3 | 1 | 5 | 2 | 2 | 2 | 1 | 1 | 1 | 1.00 | 0 | 8 | 16 |
| 0.012002162 | 0.001200216 | 1.105425272 | 0.110542528 | 2.330500165 | 1240.639844 | 0.00025 | 40 | 0.02 | 1 | 0.0005 | 0.8 | 0.04 | harmonic | 0 | 0 | 0 | 0.05 | 3 | 1 | 5 | 2 | 2 | 2 | 1 | 1 | 2 | 1.50 | 0 | 8 | 16 |
| 0.006682226 | 0.000668223 | 0.315112181 | 0.031511218 | 2.385132074 | 1241.5875 | 0.0005 | 40 | 0.01 | 1 | 0.0005 | 0.0 | 0.08 | const | 0 | 0 | 0 | | 3 | 1 | 5 | 1 | 1 | 2 | 2 | 0 | 1 | 1.00 | 0 | 8 | 8 |
| 0.025486331 | 0.002548633 | 1.723243252 | 0.172324327 | 2.37188077 | 1240.865625 | 0.00025 | 40 | 0.02 | 1 | 0.0005 | 0.0 | 0.04 | const | 0 | 0 | 0 | | 3 | 1 | 5 | 1 | 1 | 2 | 2 | 0 | 2 | 1.50 | 0 | 8 | 8 |
| 0.000513562 | 5.14e-05 | 0.091702681 | 0.009170268 | 2.325033045 | 1240.753125 | 0.0005 | 40 | 0.01 | 1 | 0.0005 | 0.8 | 0.08 | harmonic | 0 | 0 | 0 | 0.1 | 3 | 1 | 5 | 1 | 2 | 2 | 2 | 1 | 1 | 1.00 | 0 | 8 | 8 |
| 0.008697222 | 0.000869722 | 1.276544507 | 0.127654448 | 2.342204618 | 1240.599219 | 0.00025 | 40 | 0.02 | 1 | 0.0005 | 0.8 | 0.04 | harmonic | 0 | 0 | 0 | 0.05 | 3 | 1 | 5 | 1 | 2 | 2 | 2 | 1 | 2 | 1.50 | 0 | 8 | 8 |
| 0.012916667 | 0.001291667 | 0.387960172 | 0.038796017 | 2.324785471 | 1240.727344 | 0.0005 | 40 | 0.01 | 1 | 0.0005 | 0.0 | 0.08 | const | 0 | 0 | 0 | | 3 | 1 | 5 | 1 | 1 | 2 | 2 | 0 | 1 | 1.00 | 0 | 8 | 16 |
| 0.031189462 | 0.003118946 | 1.559489052 | 0.155948906 | 2.366880798 | 1240.7125 | 0.00025 | 40 | 0.02 | 1 | 0.0005 | 0.0 | 0.04 | const | 0 | 0 | 0 | | 3 | 1 | 5 | 1 | 1 | 2 | 2 | 0 | 2 | 1.50 | 0 | 8 | 16 |
| 0.002724909 | 0.000272491 | 0.117010106 | 0.011701011 | 2.305494022 | 1240.871875 | 0.0005 | 40 | 0.01 | 1 | 0.0005 | 0.8 | 0.08 | harmonic | 0 | 0 | 0 | 0.1 | 3 | 1 | 5 | 1 | 2 | 2 | 2 | 1 | 1 | 1.00 | 0 | 8 | 16 |
| 0.019667632 | 0.001966763 | 1.234394693 | 0.12343947 | 2.265299177 | 1240.741406 | 0.00025 | 40 | 0.02 | 1 | 0.0005 | 0.8 | 0.04 | harmonic | 0 | 0 | 0 | 0.05 | 3 | 1 | 5 | 1 | 2 | 2 | 2 | 1 | 2 | 1.50 | 0 | 8 | 16 |
| 0.006682226 | 0.000668223 | 0.315112181 | 0.031511218 | 2.417394686 | 1240.590625 | 0.0005 | 40 | 0.01 | 1 | 0.0005 | 0.0 | 0.08 | const | 0 | 0 | 0 | | 3 | 1 | 5 | 2 | 1 | 2 | 2 | 0 | 1 | 1.00 | 0 | 8 | 8 |
| 0.025486331 | 0.002548633 | 1.723243252 | 0.172324327 | 2.322... | 1240.888281 | 0.00025 | 40 | 0.02 | 1 | 0.0005 | 0.0 | 0.04 | const | 0 | 0 | 0 | | 3 | 1 | 5 | 2 | 1 | 2 | 2 | 0 | 2 | 1.50 | 0 | 8 | 8 |
| 0.000513562 | 5.14e-05 | 0.091702681 | 0.009170268 | 2.322550297 | 1240.583594 | 0.0005 | 40 | 0.01 | 1 | 0.0005 | 0.8 | 0.08 | harmonic | 0 | 0 | 0 | 0.1 | 3 | 1 | 5 | 2 | 2 | 2 | 2 | 1 | 1 | 1.00 | 0 | 8 | 8 |
| 0.008697222 | 0.000869722 | 1.276544507 | 0.127654448 | 2.306919479 | 1240.950781 | 0.00025 | 40 | 0.02 | 1 | 0.0005 | 0.8 | 0.04 | harmonic | 0 | 0 | 0 | 0.05 | 3 | 1 | 5 | 2 | 2 | 2 | 2 | 1 | 2 | 1.50 | 0 | 8 | 8 |
| 0.012916667 | 0.001291667 | 0.387960172 | 0.038796017 | 2.33807683 | 1241.0 | 0.0005 | 40 | 0.01 | 1 | 0.0005 | 0.0 | 0.08 | const | 0 | 0 | 0 | | 3 | 1 | 5 | 2 | 1 | 2 | 2 | 0 | 1 | 1.00 | 0 | 8 | 16 |

| welfare_last | coop_last | welfare_auc | coop_auc | time_total_sec_avg | ram_peak_mb_avg | abr_lr | abr_steps | beta_kl | cost_c | delta0 | ema | eta | eta_sched | hetero | lambda_jac | log_ucb | mult_r | mwu_grad_cap | no_plots | num_seeds | oracle_nz | oracle_period | pool_mut | pool_rand | slow | slowdown | tau | top_seed | ucb_nz | zdim |
|---|---|---|---|---|---|---|---|---|---|---|---|---|---|---|---|---|---|---|---|---|---|---|---|---|---|---|---|---|---|---|
| 0.031189462 | 0.003118946 | 1.559489052 | 0.155948906 | 2.388987923 | 1240.896094 | 0.00025 | 40 | 0.02 | 1 | 0.0005 | 0.0 | 0.04 | const | 0 | 0 | 0 | 3 |  | 1 | 5 | 2 | 1 | 2 | 2 | 0 | 2 | 1.5 | 0 | 8 | 16 |
| 0.002724909 | 0.000272491 | 0.117010106 | 0.011701011 | 2.299126863 | 1240.898438 | 0.0005 | 40 | 0.01 | 1 | 0.0005 | 0.8 | 0.08 | harmonic | 0 | 0 | 0 | 3 | 0.1 | 1 | 5 | 2 | 2 | 2 | 2 | 1 | 1 | 1.0 | 0 | 8 | 16 |
| 0.019667632 | 0.001966763 | 1.234394693 | 0.12343947 | 2.294810247 | 1240.71875 | 0.00025 | 40 | 0.02 | 1 | 0.0005 | 0.8 | 0.04 | harmonic | 0 | 0 | 0 | 3 | 0.05 | 1 | 5 | 2 | 2 | 2 | 2 | 1 | 2 | 1.5 | 0 | 8 | 16 |
| 0.007332874 | 0.000733287 | 0.285362167 | 0.028536217 | 2.389500189 | 1241.516406 | 0.0005 | 40 | 0.01 | 1 | 0.0005 | 0.0 | 0.08 | const | 0 | 0 | 0 | 3 |  | 1 | 5 | 1 | 1 | 4 | 1 | 0 | 1 | 1.00 | 0 | 8 | 8 |
| 0.03894474 | 0.003894474 | 1.836211315 | 0.18362113 | 2.413767529 | 1240.792188 | 0.00025 | 40 | 0.02 | 1 | 0.0005 | 0.0 | 0.04 | const | 0 | 0 | 0 | 3 |  | 1 | 5 | 1 | 1 | 4 | 1 | 0 | 2 | 1.5 | 0 | 8 | 8 |
| 0.000533205 | 5.33e-05 | 0.087070911 | 0.008707091 | 2.328015566 | 1241.722656 | 0.0005 | 40 | 0.01 | 1 | 0.0005 | 0.8 | 0.08 | harmonic | 0 | 0 | 0 | 3 | 0.1 | 1 | 5 | 1 | 2 | 4 | 1 | 1 | 1 | 1.0 | 0 | 8 | 8 |
| 0.008562821 | 0.000856282 | 1.201094464 | 0.120109444 | 2.327522945 | 1240.875781 | 0.00025 | 40 | 0.02 | 1 | 0.0005 | 0.8 | 0.04 | harmonic | 0 | 0 | 0 | 3 | 0.05 | 1 | 5 | 1 | 2 | 4 | 1 | 1 | 2 | 1.5 | 0 | 8 | 8 |
| 0.004823769 | 0.000482377 | 0.222542943 | 0.022254294 | 2.311411142 | 1240.721094 | 0.0005 | 40 | 0.01 | 1 | 0.0005 | 0.0 | 0.08 | const | 0 | 0 | 0 | 3 |  | 1 | 5 | 1 | 1 | 4 | 1 | 0 | 1 | 1.00 | 0 | 8 | 16 |
| 0.031591403 | 0.00315914 | 1.51600726 | 0.151600727 | 2.387298203 | 1240.904688 | 0.00025 | 40 | 0.02 | 1 | 0.0005 | 0.0 | 0.04 | const | 0 | 0 | 0 | 3 |  | 1 | 5 | 1 | 1 | 4 | 1 | 0 | 2 | 1.5 | 0 | 8 | 16 |
| 0.001338041 | 0.000133804 | 0.084246678 | 0.008424668 | 2.302635479 | 1240.71875 | 0.0005 | 40 | 0.01 | 1 | 0.0005 | 0.8 | 0.08 | harmonic | 0 | 0 | 0 | 3 | 0.1 | 1 | 5 | 1 | 2 | 4 | 1 | 1 | 1 | 1.0 | 0 | 8 | 16 |
| 0.015348874 | 0.001534887 | 1.07274501 | 0.107274502 | 2.326659441 | 1240.774219 | 0.00025 | 40 | 0.02 | 1 | 0.0005 | 0.8 | 0.04 | harmonic | 0 | 0 | 0 | 3 | 0.05 | 1 | 5 | 1 | 2 | 4 | 1 | 1 | 2 | 1.5 | 0 | 8 | 16 |
| 0.007332874 | 0.000733287 | 0.285362167 | 0.028536217 | 2.397288656 | 1241.660156 | 0.0005 | 40 | 0.01 | 1 | 0.0005 | 0.0 | 0.08 | const | 0 | 0 | 0 | 3 |  | 1 | 5 | 2 | 1 | 4 | 1 | 0 | 1 | 1.00 | 0 | 8 | 8 |
| 0.03894474 | 0.003894474 | 1.836211315 | 0.18362113 | 2.382192755 | 1241.624219 | 0.00025 | 40 | 0.02 | 1 | 0.0005 | 0.0 | 0.04 | const | 0 | 0 | 0 | 3 |  | 1 | 5 | 2 | 1 | 4 | 1 | 0 | 2 | 1.5 | 0 | 8 | 8 |
| 0.000533205 | 5.33e-05 | 0.087070911 | 0.008707091 | 2.334634495 | 1240.799219 | 0.0005 | 40 | 0.01 | 1 | 0.0005 | 0.8 | 0.08 | harmonic | 0 | 0 | 0 | 3 | 0.1 | 1 | 5 | 2 | 2 | 4 | 1 | 1 | 1 | 1.0 | 0 | 8 | 8 |
| 0.008562821 | 0.000856282 | 1.201094464 | 0.120109444 | 2.31390729 | 1241.557813 | 0.00025 | 40 | 0.02 | 1 | 0.0005 | 0.8 | 0.04 | harmonic | 0 | 0 | 0 | 3 | 0.05 | 1 | 5 | 2 | 2 | 4 | 1 | 1 | 2 | 1.5 | 0 | 8 | 8 |
| 0.004823769 | 0.000482377 | 0.222542943 | 0.022254294 | 2.310717188 | 1240.717188 | 0.0005 | 40 | 0.01 | 1 | 0.0005 | 0.0 | 0.08 | const | 0 | 0 | 0 | 3 |  | 1 | 5 | 2 | 1 | 4 | 1 | 0 | 1 | 1.00 | 0 | 8 | 16 |
| 0.031591403 | 0.00315914 | 1.51600726 | 0.151600727 | 2.371254349 | 1240.785938 | 0.00025 | 40 | 0.02 | 1 | 0.0005 | 0.0 | 0.04 | const | 0 | 0 | 0 | 3 |  | 1 | 5 | 2 | 1 | 4 | 1 | 0 | 2 | 1.5 | 0 | 8 | 16 |
| 0.001338041 | 0.000133804 | 0.084246678 | 0.008424668 | 2.333894444 | 1240.533594 | 0.0005 | 40 | 0.01 | 1 | 0.0005 | 0.8 | 0.08 | harmonic | 0 | 0 | 0 | 3 | 0.1 | 1 | 5 | 2 | 2 | 4 | 1 | 1 | 1 | 1.0 | 0 | 8 | 16 |
| 0.015348874 | 0.001534887 | 1.07274501 | 0.107274502 | 2.265339231 | 1240.628906 | 0.00025 | 40 | 0.02 | 1 | 0.0005 | 0.8 | 0.04 | harmonic | 0 | 0 | 0 | 3 | 0.05 | 1 | 5 | 2 | 2 | 4 | 1 | 1 | 2 | 1.5 | 0 | 8 | 16 |
| 0.01089445 | 0.001089445 | 0.453577777 | 0.045357777 | 2.382671642 | 1240.747656 | 0.0005 | 40 | 0.01 | 1 | 0.0005 | 0.0 | 0.08 | const | 0 | 0 | 0 | 3 |  | 1 | 5 | 1 | 1 | 4 | 2 | 0 | 1 | 1.00 | 0 | 8 | 8 |
| 0.047034564 | 0.004703456 | 2.075872362 | 0.207587238 | 2.319334459 | 1240.703906 | 0.00025 | 40 | 0.02 | 1 | 0.0005 | 0.0 | 0.04 | const | 0 | 0 | 0 | 3 |  | 1 | 5 | 1 | 1 | 4 | 2 | 0 | 2 | 1.5 | 0 | 8 | 8 |
| 0.000278009 | 2.78e-05 | 0.088678933 | 0.008867893 | 2.290198708 | 1240.641406 | 0.0005 | 40 | 0.01 | 1 | 0.0005 | 0.8 | 0.08 | harmonic | 0 | 0 | 0 | 3 | 0.1 | 1 | 5 | 1 | 2 | 4 | 2 | 1 | 1 | 1.0 | 0 | 8 | 8 |
| 0.00825585 | 0.000825585 | 1.284150191 | 0.128415016 | 2.280206776 | 1240.852344 | 0.00025 | 40 | 0.02 | 1 | 0.0005 | 0.8 | 0.04 | harmonic | 0 | 0 | 0 | 3 | 0.05 | 1 | 5 | 1 | 2 | 4 | 2 | 1 | 2 | 1.5 | 0 | 8 | 8 |
| 0.008221407 | 0.000822141 | 0.366433837 | 0.036643384 | 2.366846514 | 1240.738281 | 0.0005 | 40 | 0.01 | 1 | 0.0005 | 0.0 | 0.08 | const | 0 | 0 | 0 | 3 |  | 1 | 5 | 1 | 1 | 4 | 2 | 0 | 1 | 1.00 | 0 | 8 | 16 |
| 0.029226536 | 0.002922654 | 1.6304086 | 0.16304086 | 2.337304449 | 1240.703906 | 0.00025 | 40 | 0.02 | 1 | 0.0005 | 0.0 | 0.04 | const | 0 | 0 | 0 | 3 |  | 1 | 5 | 1 | 1 | 4 | 2 | 0 | 2 | 1.5 | 0 | 8 | 16 |
| 0.002771273 | 0.000277127 | 0.116460946 | 0.011646095 | 2.293221283 | 1240.6875 | 0.0005 | 40 | 0.01 | 1 | 0.0005 | 0.8 | 0.08 | harmonic | 0 | 0 | 0 | 3 | 0.1 | 1 | 5 | 1 | 2 | 4 | 2 | 1 | 1 | 1.0 | 0 | 8 | 16 |
| 0.012767993 | 0.001276799 | 1.113148995 | 0.1113149 | 2.383634329 | 1240.6625 | 0.00025 | 40 | 0.02 | 1 | 0.0005 | 0.8 | 0.04 | harmonic | 0 | 0 | 0 | 3 | 0.05 | 1 | 5 | 1 | 2 | 4 | 2 | 1 | 2 | 1.5 | 0 | 8 | 16 |

. . . . . . . . . . . . . . . . . . . . . . . . . . . . . . . . . . . . . . . . . . . . . . . . . . . . . . . . . . . . . . . . . . . . . . . . . . . . . . . . . . . . . . . . . . . . . . . . . .

# R   Ablation on Deceptive Mean

We conducted an ablation study for the Deceptive Message game to analyze the sensitivity of GEMS to its core hyperparameters. Our investigation focused on key parameters governing the Amortized Best-Response (ABR) training, including the learning rate (`abr_lr`), update steps (`abr_steps`), and KL-divergence coefficient (`beta_kl`), as well as the generator's latent dimension (`zdim`) and the Jacobian regularization penalty (`lambda_jac`).

Table 8: Ablation (Deceptive Mean)

| sender_last | receiver_last | sender_auc | receiver_auc | time_total_sec_mu | ram_peak_mb_mu | abr_lr | abr_steps | adv_ema | beta_kl | inner_eval | lambda_jac | meta_eval_n | zdim |
|---|---|---|---|---|---|---|---|---|---|---|---|---|---|
| 0.004882813 | 0.750976563 | 0.213747387 | 3.637925569 | 3.797561526 | 766.8417969 | 0.001 | 100.0 | 0.05 | 0.01 | 2.0 | 0.0 | 32.0 | 8.0 |
| 0.000976563 | 0.778320313 | 0.167490832 | 3.677008441 | 3.490059376 | 766.7949219 | 0.001 | 100.0 | 0.05 | 0.01 | 2.0 | 0.0 | 32.0 | 16.0 |
| 0.00390625 | 0.752929688 | 0.208329613 | 3.636479725 | 3.519120216 | 766.9082031 | 0.001 | 100.0 | 0.1 | 0.01 | 2.0 | 0.0 | 32.0 | 8.0 |
| 0.001953125 | 0.7734375 | 0.151025457 | 3.691782171 | 3.478062987 | 767.0566406 | 0.001 | 100.0 | 0.1 | 0.01 | 2.0 | 0.0 | 32.0 | 16.0 |
| 0.004882813 | 0.750976563 | 0.213747387 | 3.637925569 | 3.465744257 | 766.8632813 | 0.001 | 100.0 | 0.05 | 0.01 | 4.0 | 0.0 | 32.0 | 8.0 |
| 0.000976563 | 0.778320313 | 0.167490832 | 3.677008441 | 3.471985459 | 767.0722656 | 0.001 | 100.0 | 0.05 | 0.01 | 4.0 | 0.0 | 32.0 | 16.0 |
| 0.00390625 | 0.752929688 | 0.208329613 | 3.636479725 | 3.519359112 | 766.9570313 | 0.001 | 100.0 | 0.1 | 0.01 | 4.0 | 0.0 | 32.0 | 8.0 |
| 0.001953125 | 0.7734375 | 0.151025457 | 3.691782171 | 3.539175749 | 766.9375 | 0.001 | 100.0 | 0.1 | 0.01 | 4.0 | 0.0 | 32.0 | 16.0 |
| 0.005859375 | 0.76171875 | 0.187598763 | 3.494975376 | 4.116945505 | 766.8417969 | 0.001 | 100.0 | 0.05 | 0.01 | 2.0 | 0.0 | 64.0 | 8.0 |
| 0.005859375 | 0.766601563 | 0.200174231 | 3.545335929 | 4.110057473 | 766.9746094 | 0.001 | 100.0 | 0.05 | 0.01 | 2.0 | 0.0 | 64.0 | 16.0 |
| 0.004882813 | 0.765625 | 0.176902149 | 3.508334751 | 4.123156428 | 766.8632813 | 0.001 | 100.0 | 0.1 | 0.01 | 2.0 | 0.0 | 64.0 | 8.0 |
| 0.0078125 | 0.762695313 | 0.192496811 | 3.535875717 | 4.120264649 | 766.8925781 | 0.001 | 100.0 | 0.1 | 0.01 | 2.0 | 0.0 | 64.0 | 16.0 |
| 0.005859375 | 0.76171875 | 0.187598763 | 3.494975376 | 4.084610581 | 766.9765625 | 0.001 | 100.0 | 0.05 | 0.01 | 4.0 | 0.0 | 64.0 | 8.0 |
| 0.005859375 | 0.766601563 | 0.200174231 | 3.545335929 | 4.109117508 | 766.9492188 | 0.001 | 100.0 | 0.05 | 0.01 | 4.0 | 0.0 | 64.0 | 16.0 |
| 0.004882813 | 0.765625 | 0.176902149 | 3.508334751 | 4.129849672 | 766.8691406 | 0.001 | 100.0 | 0.1 | 0.01 | 4.0 | 0.0 | 64.0 | 8.0 |
| 0.0078125 | 0.762695313 | 0.192496811 | 3.535875717 | 4.136036158 | 766.8886719 | 0.001 | 100.0 | 0.1 | 0.01 | 4.0 | 0.0 | 64.0 | 16.0 |
| 0.030273438 | 0.689453125 | 0.398964667 | 3.1106754 | 3.488319993 | 766.9394531 | 0.0005 | 100.0 | 0.05 | 0.01 | 2.0 | 0.0 | 32.0 | 8.0 |
| 0.02734375 | 0.712890625 | 0.412892042 | 3.030212674 | 3.482948065 | 766.8671875 | 0.0005 | 100.0 | 0.05 | 0.01 | 2.0 | 0.0 | 32.0 | 16.0 |
| 0.03125 | 0.6875 | 0.397256236 | 3.113605088 | 3.475835323 | 766.8925781 | 0.0005 | 100.0 | 0.1 | 0.01 | 2.0 | 0.0 | 32.0 | 8.0 |
| 0.028320313 | 0.71875 | 0.391349073 | 3.048147321 | 3.475960135 | 767.0703125 | 0.0005 | 100.0 | 0.1 | 0.01 | 2.0 | 0.0 | 32.0 | 16.0 |
| 0.030273438 | 0.689453125 | 0.398964667 | 3.1106754 | 3.494435072 | 766.8066406 | 0.0005 | 100.0 | 0.05 | 0.01 | 4.0 | 0.0 | 32.0 | 8.0 |
| 0.02734375 | 0.712890625 | 0.412892042 | 3.030212674 | 3.538249969 | 767.0585938 | 0.0005 | 100.0 | 0.05 | 0.01 | 4.0 | 0.0 | 32.0 | 16.0 |
| 0.03125 | 0.6875 | 0.397256236 | 3.113605088 | 3.502023935 | 766.8378906 | 0.0005 | 100.0 | 0.1 | 0.01 | 4.0 | 0.0 | 32.0 | 8.0 |
| 0.028320313 | 0.71875 | 0.391349073 | 3.048147321 | 3.485627413 | 766.8632813 | 0.0005 | 100.0 | 0.1 | 0.01 | 4.0 | 0.0 | 32.0 | 16.0 |
| 0.03125 | 0.703125 | 0.41984623 | 3.008728387 | 4.091207623 | 766.8574219 | 0.0005 | 100.0 | 0.05 | 0.01 | 2.0 | 0.0 | 64.0 | 8.0 |
| 0.029296875 | 0.70703125 | 0.433813333 | 3.062614619 | 4.114554763 | 767.0214844 | 0.0005 | 100.0 | 0.05 | 0.01 | 2.0 | 0.0 | 64.0 | 16.0 |
| 0.03125 | 0.705078125 | 0.41971549 | 3.019714162 | 4.117322803 | 766.9511719 | 0.0005 | 100.0 | 0.1 | 0.01 | 2.0 | 0.0 | 64.0 | 8.0 |
| 0.032226563 | 0.704101563 | 0.440840818 | 3.053851022 | 4.145797491 | 766.9238281 | 0.0005 | 100.0 | 0.1 | 0.01 | 2.0 | 0.0 | 64.0 | 16.0 |
| 0.03125 | 0.703125 | 0.41984623 | 3.008728387 | 4.09178257 | 766.828125 | 0.0005 | 100.0 | 0.05 | 0.01 | 4.0 | 0.0 | 64.0 | 8.0 |
| 0.029296875 | 0.70703125 | 0.433813333 | 3.062614619 | 4.104922652 | 766.8183594 | 0.0005 | 100.0 | 0.05 | 0.01 | 4.0 | 0.0 | 64.0 | 16.0 |
| 0.03125 | 0.705078125 | 0.41971549 | 3.019714162 | 4.11270225 | 766.8984375 | 0.0005 | 100.0 | 0.1 | 0.01 | 4.0 | 0.0 | 64.0 | 8.0 |
| 0.032226563 | 0.704101563 | 0.440840818 | 3.053851022 | 4.102519035 | 766.8339844 | 0.0005 | 100.0 | 0.1 | 0.01 | 4.0 | 0.0 | 64.0 | 16.0 |
| 0.002929688 | 0.783203125 | 0.081438271 | 3.804808408 | 4.592607737 | 766.9003906 | 0.001 | 200.0 | 0.05 | 0.01 | 2.0 | 0.0 | 32.0 | 8.0 |
| 0.004882813 | 0.779296875 | 0.103168447 | 3.670696127 | 4.610334158 | 766.8359375 | 0.001 | 200.0 | 0.05 | 0.01 | 2.0 | 0.0 | 32.0 | 16.0 |
| 0.006835938 | 0.78125 | 0.078946951 | 3.813918651 | 4.574136138 | 766.8300781 | 0.001 | 200.0 | 0.1 | 0.01 | 2.0 | 0.0 | 32.0 | 8.0 |
| 0.004882813 | 0.778320313 | 0.094999513 | 3.680365159 | 4.598544359 | 766.796875 | 0.001 | 200.0 | 0.1 | 0.01 | 2.0 | 0.0 | 32.0 | 16.0 |
| 0.002929688 | 0.783203125 | 0.081438271 | 3.804808408 | 4.564905047 | 766.8046875 | 0.001 | 200.0 | 0.05 | 0.01 | 4.0 | 0.0 | 32.0 | 8.0 |
| 0.004882813 | 0.779296875 | 0.103168447 | 3.670696127 | 4.548526525 | 766.9375 | 0.001 | 200.0 | 0.05 | 0.01 | 4.0 | 0.0 | 32.0 | 16.0 |
| 0.006835938 | 0.78125 | 0.078946951 | 3.813918651 | 4.592769027 | 767.0585938 | 0.001 | 200.0 | 0.1 | 0.01 | 4.0 | 0.0 | 32.0 | 8.0 |
| 0.004882813 | 0.778320313 | 0.094999513 | 3.680365159 | 4.618440151 | 766.921875 | 0.001 | 200.0 | 0.1 | 0.01 | 4.0 | 0.0 | 32.0 | 16.0 |
| 0.0 | 0.805664063 | 0.02203125 | 3.879985872 | 5.202624202 | 767.0332031 | 0.001 | 200.0 | 0.05 | 0.01 | 2.0 | 0.0 | 64.0 | 8.0 |
| 0.002929688 | 0.791992188 | 0.075109614 | 3.907348489 | 5.187786818 | 766.7890625 | 0.001 | 200.0 | 0.05 | 0.01 | 2.0 | 0.0 | 64.0 | 16.0 |
| 0.0 | 0.8046875 | 0.020729167 | 3.888508007 | 5.156398416 | 766.8984375 | 0.001 | 200.0 | 0.1 | 0.01 | 2.0 | 0.0 | 64.0 | 8.0 |
| 0.002929688 | 0.791992188 | 0.079345725 | 3.900021037 | 5.170324206 | 766.9003906 | 0.001 | 200.0 | 0.1 | 0.01 | 2.0 | 0.0 | 64.0 | 16.0 |
| 0.0 | 0.805664063 | 0.02203125 | 3.879985872 | 5.196808338 | 766.9003906 | 0.001 | 200.0 | 0.05 | 0.01 | 4.0 | 0.0 | 64.0 | 8.0 |
| 0.002929688 | 0.791992188 | 0.075109614 | 3.907348489 | 5.195022225 | 766.8632813 | 0.001 | 200.0 | 0.05 | 0.01 | 4.0 | 0.0 | 64.0 | 16.0 |
| 0.0 | 0.8046875 | 0.020729167 | 3.888508007 | 5.201624393 | 767.0410156 | 0.001 | 200.0 | 0.1 | 0.01 | 4.0 | 0.0 | 64.0 | 8.0 |
| 0.002929688 | 0.791992188 | 0.079345725 | 3.900021037 | 5.167958856 | 766.84375 | 0.001 | 200.0 | 0.1 | 0.01 | 4.0 | 0.0 | 64.0 | 16.0 |
| 0.013671875 | 0.759765625 | 0.233534138 | 3.491052739 | 4.574347615 | 766.8769531 | 0.0005 | 200.0 | 0.05 | 0.01 | 2.0 | 0.0 | 32.0 | 8.0 |
| 0.008789063 | 0.760742188 | 0.237348755 | 3.381854229 | 4.594873428 | 767.046875 | 0.0005 | 200.0 | 0.05 | 0.01 | 2.0 | 0.0 | 32.0 | 16.0 |
| 0.015625 | 0.759765625 | 0.2345107 | 3.502512135 | 4.615203023 | 766.9511719 | 0.0005 | 200.0 | 0.1 | 0.01 | 2.0 | 0.0 | 32.0 | 8.0 |
| 0.01171875 | 0.755859375 | 0.240472116 | 3.358276201 | 4.573800206 | 766.9648438 | 0.0005 | 200.0 | 0.1 | 0.01 | 2.0 | 0.0 | 32.0 | 16.0 |
| 0.013671875 | 0.759765625 | 0.233534138 | 3.491052739 | 4.576252937 | 766.8632813 | 0.0005 | 200.0 | 0.05 | 0.01 | 4.0 | 0.0 | 32.0 | 8.0 |
| 0.008789063 | 0.760742188 | 0.237348755 | 3.381854229 | 4.60927546 | 767.0957031 | 0.0005 | 200.0 | 0.05 | 0.01 | 4.0 | 0.0 | 32.0 | 16.0 |
| 0.015625 | 0.759765625 | 0.2345107 | 3.502512135 | 4.562783718 | 766.9199219 | 0.0005 | 200.0 | 0.1 | 0.01 | 4.0 | 0.0 | 32.0 | 8.0 |
| 0.01171875 | 0.755859375 | 0.240472116 | 3.358276201 | 4.630032897 | 766.8847656 | 0.0005 | 200.0 | 0.1 | 0.01 | 4.0 | 0.0 | 32.0 | 16.0 |
| 0.00390625 | 0.797851563 | 0.123591093 | 3.614750257 | 5.193202019 | 766.8613281 | 0.0005 | 200.0 | 0.05 | 0.01 | 2.0 | 0.0 | 64.0 | 8.0 |

| sender_last | receiver_last | sender_auc | receiver_auc | time_total_sec_mu | ram_peak_mb_mu | abr_lr | abr_steps | adv_ema | beta_kl | inner_eval | lambda_jac | meta_eval_n | zdim |
|---|---|---|---|---|---|---|---|---|---|---|---|---|---|
| 0.002929688 | 0.7890625 | 0.241208324 | 3.576210672 | 5.175317168 | 766.8652344 | 0.0005 | 200.0 | 0.05 | 0.01 | 2.0 | 0.0 | 64.0 | 16.0 |
| 0.00390625 | 0.796875 | 0.125327204 | 3.619904337 | 5.203407407 | 766.9238281 | 0.0005 | 200.0 | 0.1 | 0.01 | 2.0 | 0.0 | 64.0 | 8.0 |
| 0.002929688 | 0.787109375 | 0.252284713 | 3.56288221 | 5.175059795 | 766.9140625 | 0.0005 | 200.0 | 0.1 | 0.01 | 2.0 | 0.0 | 64.0 | 16.0 |
| 0.00390625 | 0.797851563 | 0.123591093 | 3.614750257 | 5.206146717 | 766.9824219 | 0.0005 | 200.0 | 0.05 | 0.01 | 4.0 | 0.0 | 64.0 | 8.0 |
| 0.002929688 | 0.7890625 | 0.241208324 | 3.576210672 | 5.198497653 | 766.8398438 | 0.0005 | 200.0 | 0.05 | 0.01 | 4.0 | 0.0 | 64.0 | 16.0 |
| 0.00390625 | 0.796875 | 0.125327204 | 3.619904337 | 5.194039464 | 766.8378906 | 0.0005 | 200.0 | 0.1 | 0.01 | 4.0 | 0.0 | 64.0 | 8.0 |
| 0.002929688 | 0.787109375 | 0.252284713 | 3.56288221 | 5.209975243 | 766.8867188 | 0.0005 | 200.0 | 0.1 | 0.01 | 4.0 | 0.0 | 64.0 | 16.0 |
| 0.004882813 | 0.750976563 | 0.209971877 | 3.645173301 | 3.484597445 | 766.8574219 | 0.001 | 100.0 | 0.05 | 0.005 | 2.0 | 0.0 | 32.0 | 8.0 |
| 0.001953125 | 0.7734375 | 0.171805644 | 3.671331845 | 3.481884599 | 766.9511719 | 0.001 | 100.0 | 0.05 | 0.005 | 2.0 | 0.0 | 32.0 | 16.0 |
| 0.00390625 | 0.75390625 | 0.200136232 | 3.649067637 | 3.479956746 | 766.9042969 | 0.001 | 100.0 | 0.1 | 0.005 | 2.0 | 0.0 | 32.0 | 8.0 |
| 0.001953125 | 0.774414063 | 0.15623379 | 3.687826008 | 3.488405466 | 766.7949219 | 0.001 | 100.0 | 0.1 | 0.005 | 2.0 | 0.0 | 32.0 | 16.0 |
| 0.004882813 | 0.750976563 | 0.209971877 | 3.645173301 | 3.495933414 | 766.8652344 | 0.001 | 100.0 | 0.05 | 0.005 | 4.0 | 0.0 | 32.0 | 8.0 |
| 0.001953125 | 0.7734375 | 0.171805644 | 3.671331845 | 3.493604779 | 766.9453125 | 0.001 | 100.0 | 0.05 | 0.005 | 4.0 | 0.0 | 32.0 | 16.0 |
| 0.00390625 | 0.75390625 | 0.200136232 | 3.649067637 | 3.50344944 | 766.8417969 | 0.001 | 100.0 | 0.1 | 0.005 | 4.0 | 0.0 | 32.0 | 8.0 |
| 0.001953125 | 0.774414063 | 0.15623379 | 3.687826008 | 3.479611635 | 766.8710938 | 0.001 | 100.0 | 0.1 | 0.005 | 4.0 | 0.0 | 32.0 | 16.0 |
| 0.005859375 | 0.764648438 | 0.185098763 | 3.501440219 | 4.079826832 | 766.8730469 | 0.001 | 100.0 | 0.05 | 0.005 | 2.0 | 0.0 | 64.0 | 8.0 |
| 0.004882813 | 0.765625 | 0.19841044 | 3.547859269 | 4.141643167 | 766.9648438 | 0.001 | 100.0 | 0.05 | 0.005 | 2.0 | 0.0 | 64.0 | 16.0 |
| 0.005859375 | 0.765625 | 0.177693718 | 3.505019841 | 4.081937313 | 766.8808594 | 0.001 | 100.0 | 0.1 | 0.005 | 2.0 | 0.0 | 64.0 | 8.0 |
| 0.0078125 | 0.760742188 | 0.192957412 | 3.55072115 | 4.103323817 | 766.8632813 | 0.001 | 100.0 | 0.1 | 0.005 | 2.0 | 0.0 | 64.0 | 16.0 |
| 0.005859375 | 0.764648438 | 0.185098763 | 3.501440219 | 4.093967438 | 766.8730469 | 0.001 | 100.0 | 0.05 | 0.005 | 4.0 | 0.0 | 64.0 | 8.0 |
| 0.004882813 | 0.765625 | 0.19841044 | 3.547859269 | 4.092059493 | 766.9082031 | 0.001 | 100.0 | 0.05 | 0.005 | 4.0 | 0.0 | 64.0 | 16.0 |
| 0.005859375 | 0.765625 | 0.177693718 | 3.505019841 | 4.076184273 | 766.9199219 | 0.001 | 100.0 | 0.1 | 0.005 | 4.0 | 0.0 | 64.0 | 8.0 |
| 0.0078125 | 0.760742188 | 0.192957412 | 3.55072115 | 4.161067963 | 767.0039063 | 0.001 | 100.0 | 0.1 | 0.005 | 4.0 | 0.0 | 64.0 | 16.0 |
| 0.029296875 | 0.688476563 | 0.393779673 | 3.103399988 | 3.47068882 | 766.890625 | 0.0005 | 100.0 | 0.05 | 0.005 | 2.0 | 0.0 | 32.0 | 8.0 |
| 0.02734375 | 0.712890625 | 0.411155931 | 3.030212674 | 3.492367029 | 766.9921875 | 0.0005 | 100.0 | 0.05 | 0.005 | 2.0 | 0.0 | 32.0 | 16.0 |
| 0.03125 | 0.6875 | 0.397716837 | 3.114880598 | 3.490564466 | 766.9101563 | 0.0005 | 100.0 | 0.1 | 0.005 | 2.0 | 0.0 | 32.0 | 8.0 |
| 0.028320313 | 0.719726563 | 0.391349073 | 3.054277964 | 3.492498398 | 766.8339844 | 0.0005 | 100.0 | 0.1 | 0.005 | 2.0 | 0.0 | 32.0 | 16.0 |
| 0.029296875 | 0.688476563 | 0.393779673 | 3.103399988 | 3.50611403 | 766.8066406 | 0.0005 | 100.0 | 0.05 | 0.005 | 4.0 | 0.0 | 32.0 | 8.0 |
| 0.02734375 | 0.712890625 | 0.411155931 | 3.030212674 | 3.485206127 | 766.7988281 | 0.0005 | 100.0 | 0.05 | 0.005 | 4.0 | 0.0 | 32.0 | 16.0 |
| 0.03125 | 0.6875 | 0.397716837 | 3.114880598 | 3.504174113 | 766.8613281 | 0.0005 | 100.0 | 0.1 | 0.005 | 4.0 | 0.0 | 32.0 | 8.0 |
| 0.028320313 | 0.719726563 | 0.391349073 | 3.054277964 | 3.48001945 | 766.8085938 | 0.0005 | 100.0 | 0.1 | 0.005 | 4.0 | 0.0 | 32.0 | 16.0 |
| 0.03125 | 0.704101563 | 0.41984623 | 3.010492179 | 4.108764052 | 766.9160156 | 0.0005 | 100.0 | 0.05 | 0.005 | 2.0 | 0.0 | 64.0 | 8.0 |
| 0.03125 | 0.705078125 | 0.431778274 | 3.064138056 | 4.202890396 | 766.9355469 | 0.0005 | 100.0 | 0.05 | 0.005 | 2.0 | 0.0 | 64.0 | 16.0 |
| 0.03125 | 0.706054688 | 0.417979379 | 3.022753463 | 4.087831497 | 766.8613281 | 0.0005 | 100.0 | 0.1 | 0.005 | 2.0 | 0.0 | 64.0 | 8.0 |
| 0.03125 | 0.704101563 | 0.440352537 | 3.052575512 | 4.13163209 | 766.9726563 | 0.0005 | 100.0 | 0.1 | 0.005 | 2.0 | 0.0 | 64.0 | 16.0 |
| 0.03125 | 0.704101563 | 0.41984623 | 3.010492179 | 4.099819779 | 766.9492188 | 0.0005 | 100.0 | 0.05 | 0.005 | 4.0 | 0.0 | 64.0 | 8.0 |
| 0.03125 | 0.705078125 | 0.431778274 | 3.064138056 | 4.112385631 | 767.0527344 | 0.0005 | 100.0 | 0.05 | 0.005 | 4.0 | 0.0 | 64.0 | 16.0 |
| 0.03125 | 0.706054688 | 0.417979379 | 3.022753463 | 4.113277912 | 766.9707031 | 0.0005 | 100.0 | 0.1 | 0.005 | 4.0 | 0.0 | 64.0 | 8.0 |
| 0.03125 | 0.704101563 | 0.440352537 | 3.052575512 | 4.142390847 | 767.0449219 | 0.0005 | 100.0 | 0.1 | 0.005 | 4.0 | 0.0 | 64.0 | 16.0 |
| 0.00390625 | 0.784179688 | 0.07845433 | 3.813977245 | 4.564417839 | 766.8300781 | 0.001 | 200.0 | 0.05 | 0.005 | 2.0 | 0.0 | 32.0 | 8.0 |
| 0.002929688 | 0.783203125 | 0.112201097 | 3.684513273 | 4.55967021 | 766.9160156 | 0.001 | 200.0 | 0.05 | 0.005 | 2.0 | 0.0 | 32.0 | 16.0 |
| 0.004882813 | 0.783203125 | 0.071694878 | 3.816934612 | 4.642746329 | 766.921875 | 0.001 | 200.0 | 0.1 | 0.005 | 2.0 | 0.0 | 32.0 | 8.0 |
| 0.004882813 | 0.779296875 | 0.093263402 | 3.684325663 | 4.574479938 | 767.0039063 | 0.001 | 200.0 | 0.1 | 0.005 | 2.0 | 0.0 | 32.0 | 16.0 |
| 0.00390625 | 0.784179688 | 0.07845433 | 3.813977245 | 4.564909458 | 766.8320313 | 0.001 | 200.0 | 0.05 | 0.005 | 4.0 | 0.0 | 32.0 | 8.0 |
| 0.002929688 | 0.783203125 | 0.112201097 | 3.684513977 | 4.595101476 | 766.9492188 | 0.001 | 200.0 | 0.05 | 0.005 | 4.0 | 0.0 | 32.0 | 16.0 |
| 0.004882813 | 0.783203125 | 0.071694878 | 3.816934612 | 4.575098038 | 766.9042969 | 0.001 | 200.0 | 0.1 | 0.005 | 4.0 | 0.0 | 32.0 | 8.0 |
| 0.004882813 | 0.779296875 | 0.093263402 | 3.684325663 | 4.580031991 | 766.9023438 | 0.001 | 200.0 | 0.1 | 0.005 | 4.0 | 0.0 | 32.0 | 16.0 |
| 0.0 | 0.806640625 | 0.020295139 | 3.88961531 | 5.265115499 | 766.9238281 | 0.001 | 200.0 | 0.05 | 0.005 | 2.0 | 0.0 | 64.0 | 8.0 |
| 0.0 | 0.793945313 | 0.07736926 | 3.900982364 | 5.19207108 | 766.8710938 | 0.001 | 200.0 | 0.05 | 0.005 | 2.0 | 0.0 | 64.0 | 16.0 |
| 0.0 | 0.8046875 | 0.016822917 | 3.885627126 | 5.201019526 | 766.9609375 | 0.001 | 200.0 | 0.1 | 0.005 | 2.0 | 0.0 | 64.0 | 8.0 |
| 0.0 | 0.793945313 | 0.065121528 | 3.921266165 | 5.193174481 | 766.8808594 | 0.001 | 200.0 | 0.1 | 0.005 | 2.0 | 0.0 | 64.0 | 16.0 |
| 0.0 | 0.806640625 | 0.020295139 | 3.88961531 | 5.191734433 | 766.9140625 | 0.001 | 200.0 | 0.05 | 0.005 | 4.0 | 0.0 | 64.0 | 8.0 |
| 0.0 | 0.793945313 | 0.07736926 | 3.900982364 | 5.205765486 | 766.9570313 | 0.001 | 200.0 | 0.05 | 0.005 | 4.0 | 0.0 | 64.0 | 16.0 |
| 0.0 | 0.8046875 | 0.016822917 | 3.885627126 | 5.192848086 | 766.8886719 | 0.001 | 200.0 | 0.1 | 0.005 | 4.0 | 0.0 | 64.0 | 8.0 |
| 0.0 | 0.793945313 | 0.065121528 | 3.921266165 | 5.187784672 | 767.0644531 | 0.001 | 200.0 | 0.1 | 0.005 | 4.0 | 0.0 | 64.0 | 16.0 |
| 0.012695313 | 0.762695313 | 0.231309745 | 3.499253694 | 4.587309361 | 766.8378906 | 0.0005 | 200.0 | 0.05 | 0.005 | 2.0 | 0.0 | 32.0 | 8.0 |
| 0.009765625 | 0.76171875 | 0.233089303 | 3.394772003 | 4.590754867 | 766.9121094 | 0.0005 | 200.0 | 0.05 | 0.005 | 2.0 | 0.0 | 32.0 | 16.0 |
| 0.012695313 | 0.76171875 | 0.234781967 | 3.486716801 | 4.591512561 | 766.9394531 | 0.0005 | 200.0 | 0.1 | 0.005 | 2.0 | 0.0 | 32.0 | 8.0 |
| 0.010742188 | 0.760742188 | 0.233577585 | 3.382513951 | 4.563710451 | 767.03125 | 0.0005 | 200.0 | 0.1 | 0.005 | 2.0 | 0.0 | 32.0 | 16.0 |
| 0.012695313 | 0.762695313 | 0.231309745 | 3.499253694 | 4.560121536 | 766.859375 | 0.0005 | 200.0 | 0.05 | 0.005 | 4.0 | 0.0 | 32.0 | 8.0 |
| 0.009765625 | 0.76171875 | 0.233089303 | 3.394772003 | 4.592621326 | 766.8652344 | 0.0005 | 200.0 | 0.05 | 0.005 | 4.0 | 0.0 | 32.0 | 16.0 |
| 0.012695313 | 0.76171875 | 0.234781967 | 3.486716801 | 4.588880777 | 766.8007813 | 0.0005 | 200.0 | 0.1 | 0.005 | 4.0 | 0.0 | 32.0 | 8.0 |
| 0.010742188 | 0.760742188 | 0.233577585 | 3.382513951 | 4.57550478 | 766.9824219 | 0.0005 | 200.0 | 0.1 | 0.005 | 4.0 | 0.0 | 32.0 | 16.0 |
| 0.00390625 | 0.797851563 | 0.121854982 | 3.611278035 | 5.235417843 | 766.8730469 | 0.0005 | 200.0 | 0.05 | 0.005 | 2.0 | 0.0 | 64.0 | 8.0 |
| 0.000976563 | 0.7890625 | 0.23594463 | 3.584682894 | 5.163270712 | 766.9960938 | 0.0005 | 200.0 | 0.05 | 0.005 | 2.0 | 0.0 | 64.0 | 16.0 |
| 0.00390625 | 0.797851563 | 0.125327204 | 3.622892618 | 5.165045619 | 766.859375 | 0.0005 | 200.0 | 0.1 | 0.005 | 2.0 | 0.0 | 64.0 | 8.0 |
| 0.001953125 | 0.788085938 | 0.239114672 | 3.573882113 | 5.201363683 | 766.8847656 | 0.0005 | 200.0 | 0.1 | 0.005 | 2.0 | 0.0 | 64.0 | 16.0 |
| 0.00390625 | 0.797851563 | 0.121854982 | 3.611278035 | 5.172132373 | 766.9941406 | 0.0005 | 200.0 | 0.05 | 0.005 | 4.0 | 0.0 | 64.0 | 8.0 |
| 0.000976563 | 0.7890625 | 0.23594463 | 3.584682894 | 5.199103951 | 766.921875 | 0.0005 | 200.0 | 0.05 | 0.005 | 4.0 | 0.0 | 64.0 | 16.0 |
| 0.00390625 | 0.797851563 | 0.125327204 | 3.622892618 | 5.173853517 | 766.9179688 | 0.0005 | 200.0 | 0.1 | 0.005 | 4.0 | 0.0 | 64.0 | 8.0 |
| 0.001953125 | 0.788085938 | 0.239114672 | 3.573882113 | 5.177394509 | 766.8203125 | 0.0005 | 200.0 | 0.1 | 0.005 | 4.0 | 0.0 | 64.0 | 16.0 |

| sender_last | receiver_last | sender_auc | receiver_auc | time_total_sec_mu | ram_peak_mb_mu | abr_lr | abr_steps | adv_ema | beta_kl | inner_eval | lambda_jac | meta_eval_n | zdim |
|---|---|---|---|---|---|---|---|---|---|---|---|---|---|
| 0.008789063 | 0.73828125 | 0.234759823 | 3.582373689 | 3.485162735 | 766.953125 | 0.001 | 100.0 | 0.05 | 0.01 | 2.0 | 0.01 | 32.0 | 8.0 |
| 0.009765625 | 0.76171875 | 0.214884584 | 3.61147658 | 3.503759861 | 766.8085938 | 0.001 | 100.0 | 0.05 | 0.01 | 2.0 | 0.01 | 32.0 | 16.0 |
| 0.015625 | 0.73046875 | 0.244402282 | 3.574820543 | 3.476381302 | 766.8730469 | 0.001 | 100.0 | 0.1 | 0.01 | 2.0 | 0.01 | 32.0 | 8.0 |
| 0.002929688 | 0.771484375 | 0.165628322 | 3.683584449 | 3.504372954 | 766.9472656 | 0.001 | 100.0 | 0.1 | 0.01 | 2.0 | 0.01 | 32.0 | 16.0 |
| 0.008789063 | 0.73828125 | 0.234759823 | 3.582373689 | 3.492276192 | 766.9433594 | 0.001 | 100.0 | 0.05 | 0.01 | 4.0 | 0.01 | 32.0 | 8.0 |
| 0.009765625 | 0.76171875 | 0.214884584 | 3.61147658 | 3.477153182 | 766.8515625 | 0.001 | 100.0 | 0.05 | 0.01 | 4.0 | 0.01 | 32.0 | 16.0 |
| 0.015625 | 0.73046875 | 0.244402282 | 3.574820543 | 3.462974429 | 766.7929688 | 0.001 | 100.0 | 0.1 | 0.01 | 4.0 | 0.01 | 32.0 | 8.0 |
| 0.002929688 | 0.771484375 | 0.165628322 | 3.683584449 | 3.494990706 | 766.9550781 | 0.001 | 100.0 | 0.1 | 0.01 | 4.0 | 0.01 | 32.0 | 16.0 |
| 0.00390625 | 0.763671875 | 0.186622201 | 3.495951938 | 4.06504941 | 766.9042969 | 0.001 | 100.0 | 0.05 | 0.01 | 2.0 | 0.01 | 64.0 | 8.0 |
| 0.005859375 | 0.762695313 | 0.190373352 | 3.543969893 | 4.111203074 | 766.9824219 | 0.001 | 100.0 | 0.05 | 0.01 | 2.0 | 0.01 | 64.0 | 16.0 |
| 0.00390625 | 0.766601563 | 0.176413868 | 3.512755855 | 4.090454459 | 766.8203125 | 0.001 | 100.0 | 0.1 | 0.01 | 2.0 | 0.01 | 64.0 | 8.0 |
| 0.0078125 | 0.752929688 | 0.21913159 | 3.464269195 | 4.129907131 | 766.9765625 | 0.001 | 100.0 | 0.1 | 0.01 | 2.0 | 0.01 | 64.0 | 16.0 |
| 0.00390625 | 0.763671875 | 0.186622201 | 3.495951938 | 4.090277076 | 766.8945313 | 0.001 | 100.0 | 0.05 | 0.01 | 4.0 | 0.01 | 64.0 | 8.0 |
| 0.005859375 | 0.762695313 | 0.190373352 | 3.543969893 | 4.139833689 | 766.9960938 | 0.001 | 100.0 | 0.05 | 0.01 | 4.0 | 0.01 | 64.0 | 16.0 |
| 0.00390625 | 0.766601563 | 0.176413868 | 3.512755855 | 4.094886661 | 766.9941406 | 0.001 | 100.0 | 0.1 | 0.01 | 4.0 | 0.01 | 64.0 | 8.0 |
| 0.0078125 | 0.752929688 | 0.21913159 | 3.464269195 | 4.105113029 | 766.828125 | 0.001 | 100.0 | 0.1 | 0.01 | 4.0 | 0.01 | 64.0 | 16.0 |
| 0.01953125 | 0.711914063 | 0.406643105 | 3.163368808 | 3.509097457 | 766.8496094 | 0.0005 | 100.0 | 0.05 | 0.01 | 2.0 | 0.01 | 32.0 | 8.0 |
| 0.043945313 | 0.674804688 | 0.488205295 | 2.925471717 | 3.498527765 | 766.9296875 | 0.0005 | 100.0 | 0.05 | 0.01 | 2.0 | 0.01 | 32.0 | 16.0 |
| 0.020507813 | 0.708984375 | 0.409682407 | 3.161523083 | 3.485819697 | 766.8867188 | 0.0005 | 100.0 | 0.1 | 0.01 | 2.0 | 0.01 | 32.0 | 8.0 |
| 0.041992188 | 0.684570313 | 0.466933594 | 2.957788983 | 3.488726735 | 766.9667969 | 0.0005 | 100.0 | 0.1 | 0.01 | 2.0 | 0.01 | 32.0 | 16.0 |
| 0.01953125 | 0.711914063 | 0.406643105 | 3.163368808 | 3.499094486 | 766.8027344 | 0.0005 | 100.0 | 0.05 | 0.01 | 4.0 | 0.01 | 32.0 | 8.0 |
| 0.043945313 | 0.674804688 | 0.488205295 | 2.925471717 | 3.48725915 | 767.0683594 | 0.0005 | 100.0 | 0.05 | 0.01 | 4.0 | 0.01 | 32.0 | 16.0 |
| 0.020507813 | 0.708984375 | 0.409682407 | 3.161523083 | 3.473590493 | 767.0058594 | 0.0005 | 100.0 | 0.1 | 0.01 | 4.0 | 0.01 | 32.0 | 8.0 |
| 0.041992188 | 0.684570313 | 0.466933594 | 2.957788983 | 3.511683583 | 766.9941406 | 0.0005 | 100.0 | 0.1 | 0.01 | 4.0 | 0.01 | 32.0 | 16.0 |
| 0.03125 | 0.713867188 | 0.422857851 | 3.029866204 | 4.102545857 | 766.9316406 | 0.0005 | 100.0 | 0.05 | 0.01 | 2.0 | 0.01 | 64.0 | 8.0 |
| 0.037109375 | 0.69921875 | 0.467807097 | 3.024323448 | 4.104898214 | 767.0546875 | 0.0005 | 100.0 | 0.05 | 0.01 | 2.0 | 0.01 | 64.0 | 16.0 |
| 0.030273438 | 0.716796875 | 0.416727209 | 3.04256475 | 4.077229142 | 766.9355469 | 0.0005 | 100.0 | 0.1 | 0.01 | 2.0 | 0.01 | 64.0 | 8.0 |
| 0.040039063 | 0.694335938 | 0.483740832 | 3.006518521 | 4.094444156 | 766.8378906 | 0.0005 | 100.0 | 0.1 | 0.01 | 2.0 | 0.01 | 64.0 | 16.0 |
| 0.03125 | 0.713867188 | 0.422857851 | 3.029866204 | 4.107426643 | 766.9726563 | 0.0005 | 100.0 | 0.05 | 0.01 | 4.0 | 0.01 | 64.0 | 8.0 |
| 0.037109375 | 0.69921875 | 0.467807097 | 3.024323448 | 4.088187575 | 767.1386719 | 0.0005 | 100.0 | 0.05 | 0.01 | 4.0 | 0.01 | 64.0 | 16.0 |
| 0.030273438 | 0.716796875 | 0.416727209 | 3.04256475 | 4.133657098 | 766.84375 | 0.0005 | 100.0 | 0.1 | 0.01 | 4.0 | 0.01 | 64.0 | 8.0 |
| 0.040039063 | 0.694335938 | 0.483740832 | 3.006518521 | 4.127341628 | 766.84375 | 0.0005 | 100.0 | 0.1 | 0.01 | 4.0 | 0.01 | 64.0 | 16.0 |
| 0.0078125 | 0.779296875 | 0.086942319 | 3.802173239 | 4.544541717 | 766.9042969 | 0.001 | 200.0 | 0.05 | 0.01 | 2.0 | 0.01 | 32.0 | 8.0 |
| 0.008789063 | 0.772460938 | 0.139713054 | 3.637912371 | 4.565919995 | 766.9628906 | 0.001 | 200.0 | 0.05 | 0.01 | 2.0 | 0.01 | 32.0 | 16.0 |
| 0.002929688 | 0.783203125 | 0.073504243 | 3.863313226 | 4.57320559 | 766.9121094 | 0.001 | 200.0 | 0.1 | 0.01 | 2.0 | 0.01 | 32.0 | 8.0 |
| 0.001953125 | 0.784179688 | 0.126850641 | 3.663778832 | 4.571519375 | 766.9609375 | 0.001 | 200.0 | 0.1 | 0.01 | 2.0 | 0.01 | 32.0 | 16.0 |
| 0.0078125 | 0.779296875 | 0.086942319 | 3.802173239 | 4.599365115 | 766.90625 | 0.001 | 200.0 | 0.05 | 0.01 | 4.0 | 0.01 | 32.0 | 8.0 |
| 0.008789063 | 0.772460938 | 0.139713054 | 3.637912371 | 4.587291718 | 766.8632813 | 0.001 | 200.0 | 0.05 | 0.01 | 4.0 | 0.01 | 32.0 | 16.0 |
| 0.002929688 | 0.783203125 | 0.073504243 | 3.863313226 | 4.550252318 | 766.9023438 | 0.001 | 200.0 | 0.1 | 0.01 | 4.0 | 0.01 | 32.0 | 8.0 |
| 0.001953125 | 0.784179688 | 0.126850641 | 3.663778832 | 4.577753067 | 766.9160156 | 0.001 | 200.0 | 0.1 | 0.01 | 4.0 | 0.01 | 32.0 | 16.0 |
| 0.0 | 0.805664063 | 0.020295139 | 3.880446473 | 5.152788162 | 766.9101563 | 0.001 | 200.0 | 0.05 | 0.01 | 2.0 | 0.01 | 64.0 | 8.0 |
| 0.0 | 0.79296875 | 0.105241993 | 3.805329684 | 5.235127091 | 766.9160156 | 0.001 | 200.0 | 0.05 | 0.01 | 2.0 | 0.01 | 64.0 | 16.0 |
| 0.0 | 0.805664063 | 0.022465278 | 3.883867675 | 5.218698025 | 766.8808594 | 0.001 | 200.0 | 0.1 | 0.01 | 2.0 | 0.01 | 64.0 | 8.0 |
| 0.000976563 | 0.791992188 | 0.112056804 | 3.797186216 | 5.181479692 | 766.921875 | 0.001 | 200.0 | 0.1 | 0.01 | 2.0 | 0.01 | 64.0 | 16.0 |
| 0.0 | 0.805664063 | 0.020295139 | 3.880446473 | 5.169565082 | 766.8925781 | 0.001 | 200.0 | 0.05 | 0.01 | 4.0 | 0.01 | 64.0 | 8.0 |
| 0.0 | 0.79296875 | 0.105241993 | 3.805329684 | 5.176501393 | 766.953125 | 0.001 | 200.0 | 0.05 | 0.01 | 4.0 | 0.01 | 64.0 | 16.0 |
| 0.0 | 0.805664063 | 0.022465278 | 3.883867675 | 5.181358099 | 766.875 | 0.001 | 200.0 | 0.1 | 0.01 | 4.0 | 0.01 | 64.0 | 8.0 |
| 0.000976563 | 0.791992188 | 0.112056804 | 3.797186216 | 5.183052063 | 766.8671875 | 0.001 | 200.0 | 0.1 | 0.01 | 4.0 | 0.01 | 64.0 | 16.0 |
| 0.006835938 | 0.774414063 | 0.239691353 | 3.545346912 | 4.59584856 | 767.0449219 | 0.0005 | 200.0 | 0.05 | 0.01 | 2.0 | 0.01 | 32.0 | 8.0 |
| 0.006835938 | 0.767578125 | 0.259089737 | 3.39466137 | 4.57220912 | 766.8730469 | 0.0005 | 200.0 | 0.05 | 0.01 | 2.0 | 0.01 | 32.0 | 16.0 |
| 0.008789063 | 0.774414063 | 0.240667916 | 3.534573811 | 4.576325178 | 766.9121094 | 0.0005 | 200.0 | 0.1 | 0.01 | 2.0 | 0.01 | 32.0 | 8.0 |
| 0.006835938 | 0.764648438 | 0.259012144 | 3.382884026 | 4.618607521 | 766.8515625 | 0.0005 | 200.0 | 0.1 | 0.01 | 2.0 | 0.01 | 32.0 | 16.0 |
| 0.006835938 | 0.774414063 | 0.239691353 | 3.545346912 | 4.548846426 | 766.9101563 | 0.0005 | 200.0 | 0.05 | 0.01 | 4.0 | 0.01 | 32.0 | 8.0 |
| 0.006835938 | 0.767578125 | 0.259089737 | 3.39466137 | 4.57443428 | 766.9101563 | 0.0005 | 200.0 | 0.05 | 0.01 | 4.0 | 0.01 | 32.0 | 16.0 |
| 0.008789063 | 0.774414063 | 0.240667916 | 3.534573811 | 4.563651443 | 766.8945313 | 0.0005 | 200.0 | 0.1 | 0.01 | 4.0 | 0.01 | 32.0 | 8.0 |
| 0.006835938 | 0.764648438 | 0.259012144 | 3.382884026 | 4.570395947 | 766.8925781 | 0.0005 | 200.0 | 0.1 | 0.01 | 4.0 | 0.01 | 32.0 | 16.0 |
| 0.004882813 | 0.791992188 | 0.133063217 | 3.590356213 | 5.188795686 | 766.8652344 | 0.0005 | 200.0 | 0.05 | 0.01 | 2.0 | 0.01 | 64.0 | 8.0 |
| 0.005859375 | 0.774414063 | 0.252983542 | 3.493388295 | 5.219975233 | 766.90625 | 0.0005 | 200.0 | 0.05 | 0.01 | 2.0 | 0.01 | 64.0 | 16.0 |
| 0.00390625 | 0.793945313 | 0.131299426 | 3.608049399 | 5.20837307 | 766.8339844 | 0.0005 | 200.0 | 0.1 | 0.01 | 2.0 | 0.01 | 64.0 | 8.0 |
| 0.004882813 | 0.7734375 | 0.253770771 | 3.485601261 | 5.190756202 | 766.8964844 | 0.0005 | 200.0 | 0.1 | 0.01 | 2.0 | 0.01 | 64.0 | 16.0 |
| 0.004882813 | 0.791992188 | 0.133063217 | 3.590356213 | 5.159092546 | 766.9257813 | 0.0005 | 200.0 | 0.05 | 0.01 | 4.0 | 0.01 | 64.0 | 8.0 |
| 0.005859375 | 0.774414063 | 0.252983542 | 3.493388295 | 5.208868861 | 766.9707031 | 0.0005 | 200.0 | 0.05 | 0.01 | 4.0 | 0.01 | 64.0 | 16.0 |
| 0.00390625 | 0.793945313 | 0.131299426 | 3.608049399 | 5.236365676 | 766.9394531 | 0.0005 | 200.0 | 0.1 | 0.01 | 4.0 | 0.01 | 64.0 | 8.0 |
| 0.004882813 | 0.7734375 | 0.253770771 | 3.485601261 | 5.219882965 | 766.9980469 | 0.0005 | 200.0 | 0.1 | 0.01 | 4.0 | 0.01 | 64.0 | 16.0 |
| 0.005859375 | 0.744140625 | 0.224260116 | 3.612463241 | 3.47030735 | 766.8144531 | 0.001 | 100.0 | 0.05 | 0.005 | 2.0 | 0.01 | 32.0 | 8.0 |
| 0.010742188 | 0.76171875 | 0.215372865 | 3.617039222 | 3.490041375 | 766.8730469 | 0.001 | 100.0 | 0.05 | 0.005 | 2.0 | 0.01 | 32.0 | 16.0 |
| 0.017578125 | 0.727539063 | 0.245378844 | 3.584327921 | 3.494503975 | 766.8828125 | 0.001 | 100.0 | 0.1 | 0.005 | 2.0 | 0.01 | 32.0 | 8.0 |
| 0.00390625 | 0.770507813 | 0.166116603 | 3.689371678 | 3.497444153 | 766.9824219 | 0.001 | 100.0 | 0.1 | 0.005 | 2.0 | 0.01 | 32.0 | 16.0 |
| 0.005859375 | 0.744140625 | 0.224260116 | 3.612463241 | 3.486721992 | 766.7871094 | 0.001 | 100.0 | 0.05 | 0.005 | 4.0 | 0.01 | 32.0 | 8.0 |
| 0.010742188 | 0.76171875 | 0.215372865 | 3.617039222 | 3.518079996 | 766.8613281 | 0.001 | 100.0 | 0.05 | 0.005 | 4.0 | 0.01 | 32.0 | 16.0 |

73

0.017578125  0.727539063  0.245378844  3.584327921  3.484863997  766.8378906  0.001  100.0  0.1  0.005  4.0 0.01 32.0  8.0

. . . . . . . . . . . . . . . . . . . . . . . . . . . . . . . . . . . . . . . . . . . . . . . . . . . . . . . . . . . . . . . . . . . . . . . . . . . . . . . . . . . . . . . . . . . . . . . . . . . . . . . . . . . . . . . . . . . . . . . . . . . . . . . . . . . . . . . . . . . . . . . .

# S  Ablation on Oracle Selection

A key design choice in GEMS is the use of an Empirical-Bernstein UCB (EB-UCB) oracle (Section 3.4, Eq. 7) to select new policies. A natural question is why this specific, variance-aware bandit algorithm was chosen over simpler or alternative methods, such as standard UCB1 or Thompson Sampling.

**The "Shifting Meta" Experiment**  While the oracle in GEMS solves a stationary sub-problem within each fixed iteration (satisfying the assumptions of Theorem 3.3), the global problem across training is non-stationary. As the meta-strategy $\sigma_t$ evolves and the generator $G_\theta$ is fine-tuned, the expected value of any given latent policy $z \in \Lambda_t$ changes. The oracle must therefore be highly adaptive, capable of quickly detecting when a previously suboptimal policy has become optimal and, conversely, abandoning a previously optimal policy that is no longer effective.

To simulate this dynamic, we designed a simple "Shifting Meta" bandit game. The game consists of 3 arms.

- **Phase 1 (Time Steps $t < 1000$):** Arm 1 is the unique optimal policy with the highest expected reward.

- **Meta Shift (Time Step $t = 1000$):** The underlying meta-game abruptly changes.

- **Phase 2 (Time Steps $t \geq 1000$):** Arm 0 becomes the new unique optimal policy.

Critically, we designed Arm 0 to also have a higher reward variance than the other arms. This setup tests an algorithm's ability to abandon a "safe," well-explored, but now-suboptimal arm in favor of a "riskier," high-variance arm that has become optimal.

**Analysis of Results**  The results are shown in Figure 39.

- **Cumulative Regret (Top):** In Phase 1, all algorithms (except Greedy) successfully identify the optimal arm and achieve low regret. After the meta-shift at $t = 1000$, the limitations of non-adaptive oracles become clear. Greedy, being purely exploitative, never adapts and accumulates massive regret. Standard UCB1, which is not variance-aware, also accumulates a large amount of regret as it is too "confident" in the old optimal arm. In contrast, the variance-aware methods (EB-UCB, UCB-V) and Thompson Sampling quickly adapt, and their cumulative regret begins to decrease as they exploit the new, higher-value optimal arm.

- **Adaptability (Bottom):** This plot provides the clearest justification. After the shift, standard UCB1 takes approximately 500 time steps (from $t = 1000$ to $t = 1500$) to reliably switch to the new optimal arm. Thompson Sampling is also slow, taking over 250 steps. In stark contrast, both **EB-UCB** and **UCB-V** adapt almost **instantaneously**. Their sensitivity to variance means they never became over-confident in the old arm and were able to quickly recognize the value of the new, high-variance optimal arm.

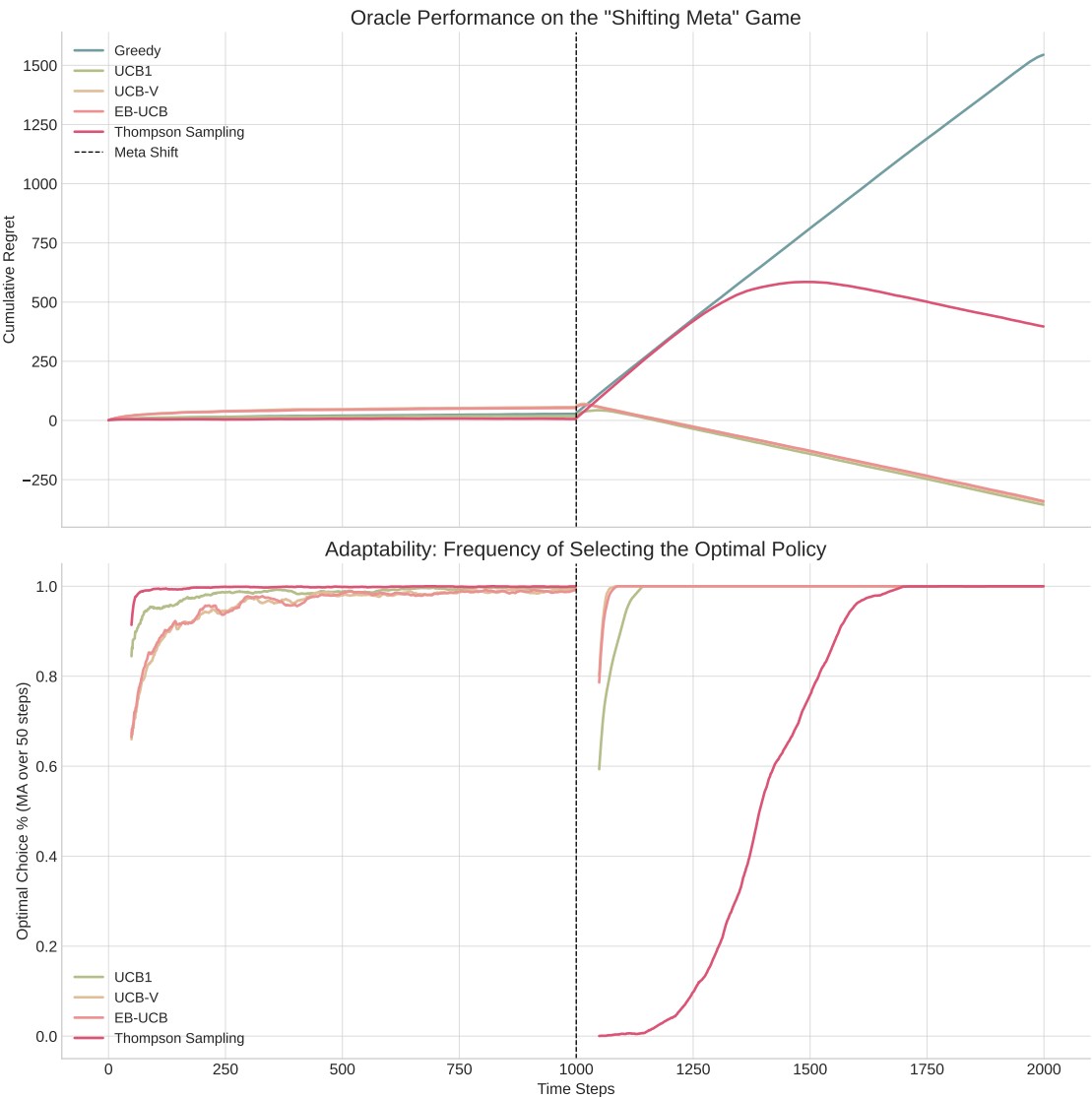

Figure 39: Performance of different oracle algorithms on the "Shifting Meta" game, averaged over 50 runs. The meta-game shifts at $t = 1000$. **(Top)** Cumulative Regret. Lower is better. Note the poor performance of Greedy and Thompson Sampling after the shift. **(Bottom)** Adaptability, measured as the 50-step moving average of picking the correct optimal arm. The variance-aware oracles (EB-UCB and UCB-V) adapt almost instantaneously, while UCB1 and Thompson Sampling are significantly slower.

**Conclusion**   This experiment demonstrates that a simple UCB1 oracle is unsuitable for the non-stationary problem faced in GEMS, as its over-confidence would cause it to lag significantly behind the evolving meta-game. The superior adaptivity of variance-aware oracles is critical. We chose EB-UCB as it not only demonstrates this state-of-the-art adaptability but also aligns directly with the theoretical concentration bounds (based on empirical Bernstein) used in our analysis (Section 3.4).

# Part IV

# Frequently Asked Questions (FAQs)

1. **Q: Why did you not benchmark GEMS on the StarCraft Multi-Agent Challenge (SMAC)?**

   **A:** Our work introduces GEMS as a direct, scalable framework to overcome the fundamental inefficiencies of the Policy-Space Response Oracles (PSRO) paradigm, namely its quadratic computation and linear memory costs. Therefore, our primary objective was to benchmark GEMS directly against classical PSRO and its most relevant variants (e.g., Double Oracle, Alpha-PSRO, A-PSRO) in domains that clearly expose these bottlenecks and test game-theoretic solution quality, such as Kuhn Poker, Deceptive Messages Game, and Multi-Agent Tag.

2. **Q: Why did you choose Optimistic Multiplicative Weights Update (OMWU) over standard Multiplicative Weights Update (MWU)?**

   **A:** We chose OMWU because it provides stronger theoretical guarantees and faster convergence in our setting. Unlike standard MWU, OMWU incorporates a predictive "hint" about the next payoff vector. This optimistic step results in an average external regret bound that scales with the cumulative *variation* of the payoff vectors ($O(\sum \|v_t - v_{t-1}\|_\infty^2)$) rather than the time horizon $T$, as shown in Proposition 3.2. Since GEMS is designed to induce a slowly changing meta-game, this property leads to faster convergence.

3. **Q: How can GEMS be considered "surrogate-free"?**

   **A:** We use the term "surrogate-free" to signify that GEMS does not maintain the two key surrogates of classical PSRO: ❶ an explicit, discrete population of $k$ policies, which requires $O(k)$ memory, and ❷ the full $k \times k$ payoff matrix, which requires $O(k^2)$ computation. Instead, GEMS replaces this entire structure with a single amortized generator ($G_\theta$) and a compact set of latent "anchor" codes ($Z_t$).

4. **Q: Does the single amortized generator ($G_\theta$) suffer from catastrophic forgetting?**

   **A:** We explicitly mitigate this risk using our Amortized Best-Response with a Trust Region (ABR-TR) objective (Section 3.5). This objective is designed to fine-tune the generator to produce new, high-performing policies while retaining its ability to generate previously effective ones. It achieves this by incorporating a KL-divergence penalty against a frozen, older version of the generator ($\theta^-$), which serves as a trust region and prevents catastrophic forgetting.

5. **Q: What is the main computational bottleneck of GEMS?**

   **A:** GEMS successfully replaces the $O(k^2)$ computational overhead of PSRO. The new computational cost is dominated by the number of Monte Carlo rollouts required per iteration. This cost scales with the number of sampled matches used to estimate the meta-game (controlled by $n_i$, $m$, and $B$ in Section 3.2) and the size of the candidate pool ($|\Lambda_t|$) evaluated by the bandit oracle (Section 3.4). This simulation-based cost is fundamentally more scalable than constructing the full payoff matrix.

6. **Q: How do the theoretical guarantees of GEMS compare to classical PSRO, given its use of approximations?**

   **A:** GEMS retains the core game-theoretic convergence guarantees of PSRO. Our overall exploitability bound (Theorem 3.4, Section 3.6) cleanly decomposes the average exploitability into four interpretable terms: ❶ the external regret of the OMWU meta-solver, ❷ the noise from Monte-Carlo estimation, ❸ the sub-optimality of the bandit oracle, and (4) the approximation error from the amortized generator. As the simulation budget ($nm$) grows and the generator training improves ($\epsilon_{BR} \to 0$), the latter three terms vanish, and the overall exploitability is driven by the no-regret property of the meta-solver, ensuring convergence.

7. **Q: Why use an EB-UCB bandit oracle instead of just computing a single, direct best response (BR)?**

   **A:** The goal of population expansion is to efficiently find new, challenging policies to add to the game (§ 3.4). Simply training a single BR can be computationally expensive. Instead, we cast this as a multi-armed bandit problem over a pool of candidate latent codes $\Lambda_t$. We use the Empirical-Bernstein UCB (EB-UCB) oracle because it efficiently balances the exploration-exploitation trade-off by using sample variance to achieve tighter confidence bounds. This allows GEMS to select promising new policies from the candidate pool more efficiently and effectively explore the latent strategy space.

8. **Q: What is the purpose of the Jacobian penalty ($\lambda_J$)?**

   **A:** The Jacobian penalty ($\lambda_J||JG_\theta(z)||_F^2$) is a regularizer applied during both the oracle selection (Eq. 7) and the generator training steps (Eq. 11). Its purpose is to encourage smoothness in the generator's latent space by penalizing large gradients of the generator's output with respect to its latent input $z$. This smoothness aids in stabilizing the optimization process.

9. **Q: Why did you benchmark against PSRO variants and not other modern MARL algorithms like MAPPO or QMIX?**

   **A:** GEMS is specifically proposed to solve the fundamental scalability bottlenecks inherent in *population-based, game-theoretic* approaches, for which PSRO is the seminal framework (Section 2). Algorithms like MAPPO or QMIX, while effective, address a different problem (e.g., decentralized execution with centralized training for a fixed number of agents) and do not typically maintain or solve a meta-game over an explicit population of policies. Therefore, to scientifically validate our claims, the most relevant and direct baselines are classical PSRO and its state-of-the-art variants.

10. **Q: How does GEMS extend from two-player zero-sum (2P-ZS) games to the n-player general-sum (NP-GS) case?**

    **A:** The core components of GEMS extend naturally to the NP-GS setting, as detailed in Section 3.7 and Appendix Part II. The main generalizations are: (1) Each of the $n$ players maintains their own independent meta-strategy $\sigma_t^{(p)}$. (2) We use a single batch of shared game rollouts and an importance-weighted estimator (Eq. 14) to efficiently compute each player's per-policy value vector $\hat{v}_t^{(p)}$ against the joint strategy of all other players. (3) Each player then independently runs their own OMWU update and EB-UCB oracle. (4) This decentralized process drives the time-averaged joint strategy toward an $\epsilon$-Coarse-Correlated Equilibrium ($\epsilon$-CCE), the standard solution concept for general-sum games.

11. **Q: The ablation tables for the Public Goods Game (Table 4) and Deceptive Messages Game (Table 5) present many results across different hyperparameter settings, but the paper doesn't explicitly state which configuration is definitively "best." What is the main takeaway from these ablations?**

    **A:** The primary purpose of the extensive ablation tables (Table 4 for PGG, Table 5 for Deceptive Messages) is to ensure **transparency** and aid **reproducibility** by documenting GEMS's sensitivity to its core hyperparameters. Identifying a single, universally optimal configuration across all games and metrics is challenging, as a full factorial sweep is computationally infeasible. These tables showcase the results of structured variations, illustrating the inherent **trade-offs** involved (e.g., between convergence speed, solution quality, and computational resources). We provide this detailed data to allow readers to observe these sensitivities directly and understand the impact of different choices. Determining the absolute optimal settings for every possible scenario remains an open area, and these tables serve as a valuable resource for guiding future work or tuning GEMS for specific applications. Our main experiments utilize configurations found to be effective for demonstrating the core advantages of GEMS.

12. **Q: What are the limitations introduced by GEMS's use of Monte Carlo estimation and an amortized generator?**

    **A:** GEMS introduces two main sources of approximation to enable $O(1)$ memory scaling:

First, **Estimation Noise**: Payoffs are estimated via Monte Carlo sampling. In sparse-reward domains, this can increase the variance of the meta-gradient. However, our results show that even noisy estimates guide the population toward effective strategies significantly faster than exact methods, as the sampling cost scales linearly ($O(k)$) rather than quadratically ($O(k^2)$).

Second, **Amortization Gap**: The generator approximates the best-response manifold. A potential limitation is that an under-parameterized generator could fail to capture niche counter-strategies. We address this by explicitly conditioning the generator on a growing set of "anchors" ($Z_t$), which forces the network to maintain diverse modes. Empirically, our 1,000-iteration Chess experiments demonstrate that the generator successfully maintains over 2,000 distinct policies without collapsing, validating its expressive capacity.

13. **Q: How does the theoretical analysis account for the non-stationarity of rewards as the meta-strategy evolves?**

    **A:** We address the global non-stationarity of the learning problem by decomposing GEMS into two distinct timescales, ensuring that the assumptions for our regret bounds are locally satisfied:

    - **Inner Loop (Stationary Sub-problem):** Within any single meta-iteration $t$, the opponent meta-strategy $\sigma_t$ is held fixed. Consequently, when the oracle selects an anchor $z \in \Lambda_t$, it faces a mathematically stationary reward distribution defined by this frozen opponent. This stationarity ensures that the instance-dependent regret bounds (Theorem 3.3) are valid for the anchor selection step.
    - **Outer Loop (Dynamic Meta-Game):** The non-stationarity arises only across iterations as $\sigma_t$ evolves. This global dynamic is explicitly handled by the OMWU meta-solver, which is designed to minimize dynamic regret in time-varying games (Proposition 3.4).

    Thus, while the overall landscape shifts (motivating our use of adaptive bandits like EB-UCB in the "Shifting Meta" experiment), the specific sub-problem addressed by the oracle at each step remains stationary and theoretically amenable to our bounds.

14. **Q: When do Monte-Carlo rollouts become a computational bottleneck, and how does GEMS handle this trade-off?**

    **A:** Monte-Carlo (MC) sampling effectively trades a "hard" computational wall for a "soft" sampling cost. We analyze this trade-off as follows:

    - **The Quadratic Bottleneck (PSRO):** Standard population-based methods require computing an exact payoff matrix. This operation scales quadratically, $\mathcal{O}(N^2)$, with the number of iterations $N$ (as the population grows indefinitely). For complex games, this quickly becomes computationally intractable.
    - **The Constant-Time Scalability (GEMS):** In contrast, GEMS maintains a compact, fixed-size anchor set ($|\Lambda| \ll N$). Consequently, the computational complexity per iteration is $\mathcal{O}(1)$ with respect to the total history of strategies.
    - **The Trade-off (Sampling Variance):** While the complexity class is constant, the wall-clock time depends on the sampling difficulty. The cost is proportional to $\mathcal{O}(k \cdot C_{eval})$, where $k$ is the number of samples and $C_{eval}$ is the cost per episode. This *can* become a bottleneck in environments with *extreme stochasticity* (requiring high $k$) or *very long episodes* (high $C_{eval}$).
    - **The Solution (EB-UCB):** To mitigate this, GEMS employs the EB-UCB oracle. It adaptively allocates the sampling budget $k$, spending compute *only* when the estimator is uncertain (high variance) or the strategy is promising. This ensures that even when $C_{eval}$ is high, GEMS avoids wasting resources on sub-optimal strategies, preserving practical scalability where matrix-based methods have limitations.

