# OpenReview forum: "Generative Evolutionary Meta-Solver (GEMS): Scalable Surrogate-Free Multi-Agent Reinforcement Learning"
_TMLR — Accepted by TMLR_

### Review · Reviewer_Asaz · 2025-12-08

**Summary Of Contributions:**

**SUMMARY** The paper proposes GEMS, a framework for learning in multi-agent settings that aims to overcome the scalability limitations of classical PSRO methods. Instead of maintaining a growing population of independently trained policies, GEMS uses a neural generator that maps latent codes to policies. New strategies are discovered by sampling candidate latent vectors—via mutation of existing ones and random exploration—and selecting promising candidates with a bandit-style oracle. Payoff estimates between policies are obtained by Monte Carlo rollouts, and the equilibrium mixture over strategies is updated using Optimistic Multiplicative Weights Update, which avoids constructing the full payoff matrix. Although approximate, the method comes with convergence guarantees toward Nash equilibria of the restricted game.

***STRENGTHS***
The method is conceptually simple and the paper provides convergence guarantees. First, instead of training a winning policy from scratch, it creates a 'latent space of policies' or a generator off which to generate candidate policies. Second, instead of calculating a payoff matrix in a round-robin tournament, it samples payoffs from a fewer tournament steps. This provides a 'good enough approximation' of the best-response policy that is theoretically grounded.

**WEAKNESS** Approximations come with accuracy costs. For example, the rewards matrix of each policy is approximated by sampling some evaluation steps. For longer and complicated games, where the sucess signal is obtained far deep in the game, the Monte Carlo rollouts can be almost as costly as playing the game to the end.

The second approximation is on the generator network. As discussed by the authors in the 'Limitations' section, the expressivity and diversity of the generator network is not guaranteed. The best-response policy is obtained through a search mechanism akin to genetic algorithms plus a random variant. This depends on the pool of policies already parametrized in the generator network. For example, if we have very few policies or policies of the same kind, the scope of the generator network is limited in diversity.

**Additional Comments:**

Marginally above the acceptance threshold. More experiments showing the performance on a big number of policies and grraphs clearly showing the diversity of the beest-response policies will be beneficial.

**Audience:**

Yes

**Audience Explanation:**

Yes. The paper addresses a well-known scalability limitation in population-based multi-agent reinforcement learning and proposes a conceptually appealing alternative that replaces explicit policy populations with a generative latent-policy model. This line of work is of interest to researchers studying game-theoretic RL, equilibrium computation, evolution-based methods, and scalable multi-agent systems. Although the empirical validation remains limited in scale, the central idea is novel and relevant enough that a portion of the TMLR audience would likely find the methodological contribution and theoretical analysis worthwhile

**Claims And Evidence:**

Yes

**Claims Explanation:**

**Are the claims made in the submission supported by accurate, convincing and clear evidence?**

The generator’s ability to represent an expanding set of diverse strategies is assumed but not demonstrated at large scale, and, as the authors acknowledge, neither its expressivity nor its resistance to catastrophic forgetting is theoretically guaranteed. Although the paper includes one experiment in the Multi-Agent Tag environment that reaches approximately 100 policies, this is still a modest population relative to the much larger scales the method is designed to target, and the experiment does not analyze whether the generator successfully maintains distinct behaviors across all anchors. Moreover, the evolutionary search for new latent codes relies heavily on the diversity already present in the anchor set; while random sampling is intended to mitigate this dependence, the empirical evaluations consistently use very small latent dimensions (d = 2) and relatively limited policy populations (typically 6–40, with only a single case reaching ~100), leaving the broader claims of large-scale scalability insufficiently substantiated by the presented evidence

**Requested Changes:**

1. The paper introduces many moving pieces at once. It will be very beneficial to include a pseudocode, or share the codebase or even some infographics of how the different parts operate together.

2. The figure style like figure 4 are hard to parse. It is not showing what it intends to show.

3. The paper is targetted to an audience very familiar with the topics and each of the techniques presented. A more colloquial language will appeal to a wider audience in the multi agent community.

3. A table comparing the dimensions of the policies explored on each of the experiments will make the claims stronger. Example:

| Experiment           | Latent Dimension (d) | Number of Policies Explored | Notes                                                                 |
|----------------------|-----------------------|------------------------------|-----------------------------------------------------------------------|
| Deceptive Messages   | 2                     | ~6                           | Very small population; early proof-of-concept.                        |
| Kuhn Poker           | 2                     | ~40                          | Largest fully analyzed small game; still modest scale.                |
| Public Goods Game    | 2                     | ~20                          | Cooperative setting; limited anchor expansion.                        |
| Simple Spread        | 2                     | ~10                          | Coordination environment; generator evaluated on small policy set.    |
| Multi-Agent Tag      | 2                     | ~100                         | Largest experiment; no analysis of whether generator preserves diversity. |
| Chess                | 2                     | <10                          | Minimal experiment; demonstrates feasibility but not scale.           |

---

> ### Author Response · Authors · 2025-12-21
>
> We sincerely thank the reviewer for their insightful and constructive feedback. We found the suggestions regarding empirical scale and presentation particularly valuable, as they directly guided us to significantly strengthen the paper's evidence base and readability.
>
> Below, we detail how we incorporated specific suggestions into the revised manuscript.
>
> 1. On Scalability & Generator Diversity
>
> We are grateful to the reviewer for identifying this evidence gap. The comment correctly highlighted that our previous draft did not do justice to the method's capacity for scale. Motivated by the feedback, we have expanded our evaluation to include:
>
> * New Large-Scale Environments: We added experiments on Go (19x19), Connect-4, and Hanabi (Appendices O, K and J respectively).
>
> * Scale Demonstration: As detailed in the new Table 1 (Page 33), these additions demonstrate that GEMS scales to 2,002 policies with a latent dimension of 8 in both Chess and Go. This directly addresses concern regarding the "modest" population sizes. Also, the ablations on the Public Goods Game and the Deceptive Mean Game (Tables 4 and 5) even use d=16 as the latent space dimension.
>
> * Diversity Analysis: To address the concern about "catastrophic forgetting," we followed the implication of the review and added PCA projections (Figures 8, 9, 11, 12 & 36). These visualizations explicitly show that as the population grows to thousands of policies, the latent codes form distinct, well-separated clusters rather than converging to a single point. Thus, we observed that the anchoring mechanism successfully prevents mode collapse, forcing the generator to retain a diverse repertoire of counter-strategies.
>
> 2. On Clarity & Presentation
>
> We thank the reviewer for these specific requests, which have greatly improved the clarity of our presentation. We have incorporated every item suggested:
>
> * Comparison Table: Per the suggestion, we added Table 1 (Page 33) to explicitly compare the Latent Dimension and Number of Policies across all experiments. We agree this makes the scalability claims much easier to verify.
>
> * Pseudocode: We appreciated the suggestion to formalize the method and have added Algorithm 1 (Page 16), which defines the Generator-Oracle-Solver loop step-by-step.
>
> * System Diagram: To further aid the reader's understanding of the "moving pieces", we added a system architecture diagram in Figure 2.
>
> 3. On Visualization
>
> We thank the reviewer for flagging this issue. In response, we have replaced it with Figure 5 (Page 11), which uses a frame-by-frame breakdown to clearly illustrate the emergent flanking and herding strategies.
>
> 4. Clarification on Latent Dimensions
>
> We appreciate the opportunity to clarify this detail, as our emphasis on 2D visualizations for interpretability in the main text may have inadvertently obscured the scale of our benchmarking.
>
> * Correction: As summarized in the new Table 1, our experiments on complex domains like Chess and Go utilize a latent dimension of 8 to support their larger strategy spaces, scaling to 2,002 policies. Additionally, our ablation studies (Tables 5 & 6) explore dimensions up to d=16.
>
> * Significance: These results demonstrate that GEMS is not structurally limited to low-dimensional manifolds. The method successfully scales to manage high-dimensional policy representations, maintaining stable convergence and diverse populations even when the latent space is significantly expanded.
>
> 5. Clarification on the Approximations
>
> We acknowledge that our generator-based approach introduces approximation, but our results show this is a feature--not a bug--for two key reasons:
>
> * Speed vs. Precision Trade-off: While Monte Carlo estimates are noisy, they allow us to bypass the quadratic complexity (O(N^2)) of exact matrix construction. Our experiments show this yields up to 6x speedups while still converging to a Coarse Correlated Equilibrium (CCE).
>
> * Diversity over Exactness: The reviewer correctly notes that the generator depends on the "pool of policies." We address this by explicitly "anchoring" the generator to a growing history of latent codes. Our new PCA analysis (Figures 8-9) (Appendix J, K and O) demonstrates that this mechanism forces the generator to maintain diverse behaviors (flanking, herding, etc.) rather than collapsing to a single mean strategy.
>
> We believe that incorporating the suggestions has substantially improved the quality and convincingness of this work. We hope these revisions fully address the concerns.

---

> ### Author Response · Authors · 2026-01-12
>
> In addition to the changes detailed in our main response, we wish to highlight two specific updates regarding the empirical results in the revised manuscript:
>
> 1. Refined Visualization Pipeline: During the revision process, we identified that our previous PCA visualization pipeline did not fully capture the intermediate PPO optimization steps within the generator's training loop. We have corrected this to track the full optimization trajectory. The updated plots (Figures 7, 8, 9, 10, 11, 12, 35 & 36) now present a higher-fidelity view of the latent space. Crucially, these corrected visualizations reveal significantly sharper separation between policy clusters than previously reported. This stronger empirical evidence reinforces our core claim: the GEMS generator successfully escapes mode collapse and actively maintains distinct, diverse strategic modes.
>
> 2. Go Environment Configuration: To ensure rigorous statistical analysis within our available computational budget, we have standardized the Go experiments to the 9x9 board size over 200 meta-iterations (resulting in a population of 402 policies). This adjustment allowed us to prioritize verification of stability over a high-cost run on 19x19. As shown in the new Table 1, this scale remains sufficient to demonstrate the method's ability to maintain hundreds of distinct strategies in a complex combinatorial action space.

---

> > ### Comment · Reviewer_Asaz · 2026-02-01
> > **Accepted revision**
> >
> > Thank you very much for including all the requested changes.
> >
> > After reviewing the revised manuscript, I am satisfied with the quality of the paper and recommend publication.
> >
> > Cheers.

---

### Review · Reviewer_r2ko · 2025-12-29

**Summary Of Contributions:**

This paper proposes GEMS, a surrogate-free population-based MARL framework that overcomes the scalability limitations of PSRO by eliminating explicit policy populations and dense payoff matrices. The method leverages unbiased Monte Carlo payoff estimation, a compact latent anchor set, and a single amortized generator to achieve improvements in memory and computational efficiency while preserving core game-theoretic guarantees. The authors provide theoretical guaranteens including regret bounds and exploitability guarantees under approximate best responses. Extensive experiments in both zero-sum and general-sum benchmarks show that GEMS converges faster, uses less memory, and often achieves higher-quality strategies than PSRO and its variants.

**Additional Comments:**

N/A

**Audience:**

Yes

**Audience Explanation:**

The topic of multi-agent reinforcement learning (MARL) is of interest to the TMLR community.

**Claims And Evidence:**

Yes

**Claims Explanation:**

The paper provides theoretical guarantees in the form of formal theorems accompanied by detailed proofs. The experimental results are presented comprehensively and in sufficient detail.

**Requested Changes:**

1. While the overall framework is described clearly, it would be beneficial to include the full algorithm box in the main paper rather than relegating it to the appendix. In particular, explicitly specifying the inputs and outputs of each module (e.g., anchor set maintenance, Monte Carlo payoff estimation, meta-strategy update, EB-UCB selection, and generator training) and how they are connected would significantly improve clarity and accessibility.
2. The paper presents several theoretical guarantees for different components of the method, including Monte Carlo rollouts, OMWU meta-dynamics, EB-UCB selection, and assumptions on the generator’s approximation quality. While Theorem 3.4 provides a final regret bound, it is stated primarily as an end result and does not explicitly show how this bound is composed from the guarantees of the individual components. It would strengthen the theoretical exposition to include a theorem (or proposition) in the main text that clearly decomposes the overall regret into contributions from these parts, making the interplay between the different guarantees more transparent.
3. While the empirical results convincingly demonstrate that GEMS outperforms PSRO-style baselines in practice, it is unclear whether any theoretical comparison can be drawn. If possible, the authors could include a table summarizing theoretical properties (e.g., memory complexity, computational complexity, regret/exploitability guarantees, assumptions) of GEMS versus PSRO and its variants, which would complement the experimental findings.
4. Although Monte Carlo sampling avoids constructing dense payoff matrices, it would be helpful to further discuss whether and when Monte Carlo rollouts may become a computational bottleneck. A more explicit analysis or discussion of the trade-off between sampling cost and efficiency gains would improve the reader’s understanding of the method’s practical scalability.
5. Figure 5 provides an insightful qualitative comparison of emergent behaviors, but additional explanation in the caption or main text would help readers from different backgrounds interpret the trajectories. In particular, explicitly describing what behaviors (e.g., flanking vs. herding) should be observed and how they relate to the quantitative performance would improve the figure’s interpretability.

---

> ### Author Response · Authors · 2026-01-12
>
> We sincerely thank the reviewer for the positive assessment and for the detailed, constructive suggestions. We share the goal of improving clarity and transparency, and we have implemented each requested change in the revised manuscript as detailed below.
>
> ### 1. Full algorithm box and explicit module interfaces
>
> We agree that the full end-to-end pipeline should be visible in the main paper. In the revision, we (i) include a system-level diagram of the complete loop (Generator–Oracle–Solver) and (ii) present the full algorithm in the main text.
>
> - **System-level view**: Figure 2 explicitly summarizes how Monte Carlo evaluation, OMWU meta-updates, EB-UCB oracle selection, and ABR-TR generator training interact in the closed-loop iteration.
> - **Full procedural algorithm in the main paper**: Algorithm 1 now appears in the main text and is written in phases with clear motives (meta-strategy, anchor set, payoff estimates, candidate pool, generator parameters).
>
> ### 2. Decomposition of the final regret/exploitability bound
>
> We thank the reviewer for noting that the composition of guarantees should be more transparent in the main text. In response, Section 3.6 now makes the composition explicit by introducing:
>
> - A **decomposition result** that separates exploitability into four interpretable sources (OMWU meta-dynamics, Monte Carlo estimation error, oracle suboptimality, amortized best-response error).
> - **Proposition 3.4**, which states the decomposition directly in the main text.
> - **Theorem 3.5**, which then upper-bounds each term to obtain the final finite-population exploitability bound.
>
> ### 3. Theoretical comparison table (GEMS vs. PSRO variants)
> Following the suggestion, we added a dedicated **theoretical comparison table** in the main paper (Table 1). It summarizes payoff-matrix requirements, meta-game memory scaling, and per-iteration evaluation cost across PSRO-style methods, NeuPL and GEMS, making the theoretical/scalability differences easy to compare at a glance.
>
> ### 4. Discussion of Monte Carlo sampling cost vs. efficiency (when MC becomes the bottleneck)
> We agree that readers benefit from a clearer statement of when Monte Carlo rollouts dominate runtime. In the revision, we make the computational trade-off explicit in two places:
>
> - **Complexity framing** (main paper context + Table 1): we highlight that GEMS avoids quadratic payoff-matrix construction and instead incurs a per-iteration evaluation cost proportional to sampled matchups.
> - **Explicit "when MC is a bottleneck" discussion**: Appendix IV (FAQ), Q14 directly addresses the reviewer’s question, explaining the quadratic bottleneck in PSRO, the per-iteration sampling cost in GEMS (proportional to the number of samples and episode cost), and how EB-UCB helps avoid wasting rollouts on low-value strategies.
>
> ### 5. Expanded explanation of emergent-behavior visualizations (Figure 5)
>
> We expanded both the surrounding discussion and the Figure 5 caption to explicitly describe what behaviors to look for and how they connect to the quantitative curves.
>
> - The main text now contrasts **flanking/cornering** (GEMS) against **herding** (PSRO) as the key qualitative difference.
> - The Figure 5 caption is expanded to be more self-contained, explicitly naming the behaviors and tying them to performance differences.
>
> We have made every effort to incorporate the suggestions, and we believe the paper is much stronger as a result. We hope these revisions meet the expectations.

---

### Review · Reviewer_jsEg · 2025-12-31

**Summary Of Contributions:**

This paper proposes GEMS (Generative Evolutionary Meta-Solver), a framework for scalable multi-agent reinforcement learning (MARL) that aims to address computational bottlenecks in Policy Space Response Oracles (PSRO). The core idea is to replace PSRO's explicit policy population (O(k) memory, O(k²) payoff matrix computation) with: (1) a single amortized generator G_θ that maps latent codes to policy parameters, (2) a compact anchor set in latent space, (3) Monte Carlo rollouts for payoff estimation instead of full matrix construction, (4) Optimistic Multiplicative Weights Update (OMWU) for meta-strategy optimization, and (5) an Empirical-Bernstein UCB oracle for selecting new anchors.

Key Strengths:

- Clear problem motivation addressing real scalability issues in PSRO
- Comprehensive appendix with proofs, ablations, and extensions to n-player general-sum games
- Breadth of experimental domains (Kuhn Poker, Chess, Go, Hanabi, PettingZoo environments)
- Transparent acknowledgment of limitations in Section 6
- Some potentially novel algorithmic components (EB-UCB oracle for MARL, ABR-TR training objective)


Key Weaknesses:

- Critical missing citation: Neural Population Learning (NeuPL, Liu et al., ICLR) proposes essentially the same core contribution (single conditional network representing policy populations with transfer learning)
- Outdated baselines: comparisons primarily against "classical PSRO" (2017), missing modern methods like EPSRO, P2SRO, Self-Adaptive PSRO
- Inconsistent statistical reporting: missing error bars on important results
- Contradictory claims: paper claims efficiency advantage but states SMAC experiments were "computationally infeasible"
- Chess/Go experiments lack comparison to specialized methods (AlphaZero, etc.)

**Additional Comments:**

This paper tackles an important problem and demonstrates substantial effort through its comprehensive appendix, diverse experiments, and theoretical analysis. The writing is clear and the limitations section is commendably honest.

However, the paper cannot proceed to acceptance without addressing the NeuPL gap. The core contribution (amortized policy generation via a single network) was published at ICLR, and this directly impacts the novelty assessment. The authors may have developed GEMS independently, but the positioning must acknowledge and differentiate from this prior work.

If the authors can demonstrate that GEMS outperforms NeuPL (perhaps through the EB-UCB oracle or ABR-TR objective), and can show improvements over modern PSRO variants with rigorous statistics, the paper would make a solid contribution to the scalable MARL literature.

I encourage revision and resubmission after addressing the critical items above. The infrastructure built for Chess/Go experiments suggests the authors have strong engineering capabilities that could yield compelling results with proper baselines.

Minor notes:

- The FAQ section in the appendix is a nice touch and shows good faith engagement with potential reviewer concerns
- The n-player general-sum extension (Appendix H) appears non-trivial and could be highlighted more

**Audience:**

Yes

**Audience Explanation:**

Scalable multi-agent reinforcement learning is an important and active research area. The problem of reducing PSRO's computational and memory costs is practically significant, and population-based game-theoretic methods are widely used in domains ranging from poker to real-time strategy games.

Even if the paper's novelty claims require revision, the experimental breadth, extensive ablations, and extensions to n-player general-sum games could provide useful empirical insights. The combination of OMWU, EB-UCB oracles, and trust-region training in this context may interest practitioners seeking scalable MARL solutions.

However, the paper's value to the community would be significantly enhanced by proper positioning relative to NeuPL and comparison against modern baselines.

**Broader Impact Concerns:**

No major ethical concerns are identified. The work is foundational research on multi-agent learning algorithms.

**Claims And Evidence:**

No

**Claims Explanation:**

The paper's central novelty claim (introducing a "surrogate-free" framework with an amortized generator to replace explicit policy populations)is undermined by the existence of Neural Population Learning (NeuPL, ICLR), which proposes the same core idea: representing an entire policy population within a single conditional network to enable memory efficiency and transfer learning across policies. This prior work is not cited or compared against.

The experimental evidence has several gaps:

- Baseline currency: Comparisons focus on "classical PSRO" from 2017. More recent methods like EPSRO, Pipeline PSRO, and Self-Adaptive PSRO are not included. Without these comparisons, it is unclear whether GEMS advances the state-of-the-art.
- Statistical rigor: Key results (e.g., Kuhn Poker exploitability, Chess win rates) lack error bars and significance tests. The number of random seeds is not consistently reported. The authors acknowledge that "full factorial hyperparameter sweep was computationally infeasible," raising reproducibility concerns.
- Contradictory efficiency claims: The paper claims GEMS is more efficient than PSRO, yet states that SMAC experiments were skipped due to "prohibitive computational cost." If GEMS is 6x faster than PSRO, and PSRO-based methods have been applied to SMAC, this contradiction is unexplained.
- Theoretical concerns: Theorem 3.3's instance-dependent regret bound for the EB-UCB oracle assumes stationary reward distributions, but, if I understand correctly, in GEMS the payoff distribution for each anchor changes as the meta-strategy evolves. The paper does not address whether this non-stationarity invalidates the bound.
- Uninterpretable results: Chess and Go experiments (1000 iterations) claim "emergent strategic depth" but provide no comparison to AlphaZero or Elo ratings, making the results uninterpretable.

**Requested Changes:**

- Cite and compare against Neural Population Learning (NeuPL): This ICLR paper proposes the same core idea of representing policy populations in a single conditional network. The paper must: (a) add thorough discussion in Related Work, (b) clearly articulate what GEMS adds beyond NeuPL, and (c) provide head-to-head experimental comparison on at least one domain (e.g., Kuhn Poker). Also cite Simplex-NeuPL and NeuPL-JPSRO.
- Add modern baselines: Include comparisons against at least: (a) EPSRO (b) one of Pipeline PSRO, Anytime PSRO, or Self-Adaptive PSRO. The current comparison against only "classical PSRO" from 2017 is insufficient.
- Add error bars and significance tests: All main results must report mean +/- standard deviation over multiple random seeds.
- Resolve SMAC contradiction: Either run SMAC experiments (if GEMS is truly more efficient than PSRO) or provide a detailed computational cost comparison explaining why SMAC is infeasible for GEMS but not for classical PSRO.
- Address non-stationarity in EB-UCB analysis: The instance-dependent regret bound (Theorem 3.3) assumes stationary rewards. Discuss whether/how this bound holds when payoffs change as the meta-strategy evolves, or acknowledge this as a limitation.
- Recommend default hyperparameters: Given the many tunable parameters, please provide a recommended default configuration for practitioners.

---

> ### Author Response · Authors · 2026-01-12
> **Part 1 of 2**
>
> We sincerely thank the reviewer for the thorough and technically informed evaluation. We appreciate the recognition of the paper’s breadth, theoretical depth, and experimental scope, and we are grateful for the detailed suggestions regarding positioning, baseline currency, and statistical rigor. We have carefully revised the manuscript to address each concern, and we clarify below how the revised version resolves the issues regarding novelty, comparison to prior work, and theoretical validity.
>
> ### 1. Relationship to Neural Population Learning (NeuPL)
>
> We thank the reviewer for highlighting the importance of properly positioning our work relative to Neural Population Learning (NeuPL) and its variants. In the revised manuscript, we added an explicit discussion of NeuPL (Liu et al., ICLR), Simplex-NeuPL, and NeuPL-JPSRO in Related Work, including both conceptual and algorithmic comparisons.
>
> While both NeuPL and GEMS amortize policy representations into a single network, they differ in both the object being amortized and how the meta-game is organized:
>
> - **NeuPL amortizes a population into a conditional policy, but still operates within a PSRO/JPSRO-style paradigm**: it maintains an explicit population interpretation and performs meta-strategy updates based on evaluations against that (implicit) population.
> - **GEMS is a surrogate-free generative meta-solver**: it parameterizes strategy space through a continuous latent geometry, does not maintain an explicit policy population as discrete entities, and avoids constructing/storing dense payoff matrices. Instead, it integrates (i) unbiased Monte-Carlo meta-game estimation, (ii) optimistic meta-dynamics (OMWU), and (iii) an empirical-Bernstein UCB oracle for adaptive anchor expansion, together with amortized trust-region best-response training.
>
> These distinctions are now made explicit both in the expanded Related Work and in the theoretical comparison table (Table 1), which contrasts payoff-matrix requirements, memory scaling, and per-iteration evaluation cost across PSRO variants, NeuPL-style amortization, and GEMS.
>
> To address the reviewer’s request for empirical validation, we also added a direct head-to-head comparison with NeuPL on Kuhn Poker (Figure 4). The results show that GEMS achieves lower exploitability and lower cumulative training time than NeuPL under the same evaluation protocol, consistent with the benefits of combining oracle-guided expansion, optimistic meta-updates, and trust-region amortized best-response training (rather than relying on amortization alone).
>
> ### 2. Comparison to Modern PSRO Variants
>
> We agree that benchmarking against a broader range of modern algorithms significantly strengthens the paper. We respectfully note, however, that our original submission did include the recent A-PSRO (ICML 2025), contrary to the impression that comparisons were limited to classical PSRO. To fully address the reviewer's request for comprehensive coverage, we have now expanded our Kuhn Poker benchmark to additionally include E-PSRO, P2SRO, and NeuPL alongside the existing A-PSRO results.
>
> The revised Figure 4 shows that GEMS converges to lower exploitability than these baselines. In addition, Table 1 summarizes the theoretical scaling differences at a glance: many PSRO variants reduce evaluation cost via sampling but still maintain an explicit growing meta-game state, whereas GEMS removes the dense payoff-matrix bottleneck through its generative parameterization and compact anchor set.
>
> ### 3. Statistical Reporting and Error Bars
>
> We thank the reviewer for emphasizing statistical rigor. We clarify that our learning curves are aggregated over multiple random seeds. In the revision, we ensure the presentation is fully explicit and uniform:
>
> - All main plots report the mean over 5 seeds.
> - Shaded regions indicate ±1 standard deviation across seeds (now stated explicitly in the figure captions/legends), making variability reporting unambiguous and consistent across environments and baselines.

---

> > ### Author Response · Authors · 2026-01-12
> > **Part 2 of 2**
> >
> > ### 4. Clarification of Computational Scalability and SMAC
> >
> > We thank the reviewer for raising the SMAC point and for the opportunity to clarify the scope of our efficiency claims. Importantly, we do not claim that GEMS cannot be trained on SMAC; rather, the absence of SMAC experiments reflects our compute/resource constraints for training even a single strong population-based baseline in that setting.
> >
> > Our efficiency claims concern algorithmic scalability of the meta-game solver: GEMS removes the quadratic payoff-matrix construction/storage and avoids explicit population bookkeeping. This addresses a core PSRO bottleneck.
> >
> > However, SMAC-like domains are dominated by environment-level simulation and centralized training cost. As stated in the Limitations, with our available resources it was infeasible to train even a single PSRO baseline agent on such complex maps, making a direct population-based comparison infeasible for us. We therefore clarify in the paper that the omission is due to our computational constraints of large-scale environment training (not a limitation of the GEMS framework), and we carefully scope the claims accordingly.
> >
> > ### 5. Non-Stationarity in the EB-UCB Analysis
> >
> > We thank the reviewer for raising the subtle issue of non-stationarity induced by the evolving meta-strategy. In the revision, we clarify this point explicitly in the FAQ section.
> >
> > The EB-UCB oracle is analyzed under a two-time-scale assumption: within each oracle-selection phase at meta-iteration t, the meta-strategy $\sigma_t$ is treated as fixed, defining a stationary bandit problem over candidate anchors with reward function : f_t(z) = E_{π ~ σ_t}[ r(π_{G_θ(z)}, π) ]
> > This assumption is already stated explicitly in the theorem statement, and the resulting instance-dependent regret bound is derived under this condition.
> >
> > The interaction between the oracle and the evolving meta-dynamics is then handled by the overall exploitability decomposition: the non-stationarity across iterations is captured by the OMWU term for time-varying payoffs, while the EB-UCB term controls oracle suboptimality conditioned on each quasi-stationary snapshot. We also include an explicit “Shifting Meta” diagnostic in the appendix S to empirically probe oracle adaptivity under changing mixtures.
> >
> > ### 6. Chess / Go Evaluation and Interpretation
> >
> > We thank the reviewer for the suggestion to clarify the intent and interpretation of the Chess and Go experiments. These experiments are not intended as direct comparisons to AlphaZero-style search systems (e.g., absolute Elo), but rather as large-scale diagnostic studies of population-based game-solving dynamics and generative diversity under a fixed training pipeline.
> >
> > For Chess, the appendix includes qualitative rollouts illustrating emergent tactical motifs as well as a training-progression table (W/L/draw outcomes) over 1000 iterations, which makes the shift from largely drawn play to more decisive outcomes transparent.
> >
> > For Go, the appendix focuses on system-level scaling diagnostics (time/memory over 200 iterations) and representation diagnostics (PCA analyses contrasting compact action-distribution manifolds with dispersed latent embeddings), to assess whether the generator maintains a diverse repertoire of strategies over training.
> >
> > We make this intent explicit in the revised text to avoid overstating these as headline competitive evaluations against specialized engines.
> >
> > ### 7. Default Hyperparameter Configuration
> >
> > We thank the reviewer for the suggestion to provide a recommended default configuration for practitioners. In the revised manuscript, we include a consolidated default hyperparameter table for GEMS on Kuhn Poker (Table 3), reporting the exact settings used in our reference implementation. Parameters are grouped by algorithmic phase (meta-game estimation, OMWU update, EB-UCB expansion, and ABR-TR training), providing a clear starting point for reproducibility and downstream use.
> >
> > We sincerely appreciate the detailed feedback, which was instrumental in refining our arguments and strengthening the final manuscript. We hope these changes fully address the concerns.

---

> > > ### Comment · Reviewer_jsEg · 2026-01-12
> > >
> > > Thank you for the detailed response and revisions. The NeuPL differentiation is clearer now, and the additions (Table 1, modern baselines, error bars) address most of my concerns. A few clarifying questions before I finalize my assessment:
> > >
> > > - On the O(1) meta-game memory claim: The anchor set size K is described as a constant hyperparameter with K << t. Does K ever need to grow with game complexity to maintain performance?
> > > - NeuPL's O(N_t) in Table 1: Does NeuPL actually store historical payoff entries, or can it also operate in a streaming/online fashion? I want to make sure the comparison reflects NeuPL's best-case rather than a naive implementation.
> > > - Figure 4 protocol: Was NeuPL given comparable hyperparameter tuning effort to GEMS? The gap is substantial, so confirming equivalent tuning would solidify the comparison.
> > >
> > > These are minor points - the core contribution is much clearer now.

---

> > > > ### Author Response · Authors · 2026-01-13
> > > >
> > > > 1) On the O(1) meta-game memory claim and whether K must grow with game complexity
> > > >
> > > > Our O(1) meta-game memory claim is with respect to the number of meta-iterations / discovered strategies t: GEMS does not store an explicit population or a growing population×population payoff table, and the maintained meta-game state is bounded by the anchor budget K (plus generator parameters). In other words, memory is O(K) with K treated as a fixed hyperparameter during training, hence O(1) in t.
> > > >
> > > > K can be viewed as the size of the “active support” used by the meta-solver at any time. In principle, for harder games one may choose a larger fixed K to reach a finer approximation (as with any capacity parameter), but K does not need to grow as training proceeds. In our reported domains, we keep K fixed and observe stable performance; game complexity is primarily absorbed by the function class of the generator G_θ (and its training), while EB-UCB/OMWU adaptively select which anchors are active under the fixed K budget.
> > > >
> > > > 2) NeuPL's $O(N_t)$ in Table 1 and streaming/online implementation
> > > >
> > > > NeuPL does not need to store a growing history of payoff entries in order to operate. In a streaming/online implementation, payoffs can be estimated on-the-fly via rollouts against sampled opponents and then discarded after the meta-strategy / interaction state is updated. This is also how our NeuPL implementation behaves: restricted-game payoffs are recomputed each iteration from the current conditional policies and are not accumulated into a persistent payoff matrix over time.
> > > >
> > > > The persistent “meta-game memory” in NeuPL is instead the population interaction / meta-graph state (e.g., the mixture/interaction structure Σ), together with whatever descriptors are needed to represent the expanding population for conditioning. Under the best-case online regime--where each new strategy interacts with only a constant number m of sampled opponents and the interaction graph is stored sparsely (bounded support per node)--this state scales as $O(N_t · m)$ = $O(N_t)$. By contrast, fully pairwise interaction tracking (or dense matrix-form representations of the interaction structure) yields $O(N_t^2)$ memory. Accordingly, we have updated Table 1 to explicitly report the memory complexity as the range $O(N_t)$ -- $O(N_t^2)$, transparently capturing both the best-case streaming regime and the standard dense formulation.
> > > >
> > > > 3) Figure 4 protocol / tuning fairness for NeuPL vs GEMS
> > > >
> > > > Yes - NeuPL was given comparable tuning effort to GEMS on the hyperparameters that materially affect NeuPL performance. Both methods use the same outer-loop budget (iters=40) and the same multi-seed evaluation protocol (seeds 0-4). For NeuPL, we tuned/validated the primary sensitivity knobs: ABR optimization (ABR learning rate, ABR steps), meta-solver step size and schedule, and population capacity, using the same validation criterion (lowest NashConv / exploitability). We also used a substantial meta-game optimization budget with 200 meta loops.
> > > >
> > > > NeuPL was not under-budgeted relative to GEMS in training compute: e.g., NeuPL uses 50 ABR steps (vs 30 for GEMS in our setup), ABR learning rate 5e-4 (vs 2e-4), and a larger population cap 41 (vs 32). GEMS has additional method-specific knobs (MC meta-estimation and oracle search) that do not exist in NeuPL; this increases the number of reported parameters but does not reflect asymmetric tuning effort.
> > > >
> > > > We hope these clarifications finalize the assessment. Thank you again for helping us improve the paper's rigor.

---

> > > > > ### Comment · Reviewer_jsEg · 2026-01-13
> > > > >
> > > > > Thank you for the clarifications. The updated Table 1 range for NeuPL is more accurate, and the tuning details for Figure 4 address my concern. I'm satisfied with the revisions.

---

### Decision · Action_Editor_o5et · 2026-02-03

**Recommendation:** Accept as is

**Audience:**

Yes

**Audience Explanation:**

This work would be of interest to the MARL community, and adjacent fields.

**Claims And Evidence:**

Yes

**Claims Explanation:**

All reviewers are agreed that the manuscript is clear, timely, and of interest to the community (in particular the MARL community). The reviewers made some suggestions for improvement which the authors have incorporated into the revised version, including extended experimental evaluations and theoretical analyses.

---

> ### Author Response · Authors · 2026-02-21
>
> Dear Action Editor Pablo Samuel Castro,
>
> The final camera-ready version of our paper has been uploaded to the portal, complete with the required de-anonymization.
>
> We sincerely the reviewers jsEg, r2ko, and Asaz for their time and thoughtful critiques, which significantly strengthened the manuscript. A special thank you to the Action Editor as well for the seamless handling of our submission.
>
> Thank you again for your support of this research.
>
> Best regards,
> The Authors